# Emergence and scaling laws in SGD learning of shallow neural networks

**Yunwei Ren**[*,1], **Eshaan Nichani**[*,1], **Denny Wu**[2,3], **Jason D. Lee**[1]

[1]Princeton University,  [2]New York University,  [3]Flatiron Institute
{yunwei.ren, eshnich, jasonlee}@princeton.edu,  dennywu@nyu.edu

## Abstract

We study the complexity of online stochastic gradient descent (SGD) for learning a two-layer neural network with $P$ neurons on isotropic Gaussian data: $f_*(\boldsymbol{x}) = \sum_{p=1}^{P} a_p \cdot \sigma(\langle \boldsymbol{x}, \boldsymbol{v}_p^* \rangle)$, $\boldsymbol{x} \sim \mathcal{N}(0, \boldsymbol{I}_d)$, where the activation $\sigma$ is an even function with information exponent $k_* > 2$ (defined as the lowest degree in Hermite expansion), $\{\boldsymbol{v}_p^*\}_{p \in [P]} \subset \mathbb{R}^d$ are orthonormal signal directions, and non-negative second-layer coefficients satisfy $\sum_p a_p^2 = 1$. We focus on the challenging "extensive-width" regime $P \gg 1$ and permit diverging condition number in the second-layer, covering as a special case the power-law scaling $a_p \asymp p^{-\beta}$ where $\beta \in \mathbb{R}_{\geq 0}$. We provide a precise analysis of SGD dynamics for the training of a student two-layer network to minimize the mean squared error (MSE) objective, and identify sharp transition times to recover each signal direction. In the power-law setting, we characterize scaling law exponents for the MSE loss with respect to the number of training samples and SGD steps, as well as the number of trainable parameters. Our analysis entails that while the learning of individual teacher neurons exhibits abrupt transitions, the juxtaposition of $P \gg 1$ emergent learning curves at different timescales leads to a smooth scaling law in the cumulative objective.

## 1 Introduction

Recent works have studied the gradient-based training of shallow neural networks for learning low-dimensional target functions (i.e., functions in $\mathbb{R}^d$ that depend on $P \ll d$ directions), such as single-index models [BAGJ21, BES+22, BBSS22, DNGL23, BMZ23, DPVLB24] and multi-index models [DLS22, AAM22, BBPV23, CWPPS23, BAGP24, TDD+24], to illustrate the adaptivity (and hence the improved statistical efficiency) of neural networks through feature learning. For such target functions on unstructured (isotropic) input data, it is known that optimization may exhibit an *emergent* risk curve: learning undergoes an extensive "search phase" during which the loss plateaus (the length of which depends on properties of the nonlinearity), followed by a sharp "descent phase" where strong recovery is achieved rapidly. For instance, when the target is a single-index model $f_*(\boldsymbol{x}) = \sigma(\boldsymbol{x} \cdot \boldsymbol{\theta})$, $\boldsymbol{\theta} \in \mathbb{R}^d$, the initial search phase of online SGD scales as $t \asymp d^{\Theta(k_*)}$, where $k_* \in \mathbb{R}_+$ is the *information exponent* of the link function $\sigma$ (defined as the index of its first nonzero Hermite coefficient [DH18, BAGJ21]), whereas the final descent phase occurs in $\eta t = \tilde{\Theta}(1)$ time.

The sharp phase transition observed in the gradient-based learning of low-dimensional target functions may seem at odds with the phenomenon of *neural scaling laws* [HNA+17, KMH+20, HBM+22], where increasing compute and data empirically leads to a predictable power-law decay in the loss. A plausible explanation lies in considering an *additive model*, where the objective can be decomposed into a large number of distinct "skills", each of which occupies only a small fraction of the trainable parameters [DDH+21, EHO+22, PSZA23]. While the acquisition of individual skills may exhibit abrupt transitions – empirically observed in [WTB+22, GHL+22] – the juxtaposition of numerous

---

[*]Equal contribution.

39th Conference on Neural Information Processing Systems (NeurIPS 2025).

emergent learning curves occurring at different timescales results in a smooth power-law rate for the cumulative objective [MLGT24, NFLL24].

Motivated by the above, we consider an idealized setting where each learning task is represented by a Gaussian single-index model, so the additive model reduces to a two-layer neural network

$$f_*(\boldsymbol{x}) = \sum_{p=1}^{P} a_p \, \sigma(\boldsymbol{v}_p^* \cdot \boldsymbol{x}), \quad \boldsymbol{x} \sim \mathcal{N}(0, \boldsymbol{I}_d),$$

where $\{\boldsymbol{v}_p^*\}_{p=1}^{P}$ are orthonormal index features, $a_1 \geq \cdots \geq a_P \geq 0$ are second-layer weights ordered in descending magnitude, and $\sigma : \mathbb{R} \to \mathbb{R}$ is an even activation function with information exponent $k_* > 2$; this implies that (online) SGD learning of each task has an emergent learning curve with $\mathrm{poly}(d)$ initial plateau. This target function is a subclass of multi-index models (with ridge-separable nonlinearity), for which the complexity of gradient-based optimization has been recently studied [OSSW24, SBH24, RL24]. We highlight the following technical challenges to be addressed.

- **Extensive width** ($P \gg 1$). Most existing results on SGD learning have focused on the "narrow-width" regime such as $P = 1$ for single-index models [BAGJ21, DNGL23, MHPG⁺22, DTA⁺24, LOSW24] and $P = O_d(1)$ for multi-index models [DLS22, BBPV23, DKL⁺23, BAGP24, ZG24]. However, to obtain a smooth power-law scaling from a sum of "discrete" learning curves, the number of tasks should be large; this motivates us to study the extensive-width regime where we allow $P \to \infty$ as $d \to \infty$, which yields an *infinite-dimensional* effective dynamics [BAGJ22].

- **Large condition number** ($\frac{a_{\max}}{a_{\min}} \gg 1$). Existing works in the extensive-width regime usually assumed identical second layer ($a_1 = ... = a_P$) [RL24, SBH24] or proved optimization complexity that scales exponentially with the condition number $\kappa = \frac{a_{\max}}{a_{\min}}$ [LMZ20, OSSW24] (to our knowledge the only exceptions are [GRWZ21, BAGP24] which considered unnatural algorithmic modifications such as Stiefel constraint or tensor deflation with re-initialization). Such exponential dependency implies that in the poly-time learnable regime $\kappa = O_d(1)$, the signal strength for individual tasks can only differ by constant, and consequently, there is insufficient timescale separation to produce a power-law risk curve. We thus focus on the challenging large condition number regime $\kappa \gg 1$.

- **Single-phase training.** Prior works on multi-index learning typically employed a layer-wise training procedure, where correlation loss SGD is first applied to the first-layer parameters to recover the index features, followed by convex optimization to solve for the optimal second layer [DLS22, BES⁺22, AAM23, OSSW24]. Such stage-wise training creates complications in the scaling law description due to the changing computational procedure. Hence we aim to characterize a natural, single-phase algorithm where both layers are updated simultaneously.

## 1.1 Our Contributions

We study the learning of an additive model (1) with orthogonal first-layer weights and even activation with information exponent $k_* > 2$, using a student two-layer network with $m$ neurons trained via online SGD to minimize the mean squared error (MSE) loss. We consider the extensive-width regime $P \gg 1$, and allow the scale of second-layer parameters of the target (teacher model) to depend polynomially on the width $P$. We establish polynomial runtime and sample complexity for single-phase SGD training and provide a sharp characterization of the recovery time for each teacher neuron.

**Theorem** ((Informal) sample complexity). *Assume the teacher model has $P \lesssim d^c$ orthogonal neurons for some small but fixed $c > 0$, and the activation $\sigma$ is an even function with information exponent $k_* > 2$. To recover the top $P_* \leq P$ teacher directions, we can train a student network (2) with $m = \tilde{\Theta}(P_*)$ neurons via online SGD with sample and runtime complexity $n \asymp T \asymp a_{P_*}^{-2} \cdot d^{k_*-1}\mathrm{poly}(P)$.*

As a corollary, we know that a student width $m = \tilde{\Theta}(P)$ and sample size $n = \tilde{\Theta}(a_{\min}^{-2} d^{k_*-1}\mathrm{poly}(P))$ are sufficient to learn all teacher neurons, where $a_{\min} := \min_{p \in [P]} a_p$. Prior to our work, [OSSW24] studied the learning of the same target function class using a layer-wise training procedure that deviates from common practice. Their analysis requires $m \gtrsim P^{\Omega(1/a_{\min})}$ student neurons, which is computationally prohibitive since $P, a_{\min}^{-1}$ can both scale with the dimensionality $d$. Interestingly, we show that this limitation can be overcome by considering an arguably more natural single-phase training algorithm. At a technical level, our analysis leverages the following key ingredients.

- *Single-stage training.* We consider a 2-homogeneous student model and simultaneously train both layers via online SGD under the MSE loss; this differs from prior layer-wise analyses where the first-layer weights are optimized under correlation loss. In our large condition number

setting, the correlation loss analysis yields super-polynomial complexity to compensate for the signal discrepancy across different tasks [OSSW24]; in contrast, our single-phase MSE dynamics circumvents this issue by automatically deflating the learned tasks from the loss.

- *Decoupled dynamics.* When $P \gg 1$, the effective dynamics of SGD cannot be captured by a finite set of summary statistics. To understand the convergence of this high-dimensional system, we show that the evolution of different signal directions can be approximately decoupled via an "automatic" deflation mechanism and carefully controlling the influence of the irrelevant coordinates.

Applying our general learnability result, we precisely characterize the scaling of the population loss along the online SGD trajectory in the following power-law setting.

**Proposition** ((Informal) scaling law). *Under the same conditions and hyperparameters as the previous theorem, and assuming $a_p \asymp p^{-\beta}$ for $\beta > 1/2$, then (ignoring logarithmic factors) we have*

(a) **Emergence.** *The $p$-th teacher neuron (where $p \lesssim m$) is recovered at time $\eta t \sim p^\beta d^{k_*/2-1}$.*

(b) **Scaling law.** *The population squared error follows a power-law decay up to approximation barrier $\mathcal{L}(t) \sim \left(t\eta d^{1-k_*/2}\right)^{\frac{1-2\beta}{\beta}} \vee m^{1-2\beta}$.*

This proposition confirms the additive model intuition from [MLGT24, NFLL24] in a high-dimensional feature learning setting, where the length of the "search phase" (plateau) for each feature direction $v_p^*$ is modulated by the magnitude of the second-layer coefficient $a_p$, and the simultaneous learning of all directions yields a power-law decay in the cumulative loss (see Figure 1). However, unlike these prior works, our problem setting does not imply that the learning of different tasks can be decoupled *a priori*, as student neurons may be attracted to multiple teacher directions and also interact with each other through the squared loss.

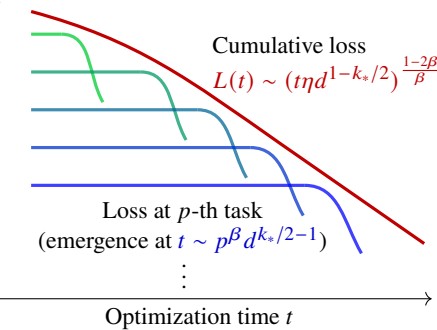

Figure 1: Power-law scaling of MSE loss as a result of superposition of emergent risk curves.

## 2 Problem Setting and Main Results

In this section, we present our main results on SGD learning and scaling laws.

### 2.1 Setting and Algorithm

**Architecture: two-layer neural network.** Let $\sigma : \mathbb{R} \to \mathbb{R}$ denote the nonlinear link function. We assume the target function is given by the following additive model

$$f_*(\boldsymbol{x}) = \sum_{p=1}^{P} a_p \sigma(\boldsymbol{v}_p^* \cdot \boldsymbol{x}), \quad \forall \boldsymbol{x} \in \mathbb{R}^d, \tag{1}$$

where $\boldsymbol{x} \sim \gamma := \mathcal{N}(0, \boldsymbol{I}_d)$ is the input, $\{\boldsymbol{v}_p^*\}_{p \in [P]} \subset \mathbb{R}^d$ are orthonormal with $P \gg 1$, $\sigma \in L^2(\gamma)$ satisfies Assumption 2.1, and $a_1 \geq \cdots \geq a_P \geq 0$ are normalized so that $\sum_p a_p^2 = 1$. Since the input distribution and our learning algorithm are rotationally invariant, we may assume w.l.o.g. that $\boldsymbol{v}_p^* = \boldsymbol{e}_p$, where $\boldsymbol{e}_p \in \mathbb{R}^d$ is the $p$-th standard basis vector. While our scaling results will assume $a_p$ follows a power law decay, no such assumptions are required for our optimization results.

**Assumption 2.1** (Link function). *Let $\{h_k\}_{k \in \mathbb{N}_{\geq 0}}$ denote the normalized Hermite polynomials.*

(a) *$\sigma$ is even and has information exponent $\mathrm{IE}(\sigma) = 2I$ for $I > 1$, that is, the Hermite expansion of $\sigma$ is given as $\sigma = \sum_{i=I}^{\infty} \hat{\sigma}_{2i} h_{2i}$, and we require $\hat{\sigma}_{2I} \geq c_\sigma$; we also assume $\|\sigma\|_{L^2(\gamma)} = 1$, and $\|\sigma'\|_{L^2(\gamma)}, \|\sigma''\|_{L^2(\gamma)} \leq C_\sigma$, where constants $c_\sigma, C_\sigma > 0$.*

(b) *$\sigma$ and $\sigma'$ have polynomial growth. That is, there exist universal constants $C, Q > 0$ such that $|\sigma(x)| \vee |\sigma'(x)| \leq C(1 + x^2)^{Q/2}$ for all $x \in \mathbb{R}$.*

**Remark.** We focus on high information exponent $\mathrm{IE}(\sigma) > 2$ link functions as in [OSSW24, SBH24, GWB25]. This setting entails that the learning of each single-index task is "hard" in the sense that online SGD exhibits a long loss plateau, and we utilize this assumption to prove (approximate) decoupling of individual tasks. The condition on even $\sigma$ simplifies the analysis by removing the $1/2$ probability of neurons initialized in the wrong hemisphere (see e.g., [BAGJ21]).

Our learner network (student model) is a width-$m$ two-layer neural network:

$$f(\boldsymbol{x}) := f\left(\boldsymbol{x}; \{\boldsymbol{v}_k\}_{k=1}^m\right) = \sum_{k=1}^m \|\boldsymbol{v}_k\|^2\, \sigma(\bar{\boldsymbol{v}}_k \cdot \boldsymbol{x}), \tag{2}$$

where $\{\boldsymbol{v}_k\}_{k=1}^m \subset \mathbb{R}^d$ are trainable parameters and $\bar{\boldsymbol{v}}_k := \boldsymbol{v}_k/\|\boldsymbol{v}_k\|$. Note that this student network is parameterized to be 2-homogeneous in each $\boldsymbol{v}_k$, i.e., the second-layer coefficients are coupled with the norm of the first-layer weights. Such 2-homogeneous parameterization has been used in many prior works [LMZ20, WWL+20, GRWZ21]; this setting originated from the analysis of training both layers of ReLU networks under balanced initialization (see e.g., [CB20]), and allows us to couple the growth of the second layer norm $\|\boldsymbol{v}_k\|^2$ with the direction convergence of $\bar{\boldsymbol{v}}_k$. We believe that a similar proof strategy can be applied to simultaneous training of networks with decoupled second-layer weights.

**Algorithm: online SGD.** The performance of the learner is measured using the mean squared error (MSE) loss. For each $\boldsymbol{x} \in \mathbb{R}^d$, the per-sample MSE loss is defined as

$$l(\boldsymbol{x}) = l\left(\boldsymbol{x}; \{\boldsymbol{v}_k\}_{k=1}^m\right) = \frac{1}{2}\left(f_*(\boldsymbol{x}) - f\left(\boldsymbol{x}; \{\boldsymbol{v}_k\}_{k=1}^m\right)\right)^2. \tag{3}$$

Using a Hermite expansion calculation ([GLM18]), one can show that the population MSE loss can be expressed as a tensor decomposition loss as follows:

$$\mathcal{L} := \underset{\mathcal{N}(0, \boldsymbol{I}_d)}{\mathbb{E}}[l(\boldsymbol{x})] = \sum_{i=I}^{\infty} \hat{\sigma}_{2i}^2\left(\frac{\|\boldsymbol{a}\|^2}{2} - \sum_{p=1}^P \sum_{k=1}^m a_p \|\boldsymbol{v}_k\|^2 \langle \bar{\boldsymbol{v}}_k, \boldsymbol{v}_p^*\rangle^{2i} + \frac{1}{2}\sum_{k,l=1}^m \|\boldsymbol{v}_k\|^2 \|\boldsymbol{v}_l\|^2 \langle \bar{\boldsymbol{v}}_k, \bar{\boldsymbol{v}}_l\rangle^{2i}\right). \tag{4}$$

We use online stochastic gradient descent (SGD) to train the learner model. Let $\{(\boldsymbol{x}_t, f_*(\boldsymbol{x}_t))\}_{t \in \mathbb{N}}$ be our dataset with $\boldsymbol{x}_t \overset{\text{i.i.d.}}{\sim} \mathcal{N}(0, \boldsymbol{I}_d)$ being the fresh sample at step $t$. We initialize the student neurons $\boldsymbol{v}_k \sim \text{Unif}(\mathbb{S}^{d-1}(\sigma_0))$, where $\sigma_0 = 1/\text{poly}(d)$ is a parameter we specify in the sequel. Let $\eta > 0$ be the step size. At each step, we update the neurons using vanilla gradient descent: $\boldsymbol{v}_k(t+1) = \boldsymbol{v}_k(t) - \eta \nabla_{\boldsymbol{v}_k} l(\boldsymbol{x}_t)$, for all $k \in [m]$, where $l$ is the per-sample loss defined in (3).

## 2.2 Complexity of SGD Learning

Our main theorem provides a sharp characterization of the sample complexity of online SGD and the recovery time of individual single-index tasks. To characterize the learning order of the first $P_* \leq P$ tasks, we introduce an ordering of student neurons $\boldsymbol{v}_1, \ldots, \boldsymbol{v}_m$ and a mapping $\pi : [P_*] \to [P]$ that specifies which student neurons converge to a particular task (teacher neuron). This mapping function is explicitly defined via the greedy maximum selection procedure (5) which we explain in Section 3.1 — intuitively speaking, after the reordering, for $p \in [P_*]$, $\boldsymbol{v}_p$ is the neuron that eventually converges to direction $\boldsymbol{v}_{\pi(p)}^*$, and the directions are learned sequentially based on the signal strength $\{a_p\}_{p=1}^P$.

Let $\bar{v}_{p,q}(t) := \langle \bar{\boldsymbol{v}}_p, \boldsymbol{v}_q^*\rangle$ denote the overlap between the $p$-th student neuron (ordered) and the $q$-th teacher neuron at time $t$. The following theorem describes the convergence of student neuron $\boldsymbol{v}_p$ to the corresponding teacher $\boldsymbol{v}_{\pi(p)}^*$ in terms of *direction*: $\bar{v}_{p,\pi(p)}^2(t) \to 1$, and *norm*: $\|\boldsymbol{v}_p(t)\|^2 \to a_{\pi(p)}$.

**Theorem 2.1** (Main theorem for online SGD). *Let $C, C' > 0$ be large universal constants, depending only on $I$ and $\sigma$, and set the initialization scale as $\sigma_0 = d^{-C}$. Let $P_* \in [P]$, $a_{\min_*} = \min_{p \in [P_*]} a_p$, and $\delta_{\mathbb{P}}^*$ be the target failure probability. Define $\Delta \simeq \frac{\delta_{\mathbb{P}}^*}{mP \max(m,P)} = o_d(1)$. Assume the dimension $d$, width $m$, learning rate $\eta$ and target accuracies $\varepsilon_D, \varepsilon_R = o_d(1)$ satisfy*

$$d \gtrsim \|\boldsymbol{a}\|_1^4 \Delta^{-8} a_{\min_*}^{-4}, \quad m \gtrsim P_*, \quad \eta \lesssim a_{\min_*} \|\boldsymbol{a}\|_1^{-2} m^{-1} P^{-1} \delta_{\mathbb{P}}^* \min(\Delta^2 d^{-I}, \varepsilon_D^2),$$

$$\Delta^6 d^{-1} \gtrsim \varepsilon_D \gtrsim \|\boldsymbol{a}\|_1 a_{\min_*}^{-1} d^{-I+1/4}, \quad P_*^{-1/2} \varepsilon_D^{1/2} \gtrsim \varepsilon_R \gtrsim \varepsilon_D,$$

*where $\lesssim, \gtrsim$ hide both constants and logarithmic factors. Then, with probability at least $1 - \delta_{\mathbb{P}}^*$, there exists an ordering of the student neurons $\boldsymbol{v}_1, \ldots, \boldsymbol{v}_m$ and a mapping $\pi : [P_*] \to [P]$ of student neurons to teacher neurons (see Equation (5)) such that, defining*

$$T_p := \left(4I(I-1)\hat{\sigma}_{2I}^2 a_{\pi(p)} \eta \bar{v}_{p,\pi(p)}^{2I-2}(0)\right)^{-1} \quad \forall p \in [P_*], \quad \text{and} \quad T_{\max} := (1 + \Delta/4) \max_{p \in [P_*]} T_p$$

*we have:*

*(a) (**Unused neurons**). $\|\boldsymbol{v}_k(t)\|^2 \leq d^{-C'} =: \sigma_1^2$ for all $k > P_*$.*

(b) (**Convergence**). $\bar{v}^2_{p,\pi(p)}(t) \geq 1-\varepsilon_D$, $\|\boldsymbol{v}_p(t)\|^2 = a_{\pi(p)}\pm\varepsilon_R$ for all $p \in [P_*]$, $(1+\Delta)T_p \leq t \leq T_{\max}$.

(c) (**Sharp Transition**). $\bar{v}^2_{p,\pi(p)}(t) \leq d^{-1/2}$, $\|\boldsymbol{v}_p(t)\|^2 \leq \sigma_1^2$ for all $p \in [P_*]$, $t \leq (1 - \Delta)T_p$.

(d) (**Loss Value**). At time $t$, the population loss of the student network can be bounded by

$$1 - \sum_{p\in[P_*]} a^2_{\pi(p)} \mathbb{1}\left\{t \geq (1-\Delta/4)T_p\right\} - O(\varepsilon_D) \leq \mathcal{L}(t) \leq 1 - \sum_{p\in[P_*]} a^2_{\pi(p)} \mathbb{1}\left\{t \geq (1+\Delta/4)T_p\right\} + O(\varepsilon_D).$$

We observe the following conclusions about Theorem 2.1.

- Points (b) and (c) suggest a *sharp transition* in the learning of the teacher neuron $\boldsymbol{v}^*_{\pi(p)}$ around time $T_p \simeq (\eta a_{\pi(p)}\langle\bar{\boldsymbol{v}}_p(0), \boldsymbol{v}^*_{\pi(p)}\rangle^{2(I-1)})^{-1}$. In particular, for time $t \leq (1 - o(1))T_p$, minimal progress is made on the learning of $\boldsymbol{v}^*_{\pi(p)}$, as $\langle\bar{\boldsymbol{v}}_p, \boldsymbol{v}^*_{\pi(p)}\rangle^2, \|\boldsymbol{v}_p\|^2/a_{\pi(p)} \ll 1$. Then, at some point during the short time interval $(1 \pm o(1))T_p$, both directional and norm convergence occur rapidly as the quantities $\langle\bar{\boldsymbol{v}}_p, \boldsymbol{v}^*_{\pi(p)}\rangle^2$ and $\|\boldsymbol{v}_p\|^2$ approach 1 and $a_{\pi(p)}$ respectively.

- The theorem implies that a student width of $m \gtrsim P_* \log(P_*)$ is sufficient to recover $P_*$ teacher neurons; this minimal (logarithmic) overparameterization allows us to establish near-optimal width dependence for the scaling laws in the ensuing section.

- Selecting $\eta = \tilde{\Theta}(a_{\min}d^{-I} \text{poly}(m, P))$, the runtime required to recover all directions $\{\boldsymbol{v}^*_k\}_{k\in[P]}$ up to $1/d$ error, and thus obtain an MSE loss of $O(1/d)$, is $T = \tilde{\Theta}(d^{2I-1} \text{poly}(P)a_{\min}^{-2}) = d^{\mathrm{IE}(\sigma)-1}P^{\Theta(1)}$, which is polynomial in all problem parameters — this contrasts with the exponential dependence on the condition number in [LMZ20, OSSW24]. Moreover, our Assumption 2.1 permits high-degree link functions; hence when $\deg(\sigma) \gg \mathrm{IE}(\sigma)$, the sample complexity in Theorem 2.1 is far superior to the $n \gtrsim d^{\deg(\sigma)}$ rate for neural networks in the kernel/lazy regime [JGH18, COB19, GMMM21].

## 2.3 Neural Scaling Laws

Now we apply Theorem 2.1 to the setting where the second-layer $a_p$ follows a power-law decay.

**Proposition 2.2** (Scaling laws). *Consider the same setting as in Theorem 2.1, and suppose $a_p = p^{-\beta}/Z$ where $\beta > 1/2$ and $Z = \sum_{p=1}^P p^{-2\beta}$ is the normalizing constant. Then, with high probability,*

(a) *For $p \leq P_* = \tilde{\Theta}(m)$, the $p$-th teacher neuron $\boldsymbol{v}^*_p$ is learned at time $t = \tilde{\Theta}(p^\beta d^{I-1}\eta^{-1})$.*

(b) *There exist constants $0 < c_\beta < C_\beta$ and $0 < c'_\beta < C'_\beta$ that can depend only on $\beta$ such that*

$$c_\beta\left[\left(\frac{m}{\log m}\right)^{1-2\beta} + \left(\frac{K_0\eta t}{d^{I-1}}\right)^{\frac{1-2\beta}{\beta}}\right] - O(\varepsilon_D) \leq \mathcal{L}(t) \leq C_\beta\left[\left(\frac{m}{\log m}\right)^{1-2\beta} + \left(\frac{K_0\eta t}{d^{I-1}}\right)^{\frac{1-2\beta}{\beta}}\right] + O(\varepsilon_D),$$

$\forall t \in [T_{\min}, T_{\max}]$, *where* $K_0 := \log^{2I-2} m/Z$, $T_{\min} = C'_\beta d^{I-1}/(K_0\eta)$, $T_{\max} = c'_\beta P^\beta d^{I-1}/(K_0\eta)$.

**Remark.** We make the following remarks.

- As in the literature on neural scaling laws [KMH+20, HBM+22, PPXP24], our scaling law in Proposition 2.2 consists of the *approximation bottleneck* $\tilde{\Theta}(m^{1-2\beta})$, governed by the width of the student network, and the *optimization bottleneck* $\Theta\left((\eta t d^{1-I})^{(1-2\beta)/\beta}\right)$, governed by the number of online SGD steps (or equivalently number of samples).

- Note that the times the first and last directions get learned are approximately $d^{I-1}/(K_0\eta)$ and $P^\beta d^{I-1}/(K_0\eta)$. Hence $[T_{\min}, T_{\max}]$ covers the time interval where most directions are learned.

- We state the risk scaling for square-summable second-layer coefficients $\beta > 1/2$ similar to prior theoretical works on scaling laws [BAP24, LWK+24]. In the "heavy-tailed" regime ($\beta < 1/2$), we can also apply Theorem 2.1 to obtain $\mathcal{L}(t) = \tilde{\Theta}\left((1 - (P/m)^{1-2\beta})_+ \vee (1 - (t\eta d^{I-1})^{(1-2\beta)/\beta})_+\right)$. Note that in this setting, the required student width is roughly proportional to the teacher width $m = \tilde{\Theta}(P)$ in order to achieve small approximation error.

**"Unstable" discretization.** Given a fixed training budget $t$, it can be quite pessimistic to choose the learning rate $\eta \propto a_{\min_*} \asymp a_{P_*}$ for $P_* = \tilde{\Theta}(m)$, since at any $t \ll (\eta a_{\pi(P_*)}\bar{v}^{2I-2}_{P_*,\pi(P_*)}(0))^{-1}$, far fewer than $P_*$ directions are learned. As such, consider pre-specifying the runtime $t$ (or equivalently the sample size $n$). If we only are interested in learning the top $p$ neurons, we can apply Theorem 2.1

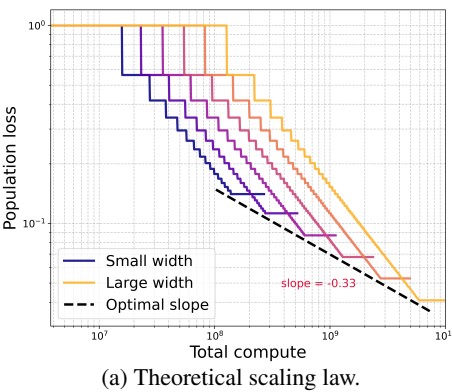
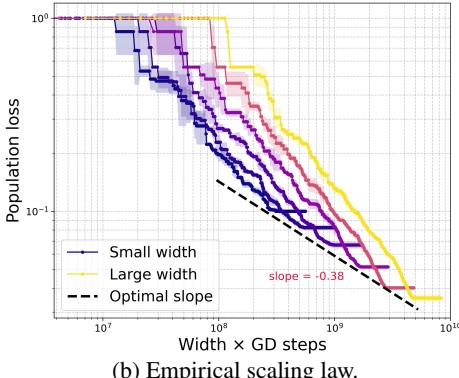

(a) Theoretical scaling law.      (b) Empirical scaling law.

Figure 2: Theoretical and empirical risk curves with $\beta = 0.8$. ($a$) Idealized scaling curves described in Section 3.1. ($b$) Empirical scaling curve of GD training on the population loss with $d = 2048, P = 1024$.

with $P_* = p$, which gives a larger learning rate of $\eta = \tilde{\Theta}(\frac{a_{\pi(p)}d^{-I}}{\text{poly}(P)})$. The $p$-th direction is now learned at $T_p = \tilde{\Theta}(a_{\pi(p)}^{-2}d^{2I-1}\text{poly}(P)) = \tilde{\Theta}(p^{2\beta}d^{2I-1}\text{poly}(P))$. This leads to an "unstable" scaling law.

**Corollary 2.3** (Unstable scaling law). *Let $m$ be the student network width and $n$ be the total number of training examples. Then, there exists a choice of learning rate $\eta$ (depending on $n$, $m$) such that with high probability the population loss after $t = n$ steps of online SGD is*

$$\mathcal{L}(n) = \tilde{\Theta}\left(m^{1-2\beta} + \left(\frac{n}{d^{2I-1}\text{poly}(P)}\right)^{(1-2\beta)/(2\beta)}\right) \pm O(\varepsilon_D).$$

We remark that the above sample size scaling matches the minimax optimal rate for Gaussian sequence models (see e.g., [Joh17]), and the exponent is consistent with existing scaling law analyses of SGD on linear models [BAP24, LWK+24, PPXP24]. Note that despite the matching exponents (in terms of the decay rate $\beta$), the underlying mechanism and our theoretical analysis differ from these prior results due to the presence of nonlinear feature learning, which is reflected, for example, by the learning rate selection in our unstable discretization — see Section 3.3 for more discussions.

### 2.4 Simulations: Compute-optimal Frontier

In Figure 2, we plot ($a$) the idealized scaling curves assuming decoupled learning and an exact emergence time for each task (see Section 3.1), and ($b$) the MSE loss curves for GD training (with fixed step size) on the population loss, where we set $d = 2048, P = 1024, \sigma = h_4$, and vary the student width. While the idealized scaling law does not exactly hold at finite $d$, the slope of MSE loss vs. compute (on logarithmic scale) is independent of the problem dimension; we therefore compare the slope of the compute-optimal frontier in ($a$)($b$). Omitting the dimensionality $d$ (which does not vary across models) in Proposition 2.2, we know that given a fixed computational budget $\mathcal{T} \asymp mt$, the compute-optimal model under constant learning rate exhibits the following scaling,

$$\mathcal{L} \sim \mathcal{T}^{(1-2\beta)/(1+\beta)}, \quad m \sim \mathcal{T}^{1/(1+\beta)}.$$

We set the power-law exponent to be $\beta = 0.8$ in Figure 2. Observe that:

- The sum of staircase-like emergent learning curves yields a smooth power-law scaling in the cumulative MSE loss towards the tail, followed by a plateau due to the approximation error.
- The compute-optimal slope (dashed black line) is roughly consistent between the theoretical and empirical risk curves. Specifically, for $\beta = 0.8$ we theoretically predict a loss scaling of $\mathcal{L} \sim (mt)^{1/3}$ for the compute-optimal model; note that the empirical slope is slightly steeper due to the finite-width truncation error of the infinite power-law sum.

## 3 Overview of Proof Ideas

We discuss the proof ideas in this section. In Section 3.1, we describe the idealized dynamics, and show that they imply a loss scaling law when the signal strength $\{a_p\}_{p=1}^{P}$ follows a power law. In

Section 3.2 we show that gradient flow approximates this idealized dynamics, and in Section 3.3 we discretize the gradient flow with online SGD. For ease of presentation, we will assume a Hermite-4 link function $\sigma = h_4$ in this section; the same argument follows for more general activations.

## 3.1 The Idealized Learning Dynamics

**Learning a single task.** First, consider the single-index setting and suppose the target function is $x \mapsto ah_4(e_1 \cdot x)$. Let $v \in \mathbb{R}^d$ denote the learner neuron. It is known that, under gradient flow, the correlation of $v$ with the ground-truth direction $e_1$ approximately follows the quadratic ODE: $\frac{d}{dt}\bar{v}_1^2 \approx 8a\bar{v}_1^4$ prior to weak recovery, i.e., when $\bar{v}_1^2 = o(1)$ [BAGJ21]. This ODE has a closed-form solution: $\bar{v}_1^2(t) = (1/\bar{v}_1^2(0) - 8at)^{-1}$. We have two immediate observations from this formula:

(i) $\bar{v}_1^2 = \langle \bar{v}, e_1 \rangle^2$ will grow from $\tilde{\Theta}(1/d)$ to a nontrivial value around time $(8a\bar{v}_1^2(0))^{-1}$.

(ii) $\bar{v}_1^2$ stays small for most of the time and then suddenly increases around time $(8a\bar{v}_1^2(0))^{-1}$.

The above claims imply an emergent learning curve for the directional recovery of the single-index task. Due to the 2-homogeneous parameterization, we can show that the norm of $v$ will not grow until strong recovery is achieved, and the norm growth occurs at a much shorter timescale than the dynamics of $\bar{v}$. Consequently, the MSE loss remains nearly constant for an extensive period of time, followed by a sharp drop by $a^2/2$ at the aforementioned critical time.

**Decoupled learning of multiple tasks.** Next consider the multi-index setting where we have $P$ orthonormal ground-truth directions $\{e_p\}_{p\in[P]}$ with signal strength $\{a_p\}_{p\in[P]}$. Assume these $P$ single-index models are fully decoupled, i.e., for each $p \in [P]$, there is exactly one learner neuron $v_p$ associated with direction $e_p$, and the learning of different directions do not interfere — in other words, we are learning $P$ single-index models independently and simultaneously. Then from our previous discussion, we know that direction $e_p$ will be learned around time $(8a_p\bar{v}_{p,p}^2(0))^{-1}$ and the MSE loss will have a sudden drop of size $a_p^2/2$. Therefore, the idealized loss can be expressed as the sum of loss decrements at different times (we omit the constant factor $1/2$ for concise presentation)

$$\tilde{L}(t) = \sum_{p=1}^{P} a_p^2 \mathbb{1}\left\{t < (8a_p\bar{v}_{p,p}^2(0))^{-1}\right\}.$$

See Figure 2(a) for illustration. Based on this heuristic, we can derive the iteration/sample scaling in Proposition 2.2. Suppose that the signal strength follows a power law $a_p = p^{-\beta}$ for some $\beta > 1/2$, and assume identical initial overlap for all neurons $\bar{v}_{p,p}^2(0) = v^2$ for all $p \in [P]$, so that direction $e_p$ is learned at exactly $t = p^\beta v^{-2}/8$. Then, when $P$ is large, we have

$$\tilde{L}(p^\beta v^{-2}/8) \approx \sum_{q=p}^{\infty} q^{-2\beta} \approx \int_{p}^{\infty} s^{-2\beta}\, ds = p^{1-2\beta}/(2\beta - 1).$$

Applying the change-of-variables $t = p^\beta v^{-2}/8$, $p = (8v^2t)^{1/\beta}$, we arrive at the idealized loss scaling $\tilde{L}(t) \approx (2\beta - 1)^{-1}(8v^2)^{(1-2\beta)/\beta} \cdot t^{-(2\beta-1)/\beta}$.

**Width scaling.** To obtain the student width dependence, we show that a width-$m$ student network can learn $\tilde{\Theta}(m)$ directions – note that this is sharp up to logarithmic factors. Hence the approximation error can be computed as a truncation of the top $\tilde{\Theta}(m)$ tasks: $\sum_{q=\tilde{\Theta}(m)}^{P} q^{-2\beta} \approx \tilde{\Theta}(m^{1-2\beta})$.

## 3.2 The Gradient Flow Dynamics

In the previous section, we assumed complete decoupling of the learning of each single-index task. We now discuss how this condition holds approximately under gradient flow.

**Re-indexing and greedy maximum selection.** To simplify notation, we first re-index the neurons based on the initial correlation with the ground-truth directions. Let $\mathcal{V} \subset \mathbb{R}^d$ be the collection of initialized neurons. Define $(\pi(1), v_1) := \mathrm{argmax}_{q\in[P],v\in\mathcal{V}} a_q\bar{v}_q^{2I-2}$. By our previous heuristic argument, we expect $e_{\pi(1)}$ to be the first direction recovered, and $v_1$ – which achieves maximal overlap (weighted by $a_{\pi(1)}$) with $e_{\pi(1)}$ at initialization – to be the student neuron that converges to this direction first. After $e_{\pi(1)}$ is fitted by $v_1$, we remove this task from the cumulative objective; assuming the remaining student neurons have not moved too much during this process, we can determine the next task to be learned and the corresponding neuron via

$$(\pi(p+1), v_{p+1}) = \mathrm{argmax}_{\substack{q\in[P]\backslash\{\pi(1),...,\pi(p)\} \\ v\in\mathcal{V}\backslash\{v_1,...,v_p\}}} a_q\bar{v}_q^{2I-2}, \quad \forall p \in [\min\{P,m\} - 1]. \quad (5)$$

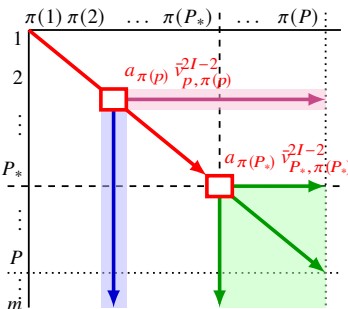

Figure 3: The greedy maximum selection matrix. The red diagonal entries represent the relevant neurons that eventually achieve overlap close to 1. The remaining irrelevant entries can be partitioned into three groups: the upper triangular entries $\bar{v}_{p,\pi(q)}$ with $p \in [P_*]$ and $p < q \in [P]$, the lower triangular entries, $\bar{v}_{k,\pi(p)}$ with $p \in [P_*]$ and $p < k \in [m]$, and the lower right block $\bar{v}_{k,\pi(q)}$ with $k > P_*, q > P_*$. We will control these blocks using the row gap (purple arrow), column gap (blue arrow), and the threshold gap (green arrows), respectively.

Finally, if $P < m$ we index the remaining unused neurons as $\{v_{P+1}, \ldots, v_m\}$, and if $m < P$ we assign $\{\pi(m+1), \ldots, \pi(P)\}$ to the unlearned teacher neurons arbitrarily so that $\pi$ is a permutation of $[P]$. Following [BAGP24], we call (5) the *greedy maximum selection* scheme and the matrix $\{a_{\pi(p)}\bar{v}_{k,\pi(p)}^{2I-2}(0)\}_{k \in [m], p \in [P]}$ the greedy maximum selection matrix (cf. Figure 3). Note that by construction, $a_{\pi(p)}\bar{v}_{p,\pi(p)}^{2I-2}$ is larger than all entries below it or on its right-hand side. We have the following quantitative estimates on the gaps between the on-diagonal and remaining entries of the maximum selection matrix at initialization. See Appendix B.2 for the formal statement and proof.

**Lemma 3.1** (Initialization (informal)). *Consider the greedy maximum selection matrix (cf. Figure 3). At initialization, with high probability, the gap between the first $P_*$ diagonal entries and all entries below them in the same column or to their right in the same row is lower bounded by $1/\text{poly } P$ (instead of $1/\text{poly } d$). The same also holds for the $(P_*, \pi(P_*))$-th entry and all entries in the lower right block.*

**Approximately decoupled dynamics.** We claim that when all irrelevant coordinates are small, the learning of different teacher directions can still be approximately decoupled. By Lemma B.1, the dynamics of the overlap $\bar{v}_{p,\pi(p)}^2$ can be decomposed into a primary signal term and the sum of contributions from the remaining coordinates:

$$\frac{d}{dt}\bar{v}_{p,\pi(p)}^2 \approx 8\big(a_{\pi(p)}\big(1 - \bar{v}_{p,\pi(p)}^2\big)\bar{v}_{p,\pi(p)}^2 - \sum_{q:q \neq p} a_{\pi(q)}\bar{v}_{p,\pi(q)}^4\big)\bar{v}_{p,\pi(p)}^2.$$

When the overlap $\bar{v}_{p,\pi(p)}^2$ is small, the signal term is of order $a_{\pi(p)}\bar{v}_{p,\pi(p)}^2 = \Omega(a_{\pi(p)}/d)$. Also, if we assume all irrelevant coordinates (i.e., $\bar{v}_{p,\pi(q)}^2$ for $q \neq p$) are small, say bounded by $d^{-0.9}$, then $\sum_{q:q \neq p} a_{\pi(q)}\bar{v}_{p,\pi(q)}^4 \leq d^{-1.8}\sum_{q:q \neq p} a_{\pi(q)} \leq P^{1/2}d^{-1.8} \ll a_{\pi(p)}/d$, as long as $a_{\min}P^{1/2} \gg d^{-0.8}$. As a result, when $\bar{v}_{p,\pi(p)}^2$ is still small, we have

$$\frac{d}{dt}\bar{v}_{p,\pi(p)}^2 \approx \big(1 \pm a_{\min}^{-1}d^{-0.8}\big) \times 8a_{\pi(p)}\bar{v}_{p,\pi(p)}^4.$$

Now suppose $a_{\min} \gg d^{-0.3}$. Then, the above implies that $\bar{v}_{p,\pi(p)}^2$ has a sharp transition around time $(1 \pm o(1))(8a_{\pi(p)}\bar{v}_{p,\pi(p)}^2)^{-1} = \tilde{\Theta}(d/a_{\pi(p)})$, and the $o(1)$ error term can be made much smaller than $1/\text{poly}(P)$ when $d$ is large — this will be useful in bounding the growth of irrelevant coordinates.

Similar to the analysis in [GRWZ21], we know that once $\bar{v}_p$ converges to $e_{\pi(p)}$, the convergence of norm $a_{\pi(p)}$ occurs within $O(\log d)$ time, and its dynamics become local in the sense that the influence of other teacher neurons becomes negligible. In addition, after $e_{\pi(p)}$ is learned, the remaining learner neurons will no longer be affected by this target direction.

**Bounding the irrelevant coordinates.** We show that the irrelevant coordinates, i.e., ones that are not in $\{\bar{v}_{p,\pi(p)}\}_{p \in [P_*]}$ (cf. Figure 3), stay small throughout training using the fact that the dynamics have sharp transitions. Here, we only consider the lower triangular entries of the greedy maximum selection matrix, i.e., $\bar{v}_{k,\pi(p)}$ with $p \in [P_*]$ and $p < k \in [m]$, which we control using the column gap. The other entries can be controlled using similar strategies – see Appendix C.2 for details. Recall that $\frac{d}{dt}\bar{v}_{k,\pi(p)}^2 \approx 8a_{\pi(p)}\bar{v}_{k,\pi(p)}^4$, which has a sharp transition around time $(8a_{\pi(p)}\bar{v}_{k,\pi(p)}^2(0))^{-1}$. From the column gap in Lemma 3.1, this implies that $\bar{v}_{k,\pi(p)}^2$ stays small before $v_p$ fits $a_{\pi(p)}e_{\pi(p)}$. After that, the signal from $a_{\pi(p)}e_{\pi(p)}$ will be close to 0, and consequently $\bar{v}_{k,\pi(p)}^2$ will cease to grow.

### 3.3 Online Stochastic Gradient Descent

We next outline the proof of Theorem 2.1, which requires converting the analysis of the gradient flow dynamics to one for the online SGD trajectory. At a high level, our proof relies on the martingale-

plus-drift argument used in prior works [BAGJ21, AAM23, DNGL23, OSSW24, RL24]. In order to rigorously handle the interdependence of the different martingale arguments, we rely on the stochastic induction arguments of [RL24]. The complete proof of Theorem 2.1 is presented in Appendix D.

**Controlling the irrelevant coordinates.** First, consider a lower triangular entry $(k, \pi(q))$ (i.e $q \in [P_*], q < k$). We wish to argue that $\bar{v}_{k,\pi(q)}^2$ stays small during the time it takes for $\bar{v}_{q,\pi(q)}^2$ to reach 1. By Lemma B.1 and a similar argument to Section 3.2, the update on $\bar{v}_{k,\pi(q)}^2$ is given by

$$\bar{v}_{k,\pi(q)}^2(t+1) \le \bar{v}_{k,\pi(q)}^2(t) + 8\eta a_{\pi(q)} \bar{v}_{k,\pi(q)}^4(t) + \xi_{t+1} + Z_{t+1},$$

where $\xi_{t+1} \ll 1$ is an error term we will ignore for ease of exposition, and $Z_{t+1}$ is the fluctuation

$$Z_{t+1} = \frac{2\eta \bar{v}_{k,\pi(q)}(t)}{\|v_k(t)\|} \langle (I - \bar{v}_k(t)\bar{v}_k(t)^\top)(\nabla_{v_k(t)} l(x_t) - \nabla_{v_k(t)} \mathcal{L}), e_{\pi(q)} \rangle.$$

By Lemma B.1, the conditional variance can be bounded as $\mathbb{E}[Z_{t+1}^2 \mid \mathcal{F}_t] \lesssim \eta^2 \bar{v}_{k,\pi(q)}^2(t)$. Hence by Doob's inequality, the total martingale term $\left|\sum_{t=1}^T Z_t\right|$ is bounded by $\tilde{\Theta}(\eta\sqrt{T/d})$ with high probability (we use the heuristic that $v_{k,\pi(q)}^2(t)$ is $\tilde{\Theta}(d^{-1})$ on average). [BAGJ21] selects $\eta \lesssim d^{-2} a_{\pi(q)}$, which bounds the martingale by $\tilde{\Theta}(d^{-1})$. $\bar{v}_{k,\pi(q)}^2(t)$ can thus be coupled to the deterministic process $\hat{x}_{t+1} = \hat{x}_t + 8\eta a_{\pi(q)} \hat{x}_t^2$ with $\hat{x}_0 = 1.5\bar{v}_{k,\pi(q)}^2(0)$, which implies the time at which $v_{k,\pi(q)}^2(t)$ ceases to stay small is within a constant factor of its corresponding gradient flow escape time $(8\eta a_{\pi(q)} \bar{v}_{k,\pi(q)}^2(0))^{-1}$. However, this is insufficient for our purposes, as the gradient flow escape time of $\bar{v}_{q,\pi(q)}^2$ is $(8\eta a_{\pi(q)} \bar{v}_{q,\pi(q)}^2(0))^{-1}$, which by Lemma 3.1 is only a $1 + 1/\text{poly}(P)$ factor smaller than the escape time of $v_{k,\pi(q)}^2(t)$. By decreasing $\eta$ by a $1/\text{poly}(P)$ factor, the total martingale term can instead be bounded by $\tilde{\Theta}(\bar{v}_{k,\pi(q)}^2(0)/\text{poly}(P))$, thus guaranteeing the online SGD escape times for $\bar{v}_{k,\pi(q)}^2, \bar{v}_{q,\pi(q)}^2$ are within $1 + 1/\text{poly}(P)$ factors of their gradient flow escape times, and hence that $\bar{v}_{k,\pi(q)}^2$ will stays small during the time it takes for $\bar{v}_{q,\pi(q)}^2$ to grow to $\approx 1$. Afterwards, the signal from $e_{\pi(q)}$ will be close to 0. The upper triangular entries ($k \in [P_*], k < q$) can be handled similarly.

**On the unstable discretization.** Next, consider the entries $\bar{v}_{k,\pi(q)}^2$ where $q > P_*$. In the argument above, since the martingale term scales as $\Theta(\eta\sqrt{T/d})$ and $e_{\pi(q)}$ is learned at time $T = T_q = \tilde{\Theta}(d\eta^{-1} a_{\pi(q)}^{-1})$, we selected a learning rate of $\eta \propto d^{-2} a_{\pi(q)}$. However, it is pessimistic to scale $\eta$ with the signal strength of a neuron which is not learned, as this can be arbitrarily small. Instead, if we are only interested in recovering the top $P_*$ directions, the martingale term only needs to be small up to time $T_{P_*} = \tilde{\Theta}(d\eta^{-1} a_{\pi(P_*)}^{-1})$. We can therefore scale $\eta$ with $a_{\pi(P_*)} \gg a_{\pi(q)}$. This can be interpreted as an *unstable discretization*: the choice of $\eta$ is too large for any of the directions $\pi(q)$ with $q > P_*$ to be learned, yet nevertheless, we can still control their growth and show that they remain small until the time that $e_{\pi(P_*)}$ is learned. Altogether, it suffices to choose $\eta \propto a_{\pi(P_*)} d^{-2}/\text{poly}(P)$.

**Controlling the relevant coordinates.** Finally, consider the growth of the relevant coordinates $\bar{v}_{p,\pi(p)}^2$ for $p \in [P_*]$. Following the argument in Section 3.2, the update on $\bar{v}_{p,\pi(p)}^2$ is approximately

$$\bar{v}_{p,\pi(p)}^2(t+1) \approx \bar{v}_{p,\pi(p)}^2(t) + 8\eta a_{\pi(p)}(1 - \bar{v}_{p,\pi(p)}^2(t))\bar{v}_{p,\pi(p)}^4(t) + Z_{t+1},$$

where the $Z_{t+1}$ satisfies $\mathbb{E}[Z_{t+1}^2 \mid \mathcal{F}_t] \lesssim \eta^2 \bar{v}_{p,\pi(p)}^2(t)$. By choosing the learning rate $\eta \lesssim d^{-2} a_{\pi(p)}/\text{poly}(P)$, we can bound $\bar{v}_{p,\pi(p)}^2(t)$ between two deterministic processes $(x_t^+)_t, (x_t^-)_t$ which satisfy $x_0^\pm = (1 \pm \frac{1}{\text{poly}(P)})\bar{v}_{p,\pi(p)}^2(0)$ and follow the updates $x_{t+1}^\pm = x_t^\pm + 8\eta a_{\pi(p)}(x_t^\pm)^2$. This guarantees that $\bar{v}_{p,\pi(p)}^2 \ll 1$ up to a time of $(1 - \frac{1}{\text{poly}(P)})(8\eta a_{\pi(p)} \bar{v}_{p,\pi(p)}^2(0))^{-1}$. Lower bounding the process $v_{p,\pi(p)}^2(t)$ is slightly more challenging, as $x_t^+$ diverges from $x_t^-$ in the time interval when the sharp transition occurs. To handle this, we partition $[\frac{1}{d}, \frac{1}{3}]$ into smaller subintervals, and rerun separate martingale-plus-drift arguments on each subinterval. We conclude by showing that once $\bar{v}_{p,\pi(p)}^2(t)$ crosses $1/3$, it rapidly converges to 1, after which $\|v_p\|^2$ rapidly converges to $a_{\pi(p)}$. Altogether, Lemma D.2 shows that $\bar{v}_p$ indeed converges to $e_{\pi(p)}$ in time $(1 \pm \frac{1}{\text{poly}(P)})(8\eta a_{\pi(p)} \bar{v}_{p,\pi(p)}^2(0))^{-1}$.

## 4 Conclusion

In this work, we study the (online) SGD training dynamics and sample complexity of learning a two-layer neural network with orthogonal ground truth weights and signal strengths $\{a_p\}_{p \in [P]} \subset \mathbb{R}_{\geq 0}$, where the width $P$ and the condition number $a_{\max}/a_{\min}$ can potentially be large. We establish a sample and runtime complexity that is polynomial in the problem dimensionality, teacher width, and condition number; as an application of our sharp analysis, when the second-layer coefficients of the teacher model follow a power law $a_p \asymp p^{-\beta}$ for $\beta > 1/2$, we derive scaling laws for the population MSE as a function of the student network width and the number of SGD steps.

Our current results assume input data with identity covariance; one interesting extension is to consider anisotropic data $x \sim \mathcal{N}(0, \Sigma)$ analogous to [MHWSE23, BQI25], and derive a two-parameter scaling law when the eigenvalues of $\Sigma$ also follow a power law. Another future direction is to consider a decaying learning rate schedule that achieves the unstable scaling law (Corollary 2.3) at any time $t$. Finally, our analysis relies on high information exponent link functions to decouple the learning of different directions, which does not cover the case of $\text{IE}(\sigma) = 2$ studied in [MBB23, RL24] — for this setting, the scaling behavior for SGD training is studied in a companion work [BAEVW25] for the special case of quadratic activation function.

#### Acknowledgments

The authors would like to thank Alberto Bietti, Theodor Misiakiewicz, Elliot Paquette and Nuri Mert Vural for discussion and feedback. JDL acknowledges support of the NSF CCF 2002272, NSF IIS 2107304, and NSF CAREER Award 2144994. This work was done in part while DW and JDL were visiting the Simons Institute for the Theory of Computing.

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

# Contents

## Limitations

One limitation of our work is that we require the input data to be drawn from the standard Gaussian. We remark that this assumption is common in prior works on learning multi-index models [BAGJ21, DLS22, AAM23, DNGL23, OSSW24, RL24]. However, it would be interesting to understand how our results generalize to other input distributions. Another limitation is that we assume $\sigma$ is an even function with information exponent $\geq 4$. One final limitation is that Corollary 2.3 requires is not anytime, and requires specifying the number of SGD steps $n$. It would be interesting to consider a decaying learning rate that can achieve Corollary 2.3 at any time $t$.

## A    Additional Related Works

**Theory of scaling laws.**    Neural scaling laws describe how the performance of deep learning models improves predictably as a power-law function of increased computational resources, data, and model size [HNA+17, KMH+20, HBM+22, BDK+24]. When the optimization algorithm is not taken into account, such scaling relations have been established for the approximation and estimation errors of deep neural networks [P+97, Suz18, SH20], as well as for the (precise) generalization error of simple closed-form estimators such as ridge regression [CLKZ21, MRS22, DLM24, AZVP24]. Recent works have also studied the loss scaling in distillation and synthetic data [IGT+24, JMS24], associative memory [CDB23, NLB24] and hierarchical models [CW24, CPT+24, ABRC24, PWL25], among other theoretical settings.

The scaling laws of SGD in sketched linear regression have been characterized in [BAP24, PPXP24, LWK+24] — this problem setting corresponds to a two-layer linear network with random, untrained first-layer weights, and is parallel to earlier works [RR17, NS20] on learning random features model under source and capacity conditions (see e.g., [CDV07, VY24]). However, this linear setup fails to capture the feature learning efficiency of neural networks. On the other hand, existing scaling analyses for the additive setting [Hut21, MLGT24, NFLL24] explicitly decompose the loss into an independent sum, simplifying the analysis due to task decoupling. We aim to understand a more natural – yet arguably more challenging – nonlinear feature learning scenario where the individual tasks are not decoupled.

**Learning shallow neural networks.**    The learning of two-layer neural networks with near-orthogonal neurons has been extensively studied in the deep learning theory literature. Existing works have studied the optimization dynamics for variants of ReLU [LMZ20, ZGJ21, Chi22], quadratic [GMMM19, MVEZ20, MBB23], and general Hermite activation functions [OSSW24, RL24, SBH24]. In the absence of the (near-)orthogonality assumption, this function class can be computationally hard to learn, as suggested by statistical query lower bounds [DKKZ20, GGJ+20]. Our target function is a subclass of additive models [Sto85, HT87], where the individual components take the form of single-index models — see [Bac17, OSSW24] for further discussion.

## B    Structure of Gradient and Initialization

### B.1    Population and Per-sample Gradients

In this subsection, we compute the population gradient and derive variance and tail bounds for the per-sample gradient. Namely, we prove the following lemma.

**Lemma B.1.** *Consider the setting described in Section 2. Assume w.l.o.g. that $\boldsymbol{v}_p^* = \boldsymbol{e}_p$ for $p \in [P]$.*
*The radial and tangent components of the population gradient are given by*

$$-\left\langle \nabla_{\boldsymbol{v}_k}\mathcal{L}, \boldsymbol{v}_k \right\rangle = 2\|\boldsymbol{v}_k\|^2 \sum_{i=I}^{\infty} \hat{\sigma}_{2i}^2 \sum_{p=1}^{P} a_p \bar{v}_{k,p}^{2i} - 2\|\boldsymbol{v}_k\|^2 \sum_{i=I}^{\infty} \hat{\sigma}_{2i}^2 \sum_{l=1}^{m} \|\boldsymbol{v}_l\|^2 \langle \bar{\boldsymbol{v}}_k, \bar{\boldsymbol{v}}_l \rangle^{2i},$$

$$-\frac{\left[(\boldsymbol{I} - \bar{\boldsymbol{v}}_k \bar{\boldsymbol{v}}_k^\top)\nabla_{\boldsymbol{v}_k}\mathcal{L}\right]_p}{\|\boldsymbol{v}_k\|} = \sum_{i=I}^{\infty} 2i\hat{\sigma}_{2i}^2 \left( a_p \bar{v}_{k,p}^{2i-2} - \sum_{q=1}^{P} a_q \bar{v}_{k,q}^{2i} \right) \bar{v}_{k,p}$$

$$-\sum_{i=I}^{\infty} 2i\hat{\sigma}_{2i}^2 \sum_{l:l \neq k} \|\boldsymbol{v}_l\|^2 \langle \bar{\boldsymbol{v}}_k, \bar{\boldsymbol{v}}_l \rangle^{2i-1} \left\langle (\boldsymbol{I} - \bar{\boldsymbol{v}}_k \bar{\boldsymbol{v}}_k^\top)\bar{\boldsymbol{v}}_l, \boldsymbol{e}_p \right\rangle.$$

*Suppose that* $\sum_{k=1}^{m} \|v_k\|^2 = O(\|a\|_1)$. *Let* $u \in \mathbb{S}^{d-1}$ *be a fixed direction. Put* $\tilde{Q} = 4(1 + Q)$. *Then, there exists a universal constant* $C \geq 1$ *such that, for any* $s \geq C$,

$$\mathbf{Var} \frac{\langle \nabla_{v_k} l(x), u \rangle}{\|v_k\|} \leq C \|a\|_1^2, \quad \mathbb{P}\left( \left| \frac{\langle \nabla_{v_k} l(x), u \rangle}{\|v_k\|} \right| \geq s \right) \leq Cm \exp\left( -C^{-1} (s/\|a\|_1)^{2/\tilde{Q}} \right).$$

*Proof.* The proof of the variance and tail bounds is essentially the same as the proof of Lemma A.5 of [RL24].[2] Now, we compute the population gradient. First, recall from (4) that the population loss is given as

$$\mathcal{L} = \sum_{i=I}^{\infty} \hat{\sigma}_i^2 \left( \frac{\|a\|^2}{2} - \sum_{p=1}^{P} \sum_{k=1}^{m} a_p \|v_k\|^2 \langle \bar{v}_k, v_p^* \rangle^i + \frac{1}{2} \sum_{k,l=1}^{m} \|v_k\|^2 \|v_l\|^2 \langle \bar{v}_k, \bar{v}_l \rangle^i \right) =: \sum_{i=I}^{\infty} \mathcal{L}_i.$$

For its gradient, first note that for each $i \geq I$,

$$\nabla_v \left( \|v\|^2 \langle \bar{v}, u \rangle^i \right) = \nabla_v \left( \frac{\langle v, u \rangle^i}{\|v\|^{i-2}} \right) = \frac{\nabla_v \langle v, u \rangle^i}{\|v\|^{i-2}} - \frac{\langle v, u \rangle^i}{\|v\|^{i-2}} \frac{\nabla_v \|v\|^{i-2}}{\|v\|^{i-2}}$$

$$= \frac{i \langle v, u \rangle^{i-1} u}{\|v\|^{i-2}} - \frac{\langle v, u \rangle^i}{\|v\|^{i-2}} \frac{(i-2) \|v\|^{i-3} \bar{v}}{\|v\|^{i-2}}$$

$$= i \langle \bar{v}, u \rangle^{i-1} \|v\| u - (i-2) \langle \bar{v}, u \rangle^i v.$$

Then, for each $k \in [m]$, we compute

$$\nabla_{v_k} \mathcal{L}_i = -\hat{\sigma}_i^2 \|v_k\| \sum_{p=1}^{P} a_p \left( i \bar{v}_{k,p}^{i-1} e_p - (i-2) \bar{v}_{k,p}^i \bar{v}_k \right)$$

$$+ 2\hat{\sigma}_i^2 \|v_k\|^2 v_k + \hat{\sigma}_i^2 \|v_k\| \sum_{l:l \neq k} \|v_l\|^2 \left( i \langle \bar{v}_k, \bar{v}_l \rangle^{i-1} \bar{v}_l - (i-2) \langle \bar{v}_k, \bar{v}_l \rangle^i \bar{v}_k \right).$$

Hence, for the radial component, we have

$$\langle \nabla_{v_k} \mathcal{L}_i, v_k \rangle = -2\hat{\sigma}_i^2 \|v_k\|^2 \sum_{p=1}^{P} a_p \bar{v}_{k,p}^i + 2\hat{\sigma}_i^2 \|v_k\|^2 \sum_{l=1}^{m} \|v_l\|^2 \langle \bar{v}_k, \bar{v}_l \rangle^i.$$

Meanwhile, for the tangent component, we have

$$(I - \bar{v}_k \bar{v}_k^\top) \nabla_{v_k} \mathcal{L}_i = -\hat{\sigma}_i^2 \|v_k\| \sum_{p=1}^{P} a_p i \bar{v}_{k,p}^{i-1} (I - \bar{v}_k \bar{v}_k^\top) e_p$$

$$+ \hat{\sigma}_i^2 \|v_k\| \sum_{l:l \neq k} \|v_l\|^2 i \langle \bar{v}_k, \bar{v}_l \rangle^{i-1} (I - \bar{v}_k \bar{v}_k^\top) \bar{v}_l$$

$$= -\hat{\sigma}_i^2 \|v_k\| \sum_{p=1}^{P} a_p i \bar{v}_{k,p}^{i-1} (e_p - \bar{v}_{k,p} \bar{v}_k)$$

$$+ \hat{\sigma}_i^2 \|v_k\| \sum_{l:l \neq k} \|v_l\|^2 i \langle \bar{v}_k, \bar{v}_l \rangle^{i-1} (\bar{v}_l - \langle \bar{v}_k, \bar{v}_l \rangle \bar{v}_k).$$

In particular, for each $p \in [P]$, we have

$$\frac{\left[ (I - \bar{v}_k \bar{v}_k^\top) \nabla_{v_k} \mathcal{L}_i \right]_p}{\|v_k\|} = -i\hat{\sigma}_i^2 \left( a_p \bar{v}_{k,p}^{i-2} - \sum_{q=1}^{P} a_q \bar{v}_q^i \right) \bar{v}_{k,p}$$

$$+ i\hat{\sigma}_i^2 \sum_{l:i \neq k} \|v_l\|^2 \langle \bar{v}_k, \bar{v}_l \rangle^{i-1} \left( \bar{v}_{l,p} - \langle \bar{v}_k, \bar{v}_l \rangle \bar{v}_{k,p} \right).$$

---

[2]Note that though Lemma A.3 of [RL24] is stated for i.i.d. random variables, the original theorem in [KC22] requires only independence and therefore applies to our setting.

Sum over $i \geq I$, and we obtain

$$\langle \nabla_{\boldsymbol{v}_k} \mathcal{L}, \boldsymbol{v}_k \rangle = -2 \|\boldsymbol{v}_k\|^2 \sum_{i=I}^{\infty} \hat{\sigma}_i^2 \sum_{p=1}^{P} a_p \bar{v}_{k,p}^i + 2 \|\boldsymbol{v}_k\|^2 \sum_{i=I}^{\infty} \hat{\sigma}_i^2 \sum_{l=1}^{m} \|\boldsymbol{v}_l\|^2 \langle \bar{\boldsymbol{v}}_k, \bar{\boldsymbol{v}}_l \rangle^i,$$

$$\frac{[(\boldsymbol{I} - \bar{\boldsymbol{v}}_k \bar{\boldsymbol{v}}_k^\top) \nabla_{\boldsymbol{v}_k} \mathcal{L}]_p}{\|\boldsymbol{v}_k\|} = -\sum_{i=I}^{\infty} i \hat{\sigma}_i^2 \left( a_p \bar{v}_{k,p}^{i-2} - \sum_{q=1}^{P} a_q \bar{v}_q^i \right) \bar{v}_{k,p}$$

$$+ \sum_{i=I}^{\infty} i \hat{\sigma}_i^2 \sum_{l:l \neq k} \|\boldsymbol{v}_l\|^2 \langle \bar{\boldsymbol{v}}_k, \bar{\boldsymbol{v}}_l \rangle^{i-1} \left( \bar{v}_{l,p} - \langle \bar{\boldsymbol{v}}_k, \bar{\boldsymbol{v}}_l \rangle \bar{v}_{k,p} \right).$$

$\square$

## B.2 The formal version of Lemma 3.1: Initialization

In this subsection, we prove the following formal version of Lemma 3.1.

**Lemma B.2** (Initialization). *Let* $\delta_{\mathbb{P}} \in (e^{-\log^2 d}, 1)$ *be the target failure probability. Suppose that* $\delta_r = \frac{\delta_{\mathbb{P}} \pi}{2mP^2}, \delta_t = \delta_c = \frac{\delta_{\mathbb{P}} \pi}{12m^2 P}, d \geq \frac{400(I-1)^2}{\delta_c^2} \log\left(\frac{2\pi}{3\delta_c}\right), m \geq 4P_* \log(P_*/\delta_{\mathbb{P}}) \vee 100 \log(P/\delta_{\mathbb{P}}), \frac{m}{\log^3 m} \geq 512 \log^2(P_*/\delta_{\mathbb{P}})$ *Then, the following holds with probability at least* $1 - O(\delta_{\mathbb{P}})$.

(a) **(Row gap).** *For any* $p \in [P_*]$ *and* $p < q \in [P]$, *we have* $a_{\pi(p)} \bar{v}_{p,\pi(p)}^{2I-2} \geq (1 + \delta_r) a_{\pi(q)} \bar{v}_{p,\pi(q)}^{2I-2}$.

(b) **(Column gap).** *For any* $p \in [P_*]$ *and* $p < k \in [m]$, *we have* $\bar{v}_{p,\pi(p)}^{2I-2} \geq (1 + \delta_c) \bar{v}_{k,\pi(p)}^{2I-2}$.

(c) **(Threshold gap).** *For any* $P_* < q \leq P$, $P_* < k \leq m$, *we have* $a_{\pi(P_*)} \bar{v}_{P_*,\pi(P_*)}^{2I-2} \geq (1 + \delta_t) a_{\pi(q)} \bar{v}_{k,\pi(q)}^{2I-2}$.

(d) **(Regularity conditions).** $\max_{k \in [m]} \|\bar{\boldsymbol{v}}_k\|_\infty^2 \leq \log^2 d/d$, $\min_{p \in [P_*]} \bar{v}_{p,\pi(p)}^2 \geq (\log P_*)/d$ *and* $\min_{q \in [P]} \max_{j > P_*} \bar{v}_{j,q}^2 \geq 1/d$.

*Proof of Lemma B.2 (row gap).* Consider an arbitrary neuron $\boldsymbol{v}$ and let $\boldsymbol{z} \sim \mathcal{N}(0, \boldsymbol{I}_d)$. Note that $\bar{\boldsymbol{v}} \overset{d}{=} \boldsymbol{z}/\|\boldsymbol{z}\|$ and therefore, for any $i \neq j$, $\bar{v}_i/\bar{v}_j \overset{d}{=} z_i/z_j$, which follows the standard Cauchy distribution. We know that $\mathbb{P}[z_i/z_j \leq z] = \pi^{-1} \arctan(z) + 1/2$. Fix $i \neq j$, we compute

$$\mathbb{P}\left[ a_i \bar{v}_i^{2I-2} \in (1 \pm \delta_r) a_j \bar{v}_j^{2I-2} \right] = 2\mathbb{P}\left[ \left((1-\delta_r)\frac{a_j}{a_i}\right)^{\frac{1}{2I-2}} \leq \frac{\bar{v}_i}{\bar{v}_j} \leq \left((1+\delta_r)\frac{a_j}{a_i}\right)^{\frac{1}{2I-2}} \right]$$

$$= \frac{2}{\pi}\left( \arctan\left( (1+\delta_r)\frac{a_j}{a_i} \right)^{\frac{1}{2I-2}} - \arctan\left( (1-\delta_r)\frac{a_j}{a_i} \right)^{\frac{1}{2I-2}} \right)$$

$$= \frac{2}{\pi} \arctan\left( \frac{\left((1+\delta_r)\frac{a_j}{a_i}\right)^{\frac{1}{2I-2}} - \left((1-\delta_r)\frac{a_j}{a_i}\right)^{\frac{1}{2I-2}}}{1 + \left((1+\delta_r)(1-\delta_r)\frac{a_j^2}{a_i^2}\right)^{\frac{1}{2I-2}}} \right),$$

where the last line comes from $\arctan a - \arctan b = \arctan \frac{a-b}{1+ab}$. Note that for any $p \in (0, 1)$, by the concavity of $z \mapsto z^p$, we have $a^p - b^p \leq pb^{p-1}(a-b)$. Therefore,

$$\left((1+\delta_r)\frac{a_j}{a_i}\right)^{\frac{1}{2I-2}} - \left((1-\delta_r)\frac{a_j}{a_i}\right)^{\frac{1}{2I-2}} \leq \frac{1}{2I-2}\left((1-\delta_r)\frac{a_j}{a_i}\right)^{\frac{1}{2I-2}-1} \delta_r \frac{a_j}{a_i} \leq \frac{1}{2I-2}\left(\frac{a_j}{a_i}\right)^{\frac{1}{2I-2}} \delta_r.$$

Recall that $\arctan z \leq z$. Thus,

$$\mathbb{P}\left[ a_i \bar{v}_i^{2I-2} \in (1 \pm \delta_r) a_j \bar{v}_j^{2I-2} \right] \leq \frac{2}{\pi} \frac{\frac{1}{2I-2}\left(\frac{a_j}{a_i}\right)^{\frac{1}{2I-2}} \delta_r}{1 + \left((1-\delta_r^2)\frac{a_j^2}{a_i^2}\right)^{\frac{1}{2I-2}}} = \frac{\delta_r}{(I-1)\pi} \frac{(a_i a_j)^{\frac{1}{2I-2}}}{(a_i^2)^{\frac{1}{2I-2}} + (1-\delta_r^2)^{\frac{1}{2I-2}}(a_j^2)^{\frac{1}{2I-2}}}$$

$$\leq \frac{\delta_r}{(I-1)\pi} \frac{\left(a_i^2 \vee a_j^2\right)^{\frac{1}{2I-2}}}{(a_i^2)^{\frac{1}{2I-2}} + (1-\delta_r^2)^{\frac{1}{2I-2}} (a_j^2)^{\frac{1}{2I-2}}}$$

$$\leq \frac{\delta_r}{(I-1)\pi(1-\delta_r^2)^{\frac{1}{2I-2}}}.$$

The last term is upper bounded by $2\delta_r/\pi$ as long as $\delta_r \leq 1/2$. Apply union bound over all $m$ neurons and all $P^2$ $(i,j)$-pairs, and we get

$$\mathbb{P}\left[\exists k \in [m], i \neq j \in [P], a_i \bar{v}_{k,i}^{2I-2} \in (1 \pm \delta_r) a_j \bar{v}_{k,j}^{2I-2}\right] \leq \frac{2mP^2}{\pi} \delta_r.$$

Choose $\delta_r = \frac{\delta_\mathbb{P} \pi}{2mP^2}$, so that the above implies $a_i \bar{v}_{k,i}^{2I-2} \notin (1 \pm \delta_r) a_j \bar{v}_{k,j}^{2I-2}$ for all $k \in [m]$ and $i \neq j \in [P]$ with probability at least $1 - \delta_\mathbb{P}$. To complete the proof, recall that by the definition of the greedy maximum selection process, we have $a_{\pi(p)} \bar{v}_{p,\pi(p)}^2 \geq a_{\pi(q)} \bar{v}_{p,\pi(q)}^2$. $\qquad\square$

*Proof of Lemma B.2 (column gap).* Let $z_1, \ldots, z_m$ be independent $\mathcal{N}(0, I_d)$ variables. Fix $k \neq l \in [m]$ and $p \in [P]$. Note that $(\bar{v}_{k,\pi(p)}, \bar{v}_{l,\pi(p)}) \overset{d}{=} (z_{k,p}/\|z_k\|, z_{l,p}/\|z_l\|)$. Hence, we can write

$$\mathbb{P}\left[\bar{v}_{k,\pi(p)}^{2I-2} \in (1 \pm \delta_c) \bar{v}_{l,\pi(p)}^{2I-2}\right] = \mathbb{P}\left[\left(\frac{z_{k,p}}{z_{k,l}}\right)^{2I-2} \in (1 \pm \delta_c) \left(\frac{\|z_k\|}{\|z_l\|}\right)^{2I-2}\right]$$

$$\leq \mathbb{P}\left[\left(\frac{z_{k,p}}{z_{k,l}}\right)^{2I-2} \in 1 \pm 3\delta_c\right] + \mathbb{P}\left[\left(\frac{\|z_k\|}{\|z_l\|}\right)^{2I-2} \notin 1 \pm \delta_c\right].$$

By our previous calculation, we know the first term is bounded by $6\delta_c/\pi$. Meanwhile, by the standard concentration results for $\mathcal{N}(0, I_d)$, we have

$$\mathbb{P}\left[\left|\frac{\|z_k\|}{\mathbb{E}\|z_k\|} - 1\right| \geq t\right] \leq 2\exp\left(-(\mathbb{E}\|z_k\|)^2 t^2/2\right) \leq 2\exp\left(-dt^2/4\right), \quad \forall t \geq 0.$$

In other words, with probability at least $1 - 4\exp\left(-dt^2/4\right)$, we have

$$\|z_k\|^{2I-2} = (1 \pm t)^{2I-2} = 1 \pm 4(I-1)t, \quad \|z_l\|^{2I-2} = 1 \pm 4(I-1)t,$$

and therefore $(\|z_k\|/\|z_l\|)^{2I-2} = 1 \pm 10(I-1)t$. Choose $t = \delta_c/(10(I-1))$, and we obtain

$$\mathbb{P}\left[\left(\frac{\|z_k\|}{\|z_l\|}\right)^{2I-2} \notin 1 \pm \delta_c\right] \leq 4\exp\left(-\frac{d}{4}\frac{\delta_c^2}{100(I-1)^2}\right).$$

As a result, we have

$$\mathbb{P}\left[\bar{v}_{k,\pi(p)}^{2I-2} \in (1 \pm \delta_c) \bar{v}_{l,\pi(p)}^{2I-2}\right] \leq \frac{6\delta_c}{\pi} + 4\exp\left(-\frac{d}{4}\frac{\delta_c^2}{100(I-1)^2}\right).$$

Take union bound over $k \neq l \in [m]$ and $p \in [P]$, and we get

$$\mathbb{P}\left[\exists k \neq l \in [m], p \in [P], \bar{v}_{k,\pi(p)}^{2I-2} \in (1 \pm \delta_c) \bar{v}_{l,\pi(p)}^{2I-2}\right] \leq m^2 P\left(\frac{6\delta_c}{\pi} + 4\exp\left(-\frac{d}{4}\frac{\delta_c^2}{100(I-1)^2}\right)\right).$$

For the RHS to be bounded by $\delta_\mathbb{P}$, it suffices to require

$$m^2 P \frac{12\delta_c}{\pi} \leq \delta_\mathbb{P} \quad \Leftarrow \quad \delta_c \leq \frac{\delta_\mathbb{P} \pi}{12m^2 P},$$

$$4\exp\left(-\frac{d}{4}\frac{\delta_c^2}{100(I-1)^2}\right) \leq \frac{6\delta_c}{\pi} \quad \Leftarrow \quad d \geq \frac{400(I-1)^2}{\delta_c^2}\log\left(\frac{2\pi}{3\delta_c}\right).$$

To complete the proof, recall that by the definition of the greedy maximum selection, we have $\bar{v}_{p,\pi(p)}^2 \geq \bar{v}_{k,\pi(p)}^2$ when $k > p$. $\qquad\square$

*Proof of Lemma B.2 (threshold gap).* Consider arbitrary $k \neq l \in [m]$ and $p \neq q \in [P]$. We estimate the gap between $a_{\pi(p)} \bar{v}_{k,\pi(p)}^{2I-2}$ and $a_{\pi(q)} \bar{v}_{l,\pi(q)}^{2I-2}$. Let $z_k, z_l$ be independent $\mathcal{N}(0, I_d)$ variables; we have $(\bar{v}_{k,\pi(p)}, \bar{v}_{l,\pi(q)}) \stackrel{d}{=} (z_{k,p}/\|z_k\|, z_{l,q}/\|z_l\|)$. As in the proof of column gap, we can write

$$\mathbb{P}\left[ a_{\pi(p)} \bar{v}_{k,\pi(p)}^{2I-2} \in (1 \pm \delta_t) a_{\pi(q)} \bar{v}_{l,\pi(p)}^{2I-2} \right] = \mathbb{P}\left[ \frac{a_{\pi(p)}}{a_{\pi(q)}} \left( \frac{z_{k,p}}{z_{k,l}} \right)^{2I-2} \in (1 \pm \delta_t) \left( \frac{\|z_k\|}{\|z_l\|} \right)^{2I-2} \right]$$

$$\leq \mathbb{P}\left[ \frac{a_{\pi(p)}}{a_{\pi(q)}} \left( \frac{z_{k,p}}{z_{k,l}} \right)^{2I-2} \in 1 \pm 3\delta_t \right] + \mathbb{P}\left[ \left( \frac{\|z_k\|}{\|z_l\|} \right)^{2I-2} \notin 1 \pm \delta_t \right].$$

By the proof of the row gap and the column gap, the last two terms are bounded by $\frac{6\delta_t}{\pi}$ and $4 \exp\left( -\frac{d}{4} \frac{\delta_t^2}{100(I-1)^2} \right)$, respectively. Note that this is the same as the bounds in the column gap proof (up to changing $\delta_c$ to $\delta_t$). Thus, we have

$$\mathbb{P}\left[ \exists k \neq l \in [m], p \in [P], \bar{v}_{k,\pi(p)}^{2I-2} \in (1 \pm \delta_c) \bar{v}_{l,\pi(p)}^{2I-2} \right] \leq \delta_{\mathbb{P}},$$

provided that

$$\delta_t \leq \frac{\delta_{\mathbb{P}} \pi}{12 m^2 P}, \quad d \geq \frac{400(I-1)^2}{\delta_t^2} \log\left( \frac{2\pi}{3\delta_t} \right).$$

To complete the proof, note that by the definition of the greedy maximum selection process, we have $a_{\pi(P_*)} \bar{v}_{P_*,\pi(P_*)}^{2I-2} \geq a_{\pi(q)} \bar{v}_{k,\pi(q)}^{2I-2}$ for all $P_* < k \leq m$ and $P_* < q \leq P$. $\qquad \square$

*Proof of Lemma B.2 (regularity conditions).* First, we consider the upper bound. Let $z_1, \ldots, z_m$ be independent $\mathcal{N}(0, I_d)$ random vectors. We have $(\bar{v}_k)_k \stackrel{d}{=} (z_k/\|z_k\|)_k$. By the standard Gaussian concentration results, we have $\mathbb{P}(\max_{k \in [m]} \|z_k\|_\infty \geq z) \leq 2md e^{-z^2/2}$ and $\mathbb{P}(\max_{k \in [m]} |\|z_k\|/\mathbb{E}\|z_1\| - 1| \geq \varepsilon) \leq 2m e^{-\varepsilon^2 d/3}$. Therefore, we have $\max_k \|\bar{v}_k\|_\infty^2 \leq \log^2 d / d$ with probability at least $1 - O(\delta_{\mathbb{P}})$.

Now, we consider the lower bound. Let $K$ be a parameter to be determined later. Our goal is to show that with high probability, $a_{\pi(p)} \bar{v}_{p,\pi(p)}^2$ is at least the $K$-th largest entry of the $\pi(p)$-th column of the greedy maximum selection matrix. In other words, at most the first $K - 1$ largest entries can be covered by the earlier neurons.

For any $k \neq l \in [m]$, the events that the $k$-th and $l$-th neurons are used by some earlier are independent. In addition, by symmetry, the probability that the $k$-th row is used by some other neuron is at most $P_*/(m - P_*)$, as we always have at least $m - P_*$ neurons remained. Meanwhile, since the coordinates of $\bar{v}_k$ are negatively correlated, conditioned on that $\bar{v}_{k,\pi(p)}^2$ is among the $K$ largest entries of that column, the probability that that row gets used is still upper bounded by $P_*/(m - P_*)$. Thus,

$$\mathbb{P}\left[ \text{all first } K \text{ largest entries of the } \pi(p)\text{-th column are used} \right] \leq \left( \frac{P_*}{m - P_*} \right)^K.$$

By union bound, the probability that one of $\{\bar{v}_{p,\pi(p)}^2\}_{p \in [P_*]}$ is not at least the $K$-th largest in that column is upper bounded by $P_* \left( \frac{P_*}{m - P_*} \right)^K$. For this to be upper bounded by $\delta_{\mathbb{P}}$, it suffices to have

$$P_* \left( \frac{P_*}{m - P_*} \right)^K \leq \delta_{\mathbb{P}} \quad \Longleftarrow \quad K \geq \frac{\log(P_*/\delta_{\mathbb{P}})}{\log((m - P_*)/P_*)} \quad \Longleftarrow \quad \begin{cases} K = \log(P_*/\delta_{\mathbb{P}}), \\ m \geq 4 P_* \log(P_*/\delta_{\mathbb{P}}). \end{cases}$$

Finally, by Lemma B.3, provided that[3]

$$\frac{m}{\log m} \geq 128\pi \log^2(P_*/\delta_{\mathbb{P}}) \quad \text{and} \quad \frac{m}{\log^3 m} \geq 512 \log^2(P_*/\delta_{\mathbb{P}}),$$

we have with probability at least $1 - \delta_{\mathbb{P}}$ that

$$\bar{v}_{p,\pi(p)}^2(0) \geq \frac{1}{d} \log\left( \frac{m}{\log(P_*/\delta_{\mathbb{P}})} \right) \geq \frac{\log P_*}{d}, \quad \forall p \in [P_*].$$

---

[3] Note that the second condition is stronger, so it suffices to keep the second one.

We conclude by establishing the last regularity condition. For fixed $j, q$, the PDF of $Z := \bar{v}_{j,q}$ is $p_Z(z) = \frac{\Gamma(\frac{d}{2})}{\sqrt{\pi}\Gamma(\frac{d-1}{2})}(1 - z^2)^{\frac{d-3}{2}}$, and therefore

$$\mathbb{P}(\bar{v}_{j,q}^2 \leq \frac{1}{d}) \leq \frac{2}{\sqrt{d}} \cdot \frac{\Gamma(\frac{d}{2})}{\sqrt{\pi}\Gamma(\frac{d-1}{2})} \leq \frac{2}{\sqrt{d}} \cdot \frac{\sqrt{d/2}}{\sqrt{\pi}} \leq \sqrt{\frac{2}{\pi}} \leq 0.8,$$

where the first inequality upper bounds the PDF by $p_Z(0)$, and the second is Gautschi's inequality. Therefore

$$\mathbb{P}(\max_{j > P_*} \bar{v}_{j,q}^2 \leq 1/d) \leq \mathbb{P}(\sum_{j \in [m]} \mathbb{1}(\bar{v}_{j,q}^2 \geq 1/d) \leq P_*).$$

Note that $\sum_{j \in [m]} \mathbb{1}(\bar{v}_{j,q}^2 \geq 1/d)$ is subGaussian with variance proxy $\leq m$. Therefore for $m \geq 10P_*$

$$\mathbb{P}(\sum_{j \in [m]} \mathbb{1}(\bar{v}_{j,q}^2 \geq 1/d) \leq P_*) \leq \exp(-(P_* - 0.2m)^2/m) = \exp(-m/100).$$

Union bounding over all $q \in [P]$, we get

$$\mathbb{P}(\min_{q \in [P]} \max_{j > P_*} \bar{v}_{j,q}^2 \leq 1/d) \leq P\exp(-m/100) \leq \delta_{\mathbb{P}}$$

for $m \geq 100\log(P/\delta_{\mathbb{P}})$. $\qquad\square$

**Lemma B.3.** *Let $Z_1, \ldots, Z_m$ be independent $\mathcal{N}(0, 1)$ variables. Suppose that*

$$\frac{m}{\log m} \geq 128\pi \log^2(1/\delta_{\mathbb{P}}) \quad and \quad \frac{m}{\log^3 m} \geq 512\pi K^2.$$

*Then, with probability at least $1 - \delta_{\mathbb{P}}$, the $K$-th largest among $Z_1, \ldots, Z_m$ is at least $\sqrt{\log(m/K)}$.*

*Proof.* Let $\Phi$ denote the CDF of $\mathcal{N}(0, 1)$. Then, the CDF $F_K$ of the $K$-th largest element among $Z_1, \ldots, Z_m$ is

$$F_K(z) = \sum_{k=1}^{K-1} \binom{m}{k}(1 - \Phi(z))^k \Phi^{m-k}(z)$$

It is well-known that the mill's ratio of $\mathcal{N}(0, 1)$ satisfies

$$\frac{1}{\sqrt{2\pi}}\frac{z}{1 + z^2}e^{-z^2/2} \leq 1 - \Phi(z) \leq \frac{1}{\sqrt{2\pi}}\frac{1}{z}e^{-z^2/2}.$$

Meanwhile, we have $\binom{m}{k} \leq m^k e^k / k^k$. As a result,

$$F_K(z) \leq \sum_{k=1}^{K-1} \left(\frac{me}{k}\right)^k \left(\frac{1}{\sqrt{2\pi}}\frac{1}{z}e^{-z^2/2}\right)^k \left(1 - \frac{1}{\sqrt{2\pi}}\frac{z}{1 + z^2}e^{-z^2/2}\right)^{m-k}$$

$$\leq \sum_{k=1}^{K-1} \left(\frac{me}{k}\frac{1}{\sqrt{2\pi}}\frac{1}{z}\right)^k \exp\left(-\frac{kz^2}{2}\right)\exp\left(-\frac{m - k}{\sqrt{2\pi}}\frac{z}{1 + z^2}e^{-z^2/2}\right).$$

Choose $z = \sqrt{(1 - \varepsilon)2\log(m/K)}$ for some $\varepsilon \in (0, 1)$. Then, we have $e^{-z^2/2} = (K/m)^{1-\varepsilon}$ and

$$F_K(z) \leq \sum_{k=1}^{K-1} \left(\frac{me}{k}\frac{1}{\sqrt{2\pi}}\frac{1}{z}\left(\frac{K}{m}\right)^{1-\varepsilon}\right)^k \exp\left(-\frac{m - k}{\sqrt{2\pi}}\frac{z}{1 + z^2}\left(\frac{K}{m}\right)^{1-\varepsilon}\right).$$

Choose $\varepsilon = 1/2$ and suppose that $K \leq m/2$. Then, we have

$$F_K(z) \leq \sum_{k=1}^{K-1} \left(m^{1/2}K^{1/2}\right)^k \exp\left(-\frac{1}{4\sqrt{2\pi}}\frac{m^{1/2}}{z}\right) \leq \sum_{k=1}^{K-1} \exp\left(\frac{k}{2}\log(mK) - \frac{1}{4\sqrt{2\pi}}\frac{m^{1/2}}{z}\right)$$

$$\leq \exp\left(2K\log m - \frac{1}{4\sqrt{2\pi}}\frac{m^{1/2}}{\sqrt{\log m}}\right).$$

To merge the first term into the second term, it suffices to require

$$2K \log m \le \frac{1}{8\sqrt{2\pi}} \frac{m^{1/2}}{\sqrt{\log m}} \quad \Longleftarrow \quad \frac{m}{\log^3 m} \ge 512\pi K^2.$$

Finally, we compute

$$\exp\left(-\frac{1}{8\sqrt{2\pi}} \frac{m^{1/2}}{\sqrt{\log m}}\right) \le \delta_{\mathbb{P}} \quad \Longleftarrow \quad \frac{m}{\log m} \ge 128\pi \log^2(1/\delta_{\mathbb{P}})$$

$$\square$$

## C   Gradient Flow Analysis

In this section, we analyze the gradient flow dynamics and show that gradient flow implements the greedy maximum selection scheme. We will assume the following on the initialization.

**Assumption C.1** (Initialization). *Suppose $P_* \le \min\{P, m\}$. We assume that the following hold at initialization.*

*(a) (Row gap) For any $p \in [P_*]$ and $p < q \in [P]$, we have $a_{\pi(p)} \bar{v}^{2I-2}_{p,\pi(p)} \ge (1 + \delta_r) a_{\pi(q)} \bar{v}^{2I-2}_{p,\pi(q)}$.*

*(b) (Column gap) For any $p \in [P_*]$ and $p < k \in [m]$, we have $\bar{v}^{2I-2}_{p,\pi(p)} \ge (1 + \delta_c)\bar{v}^{2I-2}_{k,\pi(p)}$.*

*(c) (Threshold gap) For any $P_* < k \in [m]$ and $P_* < q \in [P]$, we have $a_{\pi(P_*)} \bar{v}^{2I-2}_{P_*,\pi(P_*)} \ge (1 + \delta_t) a_{\pi(q)} \bar{v}^{2I-2}_{k,\pi(q)}$.*

*(d) (Regularity conditions) $\max_{k\in[m]} \|\bar{v}_k\|^2_\infty \le \log^2 d/d$ and $\min_{p\in[P_*]} \bar{v}^2_{p,\pi(p)} \ge 1/d$.*

**Remark.** By Lemma B.2, this assumption hold with high probability with $\delta_c, \delta_r, \delta_t = 1/\mathrm{poly}(P)$.

Now, we formally state the main theorem for gradient flow. The proof is deferred to the end of this section (cf. Section C.3). In the statement, we hide the constants that depend only on $\sigma$.

**Theorem C.1** (Main theorem for gradient flow). *Assume Assumption C.1 holds at initialization. Let $\varepsilon_D, \varepsilon_R$ be our target accuracies and $\delta_T$ be the target error in time. Put $\delta_{r,t} := \delta_r \wedge \delta_t$. Suppose that[4]*

$$\varepsilon_D \gtrsim_\sigma \frac{\|a\|_1}{a_{\min_*}} \frac{1}{d^{I-1/4}}, \quad \frac{1}{d^{I-1/4}} \lesssim_\sigma \varepsilon_R \lesssim_\sigma \frac{a^2_{\min_*}\delta_c}{(\log^2 d)^{I-1}}, \quad \frac{\|a\|_1}{a_{\min_*}} \frac{1}{d^{1/4}} \lesssim_\sigma \delta_T \lesssim_\sigma \delta_c \wedge \delta_r \wedge \delta_t,$$

$$\frac{d}{(\log^2 d)^{4I}} \gtrsim_\sigma \delta_{r,t}^{-8} \vee \left(\frac{a_{\min_*}}{\|a\|_1}\delta_{r,t}\right)^{-4} \vee \left(\frac{a^2_{\min_*}\delta_c}{\|a\|_1}\right)^{-4}.$$

*Choose the initialization scale to be*

$$\sigma_0^2 \approx_\sigma \frac{\bar{\varepsilon}^{8/(I\hat{\sigma}^2_{2I})}}{m}\left(a_{\min_*}\varepsilon_D \wedge \frac{a_{\min_*}\delta_T}{d^{I-1/2}} \wedge \varepsilon_R \wedge \frac{a_{\min_*}\delta_{r,t}}{(\log d)^{2I-2}d^{I-1/2}} \wedge \frac{a^2_{\min_*}\delta_c}{(\log^2 d)^{I-1}} \frac{1}{d^{I-1/2}}\right),$$

*where $\bar{\varepsilon} =_\sigma \varepsilon_D^2 d^{2(I-1)} \wedge \frac{\delta_T^2\delta^2_{r,t}}{d(\log d)^{4(I-1)}} \wedge \frac{\varepsilon_R}{a_{\min_*}} \wedge \frac{\delta^4_{r,t}}{d(\log d)^{4(I-1)}} \wedge \frac{a^2_{\min_*}\delta^2_c}{(\log^2 d)^{2I-2}} \frac{\delta^2_{r,t}}{d(\log d)^{4(I-1)}}$. For each $p \in [P_*]$, define*

$$T_p := \frac{1}{4I(I-1)\hat{\sigma}^2_{2I}a_{\pi(p)}\bar{v}^{2I-2}_{p,\pi(p)}(0)} = \Theta\left(\frac{1}{a_{\pi(p)}\bar{v}^{2I-2}_{p,\pi(p)}(0)}\right) = \tilde{\Theta}\left(\frac{1}{a_{\pi(p)}d^{I-1}}\right).$$

*Then, we have the following over time interval $[0, (1 + 20\delta_T)T_{P_*}]$:*

---

[4]Note that the lower bounds are $1/\mathrm{poly}(d)$, and we know from Lemma B.2 that $\delta_c, \delta_r, \delta_r$ are $1/\mathrm{poly}(P)$. Hence, the range from which $\varepsilon_D, \varepsilon_R, \delta_T$ can be chosen is not restrictive.

(a) **(Unused neurons)** $\|v_k\|^2 \leq \sigma_1^2$ for all $k > P_*$.

(b) **(Learning)** For any $p \in [P_*]$, $\bar{v}_{p,\pi(p)}^2 \geq 1 - \varepsilon_D$ and $\|v_p\|^2 = a_{\pi(p)} \pm \varepsilon_R$ for all $t \geq (1 + 20\delta_T)T_p$.

(c) **(Sharp transition)** For any $p \in [P_*]$, $\bar{v}_{p,\pi(p)}^2 \leq \left(\frac{4}{\delta_T}\right)^{\frac{1}{I-1}} \frac{\log^2 d}{d}$ and $\|v_p\|^2 \leq \sigma_1^2$ for all $t \leq (1 - 10\delta_T)T_p$.

*In words, for each $p \in [P_*]$, $\bar{v}_p$ converges to $e_{\pi(p)}$ and fit $a_{\pi(p)}$ at time $(1 \pm o(1))T_p$, and all other neurons stay small throughout training.*

Our proof will be a large (continuous) induction argument. Namely, we assume a collection of induction hypotheses, analyze the dynamics under these conditions, derive the convergence guarantees, and show that these induction hypotheses hold throughout training. One may refer to, for example, Section A.1 of [GRWZ21] or Chapter 1.3 of [Tao06] for details on this method.

We will maintain the following induction hypothesis.

**Induction Hypothesis C.2.** Let $\sigma_1 > \sigma_0$, $\bar{\varepsilon} \leq \varepsilon_0, \gamma$ be $o(1)$ parameters. We say this induction hypothesis holds at a time point if the following hold at that time point.

(a) Define $L := \{k \in [m] : \|v_k\| \geq \sigma_1\}$. For any $p \in [m]$, $v_p \in L$ implies $p \leq P_*$ and $\bar{v}_{p,\pi(p)}^2 \geq 1 - \bar{\varepsilon}$.

(b) For any $(k, \pi(q))$ that is not in $\{(p, \pi(p)) : p \in [P_*]\}$, we have $\bar{v}_{k,\pi(q)}^2 \leq \varepsilon_0 := d^{-(1-\gamma)}$.

(c) We have $\|v_p\|^2 \leq 2a_l$ for any $p \in [P \wedge m]$ and $\bar{v}_{p,\pi(p)}^2 \geq 1/d$ for any $p \in [P_*]$.

**Remark.** Condition (a) states that the norm of a neuron is large (when compared to $\sigma_0$) only if it is close to one ground-truth direction. Condition (b) means that all irrelevant coordinates stay small throughout training. Condition (c) includes some basic regularity conditions.

Before proceeding to the proofs, we state the following lemma that controls the interaction between different learner neurons. The proof is deferred to Section C.3.

**Lemma C.2.** *Suppose that Induction Hypothesis C.2 is true at time $t$. Then, at time $t$, for any $k \in [m]$ and $q \in [P]$, we have*

$$\frac{\mathrm{d}}{\mathrm{d}t}\bar{v}_{k,\pi(q)}^2 = 2\bar{v}_{k,\pi(q)}^2 \sum_{i=I}^{\infty} 2i\hat{\sigma}_{2i}^2\left(a_{\pi(q)}\bar{v}_{k,\pi(q)}^{2i-2} - \sum_{r=1}^{P} a_{\pi(r)}\bar{v}_{k,\pi(r)}^{2i}\right)$$

$$- \mathbb{1}\{k \neq q, q \in L\}\, 2\|v_q\|^2\left(1 - \bar{v}_{k,\pi(q)}^2\right)\sum_{i=I}^{\infty} 2i\hat{\sigma}_{2i}^2\bar{v}_{k,\pi(q)}^{2i}$$

$$\pm I2^{3I+6}C_\sigma^2\left|\bar{v}_{k,\pi(q)}\right|\left\{a_{\pi(q)}\bar{\varepsilon}^{1/2}\varepsilon_0^{I-1} \vee m\sigma_1^2 \vee \|a\|_1\varepsilon_0^I\right\}.$$

*In addition, for any target $\delta > 0$, we have*

$$a_{\pi(q)}\bar{\varepsilon}^{1/2}\varepsilon_0^{I-1} \vee m\sigma_1^2 \vee \|a\|_1\varepsilon_0^I \leq \delta \quad \Longleftarrow \quad \begin{cases} \bar{\varepsilon} \leq \left(\dfrac{\delta}{a_{\pi(q)}}\right)^2 d^{2(1-\gamma)(I-1)}, \\ m\sigma_1^2 \leq \delta, \\ d \geq \left(\dfrac{\delta}{\|a\|_1}\right)^{-\frac{1}{(1-\gamma)I}}. \end{cases} \tag{6}$$

The rest of this section is organized as follows. In Section C.1, we assume Induction Hypothesis C.2 and show that $v_p$ ($p \in [P_*]$) converges to $e_{\pi(p)}$ and fits $a_{\pi(p)}$ at time $(1 \pm o(1))T_p$. Then, in Section C.2, we verify Induction Hypothesis C.2. Finally, in Section C.3, we prove Lemma C.2 and Theorem C.1.

## C.1 Convergence Guarantees

In this subsection, we show under Induction Hypothesis C.2 that $v_p$ ($p \in [P_*]$) converges to $e_{\pi(p)}$ and fits $a_{\pi(p)}$ at time $(1 \pm o(1))T_p$. We will first consider the dynamics of $\bar{v}_p$ and then $\|v_p\|^2$. Our main result is the following, whose proof can be found at the end of this subsection.

**Corollary C.9** (Convergence). Let $\varepsilon_D, \varepsilon_R$ be our target accuracy in the tangent and radial directions, and $\delta_T$ the target error in time. Suppose that $\gamma < 1/(2I)$, $\delta'_v = 1/3$,

$$\varepsilon_D \geq \frac{2^{3I+7}C_\sigma^2}{(\delta'_v)^I \hat{\sigma}_{2I}^2} \frac{\|a\|_1}{a_{\min_*}} \frac{1}{d^{(1-\gamma)I}}, \quad \varepsilon_R \geq 12 \|a\|_1 2^{2I} d^{-(1-\gamma)I}, \quad \delta_T \geq \frac{2^{3I+4}C_\sigma^2}{\hat{\sigma}_{2I}^2} \frac{\|a\|_1}{a_{\min_*}} \frac{1}{d^{1/2-\gamma I}},$$

$$m\sigma_1^2 \leq \frac{\hat{\sigma}_{2I}^2 a_{\min_*}}{2^{3I+7}C_\sigma^2} \left( (\delta'_v)^I \varepsilon \wedge \frac{\delta_T}{d^{I-1/2}} \right) \wedge \frac{\varepsilon_R}{12},$$

$$\bar{\varepsilon} \leq \left( \frac{(\delta'_v)^I \hat{\sigma}_{2I}^2}{2^{3I+7}C_\sigma^2} \right)^2 \varepsilon_D^2 d^{2(1-\gamma)(I-1)} \wedge \left( \delta_T \frac{\hat{\sigma}_{2I}^2}{2^{3I+4}C_\sigma^2} \right)^2 \frac{1}{d^{1+2\gamma(I-1)}} \wedge \frac{\varepsilon_R}{12C_\sigma^2 a_{\pi(p)}}.$$

Then, for any $p \in [P_*]$, we have

$$\bar{v}_{p,\pi(p)}^2 \geq 1 - \varepsilon_D, \quad \|v_p\|^2 = a_{\pi(p)} \pm \varepsilon_R, \qquad\qquad \forall t \geq (1 + 20\delta_T)T_p,$$

$$\bar{v}_{p,\pi(p)}^2 \leq \left( \frac{4}{\delta_T} \right)^{\frac{1}{I-1}} \frac{\log^2 d}{d}, \quad \|v_p\|^2 \leq \sigma_1^2, \qquad\qquad \forall t \leq (1 - 10\delta_T)T_p,$$

where

$$T_p := \frac{1}{4I(I-1)\hat{\sigma}_{2I}^2 a_{\pi(p)} \bar{v}_{p,\pi(p)}^{2I-2}(0)} = \Theta\left( \frac{1}{a_{\pi(p)} \bar{v}_{p,\pi(p)}^{2I-2}(0)} \right) = \tilde{\Theta}\left( \frac{1}{a_{\pi(p)} d^{I-1}} \right).$$

### C.1.1 Tangent Dynamics

Here, we analyze the diagonal entries $\{\bar{v}_{p,\pi(p)}^2\}_{p \in [P_*]}$. Let $p \in [P_*]$ be fixed. For $\delta \in (0,1)$, let $T_\delta$ denote the time $\bar{v}_{p,\pi(p)}^2$ reaches $\delta$. We split the training process into $[0, T_{\delta_v}]$, $[T_{\delta_v}, T_{\delta'_v}]$ and $[T_{\delta'_v}, T_{1-\varepsilon}]$, where $\delta_v = o(1)$ and $\delta'_v = O(1)$ are two parameters to be chosen later. Our goal is to show that $\bar{v}_{p,\pi(p)}^2$ will converge to close to 1 around time $(1 \pm O(\delta_T))T_p$, where $T_p$ is the time indicated by the idealized process and $\delta_T$ is a parameter measuring the error.

**Lemma C.3** (Dynamics of the diagonal entries (Stage 1)). *Suppose that at time $t \in [0, T_{\delta_v}]$, Induction Hypothesis C.2 is true and the following hold:*

$$\delta_v \leq \frac{\delta_T}{2} \frac{2I\hat{\sigma}_{2I}^2}{C_\sigma^2}, \quad \gamma < \frac{1}{2I}, \quad m\sigma_1^2 \leq \delta_T \frac{\hat{\sigma}_{2I}^2 a_{\min_*}}{2^{3I+4}C_\sigma^2 d^{I-1/2}},$$

$$\bar{\varepsilon} \leq \left( \delta_T \frac{\hat{\sigma}_{2I}^2}{2^{3I+4}C_\sigma^2} \right)^2 \frac{1}{d^{1+2\gamma(I-1)}}, \quad d \geq \left( \frac{\hat{\sigma}_{2I}^2}{2^{3I+4}C_\sigma^2} \frac{a_{\min_*}}{\|a\|_1} \delta_T \right)^{-\frac{2}{1-2\gamma I}}.$$

*Then, at time $t \in [0, T_{\delta_v}]$, for any $p \in [P_*]$, we have*

$$\frac{\mathrm{d}}{\mathrm{d}t} \bar{v}_{p,\pi(p)}^2 = (1 \pm 3\delta_T) \times 4I\hat{\sigma}_{2I}^2 a_{\pi(p)} \bar{v}_{p,\pi(p)}^{2I}.$$

*Proof.* First, by Lemma C.2, we have

$$\frac{\mathrm{d}}{\mathrm{d}t} \bar{v}_{p,\pi(p)}^2 = 2\bar{v}_{p,\pi(p)}^2 2I\hat{\sigma}_{2I}^2 \left( a_{\pi(p)} \bar{v}_{p,\pi(p)}^{2I-2} - \sum_{r=1}^{P} a_{\pi(r)} \bar{v}_{p,\pi(r)}^{2I} \right)$$

$$+ 2\bar{v}_{p,\pi(p)}^2 \sum_{i=I+1}^{\infty} 2i\hat{\sigma}_{2i}^2 \left( a_{\pi(p)} \bar{v}_{p,\pi(p)}^{2i-2} - \sum_{r=1}^{P} a_{\pi(r)} \bar{v}_{p,\pi(r)}^{2i} \right)$$

$$\pm I2^{3I+6}C_\sigma^2 |\bar{v}_{p,\pi(p)}| \left\{ a_{\pi(p)} \bar{\varepsilon}^{1/2} \varepsilon_0^{I-1} \vee m\sigma_1^2 \vee \|a\|_1 \varepsilon_0^I \right\}$$

$$=: T_1 \left( \frac{\mathrm{d}}{\mathrm{d}t} \bar{v}_{p,\pi(p)}^2 \right) + T_2 \left( \frac{\mathrm{d}}{\mathrm{d}t} \bar{v}_{p,\pi(p)}^2 \right) + T_3 \left( \frac{\mathrm{d}}{\mathrm{d}t} \bar{v}_{p,\pi(p)}^2 \right).$$

For the signal term $T_1$, by Induction Hypothesis C.2(b), we have

$$T_1 = 4I\hat{\sigma}_{2I}^2 \left( a_{\pi(p)} \left( 1 - \bar{v}_{p,\pi(p)}^2 \right) \bar{v}_{p,\pi(p)}^{2I-2} - \sum_{r:r\neq p} a_{\pi(r)} \bar{v}_{p,\pi(r)}^{2I} \right) \bar{v}_{p,\pi(p)}^2$$

$$= 4I\hat{\sigma}_{2I}^2 \left( a_{\pi(p)} (1 \pm \delta_v) \bar{v}_{p,\pi(p)}^{2I-2} \pm \varepsilon_0^I \|\boldsymbol{a}\|_1 \right) \bar{v}_{p,\pi(p)}^2$$

$$= \left( 1 \pm \delta_v \pm \frac{\varepsilon_0^I \|\boldsymbol{a}\|_1}{a_{\pi(p)} \bar{v}_{p,\pi(p)}^{2I-2}} \right) \times 4I\hat{\sigma}_{2I}^2 a_{\pi(p)} \bar{v}_{p,\pi(p)}^{2I}.$$

We want the error terms in the coefficient to be bounded by $\delta_T$. For this to happen, we first require $\delta_v \le \delta_T/2$. Then, recall from Induction Hypothesis C.2(c) that $\bar{v}_{p,\pi(p)}^2 \ge 1/d$. Also recall $\varepsilon_0 = d^{-(1-\gamma)}$. Hence, we have

$$\frac{\varepsilon_0^I \|\boldsymbol{a}\|_1}{a_{\pi(p)} \bar{v}_{p,\pi(p)}^{2I-2}} \le \frac{\delta_T}{2} \quad \Leftarrow \quad d^{I\gamma-1} \le \frac{a_{\min_*}}{\|\boldsymbol{a}\|_1} \frac{\delta_T}{2} \quad \Leftarrow \quad \gamma < 1/I, \quad d \ge \left( \frac{a_{\min_*}}{\|\boldsymbol{a}\|_1} \frac{\delta_T}{2} \right)^{\frac{-1}{1-I\gamma}}.$$

When the above conditions hold, we have

$$\mathsf{T}_1 = (1 \pm \delta_T) \times 4I\hat{\sigma}_{2I}^2 a_{\pi(p)} \bar{v}_{p,\pi(p)}^{2I}.$$

Then, consider $\mathsf{T}_2$. We have

$$|\mathsf{T}_2| \le 2C_\sigma^2 \bar{v}_{p,\pi(p)}^2 \left( a_{\pi(p)} \bar{v}_{p,\pi(p)}^{2I} + \|\boldsymbol{a}\|_1 \varepsilon_0^I \right)$$

$$\le \left( a_{\pi(p)} \bar{v}_{p,\pi(p)}^{2I} + \|\boldsymbol{a}\|_1 \varepsilon_0^I \right) \frac{C_\sigma^2}{2I\hat{\sigma}_{2I}^2 a_{\pi(p)} \bar{v}_{p,\pi(p)}^{2I-2}} \times 4I\hat{\sigma}_{2I}^2 a_{\pi(p)} \bar{v}_{p,\pi(p)}^{2I}.$$

Again, for the coefficient to be bounded by $\delta_T$, it suffices to require

$$\frac{C_\sigma^2 a_{\pi(p)} \bar{v}_{p,\pi(p)}^{2I}}{2I\hat{\sigma}_{2I}^2 a_{\pi(p)} \bar{v}_{p,\pi(p)}^{2I-2}} \le \frac{\delta_T}{2} \quad \Leftarrow \quad \frac{C_\sigma^2 \bar{v}_{p,\pi(p)}^2}{2I\hat{\sigma}_{2I}^2} \le \frac{\delta_T}{2} \quad \Leftarrow \quad \delta_v \le \frac{\delta_T}{2} \frac{2I\hat{\sigma}_{2I}^2}{C_\sigma^2},$$

$$\frac{C_\sigma^2 \|\boldsymbol{a}\|_1 \varepsilon_0^I}{2I\hat{\sigma}_{2I}^2 a_{\pi(p)} \bar{v}_{p,\pi(p)}^{2I-2}} \le \frac{\delta_T}{2} \quad \Leftarrow \quad \varepsilon_0^I d^{I-1} \le \frac{\delta_T}{2} \frac{2I\hat{\sigma}_{2I}^2}{C_\sigma^2} \frac{a_{\min_*}}{\|\boldsymbol{a}\|_1}$$

$$\Leftarrow \quad \gamma < 1/I, \quad d \ge \left( \frac{\delta_T}{2} \frac{2I\hat{\sigma}_{2I}^2}{C_\sigma^2} \frac{a_{\min_*}}{\|\boldsymbol{a}\|_1} \right)^{\frac{-1}{1-\gamma I}}.$$

Finally, consider $\mathsf{T}_3$. We have

$$|\mathsf{T}_3| \le I2^{3I+6} C_\sigma^2 |\bar{v}_{p,\pi(p)}| \left\{ a_{\pi(p)} \bar{\varepsilon}^{1/2} \varepsilon_0^{I-1} \vee m\sigma_1^2 \vee \|\boldsymbol{a}\|_1 \varepsilon_0^I \right\}$$

$$= \left\{ a_{\pi(p)} \bar{\varepsilon}^{1/2} \varepsilon_0^{I-1} \vee m\sigma_1^2 \vee \|\boldsymbol{a}\|_1 \varepsilon_0^I \right\} \frac{2^{3I+4} C_\sigma^2 d^{I-1/2}}{\hat{\sigma}_{2I}^2 a_{\pi(p)}} \times 4I\hat{\sigma}_{2I}^2 a_{\pi(p)} \bar{v}_{p,\pi(p)}^{2I}.$$

By (6), for $a_{\pi(q)} \bar{\varepsilon}^{1/2} \varepsilon_0^{I-1} \vee m\sigma_1^2 \vee \|\boldsymbol{a}\|_1 \varepsilon_0^I \le \frac{\hat{\sigma}_{2I}^2 a_{\pi(p)}}{2^{3I+4} C_\sigma^2 d^{I-1/2}} \delta_T$ to hold, it suffices to have

$$m\sigma_1^2 \le \frac{\hat{\sigma}_{2I}^2 a_{\min_*}}{2^{3I+4} C_\sigma^2 d^{I-1/2}} \delta_T, \quad \bar{\varepsilon} \le \left( \frac{\hat{\sigma}_{2I}^2}{2^{3I+4} C_\sigma^2} \delta_T \right)^2 \frac{1}{d^{1+2\gamma(I-1)}}, \quad d \ge \left( \frac{1}{\|\boldsymbol{a}\|_1} \frac{\hat{\sigma}_{2I}^2 a_{\pi(p)}}{2^{3I+4} C_\sigma^2 d^{I-1/2}} \delta_T \right)^{-\frac{1}{(1-\gamma)I}}.$$

Note that the last condition has $d$ on both sides. Rearrange terms and it becomes

$$d^{1-\frac{I-1/2}{(1-\gamma)I}} \ge \left( \frac{\hat{\sigma}_{2I}^2}{2^{3I+4} C_\sigma^2} \frac{a_{\min_*}}{\|\boldsymbol{a}\|_1} \delta_T \right)^{-\frac{1}{(1-\gamma)I}} \quad \Leftarrow \quad \gamma < \frac{1}{2I}, \quad d \ge \left( \frac{\hat{\sigma}_{2I}^2}{2^{3I+4} C_\sigma^2} \frac{a_{\min_*}}{\|\boldsymbol{a}\|_1} \delta_T \right)^{-\frac{2}{1-2\gamma I}}.$$

Combining the above bounds, we get

$$\frac{\mathrm{d}}{\mathrm{d}t} \bar{v}_{p,\pi(p)}^2 = (1 \pm 3\delta_T) \times 4I\hat{\sigma}_{2I}^2 a_{\pi(p)} \bar{v}_{p,\pi(p)}^{2I},$$

as long as the following conditions are true:

$$\mathsf{T}_1: \quad \delta_v \le \frac{\delta_T}{2}, \quad \gamma < 1/I, \quad d \ge \left( \frac{a_{\min_*}}{\|\boldsymbol{a}\|_1} \frac{\delta_T}{2} \right)^{\frac{-1}{1-I\gamma}},$$

$$\text{T}_2: \quad \delta_v \le \frac{\delta_T}{2} \frac{2I\hat{\sigma}_{2I}^2}{C_\sigma^2}, \quad \gamma < 1/I, \quad d \ge \left( \frac{\delta_T}{2} \frac{2I\hat{\sigma}_{2I}^2}{C_\sigma^2} \frac{a_{\min_*}}{\|a\|_1} \right)^{\frac{-1}{1-\gamma I}},$$

$$\text{T}_3: \quad m\sigma_1^2 \le \delta_T \frac{\hat{\sigma}_{2I}^2 a_{\min_*}}{2^{3I+4}C_\sigma^2 d^{I-1/2}}, \quad \bar{\varepsilon} \le \left( \delta_T \frac{\hat{\sigma}_{2I}^2}{2^{3I+4}C_\sigma^2} \right)^2 \frac{1}{d^{1+2\gamma(I-1)}},$$

$$\gamma < \frac{1}{2I}, \quad d \ge \left( \frac{\hat{\sigma}_{2I}^2}{2^{3I+4}C_\sigma^2} \frac{a_{\min_*}}{\|a\|_1} \delta_T \right)^{-\frac{2}{1-2\gamma I}}.$$

Clear that the second set of conditions is stronger than the first set. In addition, since $\frac{1}{1-\gamma I} \le \frac{2}{1-2\gamma I}$, the last condition on $d$ is stronger than the first one. Hence, we can prune the above as

$$\delta_v \le \frac{\delta_T}{2} \frac{2I\hat{\sigma}_{2I}^2}{C_\sigma^2}, \quad \gamma < \frac{1}{2I}, \quad m\sigma_1^2 \le \delta_T \frac{\hat{\sigma}_{2I}^2 a_{\min_*}}{2^{3I+4}C_\sigma^2 d^{I-1/2}},$$

$$\bar{\varepsilon} \le \left( \delta_T \frac{\hat{\sigma}_{2I}^2}{2^{3I+4}C_\sigma^2} \right)^2 \frac{1}{d^{1+2\gamma(I-1)}}, \quad d \ge \left( \frac{\hat{\sigma}_{2I}^2}{2^{3I+4}C_\sigma^2} \frac{a_{\min_*}}{\|a\|_1} \delta_T \right)^{-\frac{2}{1-2\gamma I}}.$$

$\square$

We will see that the time needed for Stage 1 is much larger than all other stages combined, which allows the estimations to be looser in later stages.

**Lemma C.4** (Dynamics of the diagonal entries (Stage 2)). *Suppose that at time $t \in [T_{\delta_v}, T_{\delta_v'}]$, Induction Hypothesis C.2 is true. In addition, suppose that the conditions of Lemma C.3 holds and $\delta_v' \le 1/3$. Then, at time $t \in [T_{\delta_v}, T_{\delta_v'}]$, for any $p \in [P_*]$, we have*

$$\frac{d}{dt} \bar{v}_{p,\pi(p)}^2 \ge \frac{1}{2} \times 4I\hat{\sigma}_{2I}^2 a_{\pi(p)} \bar{v}_{p,\pi(p)}^{2I}.$$

*Proof.* Similar to the previous proof, by Lemma C.2, we have

$$\frac{d}{dt} \bar{v}_{p,\pi(p)}^2 = 2\bar{v}_{p,\pi(p)}^2 2I\hat{\sigma}_{2I}^2 \left( a_{\pi(p)} \bar{v}_{p,\pi(p)}^{2I-2} - \sum_{r=1}^{P} a_{\pi(r)} \bar{v}_{p,\pi(r)}^{2I} \right)$$

$$+ 2\bar{v}_{p,\pi(p)}^2 \sum_{i=I+1}^{\infty} 2i\hat{\sigma}_{2i}^2 \left( a_{\pi(p)} \bar{v}_{p,\pi(p)}^{2i-2} - \sum_{r=1}^{P} a_{\pi(r)} \bar{v}_{p,\pi(r)}^{2i} \right)$$

$$\pm I2^{3I+6} C_\sigma^2 |\bar{v}_{p,\pi(p)}| \left\{ a_{\pi(p)} \bar{\varepsilon}^{1/2} \varepsilon_0^{I-1} \vee m\sigma_1^2 \vee \|a\|_1 \varepsilon_0^I \right\}$$

$$=: \text{T}_1 \left( \frac{d}{dt} \bar{v}_{p,\pi(p)}^2 \right) + \text{T}_2 \left( \frac{d}{dt} \bar{v}_{p,\pi(p)}^2 \right) + \text{T}_3 \left( \frac{d}{dt} \bar{v}_{p,\pi(p)}^2 \right).$$

Since $\bar{v}_{p,\pi(p)}^2$ is larger this time, under the same conditions of Lemma C.3, we have

$$|\text{T}_3| \le \delta_T \times 4I\hat{\sigma}_{2I}^2 a_{\pi(p)} \bar{v}_{p,\pi(p)}^{2I}.$$

In addition, we have

$$\text{T}_2 \ge -2\bar{v}_{p,\pi(p)}^2 \sum_{i=I+1}^{\infty} 2i\hat{\sigma}_{2i}^2 \sum_{r:r \ne P} a_{\pi(r)} \bar{v}_{p,\pi(r)}^{2i} \ge -2C_\sigma^2 \bar{v}_{p,\pi(p)}^2 \|a\|_1 \varepsilon_0^{I+1}$$

$$= -\frac{C_\sigma^2 \|a\|_1 \varepsilon_0^{I+1}}{2I\hat{\sigma}_{2I}^2 a_{\pi(p)} \bar{v}_{p,\pi(p)}^{2I-2}} \times 4I\hat{\sigma}_{2I}^2 a_{\pi(p)} \bar{v}_{p,\pi(p)}^{2I}.$$

For the same reason, under the conditions of Lemma C.3, the coefficient is bounded by $\delta_T$. Hence

$$\frac{d}{dt} \bar{v}_{p,\pi(p)}^2 \ge \text{T}_1 \left( \frac{d}{dt} \bar{v}_{p,\pi(p)}^2 \right) - 2\delta_T \times 4I\hat{\sigma}_{2I}^2 a_{\pi(p)} \bar{v}_{p,\pi(p)}^{2I}.$$

Finally, we lower bound $\mathsf{T}_1$. To this end, we compute

$$
\begin{aligned}
\mathsf{T}_1 &= 2\bar{v}_{p,\pi(p)}^2 2I\hat{\sigma}_{2I}^2 \left( a_{\pi(p)} \left( 1 - \bar{v}_{p,\pi(p)}^2 \right) \bar{v}_{p,\pi(p)}^{2I-2} - \sum_{r:r\neq p} a_{\pi(r)} \bar{v}_{p,\pi(r)}^{2I} \right) \\
&\geq 2\bar{v}_{p,\pi(p)}^2 2I\hat{\sigma}_{2I}^2 \left( a_{\pi(p)} \left( 1 - \delta_v' \right) \bar{v}_{p,\pi(p)}^{2I-2} - \|a\|_1 \varepsilon_0^I \right) \\
&= \left( 1 - \delta_v' - \frac{\|a\|_1 \varepsilon_0^I}{a_{\pi(p)} \bar{v}_{p,\pi(p)}^{2I-2}} \right) \times 4I\hat{\sigma}_{2I}^2 a_{\pi(p)} \bar{v}_{p,\pi(p)}^{2I}.
\end{aligned}
$$

We will see that since the initial $\bar{v}_{p,\pi(p)}^2$ in Stage 2 is much larger than $1/d$, Stage 2 is much shorter than Stage 1, whence we only need the error in the coefficient to be smaller than a constant, say, $1/2$. To this end, it suffices to require $\delta_v' \leq 1/3$ and $\frac{\|a\|_1 \varepsilon_0^I}{a_{\pi(p)} \bar{v}_{p,\pi(p)}^{2I-2}} \leq \frac{1}{3}$, and the second condition is again implied by the conditions of Lemma C.3. $\qquad\square$

**Lemma C.5** (Dynamics of the diagonal entries (Stage 3)). *Suppose that at time $t \in [T_{\delta_v'}, T_{1-\varepsilon}]$, Induction Hypothesis C.2 is true. In addition, suppose that the conditions of Lemma C.3 holds and $\varepsilon \geq \frac{2^{3I+7} C_\sigma^2}{(\delta_v')^I \hat{\sigma}_{2I}^2} \left\{ \bar{\varepsilon}^{1/2} \varepsilon_0^{I-1} \vee \frac{m\sigma_1^2}{a_{\min*}} \vee \frac{\|a\|_1}{a_{\min*}} \varepsilon_0^I \right\}.$[5] Then, at time $t \in [T_{\delta_v'}, T_{1-\varepsilon}]$, for any $p \in [P_*]$, we have*

$$
\frac{\mathrm{d}}{\mathrm{d}t} \bar{v}_{p,\pi(p)}^2 \geq (\delta_v')^I I\hat{\sigma}_{2I}^2 a_{\pi(p)} \left( 1 - \bar{v}_{p,\pi(p)}^2 \right).
$$

*Proof.* By the proof of Lemma C.4, we have

$$
\frac{\mathrm{d}}{\mathrm{d}t} \bar{v}_{p,\pi(p)}^2 = \mathsf{T}_1 \left( \frac{\mathrm{d}}{\mathrm{d}t} \bar{v}_{p,\pi(p)}^2 \right) + \mathsf{T}_2 \left( \frac{\mathrm{d}}{\mathrm{d}t} \bar{v}_{p,\pi(p)}^2 \right) + \mathsf{T}_3 \left( \frac{\mathrm{d}}{\mathrm{d}t} \bar{v}_{p,\pi(p)}^2 \right),
$$

where

$$
\begin{aligned}
\mathsf{T}_1 &\geq 2\bar{v}_{p,\pi(p)}^2 2I\hat{\sigma}_{2I}^2 \left( a_{\pi(p)} \left( 1 - \bar{v}_{p,\pi(p)}^2 \right) \bar{v}_{p,\pi(p)}^{2I-2} - \|a\|_1 \varepsilon_0^I \right), \\
\mathsf{T}_2 &\geq -2C_\sigma^2 \|a\|_1 \varepsilon_0^{I+1}, \\
|\mathsf{T}_3| &\leq I2^{3I+6} C_\sigma^2 \left\{ a_{\pi(p)} \bar{\varepsilon}^{1/2} \varepsilon_0^{I-1} \vee m\sigma_1^2 \vee \|a\|_1 \varepsilon_0^I \right\}.
\end{aligned}
$$

For the first term, we compute

$$
\mathsf{T}_1 \geq \delta_v' \left( (\delta_v')^{I-1} - \frac{\|a\|_1 \varepsilon_0^I}{a_{\pi(p)} \varepsilon} \right) \times 4I\hat{\sigma}_{2I}^2 a_{\pi(p)} \left( 1 - \bar{v}_{p,\pi(p)}^2 \right)
$$

When $\varepsilon \geq \frac{2\|a\|_1 \varepsilon_0^I}{a_{\min*} (\delta_v')^{I-1}}$, we can further rewrite the above as

$$
\mathsf{T}_1 \geq \frac{(\delta_v')^I}{2} \times 4I\hat{\sigma}_{2I}^2 a_{\pi(p)} \left( 1 - \bar{v}_{p,\pi(p)}^2 \right).
$$

When $\bar{v}_{p,\pi(p)}^2 \leq 1 - \varepsilon$, the RHS is lower bounded by $\frac{(\delta_v')^I}{2} \times 4I\hat{\sigma}_{2I}^2 a_{\pi(p)}\varepsilon$. Our goal now is to show ensure $\mathsf{T}_2$ and $\mathsf{T}_3$ are both bounded by $\frac{(\delta_v')^I}{8} \times 4I\hat{\sigma}_{2I}^2 a_{\pi(p)}\varepsilon$. For $\mathsf{T}_2$, we compute

$$
-\mathsf{T}_2 \leq 2C_\sigma^2 \|a\|_1 \varepsilon_0^{I+1} \leq \frac{(\delta_v')^I}{8} \times 4I\hat{\sigma}_{2I}^2 a_{\pi(p)}\varepsilon \quad \Leftarrow \quad \varepsilon \geq \frac{4C_\sigma^2}{(\delta_v')^I I\hat{\sigma}_{2I}^2} \frac{\|a\|_1}{a_{\min*}} \varepsilon_0^{I+1}.
$$

Then, for $\mathsf{T}_3$, by (6), we

$$
a_{\pi(p)} \bar{\varepsilon}^{1/2} \varepsilon_0^{I-1} \vee m\sigma_1^2 \vee \|a\|_1 \varepsilon_0^I \leq \frac{(\delta_v')^I \hat{\sigma}_{2I}^2}{22^{3I+6} C_\sigma^2} a_{\pi(p)}\varepsilon
$$

---

[5] Note that the order of the RHS is higher than 1. This allows $\varepsilon$ to be smaller than $\varepsilon_0$ and $\bar{\varepsilon}$.

$$\Leftarrow \quad \bar{\varepsilon} \leq \left(\frac{(\delta_v')^I \hat{\sigma}_{2I}^2}{2^{23I+6}C_\sigma^2}\varepsilon\right)^2 d^{2(1-\gamma)(I-1)}, \quad m\sigma_1^2 \leq \frac{(\delta_v')^I \hat{\sigma}_{2I}^2}{2^{23I+6}C_\sigma^2}a_{\min_*}\varepsilon,$$

$$d \geq \left(\frac{1}{\|a\|_1}\frac{(\delta_v')^I \hat{\sigma}_{2I}^2}{2^{23I+6}C_\sigma^2}a_{\pi(p)}\varepsilon\right)^{-\frac{1}{(1-\gamma)I}}.$$

Then, rearrange terms so that they become conditions on $\varepsilon$:

$$\varepsilon \geq \frac{2^{3I+7}C_\sigma^2}{(\delta_v')^I \hat{\sigma}_{2I}^2}\left(\bar{\varepsilon}^{1/2}\varepsilon_0^{I-1} \vee \frac{m\sigma_1^2}{a_{\min_*}} \vee \frac{\|a\|_1}{a_{\min_*}}\varepsilon_0^I\right).$$

Combine the above results, and we obtain

$$\frac{\mathrm{d}}{\mathrm{d}t}\bar{v}_{p,\pi(p)}^2 \geq \frac{(\delta_v')^I}{4} \times 4I\hat{\sigma}_{2I}^2 a_{\pi(p)}\left(1 - \bar{v}_{p,\pi(p)}^2\right),$$

provided that

$$\varepsilon \geq \frac{2\|a\|_1 \varepsilon_0^I}{a_{\min_*}(\delta_v')^{I-1}} \vee \frac{4C_\sigma^2}{(\delta_v')^I I\hat{\sigma}_{2I}^2}\frac{\|a\|_1}{a_{\min_*}}\varepsilon_0^{I+1} \vee \frac{2^{3I+7}C_\sigma^2}{(\delta_v')^I \hat{\sigma}_{2I}^2}\left(\bar{\varepsilon}^{1/2}\varepsilon_0^{I-1} \vee \frac{m\sigma_1^2}{a_{\min_*}} \vee \frac{\|a\|_1}{a_{\min_*}}\varepsilon_0^I\right).$$

Note that (the last condition of) the third condition dominate the first two conditions. Hence, we can simplify the above condition to be

$$\varepsilon \geq \frac{2^{3I+7}C_\sigma^2}{(\delta_v')^I \hat{\sigma}_{2I}^2}\left(\bar{\varepsilon}^{1/2}\varepsilon_0^{I-1} \vee \frac{m\sigma_1^2}{a_{\min_*}} \vee \frac{\|a\|_1}{a_{\min_*}}\varepsilon_0^I\right).$$

$\square$

Now, we combine the previous lemmas and estimate the convergence rate of $\bar{v}_p$.

**Lemma C.6** (Directional convergence). *Inductively assume Induction Hypothesis C.2. Let $\varepsilon$ be the target accuracy and $\delta_T$ the target error in time. Suppose that*

$$\gamma < \frac{1}{2I}, \quad \delta_v' = \frac{1}{3},$$

$$\varepsilon \geq \exp\left(-\frac{4C_\sigma^2}{I\hat{\sigma}_{2I}^2}\frac{(\delta_v')^I}{8I}\left(\frac{d}{\log^2 d}\right)^{I+1/I-2}\right), \quad m\sigma_1^2 \leq \frac{\hat{\sigma}_{2I}^2 a_{\min_*}}{2^{3I+7}C_\sigma^2}\left((\delta_v')^I\varepsilon \wedge \frac{\delta_T}{d^{I-1/2}}\right),$$

$$d \geq \left(\frac{2^{3I+7}C_\sigma^2}{(\delta_v')^I\hat{\sigma}_{2I}^2}\frac{\|a\|_1}{a_{\min_*}}\frac{1}{\varepsilon}\right)^{\frac{1}{(1-\gamma)I}} \vee \left(\frac{\hat{\sigma}_{2I}^2}{2^{3I+4}C_\sigma^2}\frac{a_{\min_*}}{\|a\|_1}\delta_T\right)^{-\frac{2}{1-2\gamma I}},$$

$$\bar{\varepsilon} \leq \left(\frac{(\delta_v')^I\hat{\sigma}_{2I}^2}{2^{3I+7}C_\sigma^2}\right)^2\varepsilon^2 d^{2(1-\gamma)(I-1)} \wedge \left(\delta_T\frac{\hat{\sigma}_{2I}^2}{2^{3I+4}C_\sigma^2}\right)^2\frac{1}{d^{1+2\gamma(I-1)}}.$$

*Then, for any $p \in [P_*]$, the time needed for $\bar{v}_{p,\pi(p)}^2$ to reach $1 - \varepsilon$ satisfies*

$$T_{1-\varepsilon} = \frac{1 \pm 10\delta_T}{4I(I-1)\hat{\sigma}_{2I}^2 a_{\pi(p)}\bar{v}_{p,\pi(p)}^{2I-2}(0)} = \Theta\left(\frac{1}{a_{\pi(p)}\bar{v}_{p,\pi(p)}^{2I-2}(0)}\right) = \tilde{\Theta}\left(\frac{1}{a_{\pi(p)}d^{I-1}}\right).$$

*Moreover, the requirements on $d$ can be removed if we choose[6]*

$$\varepsilon \geq \frac{2^{3I+7}C_\sigma^2}{(\delta_v')^I\hat{\sigma}_{2I}^2}\frac{\|a\|_1}{a_{\min_*}}\frac{1}{d^{(1-\gamma)I}} = \Theta\left(\frac{\|a\|_1}{a_{\min_*}}\frac{1}{d^{(1-\gamma)I}}\right),$$

$$\delta_T \geq \frac{2^{3I+4}C_\sigma^2}{\hat{\sigma}_{2I}^2}\frac{\|a\|_1}{a_{\min_*}}\frac{1}{d^{1/2-\gamma I}} = \Theta\left(\frac{\|a\|_1}{a_{\min_*}}\frac{1}{d^{1/2-\gamma I}}\right).$$

---

[6]Note that this condition on $\varepsilon$ is stronger than the existing one.

*Proof (Part I): convergence rate.* By Lemma C.3, for any $t \in [0, T_{\delta_v}]$, we have

$$\frac{\mathrm{d}}{\mathrm{d}t} \bar{v}^2_{p,\pi(p)} = (1 \pm 3\delta_T) \times 4I\hat{\sigma}^2_{2I} a_{\pi(p)} \left(\bar{v}^2_{p,\pi(p)}\right)^I$$

$$\Rightarrow \quad \bar{v}^2_{p,\pi(p)}(t) = \bar{v}^{2I-2}_{p,\pi(p)}(0) \left(1 - (1 \pm 3\delta_T) \, 4I(I-1)\hat{\sigma}^2_{2I} a_{\pi(p)} \bar{v}^{2I-2}_{p,\pi(p)}(0) t\right)^{-\frac{1}{I-1}}.$$

This implies

$$\frac{1 - 4\delta_T}{4I(I-1)\hat{\sigma}^2_{2I} a_{\pi(p)} \bar{v}^{2I-2}_{p,\pi(p)}(0)} \left(1 - \left(\frac{\bar{v}^{2I-2}_{p,\pi(p)}(0)}{\delta_v}\right)^{I-1}\right) \le T_{\delta_v} \le \frac{1 + 4\delta_T}{4I(I-1)\hat{\sigma}^2_{2I} a_{\pi(p)} \bar{v}^{2I-2}_{p,\pi(p)}(0)}.$$

For the lower bound, note that

$$\left(\frac{\bar{v}^{2I-2}_{p,\pi(p)}(0)}{\delta_v}\right)^{I-1} \le \delta_T \quad \Leftarrow \quad \delta_v \ge \delta_T^{\frac{-1}{I-1}} \bar{v}^{2I-2}_{p,\pi(p)}(0) \quad \Leftarrow \quad \delta_v \ge \left(\frac{\log^2 d}{d\delta_T}\right)^{I-1}.$$

When the above condition holds, we have

$$T_{\delta_v} = \frac{1 \pm 6\delta_T}{4I(I-1)\hat{\sigma}^2_{2I} a_{\pi(p)} \bar{v}^{2I-2}_{p,\pi(p)}(0)}.$$

For Stage 2, by Lemma C.4, we have

$$\frac{\mathrm{d}}{\mathrm{d}t} \bar{v}^2_{p,\pi(p)} \ge 2I\hat{\sigma}^2_{2I} a_{\pi(p)} \left(\bar{v}^2_{p,\pi(p)}\right)^I$$

$$\Rightarrow \quad \bar{v}^2_{p,\pi(p)}(t) \ge \delta_v \left(1 - 2I(I-1)\hat{\sigma}^2_{2I} a_{\pi(p)} \delta_v^{I-1}(t - T_{\delta_v})\right)^{-\frac{1}{I-1}}$$

$$\Rightarrow \quad T_{\delta_v'} - T_{\delta_v} \le \frac{1}{2I(I-1)\hat{\sigma}^2_{2I} a_{\pi(p)} \delta_v^{I-1}} \le \frac{4\bar{v}^{2I-2}_{p,\pi(p)}(0)}{\delta_v^{I-1}} T_{\delta_v}.$$

For the coefficient to be smaller than $\delta_T$, it suffices to require

$$\frac{4\bar{v}^{2I-2}_{p,\pi(p)}(0)}{\delta_v^{I-1}} \le \delta_T \quad \Leftarrow \quad \delta_v \ge \left(\frac{4\bar{v}^{2I-2}_{p,\pi(p)}(0)}{\delta_T}\right)^{\frac{1}{I-1}} \quad \Leftarrow \quad \delta_v \ge \left(\frac{4}{\delta_T}\right)^{\frac{1}{I-1}} \frac{\log^2 d}{d}.$$

Finally, for Stage 3, by Lemma C.5, we have

$$\frac{\mathrm{d}}{\mathrm{d}t} \left(1 - \bar{v}^2_{p,\pi(p)}\right) \le - (\delta_v')^I I\hat{\sigma}^2_{2I} a_{\pi(p)} \left(1 - \bar{v}^2_{p,\pi(p)}\right)$$

$$\Rightarrow \quad 1 - \bar{v}^2_{p,\pi(p)}(t) \le \exp\left(- (\delta_v')^I I\hat{\sigma}^2_{2I} a_{\pi(p)} t\right)$$

$$\Rightarrow \quad T_{1-\varepsilon} - T_{\delta_v} \le \frac{\log(1/\varepsilon)}{(\delta_v')^I I\hat{\sigma}^2_{2I} a_{\pi(p)}} \le \frac{8I\bar{v}^{2I-2}_{p,\pi(p)}(0) \log(1/\varepsilon)}{(\delta_v')^I} T_{\delta_v}.$$

Again, for the coefficient to be smaller than $\delta_T$, it suffices to require

$$\frac{8I\bar{v}^{2I-2}_{p,\pi(p)}(0) \log(1/\varepsilon)}{(\delta_v')^I} \le \delta_T \quad \Leftarrow \quad \varepsilon \ge \exp\left(-\frac{\delta_T (\delta_v')^I}{8I\bar{v}^{2I-2}_{p,\pi(p)}(0)}\right)$$

$$\Leftarrow \quad \varepsilon \ge \exp\left(-\frac{\delta_T (\delta_v')^I}{8I} \left(\frac{d}{\log^2 d}\right)^{I-1}\right).$$

Combine the above results, and we obtain

$$T_{1-\varepsilon} = T_{\delta_v} \pm 2\delta_T T_{\delta_v} = \frac{1 \pm 10\delta_T}{4I(I-1)\hat{\sigma}^2_{2I} a_{\pi(p)} \bar{v}^{2I-2}_{p,\pi(p)}(0)},$$

provided that the conditions of Lemma C.3, C.4, C.5 hold and

$$\delta_v \ge \left(\frac{\log^2 d}{d\delta_T}\right)^{I-1} \vee \left(\frac{4}{\delta_T}\right)^{\frac{1}{I-1}} \frac{\log^2 d}{d} \quad \text{and} \quad \varepsilon \ge \exp\left(-\frac{\delta_T (\delta_v')^I}{8I} \left(\frac{d}{\log^2 d}\right)^{I-1}\right).$$

$\square$

*Proof (Part II): resolving the conditions.* We now resolve the needed conditions. For easier reference, we list the requirements of Lemma C.3, C.4, C.5, and this lemma below:

$$\delta_v \le \frac{\delta_T}{2}\frac{2I\hat{\sigma}_{2I}^2}{C_\sigma^2}, \quad \gamma < \frac{1}{2I}, \quad m\sigma_1^2 \le \delta_T\frac{\hat{\sigma}_{2I}^2 a_{\min_*}}{2^{3I+4}C_\sigma^2 d^{I-1/2}},$$

$$\bar{\varepsilon} \le \left(\delta_T\frac{\hat{\sigma}_{2I}^2}{2^{3I+4}C_\sigma^2}\right)^2 \frac{1}{d^{1+2\gamma(I-1)}}, \quad d \ge \left(\frac{\hat{\sigma}_{2I}^2}{2^{3I+4}C_\sigma^2}\frac{a_{\min_*}}{\|a\|_1}\delta_T\right)^{-\frac{2}{1-2\gamma I}},$$

$$\delta_v' \le 1/3, \tag{7}$$

$$\varepsilon \ge \frac{2^{3I+7}C_\sigma^2}{(\delta_v')^I\hat{\sigma}_{2I}^2}\left\{\bar{\varepsilon}^{1/2}\varepsilon_0^{I-1} \vee \frac{m\sigma_1^2}{a_{\min_*}} \vee \frac{\|a\|_1}{a_{\min_*}}\varepsilon_0^I\right\},$$

$$\delta_v \ge \left(\frac{\log^2 d}{d\delta_T}\right)^{I-1} \vee \left(\frac{4}{\delta_T}\right)^{\frac{1}{I-1}}\frac{\log^2 d}{d}, \quad \varepsilon \ge \exp\left(-\frac{\delta_T(\delta_v')^I}{8I}\left(\frac{d}{\log^2 d}\right)^{I-1}\right).$$

We proceed under the following principle. First, $\varepsilon$ is a given parameter, so we should have minimal restrictions on it. $\delta_T$ should be interpreted as the final output of the lemma. In other parts of the proof, we only need to be $1/\text{poly}\,P$ small, and it is relatively easy to obtain contains of form $\delta_T \ge 1/d^c$. Hence, we will try to change condition on other parameters to conditions on $\delta_T$. Finally, $\delta_v, \delta_v'$ are only used in this proof, so it suffices to ensure the existence of them.

We start with the conditions on $\varepsilon$, which are

$$\varepsilon \ge \frac{2^{3I+7}C_\sigma^2}{(\delta_v')^I\hat{\sigma}_{2I}^2}\left\{\bar{\varepsilon}^{1/2}\varepsilon_0^{I-1} \vee \frac{m\sigma_1^2}{a_{\min_*}} \vee \frac{\|a\|_1}{a_{\min_*}}\varepsilon_0^I\right\} \vee \exp\left(-\frac{\delta_T(\delta_v')^I}{8I}\left(\frac{d}{\log^2 d}\right)^{I-1}\right).$$

This can be translated into

$$\varepsilon_0^I \le \frac{(\delta_v')^I\hat{\sigma}_{2I}^2}{2^{3I+7}C_\sigma^2}\frac{a_{\min_*}}{\|a\|_1}\varepsilon, \quad m\sigma_1^2 \le \frac{(\delta_v')^I\hat{\sigma}_{2I}^2}{2^{3I+7}C_\sigma^2}a_{\min_*}\varepsilon, \quad \bar{\varepsilon}^{1/2}\varepsilon_0^{I-1} \le \frac{(\delta_v')^I\hat{\sigma}_{2I}^2}{2^{3I+7}C_\sigma^2}\varepsilon,$$

$$\delta_T \ge \frac{8I}{(\delta_v')^I}\left(\frac{\log^2 d}{d}\right)^{I-1}\log\left(\frac{1}{\varepsilon}\right).$$

Then, consider $\delta_v, \delta_v'$. We choose $\delta_v' = 1/3$. For the existence of $\delta_v$, it suffices to require (cf. the first and second last conditions of (7))

$$\left(\frac{\log^2 d}{d\delta_T}\right)^{I-1} \vee \left(\frac{4}{\delta_T}\right)^{\frac{1}{I-1}}\frac{\log^2 d}{d} \le \frac{\delta_T}{2}\frac{2I\hat{\sigma}_{2I}^2}{C_\sigma^2}$$

$$\Longleftarrow \quad \delta_T \ge \left(\frac{C_\sigma^2}{I\hat{\sigma}_{2I}^2}\right)^{1/I}\left(\frac{\log^2 d}{d}\right)^{1-1/I} \vee \left(\frac{4C_\sigma^2}{I\hat{\sigma}_{2I}^2}\frac{\log^2 d}{d}\right)^{1-1/I}$$

$$\Longleftarrow \quad \delta_T \ge \frac{4C_\sigma^2}{I\hat{\sigma}_{2I}^2}\left(\frac{\log^2 d}{d}\right)^{1-1/I}.$$

This condition will also be stronger than the previous one, as long as

$$\frac{4C_\sigma^2}{I\hat{\sigma}_{2I}^2}\left(\frac{\log^2 d}{d}\right)^{1-1/I} \ge \frac{8I}{(\delta_v')^I}\left(\frac{\log^2 d}{d}\right)^{I-1}\log\left(\frac{1}{\varepsilon}\right)$$

$$\Longleftarrow \quad \varepsilon \ge \exp\left(-\frac{4C_\sigma^2}{I\hat{\sigma}_{2I}^2}\frac{(\delta_v')^I}{8I}\left(\frac{d}{\log^2 d}\right)^{I+1/I-2}\right).$$

While this is a restriction on $\varepsilon$, it is very mild as the RHS is super-polynomially small. Now, we have replaced (7) with

$$\varepsilon \ge \exp\left(-\frac{4C_\sigma^2}{I\hat{\sigma}_{2I}^2}\frac{(\delta_v')^I}{8I}\left(\frac{d}{\log^2 d}\right)^{I+1/I-2}\right), \quad m\sigma_1^2 \le \frac{\hat{\sigma}_{2I}^2 a_{\min_*}}{2^{3I+7}C_\sigma^2}\left((\delta_v')^I\varepsilon \wedge \frac{\delta_T}{d^{I-1/2}}\right),$$

$$\delta_T \geq \frac{4C_\sigma^2}{I\hat{\sigma}_{2I}^2} \left(\frac{\log^2 d}{d}\right)^{1-1/I},$$

$$\varepsilon_0^I \leq \frac{(\delta_v')^I \hat{\sigma}_{2I}^2}{2^{3I+7}C_\sigma^2} \frac{a_{\min_*}}{\|a\|_1} \varepsilon, \quad \bar{\varepsilon}^{1/2}\varepsilon_0^{I-1} \leq \frac{(\delta_v')^I \hat{\sigma}_{2I}^2}{2^{3I+7}C_\sigma^2}\varepsilon,$$

$$\gamma < \frac{1}{2I}, \quad \bar{\varepsilon} \leq \left(\delta_T \frac{\hat{\sigma}_{2I}^2}{2^{3I+4}C_\sigma^2}\right)^2 \frac{1}{d^{1+2\gamma(I-1)}}, \quad d \geq \left(\frac{\hat{\sigma}_{2I}^2}{2^{3I+4}C_\sigma^2} \frac{a_{\min_*}}{\|a\|_1}\delta_T\right)^{-\frac{2}{1-2\gamma I}}.$$

Consider the last two lines. For the second last line, we compute

$$\varepsilon_0^I \leq \frac{(\delta_v')^I \hat{\sigma}_{2I}^2}{2^{3I+7}C_\sigma^2} \frac{a_{\min_*}}{\|a\|_1}\varepsilon \quad \Leftarrow \quad d \geq \left(\frac{2^{3I+7}C_\sigma^2}{(\delta_v')^I \hat{\sigma}_{2I}^2} \frac{\|a\|_1}{a_{\min_*}} \frac{1}{\varepsilon}\right)^{\frac{1}{(1-\gamma)I}},$$

$$\bar{\varepsilon}^{1/2}\varepsilon_0^{I-1} \leq \frac{(\delta_v')^I \hat{\sigma}_{2I}^2}{2^{3I+7}C_\sigma^2}\varepsilon \quad \Leftarrow \quad \bar{\varepsilon} \leq \left(\frac{(\delta_v')^I \hat{\sigma}_{2I}^2}{2^{3I+7}C_\sigma^2}\right)^2 \varepsilon^2 d^{2(1-\gamma)(I-1)}.$$

For the last line, we convert the conditions into conditions on $\delta_T$:

$$\bar{\varepsilon} \leq \left(\delta_T \frac{\hat{\sigma}_{2I}^2}{2^{3I+4}C_\sigma^2}\right)^2 \frac{1}{d^{1+2\gamma(I-1)}} \quad \Leftrightarrow \quad \delta_T \geq \frac{2^{3I+4}C_\sigma^2}{\hat{\sigma}_{2I}^2}\sqrt{\bar{\varepsilon}d^{1+2\gamma(I-1)}},$$

$$d \geq \left(\frac{\hat{\sigma}_{2I}^2}{2^{3I+4}C_\sigma^2} \frac{a_{\min_*}}{\|a\|_1}\delta_T\right)^{-\frac{2}{1-2\gamma I}} \quad \Leftrightarrow \quad \delta_T \geq \frac{2^{3I+4}C_\sigma^2}{\hat{\sigma}_{2I}^2} \frac{\|a\|_1}{a_{\min_*}}d^{-1/2+\gamma I}.$$

Thus, the conditions are

$$\varepsilon \geq \exp\left(-\frac{4C_\sigma^2}{I\hat{\sigma}_{2I}^2} \frac{(\delta_v')^I}{8I}\left(\frac{d}{\log^2 d}\right)^{I+1/I-2}\right), \quad m\sigma_1^2 \leq \frac{\hat{\sigma}_{2I}^2 a_{\min_*}}{2^{3I+7}C_\sigma^2}\left((\delta_v')^I \varepsilon \wedge \frac{\delta_T}{d^{I-1/2}}\right),$$

$$d \geq \left(\frac{2^{3I+7}C_\sigma^2}{(\delta_v')^I \hat{\sigma}_{2I}^2} \frac{\|a\|_1}{a_{\min_*}} \frac{1}{\varepsilon}\right)^{\frac{1}{(1-\gamma)I}}, \quad \bar{\varepsilon} \leq \left(\frac{(\delta_v')^I \hat{\sigma}_{2I}^2}{2^{3I+7}C_\sigma^2}\right)^2 \varepsilon^2 d^{2(1-\gamma)(I-1)}.$$

$$\gamma < \frac{1}{2I}, \quad \delta_T \geq \frac{4C_\sigma^2}{I\hat{\sigma}_{2I}^2}\left(\frac{\log^2 d}{d}\right)^{1-1/I} \vee \frac{2^{3I+4}C_\sigma^2}{\hat{\sigma}_{2I}^2}\sqrt{\bar{\varepsilon}d^{1+2\gamma(I-1)}} \vee \frac{2^{3I+4}C_\sigma^2}{\hat{\sigma}_{2I}^2} \frac{\|a\|_1}{a_{\min_*}}d^{-1/2+\gamma I}.$$

Note that $1/2 - \gamma I \leq 1/2 \leq 1 - 1/I$ when $I \geq 2$. Hence, the condition on $\delta_T$ is equivalent to

$$\delta_T \geq \frac{2^{3I+4}C_\sigma^2}{\hat{\sigma}_{2I}^2}\sqrt{\bar{\varepsilon}d^{1+2\gamma(I-1)}} \vee \frac{2^{3I+4}C_\sigma^2}{\hat{\sigma}_{2I}^2} \frac{\|a\|_1}{a_{\min_*}}d^{-1/2+\gamma I}.$$

To complete the proof, it suffices to revert the above conditions to conditions on $\bar{\varepsilon}$ and $\delta_T$. □

### C.1.2 Radial Dynamics

Now, we estimate the time needed for a neuron to fit the ground truth after it converges in direction.

**Lemma C.7** (Dynamics of the norm (converged)). *Suppose that Induction Hypothesis C.2 is true at time t. Then, at time t, for any $p \in [P_*]$ with $\bar{v}_{p,\pi(p)}^2 \geq 1 - \bar{\varepsilon}$, we have*

$$\frac{d}{dt}\|v_p\|^2 = 4\|v_p\|^2\left(a_{\pi(p)} - \|v_p\|^2 \pm \left(2C_\sigma^2 a_{\pi(p)}\bar{\varepsilon} + 2\|a\|_1 2^{2I}\varepsilon_0^I + 2m\sigma_1^2\right)\right).$$

*Proof.* By Lemma B.1, we have

$$\frac{1}{2}\frac{d}{dt}\|v_p\|^2 = 2\|v_p\|^2 \sum_{i=I}^{\infty}\hat{\sigma}_{2i}^2 \sum_{q=1}^{P} a_{\pi(q)}\bar{v}_{p,\pi(q)}^{2i} - 2\|v_p\|^2 \sum_{i=I}^{\infty}\hat{\sigma}_{2i}^2 \sum_{l=1}^{m}\|v_l\|^2 \langle\bar{v}_p, \bar{v}_l\rangle^{2i}$$

$$= 2\|v_p\|^2 \sum_{i=I}^{\infty}\hat{\sigma}_{2i}^2\left(\sum_{q=1}^{P} a_{\pi(q)}\bar{v}_{p,\pi(q)}^{2i} - \|v_p\|^2\right) - 2\|v_p\|^2 \sum_{i=I}^{\infty}\hat{\sigma}_{2i}^2 \sum_{l:l\neq p}\|v_l\|^2 \langle\bar{v}_p, \bar{v}_l\rangle^{2i}$$

$$=: \mathrm{T}_1 \left(\frac{1}{2}\frac{\mathrm{d}}{\mathrm{d}t}\|v_P\|^2\right) + \mathrm{T}_2\left(\frac{1}{2}\frac{\mathrm{d}}{\mathrm{d}t}\|v_P\|^2\right).$$

First, for $\mathrm{T}_1$, first recall from Assumption 2.1 that $\sum_{i=I}^{\infty}\hat{\sigma}_{2i}^2 = 1$, and $\sum_{i=I}^{\infty} 2i\hat{\sigma}_{2i}^2 \le \sum_{i=I}^{\infty} i^2\hat{\sigma}_{2i}^2 \le C_\sigma^2$. Also note that for any small $\delta \in (0,1)$ and integer $N$, we have

$$(1-\delta)^N = 1 - N\delta + \delta^2 \sum_{k=0}^{N-2}\binom{N}{k+2}(-\delta)^k$$

$$= 1 - N\delta \pm N^2\delta^2 \sum_{k=0}^{N-2}\binom{N-2}{k}(-\delta)^k = 1 - N\delta \pm N^2\delta^2.$$

Hence, we can write

$$\mathrm{T}_1 = 2\|v_P\|^2 \sum_{i=I}^{\infty}\hat{\sigma}_{2i}^2\left(a_{\pi(p)} - \|v_P\|^2\right)$$

$$+ 2\|v_P\|^2 \sum_{i=I}^{\infty}\hat{\sigma}_{2i}^2 a_{\pi(p)}\left(\bar{v}_{p,\pi(p)}^{2i} - 1\right) + 2\|v_P\|^2 \sum_{i=I}^{\infty}\hat{\sigma}_{2i}^2 \sum_{q:q\ne p} a_{\pi(q)}\bar{v}_{p,\pi(q)}^{2i}$$

$$= 2\|v_P\|^2\left(a_{\pi(p)} - \|v_P\|^2\right) \pm 4C_\sigma^2\|v_P\|^2 a_{\pi(p)}\bar{\varepsilon} \pm 2\|v_P\|^2\|a\|_1\varepsilon_0^I.$$

Meanwhile, for $\mathrm{T}_2$, by the proof of Lemma C.2, we have

$$|\mathrm{T}_2| \le 2\|v_P\|^2 \sum_{i=I}^{\infty}\hat{\sigma}_{2i}^2 \sum_{l\in L\backslash\{p\}}\|v_l\|^2\langle\bar{v}_p,\bar{v}_l\rangle^{2i} + 2\|v_P\|^2\sum_{i=I}^{\infty}\hat{\sigma}_{2i}^2\sum_{l\notin L\cup\{p\}}\|v_l\|^2\langle\bar{v}_p,\bar{v}_l\rangle^{2i}$$

$$\le 4\|v_P\|^2\sum_{i=I}^{\infty}\hat{\sigma}_{2i}^2\sum_{l\in L\backslash\{p\}}a_{\pi(l)}\left(\sqrt{\varepsilon_0}+\sqrt{2\bar{\varepsilon}}\right)^{2i} + 2\|v_P\|^2 m\sigma_1^2$$

$$\le 4\|v_P\|^2\|a\|_1 2^{2I}\varepsilon_0^I + 2\|v_P\|^2 m\sigma_1^2.$$

As a result, we have

$$\frac{\mathrm{d}}{\mathrm{d}t}\|v_P\|^2 = 4\|v_P\|^2\left(a_{\pi(p)} - \|v_P\|^2\right)$$

$$\pm 8\|v_P\|^2\left(C_\sigma^2 a_{\pi(p)}\bar{\varepsilon} + \|a\|_1\varepsilon_0^I + \|a\|_1 2^{2I}\varepsilon_0^I + m\sigma_1^2\right)$$

$$= 4\|v_P\|^2\left(a_{\pi(p)} - \|v_P\|^2 \pm \left(2C_\sigma^2 a_{\pi(p)}\bar{\varepsilon} + 2\|a\|_1 2^{2I}\varepsilon_0^I + 2m\sigma_1^2\right)\right).$$

$\square$

**Lemma C.8** (Fitting the signal). *Inductively assume Induction Hypothesis C.2. Consider $p \in [P_*]$ and $\varepsilon \ge 4\left(C_\sigma^2 a_{\pi(p)}\bar{\varepsilon} + \|a\|_1 2^{2I}\varepsilon_0^I + m\sigma_1^2\right)$. Then, after $\bar{v}_{p,\pi(p)}^2$ reaches $1-\bar{\varepsilon}$, it takes at most $\frac{3\log\left(a_{\pi(p)}^2/(\sigma_0^2\varepsilon)\right)}{a_{\pi(p)}}$ amount of time for $\|v_P\|^2$ to reach $a_{\pi(p)} \pm \varepsilon$. In addition, once it enters this range, it will stay there.*

*Proof.* Let $T_0$ be the time $\bar{v}_{p,\pi(p)}^2$ reaches $1-\bar{\varepsilon}$. By the proof of Lemma C.6, $\bar{v}_{p,\pi(p)}^2$ will stay above $1-\bar{\varepsilon}$ after time $T_0$. By Lemma C.7 and our hypothesis on $\varepsilon$, we have

$$\frac{\mathrm{d}}{\mathrm{d}t}\|v_P\|^2 = 4\|v_P\|^2\left(a_{\pi(p)} - \|v_P\|^2 \pm \frac{\varepsilon}{2}\right).$$

In particular, this implies that once $\|v_P\|^2$ reaches $a_{\pi(p)} \pm \varepsilon$, it will stay in this range. Let $T_{R,1/2}$ and $T_{R,1-\varepsilon}$ be the time $\|v_P\|^2$ reaches $a_{\pi(p)}/2$ and $1-\varepsilon$, respectively. For any $t \le T_{R,1/2}$, we have

$$\frac{\mathrm{d}}{\mathrm{d}t}\|v_P\|^2 \ge \frac{4a_{\pi(p)}}{3}\|v_P\|^2 \quad\Rightarrow\quad \|v_P(t)\|^2 \ge \sigma_0^2\exp\left(\frac{4a_{\pi(p)}}{3}(t-T_0)\right)$$

$$\Rightarrow \quad T_{R,1/2} - T_0 \leq \frac{3 \log\left(a_{\pi(p)}/\sigma_0^2\right)}{a_{\pi(p)}}.$$

After $T_{R,1/2}$ and before $T_{R,1-\varepsilon}$, we have

$$\frac{\mathrm{d}}{\mathrm{d}t} \|v_P\|^2 \geq a_{\pi(p)} \left(a_{\pi(p)} - \|v_P\|^2 \pm \frac{\varepsilon}{2}\right) \geq \frac{a_{\pi(p)}}{2} \left(a_{\pi(p)} - \|v_P\|^2\right)$$

$$\Rightarrow \quad a_{\pi(p)}(t) - \|v_P\|^2 \leq \frac{a_{\pi(p)}}{2} \exp\left(-a_{\pi(p)}(t - T_{R,1/2})/2\right)$$

$$\Rightarrow \quad T_{R,1-\varepsilon} - T_{R,1/2} \leq \frac{3 \log\left(a_{\pi(p)}/\varepsilon\right)}{a_{\pi(p)}}.$$

As a result, we have

$$T_{R,1-\varepsilon} - T_0 \leq \frac{3}{a_{\pi(p)}} \left(\log\left(a_{\pi(p)}/\sigma_0^2\right) + \log\left(a_{\pi(p)}/\varepsilon\right)\right) = \frac{3 \log\left(a_{\pi(p)}^2/(\sigma_0^2\varepsilon)\right)}{a_{\pi(p)}}.$$

$\square$

We are now ready to prove the main result of this subsection, which we restate below.

**Corollary C.9** (Convergence). *Let* $\varepsilon_D, \varepsilon_R$ *be our target accuracy in the tangent and radial directions, and* $\delta_T$ *the target error in time. Suppose that* $\gamma < 1/(2I)$, $\delta_v' = 1/3$,

$$\varepsilon_D \geq \frac{2^{3I+7}C_\sigma^2}{(\delta_v')^I \hat{\sigma}_{2I}^2} \frac{\|a\|_1}{a_{\min_*}} \frac{1}{d^{(1-\gamma)I}}, \quad \varepsilon_R \geq 12 \|a\|_1 2^{2I} d^{-(1-\gamma)I}, \quad \delta_T \geq \frac{2^{3I+4}C_\sigma^2}{\hat{\sigma}_{2I}^2} \frac{\|a\|_1}{a_{\min_*}} \frac{1}{d^{1/2-\gamma I}},$$

$$m\sigma_1^2 \leq \frac{\hat{\sigma}_{2I}^2 a_{\min_*}}{2^{3I+7}C_\sigma^2} \left((\delta_v')^I \varepsilon \wedge \frac{\delta_T}{d^{I-1/2}}\right) \wedge \frac{\varepsilon_R}{12},$$

$$\bar{\varepsilon} \leq \left(\frac{(\delta_v')^I \hat{\sigma}_{2I}^2}{2^{3I+7}C_\sigma^2}\right)^2 \varepsilon_D^2 d^{2(1-\gamma)(I-1)} \wedge \left(\delta_T \frac{\hat{\sigma}_{2I}^2}{2^{3I+4}C_\sigma^2}\right)^2 \frac{1}{d^{1+2\gamma(I-1)}} \wedge \frac{\varepsilon_R}{12C_\sigma^2 a_{\pi(p)}}.$$

*Then, for any* $p \in [P_*]$, *we have*

$$\bar{v}_{p,\pi(p)}^2 \geq 1 - \varepsilon_D, \quad \|v_P\|^2 = a_{\pi(p)} \pm \varepsilon_R, \qquad \forall t \geq (1 + 20\delta_T)T_p,$$

$$\bar{v}_{p,\pi(p)}^2 \leq \left(\frac{4}{\delta_T}\right)^{\frac{1}{I-1}} \frac{\log^2 d}{d}, \quad \|v_P\|^2 \leq \sigma_1^2, \qquad \forall t \leq (1 - 10\delta_T)T_p,$$

*where*

$$T_p := \frac{1}{4I(I-1)\hat{\sigma}_{2I}^2 a_{\pi(p)} \bar{v}_{p,\pi(p)}^{2I-2}(0)} = \Theta\left(\frac{1}{a_{\pi(p)} \bar{v}_{p,\pi(p)}^{2I-2}(0)}\right) = \tilde{\Theta}\left(\frac{1}{a_{\pi(p)} d^{I-1}}\right).$$

*Proof.* First, by Lemma C.3 (and the proof of Lemma C.6), we have

$$\bar{v}_{p,\pi(p)}^2(t) \leq \delta_v := \left(\frac{4}{\delta_T}\right)^{\frac{1}{I-1}} \frac{\log^2 d}{d}, \quad \forall t \leq \frac{1 - 10\delta_T}{4I(I-1)\hat{\sigma}_{2I}^2 a_{\pi(p)} \bar{v}_{p,\pi(p)}^{2I-2}(0)}$$

Meanwhile, by Lemma C.6, we have $\bar{v}_{p,\pi(p)}^2 \leq \delta_v \bar{v}_{p,\pi(p)}^2 \geq 1 - \varepsilon_D$ after time

$$T_T = \frac{1 \pm 10\delta_T}{4I(I-1)\hat{\sigma}_{2I}^2 a_{\pi(p)} \bar{v}_{p,\pi(p)}^{2I-2}(0)} = \Theta\left(\frac{1}{a_{\pi(p)} \bar{v}_{p,\pi(p)}^{2I-2}(0)}\right),$$

as long as $\gamma < 1/(2I)$, $\delta_v' = 1/3$, and

$$\varepsilon_D \geq \frac{2^{3I+7}C_\sigma^2}{(\delta_v')^I \hat{\sigma}_{2I}^2} \frac{\|a\|_1}{a_{\min_*}} \frac{1}{d^{(1-\gamma)I}}, \quad \delta_T \geq \frac{2^{3I+4}C_\sigma^2}{\hat{\sigma}_{2I}^2} \frac{\|a\|_1}{a_{\min_*}} \frac{1}{d^{1/2-\gamma I}},$$

$$m\sigma_1^2 \leq \frac{\hat{\sigma}_{2I}^2 a_{\min_*}}{2^{3I+7}C_\sigma^2}\left((\delta_v')^I\varepsilon \wedge \frac{\delta_T}{d^{I-1/2}}\right),$$

$$\bar{\varepsilon} \leq \left(\frac{(\delta_v')^I\hat{\sigma}_{2I}^2}{2^{3I+7}C_\sigma^2}\right)^2\varepsilon_D^2 d^{2(1-\gamma)(I-1)} \wedge \left(\delta_T\frac{\hat{\sigma}_{2I}^2}{2^{3I+4}C_\sigma^2}\right)^2\frac{1}{d^{1+2\gamma(I-1)}}.$$

By Lemma C.8, fitting $a_{\pi(p)}$ to $\pm\varepsilon_R$ takes $T_R$ amount of time, where

$$T_R := \frac{3\log\left(a_{\pi(p)}^2/(\sigma_0^2\varepsilon_R)\right)}{a_{\pi(p)}}.$$

Since $\delta_T \geq \frac{2^{3I+4}C_\sigma^2}{\hat{\sigma}_{2I}^2}\frac{\|a\|_1}{a_{\min_*}}\frac{1}{d^{1/2-\gamma I}}$, we have

$$T_R \leq \delta_T T_T \quad \Longleftarrow \quad \log\left(a_{\pi(p)}^2/(\sigma_0^2\varepsilon_R)\right) \leq \frac{\delta_T d^{I-1}}{24I(I-1)\hat{\sigma}_{2I}^2(\log d)^{2I-2}}$$

$$\Longleftarrow \quad \varepsilon_R \geq \frac{a_{\pi(p)}^2}{\sigma_0^2}\exp\left(-\frac{d^{(1-\gamma)I-1/2}}{\hat{\sigma}_{2I}^2(\log d)^{2I-2}}\right).$$

Again, this condition is mild as the RHS decays exponentially fast. To meet the conditions of Lemma C.8, it suffices to require

$$\bar{\varepsilon} \leq \frac{\varepsilon_R}{12C_\sigma^2 a_{\pi(p)}}, \quad m\sigma_1^2 \leq \frac{\varepsilon_R}{12}, \quad \varepsilon_R \geq 12\|a\|_1 2^{2I}d^{-(1-\gamma)I}.$$

Note that last condition on $\varepsilon_R$ is stronger than the previous condition on $\varepsilon_R$. $\qquad\square$

## C.2 Maintaining the Induction Hypotheses

In this subsection, we show Induction Hypothesis C.2 is true throughout training. Recall the meaning and requirements of $\varepsilon_D, \varepsilon_R, \delta_T$ from Corollary C.9.

### C.2.1 Upper Bounds on the Irrelevant Coordinates

**Lemma C.10** (Upper triangular entries (case I)). *Consider $p \in [P_*]$ and $p < q \in [P]$ with $a_{\pi(q)} \geq a_{\min_*}/(2(\log d)^{2I-2})$. Assume the conditions of Corollary C.9 and*

$$\bar{\varepsilon} \leq \left(\frac{\hat{\sigma}_{2I}^2}{2^{3I+4}C_\sigma^2}\frac{\delta_r}{24}\right)^2\frac{1}{d^{1+2\gamma(I-1)}}, \quad m\sigma_1^2 \leq \frac{\hat{\sigma}_{2I}^2}{2^{3I+4}C_\sigma^2}\frac{a_{\min_*}}{2(\log d)^{2I-2}d^{I-1/2}}\frac{\delta_r}{24},$$

$$\frac{d}{(\log^2 d)^{1/\gamma}} \geq \left(\frac{\delta_r}{4}\right)^{-\frac{1}{\gamma(I-1)}}, \quad \frac{d}{(\log^2 d)^{\frac{I-1}{1/2-\gamma I}}} \geq \left(\frac{\hat{\sigma}_{2I}^2}{2^{3I+4}C_\sigma^2}\frac{a_{\min_*}}{\|a\|_1 2^{2I-2}}\frac{\delta_r}{24}\right)^{-\frac{1}{1/2-\gamma I}}, \quad \delta_T \leq \frac{\delta_r}{240}.$$

*Then, $\bar{v}_{p,\pi(q)}^2 \leq \varepsilon_0$ throughout training.*

**Remark.** Recall from Lemma C.6 that we only need $\delta_T \geq \tilde{\Theta}(1/d^{1/2-\gamma I})$ and by Lemma B.2, $\delta_r = 1/\text{poly}(P)$. Hence, the last condition can hold as long as $d$ is large.

*Proof.* First, by Corollary C.9, we know $\bar{v}_{p,\pi(p)}^2 \geq 1 - \bar{\varepsilon}$ after time

$$T_p := \frac{1 \pm 20\delta_T}{4I(I-1)\hat{\sigma}_{2I}^2 a_{\pi(p)}\bar{v}_{p,\pi(p)}^{2I-2}(0)}.$$

This automatically implies $\bar{v}_{p,\pi(p)}^2 \leq \bar{\varepsilon} \leq \varepsilon_0$ after time $T_p$. Hence, it suffices to consider the time before $T_p$. By Lemma C.2 and the choice $\varepsilon_0 \geq \bar{\varepsilon}$, we have

$$\frac{d}{dt}\bar{v}_{p,\pi(q)}^2 \leq 2\sum_{i=I}^{\infty}2i\hat{\sigma}_{2i}^2 a_{\pi(q)}\bar{v}_{p,\pi(q)}^{2i} + I2^{3I+6}C_\sigma^2|\bar{v}_{p,\pi(q)}|\left\{a_{\pi(q)}\bar{\varepsilon}^{1/2}\varepsilon_0^{I-1} \vee m\sigma_1^2 \vee \|a\|_1\varepsilon_0^I\right\}$$

$$=: \mathsf{T}_1 \left( \frac{\mathrm{d}}{\mathrm{d}t} \bar{v}_{p,\pi(q)}^2 \right) + \mathsf{T}_2 \left( \frac{\mathrm{d}}{\mathrm{d}t} \bar{v}_{p,\pi(q)}^2 \right).$$

Since our goal is to upper bound $\bar{v}_{p,\pi(q)}^2$, we may assume w.l.o.g. that $\bar{v}_{p,\pi(p)}^2 \geq 1/d$, as we only need to track those $t$. Then, for $\mathsf{T}_2$, we have

$$\mathsf{T}_2 \leq I 2^{3I+6} C_\sigma^2 d^{I-1/2} \left\{ a_{\pi(q)} \bar{\varepsilon}^{1/2} \varepsilon_0^{I-1} \vee m \sigma_1^2 \vee \|a\|_1 \varepsilon_0^I \right\} \bar{v}_{p,\pi(q)}^{2I}.$$

Meanwhile, for $\mathsf{T}_1$, we have

$$\mathsf{T}_1 = 4I \hat{\sigma}_{2I}^2 a_{\pi(q)} \bar{v}_{p,\pi(q)}^{2I} + 2 \sum_{i=I+1}^\infty 2i \hat{\sigma}_{2i}^2 a_{\pi(q)} \bar{v}_{p,\pi(q)}^{2i}$$

$$\leq 4I \hat{\sigma}_{2I}^2 a_{\pi(q)} \bar{v}_{p,\pi(q)}^{2I} + 2 a_{\pi(q)} \bar{v}_{p,\pi(q)}^{2I} \varepsilon_0 \sum_{i=I+1}^\infty 2i \hat{\sigma}_{2i}^2$$

$$\leq 4I \hat{\sigma}_{2I}^2 a_{\pi(q)} \bar{v}_{p,\pi(q)}^{2I} + 2 C_\sigma^2 a_{\pi(q)} \bar{v}_{p,\pi(q)}^{2I} \varepsilon_0.$$

Combining the above two bounds, we obtain

$$\frac{\mathrm{d}}{\mathrm{d}t} \bar{v}_{p,\pi(q)}^2 \leq 4I \hat{\sigma}_{2I}^2 a_{\pi(q)} \bar{v}_{p,\pi(q)}^{2I} + 2 C_\sigma^2 a_{\pi(q)} \bar{v}_{p,\pi(q)}^{2I} \varepsilon_0$$

$$+ I 2^{3I+6} C_\sigma^2 d^{I-1/2} \left\{ a_{\pi(q)} \left( \bar{\varepsilon}^{1/2} \varepsilon_0^{I-1} \vee \bar{\varepsilon}^{I-1/2} \right) \vee m \sigma_1^2 \vee \|a\|_1 \varepsilon_0^I \right\} \bar{v}_{p,\pi(q)}^{2I}$$

$$\leq \left( 1 + \delta_{\mathsf{Tmp}} \right) 4I \hat{\sigma}_{2I}^2 a_{\pi(q)} \bar{v}_{p,\pi(q)}^{2I},$$

where

$$\delta_{\mathsf{Tmp}} = \frac{2 C_\sigma^2 a_{\pi(q)} \varepsilon_0}{4I \hat{\sigma}_{2I}^2 a_{\pi(q)}} + \frac{I 2^{3I+6} C_\sigma^2 d^{I-1/2} \left\{ a_{\pi(q)} \left( \bar{\varepsilon}^{1/2} \varepsilon_0^{I-1} \vee \bar{\varepsilon}^{I-1/2} \right) \vee m \sigma_1^2 \vee \|a\|_1 \varepsilon_0^I \right\}}{4I \hat{\sigma}_{2I}^2 a_{\pi(q)}}$$

$$\leq \frac{C_\sigma^2 \varepsilon_0}{2I \hat{\sigma}_{2I}^2} + \frac{2^{3I+4} C_\sigma^2 d^{I-1/2}}{\hat{\sigma}_{2I}^2 a_{\pi(q)}} \left\{ a_{\pi(q)} \bar{\varepsilon}^{1/2} \varepsilon_0^{I-1} \vee m \sigma_1^2 \vee \|a\|_1 \varepsilon_0^I \right\}$$

$$=: \delta_{\mathsf{Tmp},1} + \delta_{\mathsf{Tmp},2}.$$

As a result, for any $t \leq T_p$, we have

$$\bar{v}_{p,\pi(q)}^2(t) \leq \bar{v}_{p,\pi(q)}^2(0) \left( 1 - (I-1) \left( 1 + \delta_{\mathsf{Tmp}} \right) 4I \hat{\sigma}_{2I}^2 a_{\pi(q)} \bar{v}_{p,\pi(q)}^{2I-2}(0) t \right)^{-\frac{1}{I-1}}$$

In particular, this implies

$$\bar{v}_{p,\pi(q)}^2(t) \leq \bar{v}_{p,\pi(q)}^2(0) \left( 1 - \left( 1 + \delta_{\mathsf{Tmp}} \right) \left( 1 + 20 \delta_T \right) \frac{a_{\pi(q)} \bar{v}_{p,\pi(q)}^{2I-2}(0)}{a_{\pi(p)} \bar{v}_{p,\pi(p)}^{2I-2}(0)} \right)^{-\frac{1}{I-1}}$$

$$\leq \bar{v}_{p,\pi(q)}^2(0) \left( 1 - \frac{\left( 1 + \delta_{\mathsf{Tmp}} \right) \left( 1 + 20 \delta_T \right)}{1 + \delta_r} \right)^{-\frac{1}{I-1}}$$

$$\leq \bar{v}_{p,\pi(q)}^2(0) \left( \frac{\delta_r}{2} - 2 \delta_{\mathsf{Tmp}} - 20 \delta_T \right)^{-\frac{1}{I-1}},$$

where the second line comes from Assumption C.1(a). Now, we find conditions under which the last term is upper bounded by $\varepsilon_0 = d^{-(1-\gamma)}$. We will first find conditions under which $2\delta_{\mathsf{Tmp}} + 20\delta_T \leq \delta_r/4$ and then upper bound $\bar{v}_{p,\pi(q)}^2(0) (\delta_r/4)^{-\frac{1}{I-1}}$.

We compute

$$2\delta_{\mathsf{Tmp},1} \leq \frac{\delta_r}{12} \quad \Longleftarrow \quad d \geq \left( \frac{I \hat{\sigma}_{2I}^2}{C_\sigma^2} \frac{\delta_r}{12} \right)^{-\frac{1}{1-\gamma}},$$

$$20\delta_T \leq \frac{\delta_r}{12} \quad \Longleftarrow \quad \delta_T \leq \frac{\delta_r}{240},$$

and by (6),

$$2\delta_{\text{Tmp},2} \le \frac{\delta_r}{12} \quad \Leftarrow \quad a_{\pi(q)}\bar{\varepsilon}^{1/2}\varepsilon_0^{I-1} \vee m\sigma_1^2 \vee \|a\|_1\varepsilon_0^I \le \frac{\hat{\sigma}_{2I}^2}{2^{3I+4}C_\sigma^2}\frac{a_{\pi(q)}}{d^{I-1/2}}\frac{\delta_r}{24}$$

$$\Leftarrow \quad \bar{\varepsilon} \le \left(\frac{\hat{\sigma}_{2I}^2}{2^{3I+4}C_\sigma^2}\frac{\delta_r}{24}\right)^2 \frac{1}{d^{1+2\gamma(I-1)}}, \quad m\sigma_1^2 \le \frac{\hat{\sigma}_{2I}^2}{2^{3I+4}C_\sigma^2}\frac{a_{\pi(q)}}{d^{I-1/2}}\frac{\delta_r}{24},$$

$$d \ge \left(\frac{\hat{\sigma}_{2I}^2}{2^{3I+4}C_\sigma^2}\frac{a_{\pi(q)}}{\|a\|_1}\frac{\delta_r}{24}\right)^{-\frac{1}{1/2-\gamma I}}.$$

The above conditions ensure $\delta_r/4 \ge 2\delta_{\text{Tmp}} + 20\delta_T$. By Assumption C.1(d), $\bar{v}^2_{p,\pi(p)}(0) \le \log^2 d/d$. Hence, in order for $\bar{v}^2_{p,\pi(q)}(0)(\delta_r/4)^{-1/(I-1)}$ to be smaller than $\varepsilon_0$, it suffices to have

$$\frac{\log^2 d}{d}\left(\frac{\delta_r}{4}\right)^{-\frac{1}{I-1}} \le d^{-(1-\gamma)} \quad \Leftarrow \quad \frac{d^\gamma}{\log^2 d} \ge \left(\frac{\delta_r}{4}\right)^{-\frac{1}{I-1}}.$$

We now clean up the conditions required by this lemma, which are the conditions of Corollary C.9 and

$$\bar{\varepsilon} \le \left(\frac{\hat{\sigma}_{2I}^2}{2^{3I+4}C_\sigma^2}\frac{\delta_r}{24}\right)^2 \frac{1}{d^{1+2\gamma(I-1)}}, \quad m\sigma_1^2 \le \frac{\hat{\sigma}_{2I}^2}{2^{3I+4}C_\sigma^2}\frac{a_{\pi(q)}}{d^{I-1/2}}\frac{\delta_r}{24},$$

$$\frac{d}{(\log^2 d)^{1/\gamma}} \ge \left(\frac{\delta_r}{4}\right)^{-\frac{1}{\gamma(I-1)}}, \quad d \ge \left(\frac{\hat{\sigma}_{2I}^2}{2^{3I+4}C_\sigma^2}\frac{a_{\pi(q)}}{\|a\|_1}\frac{\delta_r}{24}\right)^{-\frac{1}{1/2-\gamma I}} \vee \left(\frac{I\hat{\sigma}_{2I}^2}{C_\sigma^2}\frac{\delta_r}{12}\right)^{-\frac{1}{1-\gamma}}, \quad \delta_T \le \frac{\delta_r}{240}.$$

For the condition on $d$, since $1/2 - \gamma I \le 1/2 \le 1 - \gamma$, the first part of it is stronger. Finally, we use the hypothesis $a_{\pi(q)} \ge a_{\min_*}/(2(\log d)^{2I-2})$ to replace (the first part of) the second condition with

$$d \ge \left(\frac{\hat{\sigma}_{2I}^2}{2^{3I+4}C_\sigma^2}\frac{a_{\min_*}}{\|a\|_1 2^{2I-2}}\frac{\delta_r}{24}\right)^{-\frac{1}{1/2-\gamma I}} (\log^2 d)^{\frac{I-1}{1/2-\gamma I}}.$$

$\square$

**Lemma C.11** (Upper triangular entries (case II)). *Consider $p \in [P_*]$ and $p < q \in [P]$ with $a_{\pi(q)} \le a_{\min_*}/(2\log^{2I-2} d)$. Suppose that the hypotheses of Lemma C.10 are true. Then, $\bar{v}^2_{p,\pi(q)} \le \varepsilon_0$ throughout training.*

*Proof.* By the proof of Lemma C.10, we have

$$\frac{\mathrm{d}}{\mathrm{d}t}\bar{v}^2_{p,\pi(q)} \le 4I\hat{\sigma}_{2I}^2 a_{\pi(q)}\bar{v}^{2I}_{p,\pi(q)} + 2C_\sigma^2 a_{\pi(q)}\bar{v}^{2I}_{p,\pi(q)}\varepsilon_0$$
$$+ I2^{3I+6}C_\sigma^2 d^{I-1/2}\left\{a_{\pi(q)}\bar{\varepsilon}^{1/2}\varepsilon_0^{I-1} \vee m\sigma_1^2 \vee \|a\|_1\varepsilon_0^I\right\}\bar{v}^{2I}_{p,\pi(q)}.$$

Suppose that $a_{\pi(q)} \le a_{\min_*}/M$ for some $M \ge 1$ to be determined later. Then, we have

$$\frac{\mathrm{d}}{\mathrm{d}t}\bar{v}^2_{p,\pi(q)} \le 4I\hat{\sigma}_{2I}^2\frac{a_{\min_*}}{M}\bar{v}^{2I}_{p,\pi(q)} + 2C_\sigma^2\frac{a_{\min_*}}{M}\bar{v}^{2I}_{p,\pi(q)}\varepsilon_0$$
$$+ I2^{3I+6}C_\sigma^2 d^{I-1/2}\left\{\frac{a_{\min_*}}{M}\bar{\varepsilon}^{1/2}\varepsilon_0^{I-1} \vee m\sigma_1^2 \vee \|a\|_1\varepsilon_0^I\right\}\bar{v}^{2I}_{p,\pi(q)}$$
$$\le \left(1 + \delta_{\text{Tmp}}\right)4I\hat{\sigma}_{2I}^2\frac{a_{\min_*}}{M}\bar{v}^{2I}_{p,\pi(q)},$$

where

$$\delta_{\text{Tmp}} = \frac{C_\sigma^2\varepsilon_0}{2I\hat{\sigma}_{2I}^2} + \frac{2^{3I+6}C_\sigma^2}{4\hat{\sigma}_{2I}^2}d^{I-1/2}\frac{M}{a_{\min_*}}\left\{\frac{a_{\min_*}}{M}\bar{\varepsilon}^{1/2}\varepsilon_0^{I-1} \vee m\sigma_1^2 \vee \varepsilon_0^I\right\}$$
$$=: \delta_{\text{Tmp},1} + \delta_{\text{Tmp},2}.$$

As a result, for any $t \leq T_p$, we have

$$\bar{v}^2_{p,\pi(q)}(t) \leq \bar{v}^2_{p,\pi(q)}(0) \frac{a_{\min_*}}{M} \left(1 - (1 + \delta_{\text{Tmp}})(1 + 20\delta_T) \frac{a_{\min_*} \bar{v}^{2I-2}_{p,\pi(q)}(0)}{M a_{\pi(p)} \bar{v}^{2I-2}_{p,\pi(p)}(0)}\right)^{-\frac{1}{I-1}}.$$

Recall from Assumption C.1 that $\bar{v}^2_{p,\pi(p)}(0) \geq 1/d$ and $\bar{v}^2_{p,\pi(q)}(0) \leq \log^2 d/d$. Hence, with $M = 2\log^{2I-2} d$, we have

$$\frac{a_{\min_*} \bar{v}^{2I-2}_{p,\pi(q)}(0)}{M a_{\pi(p)} \bar{v}^{2I-2}_{p,\pi(p)}(0)} \leq \frac{a_{\min_*}}{a_{\pi(p)}} \frac{\log^{2I-2} d}{M} \leq \frac{1}{2}.$$

Hence,

$$\bar{v}^2_{p,\pi(q)}(t) \leq \bar{v}^2_{p,\pi(q)}(0) \left(1 - \frac{(1 + \delta_{\text{Tmp}})(1 + 10\delta_T)}{2}\right)^{-\frac{1}{I-1}}.$$

As a result, to ensure $\bar{v}^2_{p,\pi(q)} \leq \varepsilon_0$ throughout training, it suffices to have $\delta_{\text{Tmp}} \leq 0.1$ and $\delta_T \leq 0.01$. The second condition clear holds under the hypotheses of Lemma C.10. For the same reason, we have $\delta_{\text{Tmp},1} \leq 0.05$ and the first term in $\delta_{\text{Tmp},2}$ will also be sufficiently small. Finally, we compute

$$d^{I-1/2} \left\{ \frac{M}{a_{\min_*}} m\sigma_1^2 \vee \frac{M \|a\|_1}{a_{\min_*}} \varepsilon_0^I \right\} \leq \frac{1}{20} \frac{4\hat{\sigma}_{2I}^2}{2^{3I+6} C_\sigma^2}$$

$$\Leftarrow \quad m\sigma_1^2 \leq \frac{1}{20} \frac{4\hat{\sigma}_{2I}^2}{2^{3I+6} C_\sigma^2} \frac{a_{\min_*}}{2\log^{2I-2} d} \frac{1}{d^{I-1/2}}, \quad \frac{d}{\log^{\frac{4I}{1-2\gamma I}} d} \geq \left(\frac{1}{40} \frac{4\hat{\sigma}_{2I}^2}{2^{3I+6} C_\sigma^2} \frac{a_{\min_*}}{\|a\|_1}\right)^{-\frac{2}{1-2\gamma I}},$$

which are also covered by the conditions of Lemma C.10. In fact, $M$ is chosen to balance the requirements of these two lemmas. $\square$

**Lemma C.12** (Lower triangular entries). *Consider $p \in [P_*]$ and $p < k \in [m]$. Assume the conditions of Corollary C.9 and*

$$\delta_T \leq \frac{\delta_c}{240}, \quad \varepsilon_R \leq \frac{1}{6} \frac{a_{\min_*}^2 \delta_c}{8(\log^2 d)^{I-1}}, \quad \bar{\varepsilon} \leq \left(\frac{1}{48} \frac{4\hat{\sigma}_{2I}^2}{2^{3I+6} C_\sigma^2}\right)^2 \frac{a_{\min_*}^2 \delta_c^2}{(\log^2 d)^{2I-2}} \frac{1}{d^{1+2\gamma(I-1)}},$$

$$m\sigma_1^2 \leq \frac{1}{48} \frac{\hat{\sigma}_{2I}^2}{2^{3I+4} C_\sigma^2} \frac{a_{\min_*}^2 \delta_c}{(\log^2 d)^{I-1}} \frac{1}{d^{I-1/2}}, \quad \frac{d}{(\log^2 d)^{\frac{I-1}{1/2-\gamma I}}} \geq \left(\frac{1}{6} \frac{4\hat{\sigma}_{2I}^2}{2^{3I+6} C_\sigma^2} \frac{a_{\min_*}^2 \delta_c}{8 \|a\|_1}\right)^{-\frac{1}{1/2-\gamma I}}.$$

*Then, we have $\bar{v}^2_{k,\pi(p)} \leq \varepsilon_0$ throughout training.*

*Proof.* First, by Lemma C.2, we have

$$\frac{d}{dt} \bar{v}^2_{k,\pi(p)} = 2\bar{v}^2_{k,\pi(p)} \sum_{i=I}^{\infty} 2i\hat{\sigma}_{2i}^2 \left(a_{\pi(p)} \bar{v}^{2i-2}_{k,\pi(p)} - \sum_{r=1}^{P} a_{\pi(r)} \bar{v}^{2i}_{k,\pi(r)}\right)$$

$$- \mathbb{1}\{p \in L\} 2 \|v_p\|^2 \left(1 - \bar{v}^2_{k,\pi(p)}\right) \sum_{i=I}^{\infty} 2i\hat{\sigma}_{2i}^2 \bar{v}^{2i}_{k,\pi(p)}$$

$$\pm I 2^{3I+6} C_\sigma^2 |\bar{v}_{k,\pi(p)}| \left\{a_{\pi(p)} \bar{\varepsilon}^{1/2} \varepsilon_0^{I-1} \vee m\sigma_1^2 \vee \|a\|_1 \varepsilon_0^I\right\}$$

$$\leq 2\left(1 - \bar{v}^2_{k,\pi(p)}\right) \sum_{i=I}^{\infty} 2i\hat{\sigma}_{2i}^2 \left(a_{\pi(p)} - \mathbb{1}\{p \in L\} \|v_p\|^2\right) \bar{v}^{2i}_{k,\pi(p)}$$

$$+ I 2^{3I+6} C_\sigma^2 |\bar{v}_{k,\pi(p)}| \left\{a_{\pi(p)} \bar{\varepsilon}^{1/2} \varepsilon_0^{I-1} \vee m\sigma_1^2 \vee \|a\|_1 \varepsilon_0^I\right\}$$

$$=: T_1 \left(\frac{d}{dt} \bar{v}^2_{k,\pi(p)}\right) + T_2 \left(\frac{d}{dt} \bar{v}^2_{k,\pi(p)}\right).$$

Similar to the proof of Lemma C.10, we assume w.l.o.g. that $\bar{v}^2_{k,\pi(p)} \geq 1/d$ and write

$$\mathrm{T}_2 \leq I2^{3I+6}C_\sigma^2 \bar{v}^{2I}_{k,\pi(p)} d^{I-1/2} \left\{ a_{\pi(p)} \bar{\varepsilon}^{1/2} \varepsilon_0^{I-1} \vee m\sigma_1^2 \vee \|a\|_1 \varepsilon_0^I \right\}.$$

For the first term, we have

$$\mathrm{T}_1 \leq 4I\hat{\sigma}_{2I}^2 \left| a_{\pi(p)} - \mathbb{1}\{p \in L\} \|v_p\|^2 \right| \bar{v}^{2I}_{k,\pi(p)} + 2\sum_{i=I+1}^{\infty} 2i\hat{\sigma}_{2i}^2 a_{\pi(p)} \bar{v}^{2i}_{k,\pi(p)}$$

$$\leq 4I\hat{\sigma}_{2I}^2 \left| a_{\pi(p)} - \mathbb{1}\{p \in L\} \|v_p\|^2 \right| \bar{v}^{2I}_{k,\pi(p)} + 2C_\sigma^2 a_{\pi(p)} \bar{v}^{2I}_{k,\pi(p)} \varepsilon_0$$

$$\leq \left( \left| 1 - \frac{\mathbb{1}\{p \in L\} \|v_p\|^2}{a_{\pi(p)}} \right| + \frac{C_\sigma^2 \varepsilon_0}{2I\hat{\sigma}_{2I}^2} \right) \times 4I\hat{\sigma}_{2I}^2 a_{\pi(p)} \bar{v}^{2I}_{k,\pi(p)}.$$

Therefore,

$$\frac{\mathrm{d}}{\mathrm{d}t} \bar{v}^2_{k,\pi(p)} \leq \left( \left| 1 - \frac{\mathbb{1}\{p \in L\} \|v_p\|^2}{a_{\pi(p)}} \right| + \delta_{\mathrm{Tmp}} \right) \times 4I\hat{\sigma}_{2I}^2 a_{\pi(p)} \bar{v}^{2I}_{k,\pi(p)},$$

where

$$\delta_{\mathrm{Tmp}} := \frac{C_\sigma^2 \varepsilon_0}{2I\hat{\sigma}_{2I}^2} + \frac{2^{3I+6}C_\sigma^2}{4\hat{\sigma}_{2I}^2} d^{I-1/2} \left\{ \bar{\varepsilon}^{1/2} \varepsilon_0^{I-1} \vee \frac{m\sigma_1^2}{a_{\min_*}} \vee \frac{\|a\|_1}{a_{\min_*}} \varepsilon_0^I \right\}$$

$$=: \delta_{\mathrm{Tmp},1} + \delta_{\mathrm{Tmp},2}.$$

By Corollary C.9, we know $p \in L$ and $\|v_p\|^2 = a_{\pi(p)} \pm \varepsilon_R$ for $\varepsilon_R$ satisfying the condition in Corollary C.9 after time

$$T_p := \frac{1 \pm 20\delta_T}{4I(I-1)\hat{\sigma}_{2I}^2 a_{\pi(p)} \bar{v}^{2I-2}_{p,\pi(p)}(0)}.$$

We now analyze the stages $[0, T_p]$ and $[T_p, T_{P_*}]$, separately. Let $\varepsilon_0' \leq \varepsilon_0$ be a parameter to be chosen later. We want to show that $\bar{v}^2_{k,\pi(p)}$ is upper bounded by $\varepsilon_0'$ in the first stage and by $\varepsilon_0$ in the second stage.

First, for $t \leq T_p$, we have $\frac{\mathrm{d}}{\mathrm{d}t} \bar{v}^2_{k,\pi(p)} \leq (1 + \delta_{\mathrm{Tmp}}) \times 4I\hat{\sigma}_{2I}^2 a_{\pi(p)} \bar{v}^{2I}_{k,\pi(p)}$ and therefore

$$\bar{v}^2_{k,\pi(p)}(t) \leq \bar{v}^2_{k,\pi(p)}(0) \left( 1 - (I-1)(1+\delta_{\mathrm{Tmp}})4I\hat{\sigma}_{2I}^2 a_{\pi(p)} \bar{v}^{2I-2}_{k,\pi(p)}(0)t \right)^{-\frac{1}{I-1}}$$

$$\leq \bar{v}^2_{k,\pi(p)}(0) \left( 1 - (1+\delta_{\mathrm{Tmp}})(1+20\delta_T) \frac{a_{\pi(p)} \bar{v}^{2I-2}_{k,\pi(p)}(0)}{a_{\pi(p)} \bar{v}^{2I-2}_{p,\pi(p)}(0)} \right)^{-\frac{1}{I-1}}$$

$$\leq \bar{v}^2_{k,\pi(p)}(0) \left( \frac{\delta_c}{2} - 2\delta_{\mathrm{Tmp},1} - 2\delta_{\mathrm{Tmp},2} - 20\delta_T \right)^{-\frac{1}{I-1}},$$

where the last line comes from Assumption C.1(b). By the proof of Lemma C.10, we have

$$2\delta_{\mathrm{Tmp},1} + 2\delta_{\mathrm{Tmp},2} + 20\delta_T \leq \frac{\delta_c}{4},$$

provided that

$$\delta_T \leq \frac{\delta_c}{240}, \quad d \geq \left( \frac{\hat{\sigma}_{2I}^2}{2^{3I+4}C_\sigma^2} \frac{a_{\min_*}}{\|a\|_1} \frac{\delta_c}{24} \right)^{-\frac{1}{1/2-\gamma I}},$$

$$\bar{\varepsilon} \leq \left( \frac{\hat{\sigma}_{2I}^2}{2^{3I+4}C_\sigma^2} \frac{\delta_c}{24} \right)^2 \frac{1}{d^{1+2\gamma(I-1)}}, \quad m\sigma_1^2 \leq \frac{\hat{\sigma}_{2I}^2}{2^{3I+4}C_\sigma^2} \frac{a_{\min_*}}{d^{I-1/2}} \frac{\delta_c}{24}.$$

Then, we compute

$$\bar{v}^2_{k,\pi(p)}(t) \leq \varepsilon_0' \quad \Longleftarrow \quad \varepsilon_0' \geq \frac{\log^2 d}{d} \left( \frac{\delta_c}{4} \right)^{-\frac{1}{I-1}} \quad \Longleftarrow \quad \varepsilon_0' = \frac{\log^2 d}{d} \left( \frac{\delta_c}{4} \right)^{-\frac{1}{I-1}}.$$

Now, consider the second stage. For $t \geq T_p$, we have

$$\frac{\mathrm{d}}{\mathrm{d}t} \bar{v}_{k,\pi(p)}^2 \leq \left(\frac{\varepsilon_R}{a_{\pi(p)}} + \delta_{\mathrm{Tmp}}\right) \times 4I\hat{\sigma}_{2I}^2 a_{\pi(p)} \bar{v}_{k,\pi(p)}^{2I}$$

$$\implies \bar{v}_{k,\pi(p)}^2(t) \leq \varepsilon_0' \left(1 - \left(\frac{\varepsilon_R}{a_{\pi(p)}} + \delta_{\mathrm{Tmp}}\right) 4I(I-1)\hat{\sigma}_{2I}^2 a_{\pi(p)} (\varepsilon_0')^{I-1}(t - T_p)\right)^{-\frac{1}{I-1}}.$$

Also, recall that the training process ends before time

$$T_{P_*} = \frac{1 \pm 20\delta_T}{4I(I-1)\hat{\sigma}_{2I}^2 a_{\pi(P_*)} \bar{v}_{P_*,\pi(P_*)}^{2I-2}(0)}.$$

For any $t \in [T_p, T_{P_*}]$, we have

$$\bar{v}_{k,\pi(p)}^2 \leq \varepsilon_0' \left(1 - \left(\frac{\varepsilon_R}{a_{\pi(p)}} + \delta_{\mathrm{Tmp}}\right)(1 + 20\delta_T)\frac{a_{\pi(p)}(\varepsilon_0')^{I-1}}{a_{\pi(P_*)} \bar{v}_{P_*,\pi(P_*)}^{2I-2}(0)}\right)^{-\frac{1}{I-1}}$$

$$\leq \varepsilon_0' \left(1 - \left(\frac{\varepsilon_R}{a_{\min_*}} + \delta_{\mathrm{Tmp}}\right)\frac{2(d\varepsilon_0')^{I-1}}{a_{\min_*}}\right)^{-\frac{1}{I-1}}$$

$$\leq \varepsilon_0' \left(1 - \left(\frac{\varepsilon_R}{a_{\min_*}} + \delta_{\mathrm{Tmp}}\right)\frac{8(\log^2 d)^{I-1}}{a_{\min_*}\delta_c}\right)^{-\frac{1}{I-1}},$$

where the last line comes form choosing (C.2.1). For the last term to be bounded by $\varepsilon_0$, it suffices to require

$$\left(\frac{\varepsilon_R}{a_{\min_*}} + \delta_{\mathrm{Tmp}}\right)\frac{8(\log^2 d)^{I-1}}{a_{\min_*}\delta_c} \leq \frac{1}{2} \quad \Longleftarrow \quad \frac{\varepsilon_R}{a_{\min_*}} + \delta_{\mathrm{Tmp},1} + \delta_{\mathrm{Tmp},2} \leq \frac{1}{2}\frac{a_{\min_*}\delta_c}{8(\log^2 d)^{I-1}},$$

which is implied by

$$\varepsilon_R \leq \frac{1}{6}\frac{a_{\min_*}^2 \delta_c}{8(\log^2 d)^{I-1}}, \quad d \geq \left(\frac{1}{6}\frac{2I\hat{\sigma}_{2I}^2}{C_\sigma^2}\frac{a_{\min_*}\delta_c}{8(\log^2 d)^{I-1}}\right)^{-\frac{1}{1-\gamma}},$$

and by (6),

$$m\sigma_1^2 \leq \frac{1}{6}\frac{4\hat{\sigma}_{2I}^2}{2^{3I+6}C_\sigma^2}\frac{a_{\min_*}^2 \delta_c}{8(\log^2 d)^{I-1}}\frac{1}{d^{I-1/2}}, \quad \bar{\varepsilon} \leq \left(\frac{1}{6}\frac{4\hat{\sigma}_{2I}^2}{2^{3I+6}C_\sigma^2}\frac{a_{\min_*}\delta_c}{8(\log^2 d)^{I-1}}\right)^2\frac{1}{d^{1+2\gamma(I-1)}},$$

$$\frac{d}{(\log^2 d)^{\frac{I-1}{1/2-\gamma I}}} \geq \left(\frac{1}{6}\frac{4\hat{\sigma}_{2I}^2}{2^{3I+6}C_\sigma^2}\frac{a_{\min_*}^2 \delta_c}{8\|a\|_1}\right)^{-\frac{1}{1/2-\gamma I}}$$

Combining the above conditions with (C.2.1), we conclude that $\bar{v}_{k,\pi(p)}^2 \leq \varepsilon_0$ throughout training, as long as the conditions of Corollary C.9 and the following conditions are true:

$$\delta_T \leq \frac{\delta_c}{240}, \quad \varepsilon_R \leq \frac{1}{6}\frac{a_{\min_*}^2 \delta_c}{8(\log^2 d)^{I-1}},$$

$$m\sigma_1^2 \leq \frac{\hat{\sigma}_{2I}^2}{2^{3I+4}C_\sigma^2}\frac{a_{\min_*}}{d^{I-1/2}}\frac{\delta_c}{24} \wedge \frac{1}{6}\frac{4\hat{\sigma}_{2I}^2}{2^{3I+6}C_\sigma^2}\frac{a_{\min_*}^2 \delta_c}{8(\log^2 d)^{I-1}}\frac{1}{d^{I-1/2}},$$

$$\bar{\varepsilon} \leq \left(\frac{\hat{\sigma}_{2I}^2}{2^{3I+4}C_\sigma^2}\frac{\delta_c}{24}\right)^2\frac{1}{d^{1+2\gamma(I-1)}} \wedge \left(\frac{1}{6}\frac{4\hat{\sigma}_{2I}^2}{2^{3I+6}C_\sigma^2}\frac{a_{\min_*}\delta_c}{8(\log^2 d)^{I-1}}\right)^2\frac{1}{d^{1+2\gamma(I-1)}},$$

$$d \geq \left(\frac{\hat{\sigma}_{2I}^2}{2^{3I+4}C_\sigma^2}\frac{a_{\min_*}}{\|a\|_1}\frac{\delta_c}{24}\right)^{-\frac{1}{1/2-\gamma I}} \vee \left(\frac{1}{6}\frac{2I\hat{\sigma}_{2I}^2}{C_\sigma^2}\frac{a_{\min_*}\delta_c}{8(\log^2 d)^{I-1}}\right)^{-\frac{1}{1-\gamma}},$$

$$\frac{d}{(\log^2 d)^{\frac{I-1}{1/2-\gamma I}}} \geq \left( \frac{1}{6} \frac{4\hat{\sigma}_{2I}^2}{2^{3I+6} C_\sigma^2} \frac{a_{\min_*}^2 \delta_c}{8 \|a\|_1} \right)^{-\frac{1}{1/2-\gamma I}}.$$

To complete the proof, it suffices to keep only the stronger one in each of the conditions on $m\sigma_1^2$, $\bar{\varepsilon}$, and $d$. □

**Lemma C.13** (Lower right block). *Consider* $k \in [m], q \in [P]$ *with* $k, q > P_*$. *Assume the conditions of Corollary* C.9 *and the conditions of Lemma* C.10, *with* $\delta_r$ *replaced by* $\delta_t$. *Then, we have* $\bar{v}_{k,\pi(q)}^2 \leq \varepsilon_0$ *throughout training.*

*Proof.* By Lemma C.2, we have

$$\frac{\mathrm{d}}{\mathrm{d}t} \bar{v}_{k,\pi(q)}^2 \leq 2 \sum_{i=I}^{\infty} 2i\hat{\sigma}_{2i}^2 a_{\pi(q)} \bar{v}_{k,\pi(q)}^{2i}$$
$$\pm I 2^{3I+6} C_\sigma^2 |\bar{v}_{k,\pi(q)}| \left\{ a_{\pi(q)} \bar{\varepsilon}^{1/2} \varepsilon_0^{I-1} \vee m\sigma_1^2 \vee \|a\|_1 \varepsilon_0^I \right\}$$
$$=: \mathrm{T}_1 \left( \frac{\mathrm{d}}{\mathrm{d}t} \bar{v}_{k,\pi(q)}^2 \right) + \mathrm{T}_2 \left( \frac{\mathrm{d}}{\mathrm{d}t} \bar{v}_{k,\pi(q)}^2 \right).$$

For the first term, we have

$$\mathrm{T}_1 = 4I\hat{\sigma}_{2I}^2 a_{\pi(q)} \bar{v}_{k,\pi(q)}^{2I} + 2 \sum_{i=I+1}^{\infty} 2i\hat{\sigma}_{2i}^2 a_{\pi(q)} \bar{v}_{k,\pi(q)}^{2i}$$
$$\leq 4I\hat{\sigma}_{2I}^2 a_{\pi(q)} \bar{v}_{k,\pi(q)}^{2I} + 2C_\sigma^2 a_{\pi(q)} \varepsilon_0 \bar{v}_{k,\pi(q)}^{2I}$$
$$= \left( 1 + \frac{C_\sigma^2 \varepsilon_0}{2I\hat{\sigma}_{2I}^2} \right) \times 4I\hat{\sigma}_{2I}^2 a_{\pi(q)} \bar{v}_{k,\pi(q)}^{2I}.$$

Similar to the previous proofs, we may assume w.l.o.g. that $\bar{v}_{k,\pi(q)}^2 \geq 1/d$. Then, for the second term, we have

$$\mathrm{T}_2 \leq I 2^{3I+6} C_\sigma^2 d^{I+1/2} \left\{ a_{\pi(q)} \bar{\varepsilon}^{1/2} \varepsilon_0^{I-1} \vee m\sigma_1^2 \vee \|a\|_1 \varepsilon_0^I \right\} \bar{v}_{k,\pi(q)}^{2I}$$
$$= \frac{2^{3I+6} C_\sigma^2}{4\hat{\sigma}_{2I}^2} d^{I+1/2} \left\{ \bar{\varepsilon}^{1/2} \varepsilon_0^{I-1} \vee \frac{m\sigma_1^2}{a_{\pi(q)}} \vee \frac{\|a\|_1}{a_{\pi(q)}} \varepsilon_0^I \right\} \times 4I\hat{\sigma}_{2I}^2 a_{\pi(q)} \bar{v}_{k,\pi(q)}^{2I}.$$

As a result, we have

$$\frac{\mathrm{d}}{\mathrm{d}t} \bar{v}_{k,\pi(q)}^2 \leq \left( 1 + \frac{C_\sigma^2 \varepsilon_0}{2I\hat{\sigma}_{2I}^2} + \frac{2^{3I+6} C_\sigma^2}{4\hat{\sigma}_{2I}^2} d^{I+1/2} \left\{ \bar{\varepsilon}^{1/2} \varepsilon_0^{I-1} \vee \frac{m\sigma_1^2}{a_{\pi(q)}} \vee \frac{\|a\|_1}{a_{\pi(q)}} \varepsilon_0^I \right\} \right)$$
$$\times 4I\hat{\sigma}_{2I}^2 a_{\pi(q)} \bar{v}_{k,\pi(q)}^{2I}.$$

Note that this is the same as the bound in the proof of Lemma C.10 and Lemma C.11. Thus, to achieve $\bar{v}_{k,\pi(q)}^2 \leq \varepsilon_0$, it suffices to require the same conditions as in those two lemmas, with $\delta_r$ replaced by $\delta_t$ (cf. Assumption C.1). □

### C.2.2 Upper Bound on the Norm Growth

Here, we verify Induction Hypothesis C.2(a).

**Lemma C.14** (Upper bound on unused neurons). *Consider* $k \in [m]$ *with* $k > P_*$. *Suppose that*

$$\gamma < \frac{1}{I}, \quad d \geq \left( \frac{a_{\min_*}}{\|a\|_1} \frac{I(I-1)\hat{\sigma}_{2I}^2}{2} \right)^{-\frac{1}{1-\gamma I}}.$$

*Then, we have* $\|v_k\|^2 \leq e\sigma_0^2$ *throughout training.*

*Proof.* First, by Lemma B.1, Induction Hypothesis C.2(b) and Assumption 2.1, we have

$$\frac{\mathrm{d}}{\mathrm{d}t} \|v_k\|^2 \le 4 \|v_k\|^2 \sum_{i=I}^{\infty} \hat{\sigma}_{2i}^2 \sum_{p=1}^{P} a_p \bar{v}_{k,p}^{2i} \le 4 \|v_k\|^2 \sum_{i=I}^{\infty} \hat{\sigma}_{2i}^2 \sum_{p=1}^{P} a_p \varepsilon_0^i \le 4 \|a\|_1 \varepsilon_0^I \|v_k\|^2 .$$

Thus, by Gronwall's lemma, we have $\|v_k(t)\|^2 \le \sigma_0^2 \exp\left(4 \|a\|_1 \varepsilon_0^I t\right) \le e\sigma_0^2$ as long as $t \le (4 \|a\|_1 \varepsilon_0^I)^{-1}$. By Lemma C.6 and Lemma C.8, the training process ends at time

$$T_{P_*} \le \frac{2}{4I(I-1)\hat{\sigma}_{2I}^2 a_{\pi(P_*)} \bar{v}_{P_*,\pi(P_*)}^{2I-2}(0)} \le \frac{d^{I-1}}{2I(I-1)\hat{\sigma}_{2I}^2 a_{\min_*}} .$$

Hence, it suffices to require

$$\frac{1}{4 \|a\|_1 \varepsilon_0^I} \ge \frac{d^{I-1}}{2I(I-1)\hat{\sigma}_{2I}^2 a_{\min_*}} \quad \Leftarrow \quad d^{\gamma I - 1} \le \frac{a_{\min_*}}{\|a\|_1} \frac{I(I-1)\hat{\sigma}_{2I}^2}{2}$$

$$\Leftarrow \quad \gamma < \frac{1}{I}, \quad d \ge \left(\frac{a_{\min_*}}{\|a\|_1} \frac{I(I-1)\hat{\sigma}_{2I}^2}{2}\right)^{-\frac{1}{1-\gamma I}} .$$

□

Then, we consider $k = p \le P_*$. Unlike those unused neurons, since $v_p$ will eventually converge to $e_{\pi(p)}$, its norm cannot stay small. Our strategy here will be coupling its norm growth with the tangent movement.

**Lemma C.15** (Upper bound on $\|v_p\|^2$ with $p \le P_*$). *Consider $p \in [P_*]$. Suppose that the hypotheses of Lemma C.14 and Lemma C.6 hold. Then, $\|v_p\|^2 \ge \sigma_1^2$ only if $\bar{v}_{p,\pi(p)}^2 \ge 1 - \bar{\varepsilon}$, where $\sigma_1^2 := 2\sigma_0^2 e^{5/\hat{\sigma}_{2I}^2} \bar{\varepsilon}^{-8/(I\hat{\sigma}_{2I}^2)}$.*

*Proof.* Again, by Lemma B.1, Induction Hypothesis C.2(b) and Assumption 2.1, we have

$$\frac{\mathrm{d}}{\mathrm{d}t} \|v_p\|^2 \le 4 \|v_p\|^2 \sum_{i=I}^{\infty} \hat{\sigma}_{2i}^2 \sum_{q=1}^{P} a_{\pi(q)} \bar{v}_{p,\pi(q)}^{2i} \le 4 \|v_p\|^2 \sum_{i=I}^{\infty} \hat{\sigma}_{2i}^2 \left(a_{\pi(p)} \bar{v}_{p,\pi(p)}^{2i} + \|a\|_1 \varepsilon_0^i\right)$$

$$\le 4 \|v_p\|^2 a_{\pi(p)} \bar{v}_{p,\pi(p)}^{2I} + 4 \|v_p\|^2 \|a\|_1 \varepsilon_0^I.$$

Hence, by Gronwall's lemma, we have

$$\|v_p(t)\|^2 \le \sigma_0^2 \exp\left(4 \|a\|_1 \varepsilon_0^I t\right) \exp\left(4a_{\pi(p)} \int_0^t \bar{v}_{p,\pi(p)}^{2I}(s) \, \mathrm{d}s\right).$$

Let $c_0 > 0$ be a small constant to be determined later and let $T_0$ be the time $\bar{v}_{p,\pi(p)}^2$ reaches $1 - c_0/I$. By the proof of Lemma C.3, we know

$$\frac{\mathrm{d}}{\mathrm{d}t} \bar{v}_{p,\pi(p)}^2 \ge (1 - (1 - c_0/I) - o(1)) 4I\hat{\sigma}_{2I}^2 a_{\pi(p)} \bar{v}_{p,\pi(p)}^{2I} \ge c_0 2\hat{\sigma}_{2I}^2 a_{\pi(p)} \bar{v}_{p,\pi(p)}^{2I}.$$

Integrate both sides, and we obtain

$$1 \ge 1 - c_0/I - \bar{v}_{p,\pi(p)}^2(0) \ge c_0 2\hat{\sigma}_{2I}^2 a_{\pi(p)} \int_0^{T_0} \bar{v}_{p,\pi(p)}^{2I}(s) \, \mathrm{d}s.$$

As a result, for $t \le T_0$, we have

$$\|v_p(t)\|^2 \le \sigma_0^2 \exp\left(4 \|a\|_1 \varepsilon_0^I T_0\right) \exp\left(\frac{4a_{\pi(p)}}{c_0 2\hat{\sigma}_{2I}^2 a_{\pi(p)}}\right)$$

$$\le \sigma_0^2 \exp\left(4 \|a\|_1 \varepsilon_0^I T_0\right) \exp\left(\frac{2}{c_0 \hat{\sigma}_{2I}^2}\right).$$

Clear that $T_0 \le T_{P_*}$ and under the conditions of Lemma C.14, we have $4\|a\|_1 \varepsilon_0^I T_{P_*} \le 1$. Therefore,

$$\|v_P(t)\|^2 \le \sigma_0^2 \exp\left(1 + \frac{2}{c_0 \hat{\sigma}_{2I}^2}\right), \quad \forall t \le T_0.$$

Now, consider the $T_0 \le t \le T_1$, where $T_1$ is the time $\bar{v}_{p,\pi(p)}^2$ reaches $1 - \bar{\varepsilon}$. By Lemma C.5 (and the proof of Lemma C.6), we know

$$\frac{\mathrm{d}}{\mathrm{d}t}\bar{v}_{p,\pi(p)}^2 \ge (1 - c_0)I\hat{\sigma}_{2I}^2 a_{\pi(p)}\left(1 - \bar{v}_{p,\pi(p)}^2\right) \quad \Rightarrow \quad T_1 - T_0 \le \frac{\log\left(c_0/\varepsilon\right)}{(1 - c_0)I\hat{\sigma}_{2I}^2 a_{\pi(p)}}.$$

Thus, for $t \in [T_0, T_1]$, we have

$$\|v_P(t)\|^2 \le \|v_P(T_0)\|^2 \exp\left(4\|a\|_1 \varepsilon_0^I (T_1 - T_0)\right) \exp\left(4a_{\pi(p)}(T_1 - T_0)\right)$$

$$\le \|v_P(T_0)\|^2 (1 + o(1)) \exp\left(4\frac{\log\left(c_0/\varepsilon\right)}{(1 - c_0)I\hat{\sigma}_{2I}^2}\right)$$

$$\le \|v_P(T_0)\|^2 2\left(\frac{c_0}{\bar{\varepsilon}}\right)^{\frac{4}{(1-c_0)I\hat{\sigma}_{2I}^2}}.$$

Choose $c_0 = 1/2$ and recall $\|v_P(T_0)\|^2 \le \sigma_0^2 \exp\left(1 + \frac{2}{c_0 \hat{\sigma}_{2I}^2}\right)$. Then, we conclude that

$$\|v_P(t)\|^2 \le 2\sigma_0^2 e^{5/\hat{\sigma}_{2I}^2} \bar{\varepsilon}^{-8/(I\hat{\sigma}_{2I}^2)} =: \sigma_1^2,$$

for all $t \le T_1$. Recall from Lemma C.6 that once $\bar{v}_{p,\pi(p)}^2$ reaches $1 - \bar{\varepsilon}$, it will stay above $1 - \bar{\varepsilon}$. Thus, this implies that $\|v_P\|^2 \ge \sigma_1^2$ only if $\bar{v}_{p,\pi(p)}^2 \ge 1 - \bar{\varepsilon}$. $\qquad\square$

## C.3 Deferred Proofs

### C.3.1 Proof of Lemma C.2

*Proof of Lemma C.2.* Recall from Lemma B.1 that

$$-\frac{\left[(I - \bar{v}_k \bar{v}_k^\top)\nabla_{v_k}\mathcal{L}\right]_p}{\|v_k\|} = \sum_{i=I}^\infty 2i\hat{\sigma}_{2i}^2 \left(a_p \bar{v}_{k,p}^{2i-2} - \sum_{r=1}^P a_r \bar{v}_{k,r}^{2i}\right) \bar{v}_{k,p}$$

$$- \sum_{i=I}^\infty 2i\hat{\sigma}_{2i}^2 \sum_{l:l\ne k} \|v_l\|^2 \langle \bar{v}_k, \bar{v}_l \rangle^{2i-1} \left\langle (I - \bar{v}_k \bar{v}_k^\top)\bar{v}_l, e_p \right\rangle.$$

Re-index the summation as $\sum_{r=1}^P a_{\pi(r)} \bar{v}_{k,\pi(r)}^{2i}$, replace $p$ with $\pi(q)$, and we obtain

$$\dot{\bar{v}}_{k,\pi(q)} = \sum_{i=I}^\infty 2i\hat{\sigma}_{2i}^2 \left(a_{\pi(q)} \bar{v}_{k,\pi(q)}^{2i-2} - \sum_{r=1}^P a_{\pi(r)} \bar{v}_{k,\pi(r)}^{2i}\right) \bar{v}_{k,\pi(q)}$$

$$- \sum_{i=I}^\infty 2i\hat{\sigma}_{2i}^2 \sum_{l:l\ne k} \|v_l\|^2 \langle \bar{v}_k, \bar{v}_l \rangle^{2i-1} \left\langle (I - \bar{v}_k \bar{v}_k^\top)\bar{v}_l, e_{\pi(q)} \right\rangle.$$

Therefore, we have

$$\frac{\mathrm{d}}{\mathrm{d}t}\bar{v}_{k,\pi(q)}^2 = 2\bar{v}_{k,\pi(q)}^2 \sum_{i=I}^\infty 2i\hat{\sigma}_{2i}^2 \left(a_{\pi(q)} \bar{v}_{k,\pi(q)}^{2i-2} - \sum_{r=1}^P a_{\pi(r)} \bar{v}_{k,\pi(r)}^{2i}\right)$$

$$- \mathbb{1}\{k \ne q\} 2\bar{v}_{k,\pi(q)} \sum_{i=I}^\infty 2i\hat{\sigma}_{2i}^2 \|v_q\|^2 \langle \bar{v}_k, \bar{v}_q \rangle^{2i-1} \left\langle (I - \bar{v}_k \bar{v}_k^\top)\bar{v}_q, e_{\pi(q)} \right\rangle$$

$$- 2\bar{v}_{k,\pi(q)} \sum_{i=I}^\infty 2i\hat{\sigma}_{2i}^2 \sum_{l\notin\{k,q\}} \|v_l\|^2 \langle \bar{v}_k, \bar{v}_l \rangle^{2i-1} \left\langle (I - \bar{v}_k \bar{v}_k^\top)\bar{v}_l, e_{\pi(q)} \right\rangle$$

$$=: \mathrm{T}_1\left(\frac{\mathrm{d}}{\mathrm{d}t}\bar{v}_{k,\pi(q)}^2\right) + \mathrm{T}_2\left(\frac{\mathrm{d}}{\mathrm{d}t}\bar{v}_{k,\pi(q)}^2\right) + \mathrm{T}_3\left(\frac{\mathrm{d}}{\mathrm{d}t}\bar{v}_{k,\pi(q)}^2\right).$$

We keep $\mathrm{T}_1$ as it is, and simplify $\mathrm{T}_2$ and $\mathrm{T}_3$ as follows. Consider $\mathrm{T}_2$. When $q \notin L$, we have $\left\|v_q\right\|^2 \le \sigma_1^2$, and therefore,

$$(\text{When } q \notin L) \quad |\mathrm{T}_2| \le 2\left|\bar{v}_{k,\pi(q)}\right|\sum_{i=I}^{\infty} 2i\hat{\sigma}_{2i}^2\sigma_1^2 \le 2\left|\bar{v}_{k,\pi(q)}\right|C_\sigma^2\sigma_1^2,$$

where the last inequality comes from Assumption 2.1. Now, suppose that $q \in L$. In this case, we have $\bar{v}_q \approx s_q e_{\pi(q)}$ where $s_q := \operatorname{sgn}\bar{v}_{q,\pi(q)}$. This suggests writing

$$\left\langle\bar{v}_k,\bar{v}_q\right\rangle^{2i-1}\left\langle(I-\bar{v}_k\bar{v}_k^\top)\bar{v}_q,e_{\pi(q)}\right\rangle = \left\langle\bar{v}_k,\bar{v}_q\right\rangle^{2i-1}\left(\left\langle\bar{v}_q,e_{\pi(q)}\right\rangle - \left\langle\bar{v}_k,\bar{v}_q\right\rangle\left\langle\bar{v}_k,e_{\pi(q)}\right\rangle\right)$$
$$= \left\langle\bar{v}_k,\bar{v}_q\right\rangle^{2i-1}\bar{v}_{q,\pi(q)} - \left\langle\bar{v}_k,\bar{v}_q\right\rangle^{2i}\bar{v}_{k,\pi(q)}.$$

By Induction Hypothesis C.2(a), we have $\bar{v}_{q,\pi(q)}^2 \ge 1-\bar{\varepsilon}$. First, this implies $|\bar{v}_{q,\pi(q)}| \ge \sqrt{1-\bar{\varepsilon}} \ge 1-\bar{\varepsilon}$. Hence, $\bar{v}_{q,\pi(q)} = s_q \pm \bar{\varepsilon}$. In addition, we have

$$\left\|s_q e_{\pi(q)} - \bar{v}_q\right\| = \sqrt{2 - 2\left\langle s_q e_{\pi(q)},\bar{v}_q\right\rangle} = \sqrt{2 - 2s_q(s_q \pm \bar{\varepsilon})} \le \sqrt{2\bar{\varepsilon}}.$$

As a result, we have

$$\left\langle\bar{v}_k,\bar{v}_q\right\rangle = \left\langle\bar{v}_k,s_q e_{\pi(q)}\right\rangle + \left\langle\bar{v}_k,s_q e_{\pi(q)} - \bar{v}_q\right\rangle = s_q\bar{v}_{k,\pi(q)} \pm \left\|s_q e_{\pi(q)} - \bar{v}_q\right\| = s_q\bar{v}_{k,\pi(q)} \pm \sqrt{2\bar{\varepsilon}}.$$

Combine these estimations with the previous identity, and we obtain

$$\left\langle\bar{v}_k,\bar{v}_q\right\rangle^{2i-1}\left\langle(I-\bar{v}_k\bar{v}_k^\top)\bar{v}_q,e_{\pi(q)}\right\rangle = \left\langle\bar{v}_k,\bar{v}_q\right\rangle^{2i-1}\bar{v}_{q,\pi(q)} - \left\langle\bar{v}_k,\bar{v}_q\right\rangle^{2i}\bar{v}_{k,\pi(q)}$$
$$= \left(s_q\bar{v}_{k,\pi(q)} \pm \sqrt{2\bar{\varepsilon}}\right)^{2i-1}\left(s_q \pm \bar{\varepsilon}\right) - \left(s_q\bar{v}_{k,\pi(q)} \pm \sqrt{2\bar{\varepsilon}}\right)^{2i}\bar{v}_{k,\pi(q)}.$$

Note that, for any $a,\delta \in \mathbb{R}$ and integer $N$, we have

$$(a+\delta)^N = a^N + \sum_{n=1}^{N}\binom{N}{n}a^{N-n}\delta^n = a^N + \delta\sum_{n=0}^{N-1}\binom{N}{n+1}a^{N-n-1}\delta^n$$
$$= a^N + \delta\sum_{n=0}^{N-1}\binom{N-1}{n}\frac{N}{n+1}a^{(N-1)-n}\delta^n$$
$$= a^N \pm \delta N\left(|a|+|\delta|\right)^{N-1}$$
$$= a^N \pm N2^{N-1}\left(\delta|a|^{N-1}\vee|\delta|^N\right).$$

Thus, we can further rewrite the above as

$$\left\langle\bar{v}_k,\bar{v}_q\right\rangle^{2i-1}\left\langle(I-\bar{v}_k\bar{v}_k^\top)\bar{v}_q,e_{\pi(q)}\right\rangle$$
$$= \left(s_q^{2i-1}\bar{v}_{k,\pi(q)}^{2i-1} \pm i2^{3i}\left(\bar{\varepsilon}^{1/2}\bar{v}_{k,\pi(q)}^{2i-2}\vee\bar{\varepsilon}^{i-1/2}\right)\right)\left(s_q \pm \bar{\varepsilon}\right)$$
$$\quad - \left(\bar{v}_{k,\pi(q)}^{2i} \pm i2^{3i}\left(\bar{\varepsilon}^{1/2}\left|\bar{v}_{k,\pi(q)}\right|^{2i-1}\vee\bar{\varepsilon}^i\right)\right)\bar{v}_{k,\pi(q)}$$
$$= \left(1 - \bar{v}_{k,\pi(q)}^2\right)\bar{v}_{k,\pi(q)}^{2i-1}$$
$$\quad \pm \bar{v}_{k,\pi(q)}^{2i-1}\bar{\varepsilon} \pm 2i2^{3i}\left(\bar{\varepsilon}^{1/2}\bar{v}_{k,\pi(q)}^{2i-2}\vee\bar{\varepsilon}^{i-1/2}\right) \pm i2^{3i}\bar{v}_{k,\pi(q)}\left(\bar{\varepsilon}^{1/2}\left|\bar{v}_{k,\pi(q)}\right|^{2i-1}\vee\bar{\varepsilon}^i\right).$$

For the last three terms, clear that the second one is the largest as it has the smallest exponents on both $\bar{\varepsilon}$ and $\bar{v}_{k,\pi(q)}$. Also recall from Induction Hypothesis C.2(b) that $|\bar{v}_{k,\pi(q)}| \le \varepsilon_0$. Thus, we have

$$\left\langle\bar{v}_k,\bar{v}_q\right\rangle^{2i-1}\left\langle(I-\bar{v}_k\bar{v}_k^\top)\bar{v}_q,e_{\pi(q)}\right\rangle = \left(1 - \bar{v}_{k,\pi(q)}^2\right)\bar{v}_{k,\pi(q)}^{2i-1} \pm 3i2^{3i}\left(\bar{\varepsilon}^{1/2}\varepsilon_0^{i-1}\vee\bar{\varepsilon}^{i-1/2}\right).$$

As a result, we have

$$(\text{When } q \in L)$$

$$\mathrm{T}_2 = -\mathbb{1}\{k \neq q\}\, 2\bar{v}_{k,\pi(q)} \sum_{i=I}^{\infty} 2i\hat{\sigma}_{2i}^2 \|v_q\|^2 \left(\left(1 - \bar{v}_{k,\pi(q)}^2\right) \bar{v}_{k,\pi(q)}^{2i-1} \pm 3i2^{3i}\left(\bar{\varepsilon}^{1/2}\varepsilon_0^{i-1} \vee \bar{\varepsilon}^{i-1/2}\right)\right)$$

$$= -\mathbb{1}\{k \neq q\}\, 2 \sum_{i=I}^{\infty} 2i\hat{\sigma}_{2i}^2 \|v_q\|^2 \left(1 - \bar{v}_{k,\pi(q)}^2\right) \bar{v}_{k,\pi(q)}^{2i}$$

$$\pm 2\bar{v}_{k,\pi(q)} 3I2^{3I}\left(\bar{\varepsilon}^{1/2}\varepsilon_0^{I-1} \vee \bar{\varepsilon}^{I-1/2}\right) \sum_{i=I}^{\infty} 2i\hat{\sigma}_{2i}^2 \|v_q\|^2$$

$$= -\mathbb{1}\{k \neq q\}\, 2 \|v_q\|^2 \left(1 - \bar{v}_{k,\pi(q)}^2\right) \sum_{i=I}^{\infty} 2i\hat{\sigma}_{2i}^2 \bar{v}_{k,\pi(q)}^{2i}$$

$$\pm 12I2^{3I}C_\sigma^2 a_{\pi(q)}\bar{v}_{k,\pi(q)}\left(\bar{\varepsilon}^{1/2}\varepsilon_0^{I-1} \vee \bar{\varepsilon}^{I-1/2}\right).$$

Combining the cases $q \in L$ and $q \notin L$, we obtain

$$\mathrm{T}_2 = -\mathbb{1}\{k \neq q, q \in L\}\, 2 \|v_q\|^2 \left(1 - \bar{v}_{k,\pi(q)}^2\right) \sum_{i=I}^{\infty} 2i\hat{\sigma}_{2i}^2 \bar{v}_{k,\pi(q)}^{2i}$$

$$\pm 12I2^{3I}C_\sigma^2 a_{\pi(q)}\bar{v}_{k,\pi(q)}\left(\bar{\varepsilon}^{1/2}\varepsilon_0^{I-1} \vee \bar{\varepsilon}^{I-1/2}\right) \pm 2\left|\bar{v}_{k,\pi(q)}\right|C_\sigma^2\sigma_1^2.$$

Now, we estimate

$$\mathrm{T}_3 := -2\bar{v}_{k,\pi(q)} \sum_{i=I}^{\infty} 2i\hat{\sigma}_{2i}^2 \sum_{l \notin \{k,q\}} \|v_l\|^2 \langle \bar{v}_k, \bar{v}_l\rangle^{2i-1} \left\langle (I - \bar{v}_k\bar{v}_k^\top)\bar{v}_l, e_{\pi(q)}\right\rangle$$

$$:= -2\bar{v}_{k,\pi(q)} \sum_{i=I}^{\infty} 2i\hat{\sigma}_{2i}^2 \sum_{l \notin L \cup \{k,q\}} \|v_l\|^2 \langle \bar{v}_k, \bar{v}_l\rangle^{2i-1} \left\langle (I - \bar{v}_k\bar{v}_k^\top)\bar{v}_l, e_{\pi(q)}\right\rangle$$

$$- 2\bar{v}_{k,\pi(q)} \sum_{i=I}^{\infty} 2i\hat{\sigma}_{2i}^2 \sum_{l \in L \setminus \{k,q\}} \|v_l\|^2 \langle \bar{v}_k, \bar{v}_l\rangle^{2i-1} \left\langle (I - \bar{v}_k\bar{v}_k^\top)\bar{v}_l, e_{\pi(q)}\right\rangle$$

$$=: \mathrm{T}_{3.1} + \mathrm{T}_{3.2}.$$

Similar to the previous analysis, for $\mathrm{T}_{3.1}$, we have

$$|\mathrm{T}_{3.1}| \leq 2\left|\bar{v}_{k,\pi(q)}\right| \sum_{i=I}^{\infty} 2i\hat{\sigma}_{2i}^2 \sum_{l \notin L \cup \{k,q\}} \sigma_1^2 \leq 2C_\sigma^2\left|\bar{v}_{k,\pi(q)}\right|(m-1)\sigma_1^2.$$

Consider $\mathrm{T}_{3.2}$. Note that by our previous analysis, for any $l \in L \setminus \{k, q\}$, we have

$$\left|\langle \bar{v}_k, \bar{v}_l\rangle^{2i-1} \left\langle (I - \bar{v}_k\bar{v}_k^\top)\bar{v}_l, e_\pi(q)\right\rangle\right|$$

$$\leq \left|\left(s_l\bar{v}_{k,\pi(l)} \pm \sqrt{2\bar{\varepsilon}}\right)^{2i-1} \bar{v}_{l,\pi(q)}\right| + \left|\left(s_l\bar{v}_{k,\pi(l)} \pm \sqrt{2\bar{\varepsilon}}\right)^{2i} \bar{v}_{k,\pi(q)}\right|$$

$$\leq \left(\sqrt{\varepsilon_0} + \sqrt{2\bar{\varepsilon}}\right)^{2i-1} \sqrt{\varepsilon_0} + \left(\sqrt{\varepsilon_0} + \sqrt{2\bar{\varepsilon}}\right)^{2i}.$$

Note that $\sqrt{\varepsilon_0}^{2i} \vee \sqrt{\bar{\varepsilon}}^{2i-1}\sqrt{\varepsilon_0} \vee \sqrt{\bar{\varepsilon}}^{2i} = \varepsilon_0^i \vee \bar{\varepsilon}^i$. Hence, we can bound the last term as

$$\left|\langle \bar{v}_k, \bar{v}_l\rangle^{2i-1} \left\langle (I - \bar{v}_k\bar{v}_k^\top)\bar{v}_l, e_\pi(q)\right\rangle\right| \leq 2^{i+2}\left(\varepsilon_0^i \vee \bar{\varepsilon}^i\right).$$

Therefore,

$$|\mathrm{T}_{3.2}| \leq 2\bar{v}_{k,\pi(q)} \sum_{i=I}^{\infty} 2i\hat{\sigma}_{2i}^2 \sum_{l \in L \setminus \{k,q\}} \|v_l\|^2\, 2^{i+2}\left(\varepsilon_0^i \vee \bar{\varepsilon}^i\right) \leq 2^{I+5}C_\sigma^2 \|a\|_1 \left|\bar{v}_{k,\pi(q)}\right|\left(\varepsilon_0^I \vee \bar{\varepsilon}^I\right).$$

As a result, for $\mathrm{T}_3$, we have

$$|\mathrm{T}_3| \leq 2C_\sigma^2\left|\bar{v}_{k,\pi(q)}\right|(m-1)\sigma_1^2 + 2^{i+5}C_\sigma^2 \|a\|_1 \left|\bar{v}_{k,\pi(q)}\right|\left(\varepsilon_0^I \vee \bar{\varepsilon}^I\right).$$

Combine our bounds for $T_2$ and $T_3$, and we get

$$\frac{d}{dt}\bar{v}_{k,\pi(q)}^2 = 2\bar{v}_{k,\pi(q)}^2 \sum_{i=I}^{\infty} 2i\hat{\sigma}_{2i}^2 \left( a_{\pi(q)} \bar{v}_{k,\pi(q)}^{2i-2} - \sum_{r=1}^{P} a_{\pi(r)} \bar{v}_{k,\pi(r)}^{2i} \right)$$

$$- \mathbb{1}\{k \neq q, q \in L\} 2 \|v_q\|^2 \left(1 - \bar{v}_{k,\pi(q)}^2\right) \sum_{i=I}^{\infty} 2i\hat{\sigma}_{2i}^2 \bar{v}_{k,\pi(q)}^{2i}$$

$$\pm 12I2^{3I} C_\sigma^2 a_{\pi(q)} \bar{v}_{k,\pi(q)} \left(\bar{\varepsilon}^{1/2}\varepsilon_0^{I-1} \vee \bar{\varepsilon}^{I-1/2}\right) \pm 2|\bar{v}_{k,\pi(q)}| C_\sigma^2 \sigma_1^2$$

$$\pm C_\sigma^2 |\bar{v}_{k,\pi(q)}|(m-1)\sigma_1^2 \pm 2^{I+5} C_\sigma^2 \|a\|_1 |\bar{v}_{k,\pi(q)}| \left(\varepsilon_0^I \vee \bar{\varepsilon}^I\right).$$

For the last four error terms, clear that we can merge the second and the third terms, which leads to $2C_\sigma^2 |\bar{v}_{k,\pi(q)}| m\sigma_1^2$. Meanwhile, the largest coefficient is $12I2^{3I}C_\sigma^2$. Thus,

$$\frac{d}{dt}\bar{v}_{k,\pi(q)}^2 = 2\bar{v}_{k,\pi(q)}^2 \sum_{i=I}^{\infty} 2i\hat{\sigma}_{2i}^2 \left( a_{\pi(q)} \bar{v}_{k,\pi(q)}^{2i-2} - \sum_{r=1}^{P} a_{\pi(r)} \bar{v}_{k,\pi(r)}^{2i} \right)$$

$$- \mathbb{1}\{k \neq q, q \in L\} 2 \|v_q\|^2 \left(1 - \bar{v}_{k,\pi(q)}^2\right) \sum_{i=I}^{\infty} 2i\hat{\sigma}_{2i}^2 \bar{v}_{k,\pi(q)}^{2i}$$

$$\pm I2^{3I+6} C_\sigma^2 |\bar{v}_{k,\pi(q)}| \left\{ a_{\pi(q)} \left(\bar{\varepsilon}^{1/2}\varepsilon_0^{I-1} \vee \bar{\varepsilon}^{I-1/2}\right) \vee m\sigma_1^2 \vee \|a\|_1 \left(\varepsilon_0^I \vee \bar{\varepsilon}^I\right) \right\}.$$

Finally, recall that $\bar{\varepsilon} \leq \varepsilon_0$. Hence, $\bar{\varepsilon}^{1/2}\varepsilon_0^{I-1} \vee \bar{\varepsilon}^{I-1/2} = \bar{\varepsilon}^{1/2}\varepsilon_0^{I-1}$ and $\varepsilon_0^I \vee \bar{\varepsilon}^I = \varepsilon_0^I$.

Now, consider the second part of the lemma. In order for $a_{\pi(q)}\bar{\varepsilon}^{1/2}\varepsilon_0^{I-1} \vee m\sigma_1^2 \vee \|a\|_1 \varepsilon_0^I \leq \delta$, clear that we need $m\sigma_1^2 \leq \delta$. Meanwhile, for the last condition, we have

$$\|a\|_1 \varepsilon_0^I \leq \delta \quad \Longleftarrow \quad d^{-(1-\gamma)I} \leq \frac{\delta}{\|a\|_1} \quad \Longleftarrow \quad d \geq \left(\frac{\delta}{\|a\|_1}\right)^{-\frac{1}{(1-\gamma)I}}.$$

For the first condition, we have

$$a_{\pi(q)}\bar{\varepsilon}^{1/2}\varepsilon_0^{I-1} \leq \delta \quad \Longleftarrow \quad \bar{\varepsilon} \leq \left(\frac{\delta}{a_{\pi(q)}}\right)^2 d^{2(1-\gamma)(I-1)}.$$

$\square$

### C.3.2  Proof of Theorem C.1

*Proof of Theorem C.1.* By Corollary C.9, Lemma C.10, C.11, C.12, C.13, C.14, and C.15. Induction Hypothesis C.2 holds throughout training and the conclusions of Theorem C.1 are true, provided that all the conditions of these lemmas are met.

For easier reference, we collect the conditions of all above lemmas below:

$$\gamma < 1/(2I), \quad \delta_v' = 1/3, \quad \delta_{r,t} = \delta_r \wedge \delta_t,$$

$$\varepsilon_D \geq \frac{2^{3I+7}C_\sigma^2 \|a\|_1}{(\delta_v')^I \hat{\sigma}_{2I}^2} \frac{1}{a_{\min_*}} \frac{1}{d^{(1-\gamma)I}}, \quad \varepsilon_R \geq 12\|a\|_1 2^{2I} d^{-(1-\gamma)I}, \quad \delta_T \geq \frac{2^{3I+4}C_\sigma^2 \|a\|_1}{\hat{\sigma}_{2I}^2} \frac{1}{a_{\min_*}} \frac{1}{d^{1/2-\gamma I}},$$

$$m\sigma_1^2 \leq \frac{\hat{\sigma}_{2I}^2 a_{\min_*}}{2^{3I+7}C_\sigma^2} \left((\delta_v')^I \varepsilon \wedge \frac{\delta_T}{d^{I-1/2}}\right) \wedge \frac{\varepsilon_R}{12},$$

$$\bar{\varepsilon} \leq \left(\frac{(\delta_v')^I \hat{\sigma}_{2I}^2}{2^{3I+7}C_\sigma^2}\right)^2 \varepsilon_D^2 d^{2(1-\gamma)(I-1)} \wedge \left(\delta_T \frac{\hat{\sigma}_{2I}^2}{2^{3I+4}C_\sigma^2}\right)^2 \frac{1}{d^{1+2\gamma(I-1)}} \wedge \frac{\varepsilon_R}{12C_\sigma^2 a_{\pi(p)}},$$

$$\bar{\varepsilon} \leq \left(\frac{\hat{\sigma}_{2I}^2}{2^{3I+4}C_\sigma^2} \frac{\delta_{r,t}}{24}\right)^2 \frac{1}{d^{1+2\gamma(I-1)}}, \quad m\sigma_1^2 \leq \frac{\hat{\sigma}_{2I}^2}{2^{3I+4}C_\sigma^2} \frac{a_{\min_*}}{2(\log d)^{2I-2}d^{I-1/2}} \frac{\delta_{r,t}}{24},$$

$$\frac{d}{(\log^2 d)^{1/\gamma}} \geq \left(\frac{\delta_{r,t}}{4}\right)^{-\frac{1}{\gamma(I-1)}}, \quad \frac{d}{(\log^2 d)^{\frac{I-1}{1/2-\gamma I}}} \geq \left(\frac{\hat{\sigma}_{2I}^2}{2^{3I+4}C_\sigma^2} \frac{a_{\min_*}}{\|a\|_1 2^{2I-2}} \frac{\delta_{r,t}}{24}\right)^{-\frac{1}{1/2-\gamma I}}, \quad \delta_T \leq \frac{\delta_{r,t}}{240},$$

$$\delta_T \le \frac{\delta_c}{240}, \quad \varepsilon_R \le \frac{1}{6} \frac{a_{\min_*}^2 \delta_c}{8(\log^2 d)^{I-1}}, \quad \bar{\varepsilon} \le \left(\frac{1}{48} \frac{4\hat{\sigma}_{2I}^2}{2^{3I+6} C_\sigma^2}\right)^2 \frac{a_{\min_*}^2 \delta_c^2}{(\log^2 d)^{2I-2}} \frac{1}{d^{1+2\gamma(I-1)}},$$

$$m\sigma_1^2 \le \frac{1}{48} \frac{\hat{\sigma}_{2I}^2}{2^{3I+4} C_\sigma^2} \frac{a_{\min_*}^2 \delta_c}{(\log^2 d)^{I-1}} \frac{1}{d^{I-1/2}}, \quad \frac{d}{(\log^2 d)^{\frac{I-1}{1/2-\gamma I}}} \ge \left(\frac{1}{6} \frac{4\hat{\sigma}_{2I}^2}{2^{3I+6} C_\sigma^2} \frac{a_{\min_*}^2 \delta_c}{8\|a\|_1}\right)^{-\frac{1}{1/2-\gamma I}}.$$

In the following, for notational simplicity, we will use $\lesssim_\sigma$ and $\gtrsim_\sigma$ to hide constant that can only depend on $\sigma$. First, we consider the conditions on $\gamma$, which are

$$\gamma < \frac{1}{2I} \quad \text{and} \quad \frac{d}{(\log^2 d)^{1/\gamma}} \ge \left(\frac{\delta_{r,t}}{4}\right)^{-\frac{1}{\gamma(I-1)}}.$$

For concreteness, we will require $\gamma \le 1/(4I)$ and choose $\gamma$ such that

$$\frac{d^\gamma}{\log^2 d} = \left(\frac{\delta_{r,t}}{4}\right)^{-\frac{1}{I-1}}.$$

For such a $\gamma$ to exist, it suffices to have

$$\frac{d^{1/(4I)}}{\log^2 d} \ge \left(\frac{\delta_{r,t}}{4}\right)^{-\frac{1}{I-1}} \quad \Leftarrow \quad \frac{d}{\log^{8I} d} \gtrsim \delta_{r,t}^{-8}.$$

First, for the conditions on the target accuracy $\varepsilon_D, \varepsilon_R$ and error in time $\delta_T$, we need

$$\varepsilon_D \gtrsim_\sigma \frac{\|a\|_1}{a_{\min_*}} \frac{1}{d^{I-1/4}}, \quad \frac{1}{d^{I-1/4}} \lesssim_\sigma \varepsilon_R \lesssim_\sigma \frac{a_{\min_*}^2 \delta_c}{(\log^2 d)^{I-1}}, \quad \frac{\|a\|_1}{a_{\min_*}} \frac{1}{d^{1/4}} \lesssim_\sigma \delta_T \lesssim_\sigma \delta_c \wedge \delta_r \wedge \delta_t.$$

Then, for $\bar{\varepsilon}$, we choose

$$\bar{\varepsilon} =_\sigma \varepsilon_D^2 d^{2(I-1)} \wedge \frac{\delta_T^2 \delta_{r,t}^2}{d(\log d)^{4(I-1)}} \wedge \frac{\varepsilon_R}{a_{\min_*}} \wedge \frac{\delta_{r,t}^4}{d(\log d)^{4(I-1)}} \wedge \frac{a_{\min_*}^2 \delta_c^2}{(\log^2 d)^{2I-2}} \frac{\delta_{r,t}^2}{d(\log d)^{4(I-1)}}.$$

The condition on $m\sigma_1^2$ is

$$m\sigma_1^2 \lesssim_\sigma a_{\min_*} \varepsilon_D \wedge \frac{a_{\min_*} \delta_T}{d^{I-1/2}} \wedge \varepsilon_R \wedge \frac{a_{\min_*} \delta_{r,t}}{(\log d)^{2I-2} d^{I-1/2}} \wedge \frac{a_{\min_*}^2 \delta_c}{(\log^2 d)^{I-1}} \frac{1}{d^{I-1/2}}$$

Since $\sigma_1^2 := 2\sigma_0^2 e^{5/\hat{\sigma}_{2I}^2} \bar{\varepsilon}^{-8/(I\hat{\sigma}_{2I}^2)}$, this is equivalent to

$$\sigma_0^2 \lesssim_\sigma \frac{\bar{\varepsilon}^{8/(I\hat{\sigma}_{2I}^2)}}{m} \left(a_{\min_*} \varepsilon_D \wedge \frac{a_{\min_*} \delta_T}{d^{I-1/2}} \wedge \varepsilon_R \wedge \frac{a_{\min_*} \delta_{r,t}}{(\log d)^{2I-2} d^{I-1/2}} \wedge \frac{a_{\min_*}^2 \delta_c}{(\log^2 d)^{I-1}} \frac{1}{d^{I-1/2}}\right).$$

Finally, the conditions on $d$ are

$$\frac{d}{\log^{8I} d} \gtrsim \delta_{r,t}^{-8}, \quad \frac{d}{(\log^2 d)^{4(I-1)}} \ge \left(\frac{a_{\min_*}}{\|a\|_1} \delta_{r,t}\right)^{-\frac{1}{1/4}} \vee \left(\frac{a_{\min_*}^2 \delta_c}{\|a\|_1}\right)^{-\frac{1}{1/4}},$$

which can be merged into

$$\frac{d}{(\log^2 d)^{4I}} \gtrsim_\sigma \delta_{r,t}^{-8} \vee \left(\frac{a_{\min_*}}{\|a\|_1} \delta_{r,t}\right)^{-4} \vee \left(\frac{a_{\min_*}^2 \delta_c}{\|a\|_1}\right)^{-4},$$

$\square$

# D  Online SGD Dynamics

Our goal in this section is to prove Theorem 2.1, which we restate below for convenience:

**Theorem 2.1** (Main theorem for online SGD). *Let $C, C' > 0$ be large universal constants, depending only on $I$ and $\sigma$, and set the initialization scale as $\sigma_0 = d^{-C}$. Let $P_* \in [P]$, $a_{\min_*} = \min_{p \in [P_*]} a_p$, and $\delta_{\mathbb{P}}^*$ be the target failure probability. Define $\Delta \simeq \frac{\delta_{\mathbb{P}}^*}{mP\max(m,P)} = o_d(1)$. Assume the dimension $d$, width $m$, learning rate $\eta$ and target accuracies $\varepsilon_D, \varepsilon_R = o_d(1)$ satisfy*

$$d \gtrsim \|a\|_1^4 \Delta^{-8} a_{\min_*}^{-4}, \quad m \gtrsim P_*, \quad \eta \lesssim a_{\min_*} \|a\|_1^{-2} m^{-1} P^{-1} \delta_{\mathbb{P}}^* \min(\Delta^2 d^{-I}, \varepsilon_D^2),$$

$$\Delta^6 d^{-1} \gtrsim \varepsilon_D \gtrsim \|a\|_1 a_{\min_*}^{-1} d^{-I+1/4}, \quad P_*^{-1/2} \varepsilon_D^{1/2} \gtrsim \varepsilon_R \gtrsim \varepsilon_D,$$

*where $\lesssim, \gtrsim$ hide both constants and logarithmic factors. Then, with probability at least $1 - \delta_{\mathbb{P}}^*$, there exists an ordering of the student neurons $v_1, \ldots, v_m$ and a mapping $\pi : [P_*] \to [P]$ of student neurons to teacher neurons (see Equation (5)) such that, defining*

$$T_p := \left(4I(I-1)\hat{\sigma}_{2I}^2 a_{\pi(p)} \eta \bar{v}_{p,\pi(p)}^{2I-2}(0)\right)^{-1} \quad \forall p \in [P_*], \quad and \quad T_{\max} := (1 + \Delta/4) \max_{p \in [P_*]} T_p$$

*we have:*

(a) *(**Unused neurons**). $\|v_k(t)\|^2 \le d^{-C'} =: \sigma_1^2$ for all $k > P_*$.*

(b) *(**Convergence**). $\bar{v}_{p,\pi(p)}^2(t) \ge 1 - \varepsilon_D$, $\|v_p(t)\|^2 = a_{\pi(p)} \pm \varepsilon_R$ for all $p \in [P_*]$, $(1+\Delta)T_p \le t \le T_{\max}$.*

(c) *(**Sharp Transition**). $\bar{v}_{p,\pi(p)}^2(t) \le d^{-1/2}$, $\|v_p(t)\|^2 \le \sigma_1^2$ for all $p \in [P_*]$, $t \le (1 - \Delta)T_p$.*

(d) *(**Loss Value**). At time $t$, the population loss of the student network can be bounded by*

$$1 - \sum_{p \in [P_*]} a_{\pi(p)}^2 \mathbb{1}\left\{t \ge (1-\Delta/4)T_p\right\} - O(\varepsilon_D) \le \mathcal{L}(t) \le 1 - \sum_{p \in [P_*]} a_{\pi(p)}^2 \mathbb{1}\left\{t \ge (1+\Delta/4)T_p\right\} + O(\varepsilon_D).$$

Similarly to the gradient flow setting, our proof will proceed by maintaining Induction Hypothesis C.2 with high probability throughout training. We will additionally maintain the following induction hypothesis on the growth of $\|v_p\|^2$.

**Induction Hypothesis D.1.** *The neuron $v_p$ learns at time $(1 \pm o(1))T_p$; that is*

(a) $\bar{v}_{p,\pi(p)}^2(t) \ge 1 - \varepsilon_D$ *for all* $t \in \left[(1 + \frac{\Delta}{8})T_p, T_{\max}\right]$.

(b) $\left|\|v_p\|^2 - a_p\right| \le \varepsilon_R$ *for all* $t \in \left[(1 + \frac{\Delta}{4})T_p, T_{\max}\right]$

To maintain these induction hypotheses, we rely on the following stochastic induction argument from [RL24]. Suppose that the goal is to show a stochastic process $X_t$ stays close to its deterministic counterpart $x_t$ with high probability. First, we assume $X_t \approx x_t$ and use this induction hypothesis to obtain estimations on the related quantities, such as the variance of the noises. Then, using these estimations, we show that when $X_t$ is still close to $x_t$, the probability that $X_t$ will drift away from $x_t$ is small. This argument can be viewed as the stochastic counterpart of the continuity argument, and can be made rigorous by considering the stopping time $\tau$ that $X_t$ is no longer close to $x_t$ and analyzing the stopped process $(X_{t \wedge \tau})_t$. One may refer to Section F.2 of [RL24] for more details on this technique. Finally, we remark that this argument can be easily generalized to cases with multiple induction hypotheses by considering the stopping time that any of them is violated.

## D.1  Preliminaries

The following lemma decomposes the online SGD dynamics into the update on the radial component $\|v_k(t)\|^2$ and the tangent component $\bar{v}_{k,p}^2(t+1)$.

**Lemma D.1.** *Fix $k \in [m]$, $p \in [P]$ and $t > 0$. Let $\delta_{\mathbb{P},\xi} \in (0,1)$ be target failure probability at this step. Let $C > 0$ be a large universal constant. Suppose that $\eta \le 2\left(C\|a\|_1 d \log^{\tilde{Q}/2}(md/\delta_{\mathbb{P}})\right)^{-1}$ and let $H_k(t+1) := \hat{\nabla}_{v_k} l - \nabla_{v_k}\mathcal{L}$ denote the difference between the mini-batch gradient and the*

*population at this step. Then, we have (denoting $\boldsymbol{v}_k := \boldsymbol{v}_k(t)$):*

$$\|\boldsymbol{v}_k(t+1)\|^2 = \|\boldsymbol{v}_k\|^2 + 4\eta \left( \sum_{i=I}^{\infty} \hat{\sigma}_{2i}^2 \sum_{p=1}^{P} a_p \bar{v}_{k,p}^{2i} - \sum_{i=I}^{\infty} \hat{\sigma}_{2i}^2 \sum_{l=1}^{m} \|\boldsymbol{v}_l\|^2 \langle \bar{\boldsymbol{v}}_k, \bar{\boldsymbol{v}}_l \rangle^{2i} \right) \|\boldsymbol{v}_k\|^2$$
$$- 2\eta \langle \boldsymbol{v}_k, \boldsymbol{H}_k \rangle + \xi_{k,R}(t+1),$$

$$\bar{v}_{k,p}^2(t+1) = \bar{v}_{k,p}^2 + 2\eta \bar{v}_{k,p}^2 \cdot \sum_{i=I}^{\infty} 2i\hat{\sigma}_{2i}^2 \left( a_p \bar{v}_{k,p}^{2i-2} - \sum_{q=1}^{P} a_q \bar{v}_{k,q}^{2i} \right)$$
$$- 2\eta \bar{v}_{k,p} \sum_{i=I}^{\infty} 2i\hat{\sigma}_{2i}^2 \sum_{l:l\neq k} \|\boldsymbol{v}_l\|^2 \langle \bar{\boldsymbol{v}}_k, \bar{\boldsymbol{v}}_l \rangle^{2i-1} \left\langle (\boldsymbol{I} - \bar{\boldsymbol{v}}_k \bar{\boldsymbol{v}}_k^\top) \bar{\boldsymbol{v}}_l, \boldsymbol{e}_p \right\rangle$$
$$- 2\eta \bar{v}_{k,p} \frac{\left\langle (\boldsymbol{I} - \bar{\boldsymbol{v}}_k \bar{\boldsymbol{v}}_k^\top) \boldsymbol{H}_k, \boldsymbol{e}_p \right\rangle}{\|\boldsymbol{v}_k\|} + \xi_{k,p}(t+1),$$

*where $\xi_{k,R}(t+1)$ and $\xi_{k,p}(t+1)$ satisfy*

$$|\xi_{k,R}(t+1)| \leq C\eta^2 d \|\boldsymbol{a}\|_1^2 \log^{\tilde{Q}} \left( \frac{md}{\delta_{\mathbb{P},\xi}} \right) \|\boldsymbol{v}_k\|^2, \quad |\xi_{k,p}(t+1)| \leq C\eta^2 \left( 1 \vee \bar{v}_{k,p}^2 d \right) \|\boldsymbol{a}\|_1^2 \log^{\tilde{Q}} \left( \frac{md}{\delta_{\mathbb{P},\xi}} \right)$$

*with probability at least $1 - \delta_{\mathbb{P},\xi}$.*

*Proof.* Let $k \in [m]$ be fixed and $t > 0$. We write

$$\hat{\nabla}_{\boldsymbol{v}_k} l = \nabla_{\boldsymbol{v}_k} \mathcal{L} + \left( \hat{\nabla}_{\boldsymbol{v}_k} l - \nabla_{\boldsymbol{v}_k} \mathcal{L} \right) =: \nabla_{\boldsymbol{v}_k} \mathcal{L} + \boldsymbol{H}_k,$$

where $\hat{\nabla}$ denotes the mini-batch gradient. First, consider the dynamics of $\|\boldsymbol{v}_k\|^2$. By Lemma B.1, we have that

$$\|\boldsymbol{v}_k(t+1)\|^2 = \left\| \boldsymbol{v}_k - \eta \hat{\nabla}_{\boldsymbol{v}_k} l \right\|^2$$
$$= \|\boldsymbol{v}_k\|^2 - 2\eta \langle \boldsymbol{v}_k, \nabla_{\boldsymbol{v}_k} \mathcal{L} \rangle - 2\eta \langle \boldsymbol{v}_k, \boldsymbol{H}_k \rangle + \eta^2 \left\| \hat{\nabla}_{\boldsymbol{v}_k} l \right\|^2$$
$$= \|\boldsymbol{v}_k\|^2 + 4\eta \left( \sum_{i=I}^{\infty} \hat{\sigma}_{2i}^2 \sum_{p=1}^{P} a_p \bar{v}_{k,p}^{2i} - \sum_{i=I}^{\infty} \hat{\sigma}_{2i}^2 \sum_{l=1}^{m} \|\boldsymbol{v}_l\|^2 \langle \bar{\boldsymbol{v}}_k, \bar{\boldsymbol{v}}_l \rangle^{2i} \right) \|\boldsymbol{v}_k\|^2$$
$$- 2\eta \langle \boldsymbol{v}_k, \boldsymbol{H}_k \rangle + \eta^2 \left\| \hat{\nabla}_{\boldsymbol{v}_k} l \right\|^2.$$

By the tail bound in Lemma B.1, for any given direction $\boldsymbol{u} \in \mathbb{S}^{d-1}$, with probability at least $1 - \delta_{\mathbb{P}}$, we have $\left| \langle \hat{\nabla}_{\boldsymbol{v}_k} l, \boldsymbol{u} \rangle \right| \leq C \|\boldsymbol{a}\|_1 \log^{\tilde{Q}/2}(m/\delta_{\mathbb{P}}) \|\boldsymbol{v}_k\|$, for some universal constant $C > 0$. Take $\boldsymbol{u}$ to be $\boldsymbol{v}_k$ and $\boldsymbol{e}_1, \dots, \boldsymbol{e}_d$, and replace $\delta_{\mathbb{P}}$ with $\delta_{\mathbb{P}}/(2d)$. Then, we obtain

$$\left| \langle \boldsymbol{v}_k, \hat{\nabla}_{\boldsymbol{v}_k} l \rangle \right| \leq C \|\boldsymbol{a}\|_1 \log^{\tilde{Q}/2}(md/\delta_{\mathbb{P}}) \|\boldsymbol{v}_k\|^2, \quad \left\| \hat{\nabla}_{\boldsymbol{v}_k} l \right\|^2 \leq C^2 d \|\boldsymbol{a}\|_1^2 \log^{\tilde{Q}}(md/\delta_{\mathbb{P}}) \|\boldsymbol{v}_k\|^2,$$

for some universal constant $C > 0$ with probability at least $1 - \delta_{\mathbb{P}}$. Plugging in the bound for $\left\| \hat{\nabla}_{\boldsymbol{v}_k} l \right\|^2$ yields the desired update for $\|\boldsymbol{v}_k(t+1)\|^2$.

We next analyze the dynamics of $\bar{v}_{k,p}^2$ where $p \in [P]$. To this end, first we estimate $1/\|\boldsymbol{v}_k(t+1)\|^2$. With probability $1 - \delta_{\mathbb{P}}$ we have that,

$$\|\boldsymbol{v}_k(t+1)\|^2 = \|\boldsymbol{v}_k\|^2 - 2\eta \langle \nabla_{\boldsymbol{v}_k} l, \boldsymbol{v}_k \rangle + \eta^2 \left\| \nabla_{\boldsymbol{v}_k} l \right\|^2$$
$$= \|\boldsymbol{v}_k\|^2 \left( 1 \pm 2\eta C \|\boldsymbol{a}\|_1 \log^{\tilde{Q}/2}(md/\delta_{\mathbb{P}}) \pm C^2 \eta^2 d \|\boldsymbol{a}\|_1^2 \log^{\tilde{Q}}(md/\delta_{\mathbb{P}}) \right).$$

When $\eta \leq 2 \left( C \|\boldsymbol{a}\|_1 d \log^{\tilde{Q}/2}(md/\delta_{\mathbb{P}}) \right)^{-1}$, we have

$$C^2 \eta^2 d \|\boldsymbol{a}\|_1^2 \log^{\tilde{Q}}(md/\delta_{\mathbb{P}}) \leq 2\eta C \|\boldsymbol{a}\|_1 \log^{\tilde{Q}/2}(md/\delta_{\mathbb{P}}) \leq \frac{1}{4}.$$

Hence, we can use the identity

$$\frac{1}{1+\delta} = 1 - \delta \pm 2\delta^2, \quad \forall \, |\delta| \le 1/2,$$

to obtain

$$\frac{1}{\|\boldsymbol{v}_k(t+1)\|^2} = \frac{1}{\|\boldsymbol{v}_k\|^2} \left( 1 + \frac{2\eta \langle \boldsymbol{v}_k, \hat{\nabla}_{\boldsymbol{v}_k} l \rangle}{\|\boldsymbol{v}_k\|^2} + \frac{\eta^2 \|\hat{\nabla}_{\boldsymbol{v}_k} l\|^2}{\|\boldsymbol{v}_k\|^2} \pm 8C^2 \eta^2 \|\boldsymbol{a}\|_1^2 \log^{\tilde{Q}} \left( \frac{md}{\delta_{\mathbb{P}}} \right) \right)$$

$$= \frac{1}{\|\boldsymbol{v}_k\|^2} \left( 1 + \frac{2\eta \langle \boldsymbol{v}_k, \hat{\nabla}_{\boldsymbol{v}_k} l \rangle}{\|\boldsymbol{v}_k\|^2} \pm 2C^2 \eta^2 d \|\boldsymbol{a}\|_1^2 \log^{\tilde{Q}} \left( \frac{md}{\delta_{\mathbb{P}}} \right) \right).$$

Therefore the update for $\boldsymbol{v}_{k,p}(t+1)$ is

$$\bar{v}_{k,p}^2(t+1) = \frac{v_{k,p}^2 - 2\eta v_{k,p} \langle \hat{\nabla}_{\boldsymbol{v}_k} l, \boldsymbol{e}_p \rangle + \eta^2 \langle \hat{\nabla}_{\boldsymbol{v}_k} l, \boldsymbol{e}_p \rangle^2}{\|\boldsymbol{v}_k(t+1)\|^2}$$

$$= \left( \bar{v}_{k,p}^2 - 2\eta \bar{v}_{k,p} \frac{\langle \hat{\nabla}_{\boldsymbol{v}_k} l, \boldsymbol{e}_p \rangle}{\|\boldsymbol{v}_k\|} \pm C^2 \eta^2 \|\boldsymbol{a}\|_1^2 \log^{\tilde{Q}} \left( \frac{md}{\delta_{\mathbb{P}}} \right) \right)$$

$$\times \left( 1 + \frac{2\eta \langle \boldsymbol{v}_k, \hat{\nabla}_{\boldsymbol{v}_k} l \rangle}{\|\boldsymbol{v}_k\|^2} \pm 2C^2 \eta^2 d \|\boldsymbol{a}\|_1^2 \log^{\tilde{Q}} \left( \frac{md}{\delta_{\mathbb{P}}} \right) \right)$$

$$= \bar{v}_{k,p}^2 - 2\eta \bar{v}_{k,p} \frac{\langle \hat{\nabla}_{\boldsymbol{v}_k} l, \boldsymbol{e}_p \rangle}{\|\boldsymbol{v}_k\|} + \frac{2\eta \langle \boldsymbol{v}_k, \hat{\nabla}_{\boldsymbol{v}_k} l \rangle}{\|\boldsymbol{v}_k\|^2} \bar{v}_{k,p}^2 \pm O\left( \eta^2 \left( 1 \vee \bar{v}_{k,p}^2 d \right) \|\boldsymbol{a}\|_1^2 \log^{\tilde{Q}} \left( \frac{md}{\delta_{\mathbb{P}}} \right) \right)$$

$$= \bar{v}_{k,p}^2 - 2\eta \bar{v}_{k,p} \frac{\langle (\boldsymbol{I} - \bar{\boldsymbol{v}}_k \bar{\boldsymbol{v}}_k^\top) \hat{\nabla} l, \boldsymbol{e}_p \rangle}{\|\boldsymbol{v}_k\|} \pm O\left( \eta^2 \left( 1 \vee \bar{v}_{k,p}^2 d \right) \|\boldsymbol{a}\|_1^2 \log^{\tilde{Q}} \left( \frac{md}{\delta_{\mathbb{P}}} \right) \right).$$

Finally, write $\hat{\nabla}_{\boldsymbol{v}_k} l = \nabla \mathcal{L} + \boldsymbol{H}_k$, use our previous formula from Lemma B.1 for the tangent term of $\nabla \mathcal{L}$, and we obtain

$$\bar{v}_{k,p}^2(t+1) = \bar{v}_{k,p}^2 + 2\eta \bar{v}_{k,p}^2 \cdot \sum_{i=I}^{\infty} 2i \hat{\sigma}_{2i}^2 \left( a_p \bar{v}_{k,p}^{2i-2} - \sum_{q=1}^{P} a_q \bar{v}_{k,q}^{2i} \right)$$

$$- 2\eta \bar{v}_{k,p} \sum_{i=I}^{\infty} 2i \hat{\sigma}_{2i}^2 \sum_{l:l \neq k} \|\boldsymbol{v}_l\|^2 \langle \bar{\boldsymbol{v}}_k, \bar{\boldsymbol{v}}_l \rangle^{2i-1} \langle (\boldsymbol{I} - \bar{\boldsymbol{v}}_k \bar{\boldsymbol{v}}_k^\top) \bar{\boldsymbol{v}}_l, \boldsymbol{e}_p \rangle$$

$$- 2\eta \bar{v}_{k,p} \frac{\langle (\boldsymbol{I} - \bar{\boldsymbol{v}}_k \bar{\boldsymbol{v}}_k^\top) \boldsymbol{H}_k, \boldsymbol{e}_p \rangle}{\|\boldsymbol{v}_k\|} \pm O\left( \eta^2 \left( 1 \vee \bar{v}_{k,p}^2 d \right) \|\boldsymbol{a}\|_1^2 \log^{\tilde{Q}} \left( \frac{md}{\delta_{\mathbb{P}}} \right) \right).$$

$\square$

For notational convenience, we will define the quantity $\Delta := \min(\delta_c, \delta_r, \delta_t)$.

## D.2 Convergence Guarantees

In this subsection, we show under Induction Hypothesis C.2 that for all $p \in [P_*]$, $\bar{v}_{p,\pi(p)}^2$ reaches 1 in time $(1 \pm o(1)) T_p$.

### D.2.1 Tangent Dynamics

We begin by tracking the growth of the signal term $\bar{v}_{p,\pi(p)}^2$, for $p \in [P_*]$. Our goal is to prove the following lemma.

**Lemma D.2** (Directional Convergence). *Let $p \in [P_*]$. Inductively assume Induction Hypothesis C.2, and that the conditions on Lemma C.3 hold. Let the target accuracy $\varepsilon_D$ satisfy $\varepsilon_D \ge \frac{2^{3I+7} 3^I C_\sigma^2}{\hat{\sigma}_{2I}^2} \left\{ \bar{\varepsilon}^{1/2} \varepsilon_0^{I-1} \vee \frac{m \sigma_1^2}{a_{\min_*}} \vee \frac{\|\boldsymbol{a}\|_1}{a_{\min_*}} \varepsilon_0^I \right\}$, the dimension $d$ satisfy*

$$\frac{d}{\log^4 d} \ge 2^{20} I^2 \Delta^{-2}, \quad d \ge \frac{C^2 I^2 C_\sigma^4 \Delta^{-4}}{\hat{\sigma}_{2I}^4},$$

*the learning rate $\eta$ satisfy*

$$\eta \le \frac{a_{\pi(p)}\hat{\sigma}_{2I}^2 \|\boldsymbol{a}\|_1^{-2}\delta_{\mathbb{P}}}{C\log(512I/\Delta)\log^{\tilde{Q}}\left(\frac{md}{\delta_{\mathbb{P},\xi}}\right)}\min(d^{-I}\Delta^2, 3^{-I}d^{-1}\varepsilon_D, 3^{-I}\varepsilon_D^2)$$

*for sufficiently large constant $C$. Then, with probability $1 - T_{max}\delta_{\mathbb{P},\xi} - \delta_{\mathbb{P}}\cdot\log\log d$, we have*

$$\bar{v}_{p,\pi(p)}^2(t) \le \frac{1}{\sqrt{d}}, \quad \forall t \le \frac{1 - \Delta/256}{4I(I-1)\hat{\sigma}_{2I}^2\eta a_{\pi(p)}\bar{v}_{p,\pi(p)}^{2I-2}(0)}$$

$$\bar{v}_{p,\pi(p)}^2(t) \ge 1 - \varepsilon_D, \quad \forall\frac{1 + \Delta/8}{4I(I-1)\hat{\sigma}_{2I}^2\eta a_{\pi(p)}\bar{v}_{p,\pi(p)}^{2I-2}(0)} \le t \le T_{max}.$$

The proof of Lemma D.2 is split into stages based on the size of $\bar{v}_{p,\pi(p)}^2$. We first consider the case when $\bar{v}_{p,\pi(p)}^2$ is small. The update is given by the following:

**Lemma D.3.** *Assume that Induction Hypothesis C.2 holds, and moreover that $\bar{v}_{p,\pi(p)}^2 \le \delta_{\bar{v}}$ for some $\delta_{\bar{v}} > 0$. Let $\delta_T \ge \frac{C_\sigma^2\delta_{\bar{v}}}{I\hat{\sigma}_{2I}^2}$. Then, under the same conditions as Lemma C.3, we have*

$$\bar{v}_{p,\pi(p)}^2(t+1) = \bar{v}_{p,\pi(p)}^2(t) + 4I\hat{\sigma}_{2I}^2\eta a_{\pi(p)}\bar{v}_{p,\pi(p)}^{2I}(t) + Z(t+1) + \xi(t+1),$$

*where $\mathbb{E}[Z(t+1)\mid\mathcal{F}_t] \lesssim \eta^2\|\boldsymbol{a}\|_1^2\bar{v}_{p,\pi(p)}^2$, and with probability $1 - \delta_{\mathbb{P},\xi}$.*

$$|\xi(t+1)| \lesssim \eta^2(1\vee\bar{v}_{p,\pi(p)}^2 d)\|\boldsymbol{a}\|_1^2\log^{\tilde{Q}}\left(\frac{md}{\delta_{\mathbb{P},\xi}}\right) + \eta\delta_T I\hat{\sigma}_{2I}^2 a_{\pi(p)}\bar{v}_{p,\pi(p)}^{2I}$$

*Proof.* This follows directly from Lemma C.3 and Lemma D.1. $\qquad\square$

This motivates the following stochastic induction helper lemma, with proof deferred to Appendix D.5

**Lemma D.4.** *Let $(X_t)_t$ satisfy*

$$X_{t+1} = X_t + \alpha X_t^I + \xi_{t+1} + Z_{t+1}, \quad X_0 = x_0, \tag{8}$$

*where $(\xi_t)_t$ is an adapted process and $(Z_t)_t$ is a martingale difference sequence. Define the processes $(x_t^+)_t, (x_t^-)_t$ by*

$$x_{t+1}^+ = \left(1 + \alpha\left(x_t^+\right)^{I-1}\right)x_t^+, \quad x_0^+ = (1 + \varepsilon)x_0$$

$$x_{t+1}^- = \left(1 + \alpha\left(x_t^-\right)^{I-1}\right)x_t^-, \quad x_0^- = (1 - \varepsilon)x_0.$$

*Suppose that when $X_t \in [x_t^-, x_t^+]$ we have $|\xi_{t+1}| \le X_t^I\Xi_1 + X_t\Xi_2 + \Xi_3$ with probability $1 - \delta_{\mathbb{P},\xi}$, and $\mathbb{E}[Z_{t+1}\mid\mathcal{F}_t] \le X_t\sigma_Z^2$. Then, if*

$$\Xi_1 \le \frac{\varepsilon x_0}{6\sum_{t=0}^{T-1}\hat{x}_t^I}, \quad \Xi_2 \le \frac{\varepsilon x_0}{6\sum_{t=0}^{T-1}\hat{x}_t}, \quad \Xi_3 \le \frac{\varepsilon x_0}{6T}, \quad \text{and} \quad \sigma_Z^2 \le \frac{x_0^2\varepsilon^2\delta_{\mathbb{P}}}{4\sum_{t=0}^{T-1}\hat{x}_t},$$

*we have $X_t \in [x_t^-, x_t^+]$ for all $t \le T$, with probability $1 - T\delta_{\mathbb{P},\xi} - \delta_{\mathbb{P}}$.*

We can use this lemma to bound the time it takes for $\bar{v}_{p,\pi(p)}^2$ to reach some $\omega(1/d)$ quantity.

**Lemma D.5** (Weak Recovery). *Assume that the learning rate $\eta$ satisfies*

$$\eta \ll \frac{a_{\pi(p)}\hat{\sigma}_{2I}^2 d^{-I}\|\boldsymbol{a}\|_1^{-2}\Delta^2\delta_{\mathbb{P}}}{\log(512I/\Delta)\log^{\tilde{Q}}\left(\frac{md}{\delta_{\mathbb{P},\xi}}\right)}.$$

*Moreover, assume that the conditions of Lemma C.3 hold for $\delta_v = d^{-1/2}, \delta_T = \frac{\Delta^2}{CI^2}$ for sufficiently large constant $C$, and also that*

$$\frac{d}{\log^4 d} \ge 2^{20}I^2\Delta^{-2}, \quad d \ge \frac{C^2C_\sigma^4 I^2\Delta^{-4}}{\hat{\sigma}_{2I}^4}.$$

*Define $T^+$ by*

$$T^+ := (1 - \Delta/256)T_p = \frac{1 - \Delta/256}{4I(I-1)\hat{\sigma}_{2I}^2 \eta a_{\pi(p)} \bar{v}_{p,\pi(p)}^{2I-2}(0)}.$$

*Then with probability $1 - T^+\delta_{\mathbb{P},\xi} - \delta_{\mathbb{P}}$,*

$$\sup_{t \leq T^+} v_{p,\pi(p)}^2(t) \leq \frac{1}{\sqrt{d}} \quad and \quad (2/\Delta)^{\frac{1}{I-1}} \cdot \bar{v}_{p,\pi(p)}^2(0) \leq \bar{v}_{p,\pi(p)}^2(T^+).$$

*Proof.* We will apply Lemma D.4 to the process with $X_t = \bar{v}_{p,\pi(p)}^2(t)$, $\alpha = 4I\hat{\sigma}_{2I}^2 \eta a_{\pi(p)}$, $\varepsilon = \frac{\Delta}{256I}$. By Lemma D.23, the process $(x_t^+)_t$ satisfies

$$x_t^+ \leq \frac{(1+\varepsilon)\bar{v}_{p,\pi(p)}^2(0)}{\left(1 - 4I(I-1)\hat{\sigma}_{2I}^2 \eta a_{\pi(p)}(1+\varepsilon)^{I-1}\bar{v}_{p,\pi(p)}^{2I-2}(0) \cdot t\right)^{\frac{1}{I-1}}}$$

Therefore for

$$t \leq T^+ \leq \frac{1 - I\varepsilon}{4I(I-1)\hat{\sigma}_{2I}^2 \eta a_{\pi(p)} \bar{v}_{p,\pi(p)}^{2I-2}(0)},$$

we have

$$\begin{aligned}
(I-1)\alpha \left(\hat{x}_0^+\right)^{I-1} \cdot t &= 4I(I-1)\hat{\sigma}_{2I}^2 \eta a_{\pi(p)}(1+\varepsilon)^{I-1}\bar{v}_{p,\pi(p)}^{2I-2}(0) \cdot t \\
&\leq (1+\varepsilon)^{I-1}(1 - I\varepsilon) \\
&\leq \exp(-\varepsilon) \\
&\leq 1 - \varepsilon/2.
\end{aligned}$$

Altogether, we can upper bound $\hat{x}_t^+$ as

$$x_t^+ \leq \frac{(1+\varepsilon)\bar{v}_{p,\pi(p)}^2(0)}{(\varepsilon/2)^{\frac{1}{I-1}}} \leq 4\varepsilon^{-1}\bar{v}_{p,\pi(p)}^2(0) \leq \frac{1}{\sqrt{d}},$$

as long as $\frac{d}{\log^4 d} \geq 2^{20}I^2\Delta^{-2}$. As such, if $X_t \leq x_t^+$ at time $t$, then the update in Lemma D.3 holds for $\delta_v = 1/\sqrt{d}$. This update is indeed of the form (8); we must now verify that the conditions on $\sigma_Z^2, \Xi_1, \Xi_2, \Xi_3$ indeed hold. Recall that

$$1 - (I-1)\alpha \left(x_0^+\right)^{I-1} T \geq \varepsilon/2 = \frac{\Delta}{512I}.$$

We therefore have that

$$\begin{aligned}
\sum_{t=0}^{T-1} x_t^+ &\leq \int_0^T \frac{x_0^+}{\left(1 - \alpha(I-1)\left(x_0^+\right)^{I-1}t\right)^{\frac{1}{I-1}}} dt \\
&\leq \begin{cases} \alpha^{-1}\log\left(\frac{1}{1-\alpha x_0^+ T}\right) & I = 2 \\ \frac{1}{(I-2)\alpha\left(x_0^+\right)^{I-2}}\left[1 - \left(1 - \alpha(I-1)\left(x_0^+\right)^{I-1}T^+\right)^{\frac{I-2}{I-1}}\right] & I > 2 \end{cases} \\
&\leq \begin{cases} \alpha^{-1}\log(512I/\Delta) & I = 2 \\ (I-2)^{-1}\alpha^{-1}(x_0^+)^{2-I} & I > 2 \end{cases}.
\end{aligned}$$

and

$$\sum_{t=0}^{T-1} (x_t^+)^I \leq \int_0^T \frac{\left(x_0^+\right)^I}{\left(1 - \alpha(I-1)\left(x_0^+\right)^{I-1}t\right)^{\frac{I}{I-1}}} dt$$

$$= x_0^+ \alpha^{-1} \left( \frac{1}{\left( 1 - \alpha(I-1)\left(x_0^+\right)^{I-1} T \right)^{\frac{1}{I-1}}} - 1 \right)$$

$$\leq x_0^+ \alpha^{-1} (\varepsilon/2)^{-\frac{1}{I-1}}.$$

The condition on $\sigma_Z^2$ is

$$\sigma_Z^2 \leq \frac{x_0^2 \varepsilon^2 \delta_{\mathbb{P}}}{4 \sum_{t=0}^{T-1} x_t^+} \Longleftarrow \sigma_Z^2 \lesssim x_0^I \Delta^2 I^{-2} \delta_{\mathbb{P}} \alpha \cdot \left( \frac{1}{\log(512I/\Delta)} \vee (I-2) \right)$$

Since $\sigma_Z^2 \lesssim \eta^2 \|a\|_1^2$, this is satisfied if we take

$$\eta \lesssim \frac{a_{\pi(p)} \hat{\sigma}_{2I}^2 d^{-I} \|a\|_1^{-2} \Delta^2 \delta_{\mathbb{P}}}{\log(512I/\Delta)}.$$

Next, observe that $\Xi_1 \lesssim \delta_T \cdot \eta a_{\pi(p)} I \hat{\sigma}_{2I}^2$. We observe that

$$\frac{\varepsilon x_0}{6 \sum_{t=0}^{T-1} x_t^{+I}} \gtrsim \frac{\varepsilon^{\frac{I}{I-1}} x_0 \alpha}{x_0^+} \gtrsim \Delta^{\frac{I}{I-1}} I^{-\frac{I}{I-1}} \cdot \eta a_{\pi(p)} I \hat{\sigma}_{2I}^2 \gg \Xi_1,$$

and thus the condition on $\Xi_1$ is satisfied since $\delta_T = \frac{\Delta^2}{CI^2}$ for a sufficiently large constant $C$. Next, we see that $\Xi_2 = \eta^2 d \|a\|_1^2 \log^{\tilde{Q}} \left( \frac{md}{\delta_{\mathbb{P}, \xi}} \right)$, and thus we require

$$\Xi_2 \leq \frac{\varepsilon x_0}{6 \sum_{t=1}^{T} \hat{x}_t} \Longleftarrow \Xi_2 \lesssim \frac{\Delta I^{-1} x_0^{I-1} \alpha}{\log(512I/\Delta)}$$

$$\Longleftarrow \eta^2 d \|a\|_1^2 \log^{\tilde{Q}} \left( \frac{md}{\delta_{\mathbb{P}, \xi}} \right) \ll \frac{\Delta d^{-(I-1)} \eta a_{\pi(p)} \hat{\sigma}_{2I}^2}{\log(512I/\Delta)}$$

$$\Longleftarrow \eta \ll \frac{a_{\pi(p)} \hat{\sigma}_{2I}^2 d^{-I} \|a\|_1^{-2} \Delta}{\log(512I/\Delta) \log^{\tilde{Q}} \left( \frac{md}{\delta_{\mathbb{P}, \xi}} \right)},$$

which is indeed satisfied from our choice of $\eta$. Finally, we see that $\Xi_3 = \eta^2 \|a\|_1^2 \log^{\tilde{Q}} \left( \frac{md}{\delta_{\mathbb{P}, \xi}} \right)$, and thus we require

$$\Xi_3 \leq \frac{\varepsilon x_0}{6T} \Longleftarrow \eta^2 \|a\|_1^2 \log^{\tilde{Q}} \left( \frac{md}{\delta_{\mathbb{P}, \xi}} \right) \lesssim \Delta(I-1) \hat{\sigma}_{2I}^2 \eta a_{\pi(p)} x_0^I$$

$$\Longleftarrow \eta \ll \frac{a_{\pi(p)} (I-1) \hat{\sigma}_{2I}^2 d^{-I} \|a\|_1^{-2} \Delta}{\log^{\tilde{Q}} \left( \frac{md}{\delta_{\mathbb{P}, \xi}} \right)}.$$

which is again satisfied by our choice of $\eta$. Therefore the conditions of Lemma D.4 are satisfied, and so with probability $1 - T^+ \delta_{\mathbb{P}, \xi} - \delta_{\mathbb{P}}$ we have $X_t \in [x_t^-, x_t^+]$ for all $t \leq T^+$.

We conclude by lower bounding $x_t^-$. By Lemma D.24,

$$x_t^- \geq \frac{x_0^-}{\left( 1 - \alpha(I-1) \exp(-\alpha I) \left(x_0^-\right)^{I-1} t \right)^{\frac{1}{I-1}}}.$$

Plugging in $\alpha = 4I \hat{\sigma}_{2I}^2 \eta a_{p, \pi(p)} \leq \varepsilon$, we see that

$$\alpha(I-1) \exp(-\alpha I) \left(x_0^-\right)^{I-1} T^+ \geq \exp(-\alpha I) \left( \frac{x_0^-}{x_0} \right)^{I-1} \geq \exp(-\alpha I)(1-\varepsilon)^I \geq 1 - 2I\varepsilon,$$

and therefore

$$x_{T^+}^- \geq \frac{(1-\varepsilon)x_0}{(2I\varepsilon)^{\frac{1}{I-1}}} \geq x_0 \cdot \frac{\exp(-\Delta/(128I))}{(\Delta/128)^{\frac{1}{I-1}}} \geq (64/\Delta)^{\frac{1}{I-1}} x_0,$$

as desired.

$\square$

Next, we bound the time that $\bar{v}_{p,\pi(p)}^2(t)$ grows to $1/3$. We first introduce the following helper lemma, with proof deferred to Appendix D.5.

**Lemma D.6.** *Let* $(X_t)_t$ *satisfy*

$$X_{t+1} \geq X_t + \alpha X_t^I + \xi_{t+1} + Z_{t+1}, \quad X_0 > x_0.$$

*where* $(\xi_t)_t$ *is an adapted process and* $(Z_t)_t$ *is a martingale difference sequence. Define the process* $\hat{x}_t$ *by*

$$\hat{x}_{t+1} = (1 + \alpha \hat{x}_t^{I-1}) \hat{x}_t, \quad \hat{x}_0 = x_0/2.$$

*Suppose that when* $\hat{x}_t \leq X_t \leq \delta$, *we have* $|\xi_{t+1}| \leq \Xi$ *with probability* $1 - \delta_{\mathbb{P},\xi}$ *and* $\mathbb{E}[Z_{t+1} \mid \mathcal{F}_t] \leq \sigma_Z^2$. *Then if*

$$\Xi \leq \frac{x_0}{4T}, \quad and \quad \sigma_Z^2 \leq \frac{x_0^2 \delta_{\mathbb{P}}}{16T},$$

*we with probability* $1 - T\delta_{\mathbb{P},\xi} - \delta_{\mathbb{P}}$ *either have* $X_t \geq \hat{x}_t$ *for all* $t \leq T$, *or* $\sup_{t \leq T} X_t > \delta$.

The following lemma bounds the time it takes for $\bar{v}_{p,\pi(p)}(t)$ to grow slightly.

**Lemma D.7** (Intermediate growth)**.** *Let* $\delta > 1$. *Assume that for some* $T_{\delta/d}$, $\bar{v}_{p,\pi(p)}^2(T_{\delta/d}) \geq \delta/d$. *Assume that the learning rate* $\eta$ *satisfies*

$$\eta \ll \frac{a_{\pi(p)} I(I-1) \hat{\sigma}_{2I}^2 d^{-I} \|a\|_1^{-2} \delta_{\mathbb{P}}}{\log^{\tilde{Q}}\left(\frac{md}{\delta_{\mathbb{P},\xi}}\right)}.$$

*Moreover, assume that Induction Hypothesis C.2 and the same conditions as Lemma C.3 hold. Then, with probability* $1 - T_\delta^* \delta_{\mathbb{P},\xi} - \delta_P$, *there exists some* $t \leq \frac{d^{I-1}}{2I(I-1)\hat{\sigma}_{2I}^2 \eta a_{\pi(p)} \delta^{I-1}} =: T_\delta^*$ *such that*

$$\bar{v}_{p,\pi(p)}^2(T_{\delta/d} + t) > \min\left(\frac{\delta^I}{d}, \frac{1}{3}\right)$$

*Proof.* Define $X_t = \bar{v}_{p,\pi(p)}^2(T_{\delta/d} + t)$, so that $X_0 \geq \delta/d =: x_0$. For notational convenience, let us define $\bar{\delta} := \min(\delta^I/d, \frac{1}{3})$. Let $T$ be the last time at which $\hat{x}_t \leq \bar{\delta}$. For $t \leq T$, if $X_t \leq \bar{\delta}$, then by Lemma C.2 and Lemma D.1, we have

$$\bar{v}_{p,\pi(p)}^2(t+1) \geq \bar{v}_{p,\pi(p)}^2(t) + 2\eta a_{\pi(p)} I \hat{\sigma}_{2I}^2 \bar{v}_{p,\pi(p)}^{2I}(t) + Z(t+1) + \xi(t+1),$$

where $\mathbb{E}[Z(t+1) \mid \mathcal{F}_t] \lesssim \bar{\delta}\eta^2 \|a\|_1^2$ and $|\xi(t+1)| \lesssim \eta^2 d\bar{\delta} \|a\|_1^2 \log^{\tilde{Q}}\left(\frac{md}{\delta_{\mathbb{P},\xi}}\right)$. We would like to apply Lemma D.6 with $\alpha = 2\eta a_{\pi(p)} I \hat{\sigma}_{2I}^2$.

By Lemma D.24,

$$\bar{\delta} \geq \hat{x}_T \geq \frac{\hat{x}_0}{\left(1 - \alpha(I-1)\exp(-\alpha I)\hat{x}_0^{I-1}T\right)^{\frac{1}{I-1}}},$$

and thus

$$T \leq \frac{\exp(\alpha I)}{\alpha(I-1)\hat{x}_0^{I-1}} \leq \frac{d^{I-1}}{2I(I-1)\hat{\sigma}_{2I}^2 \eta a_{\pi(p)} \delta^{I-1}}.$$

We next verify the conditions of the lemma. We first require $\sigma_Z^2 \leq \frac{x_0^2 \delta_{\mathbb{P}}}{16T}$, or equivalently

$$\eta^2 \|a\|_1^2 \bar{\delta} \lesssim \frac{\delta^2 \delta_{\mathbb{P}}}{d^2 T} \Longleftarrow \eta^2 \bar{\delta} \lesssim d^{-(I+1)} \delta^{I+1} \|a\|_1^{-2} \delta_{\mathbb{P}} \cdot I(I-1)\hat{\sigma}_{2I}^2 \eta a_{\pi(p)}$$

$$\Longleftarrow \eta \lesssim \bar{\delta}^{-1} d^{-(I+1)} \delta^{I+1} \|a\|_1^{-2} \delta_{\mathbb{P}} \cdot I(I-1)\hat{\sigma}_{2I}^2 \eta a_{\pi(p)}$$

$$\Longleftarrow \eta \lesssim a_{\pi(p)} I(I-1)\hat{\sigma}_{2I}^2 d^{-I} \delta \|a\|_1^{-2} \delta_{\mathbb{P}}$$

We additionally require $\Xi \leq \frac{x_0}{4T}$. Plugging in $\Xi, x_0, T$, it suffices to take

$$\eta^2 d \|a\|_1^2 \bar{\delta} \log^{\tilde{Q}}\left(\frac{md}{\delta_{\mathbb{P},\xi}}\right) \ll \delta^I d^{-I} I(I-1)\hat{\sigma}_{2I}^2 \eta a_{\pi(p)}$$

$$\Longleftarrow \eta \ll \frac{a_{\pi(p)} I(I-1)\hat{\sigma}_{2I}^2 d^{-I} \|a\|_1^{-2}}{\log^{\tilde{Q}}\left(\frac{md}{\delta_{\mathbb{P},\xi}}\right)},$$

where we have used the fact that $\bar{\delta} \leq \delta^I/d$. Therefore by Lemma D.6, with high probability we have $X_t \geq \hat{x}_t$ for all $t \leq T$. But this implies that we actually must have $X_t > \bar{\delta}$ for some $t \leq T$, as desired. $\qquad\square$

Putting everything together, we can now bound the total time it takes for $\bar{v}_{p,\pi(p)}^2(t)$ to reach $1/3$.

**Lemma D.8.** *Assume that the conditions of Lemma D.5 hold. Then, with high probability, there exists some $t \leq T = \frac{1+\Delta/16}{4I(I-1)\hat{\sigma}_{2I}^2 \eta a_{\pi(p)} \bar{v}_{p,\pi(p)}^{2I-2}(0)}$ such that $\bar{v}_{p,\pi(p)}^2(t) \geq \frac{1}{3}$.*

*Proof.* On the event that Lemma D.5 holds, at time $T^+$, we have the bound

$$\bar{v}_{p,\pi(p)}^2(T^+) \geq (64/\Delta)^{\frac{1}{I-1}} \bar{v}_{p,\pi(p)}^2(0) =: \delta_0/d,$$

for $\delta_0 := (64/\Delta)^{\frac{1}{I-1}} d \bar{v}_{p,\pi(p)}^2(0)$. By Lemma D.7, with probability $1 - T_\delta^* \delta_{\mathbb{P},\xi} - \delta_{\mathbb{P}}$, $v_{p,\pi(p)}^2(t)$ grows to a value of $\delta_0^I/d$ in time $t \leq \frac{d^{I-1}}{2I(I-1)\hat{\sigma}_{2I}^2 \eta a_{\pi(p)} \delta_0^{I-1}}$. Repeatedly applying this lemma for at most $\log\log d$ iterations we get that $v_{p,\pi(p)}^2(t)$ grows to be at least $\frac{1}{3}$ in time

$$\sum_{k=0}^{\infty} \frac{d^{I-1}}{2I(I-1)\hat{\sigma}_{2I}^2 \eta a_{\pi(p)} \delta_0^{(I-1)I^k}} = \frac{d^{I-1}}{2I(I-1)\hat{\sigma}_{2I}^2 \eta a_{\pi(p)}} \sum_{k=0}^{\infty} \delta_0^{-(I-1)I^k}$$

$$\leq \frac{d^{I-1}}{I(I-1)\hat{\sigma}_{2I}^2 \eta a_{\pi(p)} \delta_0^{I-1}}$$

$$= \frac{\Delta/64}{I(I-1)\hat{\sigma}_{2I}^2 \eta a_{\pi(p)} \bar{v}_{p,\pi(p)}^{2I-2}(0)}$$

$$\leq \frac{\Delta/16}{4I(I-1)\hat{\sigma}_{2I}^2 \eta a_{\pi(p)} \bar{v}_{p,\pi(p)}^{2I-2}(0)}$$

with total failure probability at most $T\delta_{\mathbb{P},\xi} + \delta_{\mathbb{P}} \log\log d$. $\qquad\square$

Finally, we can lower bound the time it takes for $\bar{v}_{p,\pi(p)}^2$ to grow from $\frac{1}{2}$ to $1 - \varepsilon_D$. The proof of the following is deferred to Appendix D.5.

**Lemma D.9.** *Let $(X_t)_t \geq 0$ satisfy*

$$X_{t+1} \leq (1-\alpha)X_t + \xi_{t+1} + Z_{t+1}, \quad X_0 = x_0$$

*where $(\xi_t)_t$ is an adapted process and $(Z_t)_t$ is a martingale difference sequence, and with probability $1 - \delta_{\mathbb{P},\xi}$ we have $|\xi_{t+1}| \leq \Xi$ and $\mathbb{E}[Z_{t+1} \mid \mathcal{F}_t] \leq \sigma_Z^2$ when $X_t \leq 1.5x_0$. Then, if*

$$\Xi \leq \frac{\varepsilon\alpha}{4}, \quad \sigma_Z^2 \leq \frac{\varepsilon^2 \alpha \delta_{\mathbb{P}}}{16}$$

*we have with probability $1 - T\delta_{\mathbb{P},\xi} - \delta_{\mathbb{P}}$.*

$$X_t \leq (1-\alpha)^t x_0 + \varepsilon/2 \leq 1.5x_0$$

*for all $t \leq T$.*

**Lemma D.10** (Strong Recovery). *Let us assume that Lemma D.8 holds, i.e for some time $T_{1/3}$, $\bar{v}^2_{p,\pi(p)}(T_{1/3}) \geq \frac{1}{3}$. Let the target accuracy $\varepsilon_D$ satisfy the same condition as in Lemma C.5. Choose $\eta$ so that*

$$\eta \ll \frac{a_{\pi(p)} I \hat{\sigma}^2_{2I} 3^{-I} \|a\|_1^{-2} \delta_{\mathbb{P}}}{\log^{\tilde{Q}}\left(\frac{md}{\delta_{\mathbb{P},\xi}}\right)} \min(d^{-1}\varepsilon_D, \varepsilon_D^2)$$

*Then with probability $1 - T\delta_{\mathbb{P},\xi} - \delta_{\mathbb{P}}$, we have*

$$\bar{v}^2_{p,\pi(p)}(t) \geq 1 - \varepsilon_D, \quad \forall \frac{3^I}{I\hat{\sigma}^2_{2I}\eta a_{\pi(p)}} \log(2/\varepsilon_D) \leq t \leq T.$$

*Proof.* By Lemma D.1 and Lemma C.5, when $\bar{v}^2_{p,\pi(p)}(t) \geq \frac{1}{3}$ we have

$$\bar{v}^2_{p,\pi(p)}(t+1) \geq \bar{v}^2_{p,\pi(p)}(t) + 3^{-I} I \hat{\sigma}^2_{2I} \eta a_{\pi(p)} (1 - \bar{v}^2_{p,\pi(p)}(t)) + \xi_{t+1} + Z_{t+1}$$

where

$$|\xi_{t+1}| \lesssim \eta^2 \|a\|_1^2 d \log^{\tilde{Q}}\left(\frac{md}{\delta_{\mathbb{P},\xi}}\right), \quad \mathbb{E}[Z_{t+1}^2 \mid \mathcal{F}_t] \lesssim \eta^2 \|a\|_1^2.$$

We would like to apply Lemma D.9, with $\alpha = 3^{-I} I \hat{\sigma}^2_{2I} \eta a_{\pi(p)}$ and $X_t = 1 - \bar{v}^2_{p,\pi(p)}(T_{1/2}+t)$, $\varepsilon = \varepsilon_D$. We first require $\Xi \leq \frac{\varepsilon\alpha}{4}$, which is satisfied by taking

$$\eta^2 \|a\|_1^2 d \log^{\tilde{Q}}\left(\frac{md}{\delta_{\mathbb{P},\xi}}\right) \lesssim 3^{-I} I \hat{\sigma}^2_{2I} \eta a_{\pi(p)} \varepsilon$$

$$\Longleftarrow \eta \ll \frac{a_{\pi(p)} I \hat{\sigma}^2_{2I} 3^{-I} d^{-1} \|a\|_1^{-2} \varepsilon}{\log^{\tilde{Q}}\left(\frac{md}{\delta_{\mathbb{P},\xi}}\right)}.$$

Next, we require $\sigma_Z^2 \leq \varepsilon^2 \alpha \delta_{\mathbb{P}}/16$, which is obtained by taking

$$\eta^2 \|a\|_1^2 \lesssim \varepsilon^2 3^{-I} I \hat{\sigma}^2_{2I} \eta a_{\pi(p)} \delta_{\mathbb{P}} \Longleftarrow \eta \lesssim a_{\pi(p)} I \hat{\sigma}^2_{2I} 3^{-I} \|a\|_1^{-2} \varepsilon^2 \delta_{\mathbb{P}}.$$

Altogether, with high probability,

$$1 - \bar{v}^2_{p,\pi(p)}(T_{1/2}+t) \leq (1-\alpha)^t \cdot \frac{1}{2} + \varepsilon/2 \leq \varepsilon$$

for $t \geq \alpha^{-1} \log(2/\varepsilon) = \frac{3^I}{I\hat{\sigma}^2_{2I}\eta a_{\pi(p)}} \log(2/\varepsilon)$. $\qquad\square$

*Proof of Theorem D.2.* This follows directly from combining Lemma D.5, Lemma D.8, and Lemma D.10, and noting that

$$\frac{3^I}{I\hat{\sigma}^2_{2I}\eta a_{\pi(p)}} \leq \frac{\Delta/16}{4I(I-1)\hat{\sigma}^2_{2I}\eta a_{\pi(p)} \bar{v}^{2I-2}_{p,\pi(p)}(0)}.$$

$\qquad\square$

### D.2.2 Radial Dynamics

In this subsection, we analyze the dynamics of $\|v_P\|^2$, when $\bar{v}^2_{p,\pi(p)}(t) \geq 1 - \bar{\varepsilon}$. In this regime, the update on the norm is given by the following.

**Lemma D.11.** *Assume that $\bar{v}^2_{p,\pi(p)}(t) \geq 1 - \bar{\varepsilon}$. Then*

$$\|v_p(t+1)\|^2 = \|v_p(t)\|^2 + 4\eta \|v_p(t)\|^2 \left(a_{\pi(p)} - \|v_p(t)\|^2\right) + Z_{t+1} + \xi_{t+1}$$

*where with probability $1 - \delta_{\mathbb{P},\xi}$*

$$\mathbb{E}[Z_{t+1}^2 \mid \mathcal{F}_t] \lesssim \eta^2 \|a\|_1^2 \|v_p(t)\|^4$$

$$|\xi_{t+1}| \lesssim \left(\eta^2 d \|a\|_1^2 \log^{\tilde{Q}}(md/\delta_{\mathbb{P},\xi}) + \eta(C_\sigma^2 a_{\pi(p)} \bar{\varepsilon} + \|a\|_1 2^{2I} \varepsilon_0^I + m\sigma_1^2)\right) \|v_p(t)\|^2.$$

*Proof.* This follows directly from Lemma C.7 and Lemma D.1. $\qquad\square$

We would like to prove that Inductive Hypothesis D.1(b) holds, assuming that D.1(a) holds. This is given by the following result.

**Lemma D.12.** *Assume that Inductive Hypothesis C.2 and Inductive Hypothesis D.1(a) hold. Let* $T_{1-\bar{\varepsilon}} \leq \frac{1+\Delta/8}{4I(I-1)\hat{\sigma}_{2I}^2 \eta a_{\pi(p)} v_{p,\pi(p)}^{2I-2}(0)}$ *be some time at which* $\bar{v}_{p,\pi(p)}^2 \geq 1 - \bar{\varepsilon}$. *Let the learning rate* $\eta$ *and target accuracy* $\varepsilon_R$ *satisfy*

$$\eta \lesssim \frac{\|a\|_1^{-2}}{\log(2a_k/\sigma_0^2)} \min\left(\frac{a_{\min_*}d^{-1}\varepsilon_R}{\log^{\tilde{Q}}(md/\delta_{\mathbb{P},\xi})}, \varepsilon_R^2\delta_{\mathbb{P}}\right), \quad \varepsilon_R \gtrsim \log(2a_k/\sigma_0^2)\left(C_\sigma^2 a_{\pi(p)}\bar{\varepsilon} + \|a\|_1 2^{2I}\varepsilon_0^I + m\sigma_1^2\right),$$

*Then, with probability* $1 - T_{max}\delta_{\mathbb{P},\xi} - \delta_{\mathbb{P}}$,

$$\left|\|v_p(t)\|^2 - a_k\right| \leq \varepsilon_R, \quad \forall\ T_{1-\bar{\varepsilon}} + \frac{\Delta/8}{4I(I-1)\hat{\sigma}_{2I}^2 \eta a_{\pi(p)}\bar{v}_{p,\pi(p)}^{2I-2}(0)} \leq t \leq T_{1-\bar{\varepsilon}} + T_{max}.$$

To prove this lemma, we first lower bound the time it takes for $\|v_p(t)\|^2$ to reach $\delta a_{\pi(p)}$ for some small quantity $\delta a_p$. We start by proving the following helper lemma, which resembles Lemma F.6 from [RL24] and whose proof is deferred to Appendix D.5.

**Lemma D.13.** *Let* $(X_t)_t$ *satisfy*

$$X_{t+1} = (1+\alpha)X_t + \xi_{t+1} + Z_{t+1}, \quad X_0 = x_0 > 0,$$

*where* $(\xi_t)_t$ *is an adapted process and* $(Z_t)_t$ *is a martingale difference sequence. Define* $x_t = (1+\alpha)^t x_0$. *Suppose that if* $X_t = (1 \pm 0.5)x_t$, *then* $|\xi_{t+1}| \leq x_t \Xi$ *with probability* $1 - \delta_{\mathbb{P},\xi}$ *and* $\mathbb{E}[Z_{t+1}^2 \mid \mathcal{F}_t] \leq x_t^2\sigma_Z^2$. *Then, if*

$$\Xi \leq \frac{1}{4T}, \quad \sigma_Z^2 \leq \frac{\delta_{\mathbb{P}}}{16T}.$$

*then we have with probability* $1 - T\delta_{\mathbb{P},\xi} - \delta_{\mathbb{P}}$ *that* $X_t = (1 \pm 0.5)x_t$ *for all* $t \leq T$.

The following lemma then lower bounds the escape time.

**Lemma D.14.** *Let* $\delta = \frac{1}{C\log(2a_{\pi(p)}/\sigma_0^2)}$, *for sufficiently large constant* $C$. *Define* $T = \frac{\log(2\delta a_{\pi(p)}/\sigma_0^2)}{4\eta a_{\pi(p)}} \leq$ $\frac{\delta^{-1}}{4C\eta a_{\pi(p)}}$. *Let the learning rate satisfy* $\eta \lesssim \frac{a_{\pi(p)}d^{-1}\|a\|_1^{-2}\delta_{\mathbb{P}}\delta}{\log^{\tilde{Q}}(md/\delta_{\mathbb{P},\xi})}$. *With probability* $1 - \delta_{\mathbb{P},\xi} - T\delta_{\mathbb{P}}$, *we have* $\sup_{t \leq T}\|v_p(T_{1-\bar{\varepsilon}} + t)\|^2 \geq \delta a_{\pi(p)}$.

*Proof.* When $\|v_p(t)\|^2 \leq \delta a_{\pi(p)}$, we can bound

$$\|v_p(t+1)\|^2 = \|v_p(t)\|^2 + 4\eta a_{\pi(p)}\|v_p(t)\|^2 + Z_{t+1} + \xi_{t+1},$$

where

$$|\xi_{t+1}| \lesssim \eta\delta a_{\pi(p)}\|v_p(t)\|^2, \quad \mathbb{E}[Z_{t+1}^2 \mid \mathcal{F}_t] \lesssim \eta^2\|a\|_1^2\|v_p(t)\|^4,$$

provided that

$$\delta \gtrsim a_{\min_*}^{-1}\left(\eta d\|a\|_1^2\log^{\tilde{Q}}(md/\delta_{\mathbb{P},\xi}) + C_\sigma^2 a_{\pi(p)}\bar{\varepsilon} + \|a\|_1 2^{2I}\varepsilon_0^I + m\sigma_1^2\right)$$

$$\Longleftarrow \eta \lesssim \frac{a_{\min_*}d^{-1}\delta\|a\|_1^{-2}}{\log^{\tilde{Q}}(md/\delta_{\mathbb{P},\xi})}, \quad \delta^{-1}\left(C_\sigma^2 a_{\pi(p)}\bar{\varepsilon} + \|a\|_1 2^{2I}\varepsilon_0^I + m\sigma_1^2\right) \lesssim 1.$$

Define the process $X_t = \|v_p(T_{1-\bar{\varepsilon}} + t)\|^2$, where $x_0 = \|v_p(T_{1-\bar{\varepsilon}})\|^2$ and $\alpha = 4\eta a_{\pi(p)}$. Assume that $\sup_{t \leq T} X_t < \delta a_{\pi(p)}$. We can thus apply Lemma D.13, since the conditions on $\sigma_Z^2, \Xi$ are indeed met:

$$\sigma_Z^2 \leq \frac{\delta_{\mathbb{P}}}{16T} \Longleftarrow \eta \ll a_{\pi(p)}\|a\|_1^{-2}\delta_{\mathbb{P}}\delta$$

$$\Xi \leq \frac{1}{4T} \Longleftarrow 1 \ll C.$$

But recall that for the process $x_t = (1+\alpha)^t x_0$, for $T = \alpha^{-1}\log(2\delta a_{\pi(p)}/x_0)$ we have $x_T \geq 2\delta a_{\pi(p)}$ and thus $X_T > \delta a_{\pi(p)}$, a contradiction. Therefore there exists $t \leq T$ such that $X_t \geq \delta a_{\pi(p)}$, as desired. $\qquad\square$

We next introduce the following helper lemma, with proof deferred to Appendix D.5.

**Lemma D.15.** *Let* $(X_t)_t$ *satisfy*

$$X_{t+1} = (1 - \alpha(X_t))X_t + \xi_{t+1} + Z_{t+1},$$

*where* $(\xi_t)_t$ *is an adapted process and* $(Z_t)_t$ *is a martingale difference sequence, and with probability* $1 - \delta_{\mathbb{P},\xi}$ *we have* $\alpha(X_t) \in [\alpha_-, \alpha_+]$, $|\xi_{t+1}| \leq \Xi$ *and* $\mathbb{E}[Z_{t+1}^2 \mid \mathcal{F}_t] \leq \sigma_Z^2$ *when* $X_t \in [-\varepsilon/2, x_0 + \varepsilon/2]$. *Then, if for some* $\varepsilon \in (0, x_0)$

$$\Xi \leq \frac{\varepsilon\alpha_-}{4}, \quad \sigma_Z^2 \leq \frac{\varepsilon^2\alpha_-\delta_{\mathbb{P}}}{16},$$

*we have with probability* $1 - T\delta_{\mathbb{P},\xi} - \delta_{\mathbb{P}}$ *that*

$$(1 - \alpha_+)^t x_0 - \varepsilon/2 \leq X_t \leq (1 - \alpha_-)^t x_0 + \varepsilon/2$$

*for all* $t \leq T$.

The following lemma bounds the time it takes for the norm to grow from $\delta a_{\pi(k)}$ to approximately $a_{\pi(k)}$, and furthermore establishes that it stays close to $a_{\pi(k)}$

**Lemma D.16.** *Inductively assume that Induction Hypothesis C.2 and Induction Hypothesis D.1(a) are true. Pick* $\delta > 0$, *and let* $T^*$ *be some time at which* $\|v_k(T^*)\|^2 \in [\delta a_k, a_k/2]$. *Let* $\varepsilon_R > 0$ *be the target accuracy. If*

$$\eta \lesssim \varepsilon_R^2 \|a\|_1^{-2} \delta_{\mathbb{P}}\delta \wedge \frac{\varepsilon_R d^{-1} \|a\|_1^{-2}\delta}{\log^{\tilde{Q}}(md/\delta_{\mathbb{P},\xi})}, \quad \varepsilon_R \gtrsim \delta^{-1}\left(C_\sigma^2 a_{\pi(p)}\bar{\varepsilon} + \|a\|_1 2^{2I}\varepsilon_0^I + m\sigma_1^2\right),$$

*then we have with probability* $1 - T_{max}\delta_{\mathbb{P},\xi} - \delta_{\mathbb{P}}$ *that*

$$\left\|v_p(t)\right\|^2 \in [a_{\pi(p)} - \varepsilon_R, a_{\pi(p)} + \varepsilon_R] \ \forall \ T^* + \frac{2\log(a_{\pi(p)}/\varepsilon_R)}{\delta\eta a_{\pi(p)}} \leq t \leq T^* + T_{max}.$$

*Proof.* Assume that the inductive hypothesis holds at time $t$. By Lemma D.11, we have that

$$\left\|v_p(t+1)\right\|^2 = \left\|v_p(t)\right\|^2 + 4\eta\left\|v_p(t)\right\|^2(a_{\pi(p)} - \left\|v_p(t)\right\|^2) + Z_{t+1} + \xi_{t+1}$$

for $\mathbb{E}[Z_{t+1}^2 \mid \mathcal{F}_t] \lesssim \eta^2 \|a\|_1^2 \|v_p(t)\|^4 \lesssim \eta^2 \|a\|_1^2 a_{\pi(p)}^2$ and

$$|\xi_{t+1}| \lesssim \left(\eta^2 d \|a\|_1^2 \log^{\tilde{Q}}(md/\delta_{\mathbb{P},\xi}) + \eta(C_\sigma^2 a_{\pi(p)}\bar{\varepsilon} + \|a\|_1 2^{2I}\varepsilon_0^I + m\sigma_1^2)\right) a_{\pi(p)}.$$

Therefore

$$a_{\pi(p)} - \left\|v_p(t+1)\right\|^2 = \left(1 - 4\eta\left\|v_p(t)\right\|^2\right)\left(a_{\pi(p)} - \left\|v_p(t)\right\|^2\right) + Z_{t+1} + \xi_{t+1}.$$

We thus would like to apply Lemma D.15 to the process $X_t = a_{\pi(p)} - \left\|v_p(t+T^*)\right\|^2$, with $\varepsilon = \varepsilon_R$. We see that $x_0 \in [a_{\pi(p)}/2, (1-\delta)a_{\pi(p)}]$, so for $X_t \in [-\varepsilon_R/2, (1-\delta/2)a_{\pi(p)}]$ we can bound

$$\frac{\delta a_{\pi(p)}}{2} \leq \left\|v_p(t)\right\|^2 \leq 2a_{\pi(p)}.$$

Therefore the conditions of Lemma D.15 are indeed satisfied. It thus suffices to take

$$\Xi \leq \frac{\varepsilon\alpha_-}{4} \Longleftarrow \left(\eta^2 d \|a\|_1^2 \log^{\tilde{Q}}(md/\delta_{\mathbb{P},\xi}) + \eta(C_\sigma^2 a_{\pi(p)}\bar{\varepsilon} + \|a\|_1 2^{2I}\varepsilon_0^I + m\sigma_1^2)\right) a_{\pi(p)} \lesssim \eta\delta a_{\pi(p)}\varepsilon_R$$

$$\Longleftarrow \eta \lesssim \frac{\varepsilon_R d^{-1} \|a\|_1^{-2}\delta}{\log^{\tilde{Q}}(md/\delta_{\mathbb{P},\xi})}, \quad \varepsilon_R \gtrsim \delta^{-1}\left(C_\sigma^2 a_{\pi(p)}\bar{\varepsilon} + \|a\|_1 2^{2I}\varepsilon_0^I + m\sigma_1^2\right).$$

as well as

$$\sigma_Z^2 \leq \frac{\varepsilon_R^2\alpha_-\delta_{\mathbb{P}}}{16} \Longleftarrow \eta^2 \|a\|_1^2 a_{\pi(p)}^2 \lesssim \varepsilon_R^2 \eta a_{\pi(p)}\delta\delta_{\mathbb{P}}$$

$$\Longleftarrow \eta \lesssim \frac{\varepsilon_R^2 \|a\|_1^{-2}\delta_{\mathbb{P}}\delta}{a_{\pi(p)}}.$$

Altogether, by Lemma D.15 with high probability we have

$$(1 - \alpha_+)^t x_0 - \varepsilon_R/2 \leq X_t \leq (1 - \alpha_-)^t x_0 + \varepsilon_R/2$$

Naively, we have the bound $X_t \geq -\varepsilon_R/2$, which implies $\left\|v_p(t)\right\|^2 \leq a_{\pi(p)} + \varepsilon_R/2$. Moreover, for $t \geq \frac{2\log(a_p/\varepsilon_R)}{\delta\eta a_{\pi(p)}} \geq \alpha_-^{-1}\log(2x_0/\varepsilon_R)$, we have $X_t \leq \varepsilon_R$. $\qquad\square$

Putting everything together, we can prove Lemma D.12.

*Proof of Lemma D.12.* We apply Lemma D.14 and Lemma D.16 with $\delta = \frac{1}{C \log(2a_k/\delta_0^2)}$. The conditions on $\eta, \bar{\varepsilon}$ are indeed satisfied, and moreover $\|v_p(t)\|^2$ reaches the interval $[a_{\pi(p)} - \varepsilon_R, a_{\pi(p)} + \varepsilon_R]$ within a time of

$$\frac{\log(2\delta a_{\pi(p)}/\sigma_0^2)}{4\eta a_{\pi(p)}} + \frac{2\log(a_{\pi(p)}/\varepsilon_R)}{\delta \eta a_{\pi(p)}} \leq \frac{\log(a_{\pi(p)}/\sigma_0^2) + 2C\log(2a_{\pi(p)}/\sigma_0^2)\log(a_k/\varepsilon_R)}{\eta a_{\pi(p)}}$$

$$\ll \frac{\Delta/8}{4I(I-1)\hat{\sigma}_{2I}^2 \eta a_{\pi(p)} \bar{v}_{p,\pi(p)}^{2I-2}(0)}.$$

$\square$

## D.3  Maintaining the Induction Hypotheses

### D.3.1  Upper Bounds on the Irrelevant Coordinates

We first track the growth of a failed coordinate $\bar{v}_{k,\pi(q)}$ for $(k, \pi(q)) \notin \{(p, \pi(p))\}_{p \in [P_*]}$. The update on $\bar{v}_{k,\pi(q)}(t)$ is given by the following.

**Lemma D.17.** *Assume that Induction Hypothesis C.2 holds at time t. Then*

$$\bar{v}_{k,\pi(q)}^2(t+1) \leq \bar{v}_{k,\pi(q)}^2(t) + 4I\hat{\sigma}_{2I}^2 \eta \bar{v}_{k,\pi(q)}^{2I} \left| a_{\pi(q)} - \mathbb{1}(q \in [m], q \in L) \|v_q\|^2 \right| + Z(t+1) + \xi(t+1),$$

*where* $\mathbb{E}[Z(t+1) \mid \mathcal{F}_t] \lesssim \eta^2 \|a\|_1^2 \bar{v}_{k,\pi(q)}^2(t)$, *and*

$$\left| \xi(t+1) \right| \lesssim \eta^2 (1 + \bar{v}_{k,\pi(q)}^2(t)d) \|a\|_1^2 \log^{\tilde{Q}}(md/\delta_{\mathbb{P},\xi}) + C_\sigma^2 \eta a_{\pi(q)} \bar{v}_{k,\pi(q)}^{2I} \varepsilon_0 + \eta \left| \bar{v}_{k,\pi(q)}(t) \right| \delta_{\mathrm{error}},$$

*where*

$$\delta_{\mathrm{error}} := I2^{3I+6} C_\sigma^2 \left( a_{\pi(q)} \bar{\varepsilon}^{1/2} \varepsilon_0^{I-1} \vee m\sigma_1^2 \vee \|a\|_1 \varepsilon_0^I \right)$$

*Proof.* From the proof of Lemma C.10, we have that

$$\frac{d}{dt} \bar{v}_{k,\pi(q)}^2 \leq 4I\hat{\sigma}_{2I}^2 a_{\pi(q)} \bar{v}_{k,\pi(q)}^{2I} + 2C_\sigma^2 a_{\pi(q)} \bar{v}_{k,\pi(q)}^{2I} \varepsilon_0 + \left| \bar{v}_{k,\pi(q)} \right| \delta_{\mathrm{error}},$$

and so the desired result follows directly from combining the above with Lemma D.1. $\square$

We will next require the following stochastic induction helper lemma, with proof deferred to Appendix D.5.

**Lemma D.18.** *Suppose that* $(X_t)_t \geq 0$ *satisfies*

$$X_{t+1} \leq X_t + \alpha X_t^I + \xi_{t+1} + Z_{t+1}, \quad X_0 \leq x_0, \tag{9}$$

*where* $(\xi_t)_t$ *is an adapted process and* $(Z_t)_t$ *is a martingale difference sequence. Let* $\hat{x}_t$ *be a solution to the recurrence*

$$\hat{x}_{t+1} = \hat{x}_t + \alpha \hat{x}_t^I, \quad \hat{x}_0 = (1 + \epsilon)x_0$$

*Suppose that when* $X_t \leq \hat{x}_t$, *we have* $|\xi_{t+1}| \leq X_t^{1/2} \Xi_1 + X_t \Xi_2 + X_t^I \Xi_3 + \Xi_4$ *with probability* $1 - \delta_{\mathbb{P},\xi}$ *and* $\mathbb{E}[Z(t+1) \mid \mathcal{F}_t] \leq X_t \sigma_Z^2$. *Then if*

$$\Xi_1 \leq \frac{\epsilon x_0}{8 \sum_{t=0}^{T-1} \hat{x}_t^{1/2}}, \quad \Xi_2 \leq \frac{\epsilon x_0}{8 \sum_{t=0}^{T-1} \hat{x}_t}, \quad \Xi_3 \leq \frac{\epsilon x_0}{8 \sum_{t=0}^{T-1} \hat{x}_t^I}, \quad \Xi_4 \leq \frac{\epsilon x_0}{8T}, \quad and \quad \sigma_Z^2 \leq \frac{x_0^2 \epsilon^2 \delta_{\mathbb{P}}}{4 \sum_{t=0}^{T-1} \hat{x}_t}(1,0)$$

*we have* $X_t \leq \hat{x}_t$ *for all* $t \leq T$ *with probability* $1 - T\delta_{\mathbb{P},\xi} - \delta_{\mathbb{P}}$.

We can now control the growth of $\bar{v}_{k,\pi(q)}$ by applying Lemma D.18 with $X_t = \bar{v}^2_{k,\pi(q)}(t)$. For $(k,\pi(q)) \notin \{(p,\pi(p))\}_{p\in[P_*]}$, define the time $T_{(k,\pi(q))}$ by

$$T_{(k,\pi(q))} := \begin{cases} T_k & k < q, k \in [P^*] \\ T_q & q < k, q \in [P^*] \\ T_{P_*} & k,q > P^* \end{cases}.$$

By Assumption C.1(d), we have that

$$T_{(k,\pi(q))} \le \frac{(1+\Delta/4)d^{I-1}}{4I(I-1)\hat{\sigma}^2_{2I}\eta a_{\min_*}}.$$

**Lemma D.19** (Total growth of failed coordinates). *Let $(k,\pi(q)) \notin \{(p,\pi(p))\}_{p\in[P_*]}$. Assume that the learning rate $\eta$ satisfies*

$$\eta \le \frac{a_{\min_*} I \hat{\sigma}^2_{2I} d^{-I} \|a\|_1^{-2} \Delta^2 \delta_{\mathbb{P}}}{I \log(4/\Delta) \log^{\tilde{Q}}(md/\delta_{\mathbb{P},\xi})}$$

*for some sufficiently large constant C. Furthermore, suppose that*

$$\bar{\varepsilon}^{1/2}\varepsilon_0^{I-1} \vee m\sigma_1^2 \vee \|a\|_1 \varepsilon_0^I \ll \frac{a_{\min_*}\hat{\sigma}^2_{2I}d^{-I+1/2}\Delta}{I2^{3I+6}C_\sigma^2}, \quad \varepsilon_R \lesssim \frac{a_{\min_*}}{1.5^I d^{\gamma(I-1)}}, \quad \frac{d}{\log^{2/\gamma} d} \ge 2^{1/\gamma}(4/\Delta)^{\frac{1}{\gamma(I-1)}}.$$

*Then, with probability $1 - T_{P_*}\delta_{\mathbb{P},\xi} - 2\delta_{\mathbb{P}}$, we have that $\bar{v}^2_{k,\pi(q)} \le \varepsilon_0$ (and hence Induction Hypothesis C.2(b) is true) for all $t \le T_{P_*}$.*

*Proof.* First, we will show that $\bar{v}^2_{k,\pi(q)}(t) \le \varepsilon_0/2$ up to time $T_{(k,\pi(q))}$. Next, we will show that $v^2_{k,\pi(q)}(t)$ does not grow too much more in the interval $[T_{(k,\pi(q))}, T_{P_*}]$.

**Part 1** ($t \le T_{(k,\pi(q))}$). Our goal will be to apply Lemma D.18 up to time $T = T_{(k,\pi(q))}$, to the process $X_t = \bar{v}^2_{k,\pi(q)}(t)$, with $\alpha = 4I\hat{\sigma}^2_{2I}\eta a_{\pi(q)}$, $\varepsilon = \frac{\Delta}{4I}$, and $x_0 = \max(\frac{1}{2d}, \bar{v}^2_{k,\pi(q)})$.

We first aim to bound the quantity $\alpha(I-1)\hat{x}_0^{I-1}T$. We begin by considering the upper triangular entries, i.e those where $k < q$ and $k \in [P^*]$, in which case $T_{(k,\pi(q))} = T_k$. We have that

$$\alpha(I-1)\hat{x}_0^{I-1}T = 4I(I-1)\hat{\sigma}^2_{2I}\eta a_{\pi(q)} \cdot (1+\varepsilon)^{I-1}\max\left(\frac{1}{2d}, \bar{v}^2_{k,\pi(q)}\right)^{I-1} \cdot \frac{1+\Delta/4}{4I(I-1)\hat{\sigma}^2_{2I}\eta a_{\pi(k)}\bar{v}^{2I-2}_{k,\pi(k)}(0)}$$

$$\le (1+\varepsilon)^{I-1}(1+\Delta/4)\frac{a_{\pi(q)}\max\left(\left(\frac{1}{2d}\right)^{I-1}, \bar{v}^{2I-2}_{k,\pi(q)}\right)}{a_{\pi(k)}\bar{v}^{2I-2}_{k,\pi(k)}(0)}.$$

By the bound on the row gap in Assumption C.1(a), we have that $\frac{a_{\pi(q)}v^{2I-2}_{k,\pi(q)}}{a_{\pi(k)}\bar{v}^{2I-2}_{k,\pi(k)}(0)} \le \frac{1}{1+\Delta}$. Moreover, by the definition of the greedy maximum selection process along with Assumption C.1(d), $a_{\pi(k)}\bar{v}^{2I-2}_{k,\pi(k)}(0) \ge a_{\pi(q)}\max_{j>k}\bar{v}^{2I-2}_{j,\pi(q)}(0) \ge a_{\pi(q)}/d^{I-1}$, and thus $\frac{a_{\pi(q)}\cdot 1/(2d)^{I-1}}{a_{\pi(k)}\bar{v}^{2I-2}_{k,\pi(k)}(0)} \le \frac{1}{2^{I-1}} \le \frac{1}{1+\Delta}$. Altogether,

$$\alpha(I-1)\hat{x}_0^{I-1}T \le \frac{(1+\varepsilon)^{I-1}(1+\Delta/4)}{1+\Delta} \le \frac{\exp(\Delta/2)}{1+\Delta} \le 1 - \Delta/4,$$

since $\varepsilon = \frac{\Delta}{4I}$ and $\Delta \le 1/2$.

Next, consider the lower triangular entries, with $q < k$, $q \in [P^*]$. We have that $T_{(k,\pi(q))} = T_q$, and thus

$$\alpha(I-1)\hat{x}_0^{I-1}T = (1+\varepsilon)^{I-1}(1+\Delta/4)\frac{\max\left(\frac{1}{2d}, \bar{v}^2_{k,\pi(q)}(0)\right)^{I-1}}{\bar{v}^{2I-2}_{q,\pi(q)}(0)}.$$

By the bound on the column gap in Assumption C.1(b), we have $\frac{\bar{v}_{k,\pi(q)}^{2I-2}(0)}{\bar{v}_{q,\pi(q)}^{2I-2}(0)} \leq \frac{1}{1+\Delta}$. Moreover, by Assumption C.1(d), we have $\frac{1/(2d)}{\bar{v}_{q,\pi(q)}^2(0)} \leq \frac{1}{2} \leq \frac{1}{1+\Delta}$. Therefore $\alpha(I-1)\hat{x}_0^{I-1}T \leq \frac{(1+\varepsilon)^{I-1}(1+\Delta/4)}{1+\Delta} \leq 1 - \Delta/4$ as well.

Finally, we consider the lower right block, with $k, q > P_*$, in which case $T_{(k,p)} = T_{P_*}$. We see that

$$\alpha(I-1)\hat{x}_0^{I-1}T = (1+\varepsilon)(1+\Delta/4)\frac{a_{\pi(q)}\max(\frac{1}{2d}, \bar{v}_{k,\pi(q)}^2(0))}{a_{\pi(P_*)}\bar{v}_{P_*,\pi(P_*)}^2(0)}.$$

By the bound on the threshold gap in C.1(c), we have $\frac{\bar{v}_{k,\pi(q)}^{2I-2}(0)}{\bar{v}_{P_*,\pi(P_*)}^{2I-2}(0)} \leq \frac{1}{1+\Delta}$. Moreover, by the definition of the greedy maximum selection process along with Assumption C.1(d), we have that $a_{\pi(P_*)}\bar{v}_{P_*,\pi(P_*)}^2(0) \geq a_{\pi(q)}\max_{j>P_*}\bar{v}_{j,\pi(q)}^2(0) \geq a_{\pi(q)}/d$, and thus $\frac{a_{\pi(q)}\cdot 1/(2d)^{I-1}}{a_{\pi(P_*)}\bar{v}_{P_*,\pi(P_*)}^{2I-2}(0)} \leq \frac{1}{2^{I-1}} \leq \frac{1}{1+\Delta}$. Altogether, $\alpha(I-1)\hat{x}_0^{I-1}T \leq \frac{(1+\varepsilon)^{(I-1)}(1+\Delta/4)}{1+\Delta} \leq 1 - \Delta/4$.

In all cases, we have $\alpha(I-1)\hat{x}_0^{I-1}T \leq 1 - \Delta/4$. Thus by Lemma D.23, we can bound $\hat{x}_T$ by

$$\hat{x}_T \leq \frac{\hat{x}_0}{\left(1 - \alpha(I-1)\hat{x}_0^{I-1}T\right)^{\frac{1}{I-1}}} \leq \hat{x}_0(\Delta/4)^{-\frac{1}{I-1}} \leq d^{-1+\gamma}/2 =: \varepsilon_0/2,$$

provided that $\frac{d}{\log^{2/\gamma}d} \geq 2^{1/\gamma}(4/\Delta)^{\frac{1}{\gamma(I-1)}}$.

Therefore by Lemma D.17, the update for $\bar{v}_{k,\pi(q)}^2(t)$ is

$$\bar{v}_{k,\pi(q)}^2(t+1) \leq \bar{v}_{k,\pi(q)}^2(t) + 4I\hat{\sigma}_{2I}^2\eta a_{\pi(q)}\bar{v}_{k,\pi(q)}^{2I} + Z(t+1) + \xi(t+1),$$

which is indeed of the form (9) for $\sigma_Z^2 \lesssim \eta^2\|a\|_1^2$ and $\Xi_1 \lesssim \eta\delta_{\text{error}}, \Xi_2 \lesssim \eta^2 d\|a\|_1^2 \log^{\tilde{Q}}(md/\delta_{\mathbb{P},\xi}), \Xi_3 \lesssim C_\sigma^2\eta a_{\pi(q)}\varepsilon_0, \Xi_4 \lesssim \eta^2\|a\|_1^2\log^{\tilde{Q}}(md/\delta_{\mathbb{P},\xi})$.

Next, we verify that the conditions on $\Xi, \sigma_Z^2$, in (10) hold. We first bound the quantity $\sum_{t=0}^{T-1}\hat{x}_t$.

$$\sum_{t=0}^{T-1}\hat{x}_t \leq \int_0^T \frac{x_0}{\left(1 - \alpha(I-1)(\hat{x}_0)^{I-1}t\right)^{\frac{1}{I-1}}}dt$$

$$\leq \begin{cases} \alpha^{-1}\log\left(\frac{1}{1-\alpha\hat{x}_0T}\right) & I = 2 \\ \frac{1}{(I-2)\alpha(\hat{x}_0)^{I-2}}\left[1 - (1-\alpha(I-1)\hat{x}_0^{I-1}T^+)^{\frac{I-2}{I-1}}\right] & I > 2 \end{cases}$$

$$\leq \begin{cases} \alpha^{-1}\log(4/\Delta) & I = 2 \\ (I-2)^{-1}\alpha^{-1}\hat{x}_0^{2-I} & I > 2 \end{cases}.$$

Therefore

$$\sum_{t=0}^{T-1}\hat{x}_t \leq \alpha^{-1}\hat{x}_0^{2-I}\min((I-2)^{-1}, \log(4/\Delta)) \tag{11}$$

Next, we can bound the quantity $\sum_{t=0}^{T-1}\hat{x}_t^{1/2}$:

$$\sum_{t=0}^{T-1}\hat{x}_t^{1/2} \leq \int_0^T \frac{\hat{x}_0^{1/2}}{\left(1 - \alpha(I-1)(\hat{x}_0)^{I-1}t\right)^{\frac{1}{2(I-1)}}}dt$$

$$= \frac{2\hat{x}_0^{1/2}}{\alpha(2I-3)\hat{x}_0^{I-1}}\left(1 - \left(1 - \alpha(I-1)\hat{x}_0^{I-1}T\right)^{\frac{2I-3}{2(I-1)}}\right)$$

$$\leq 2\hat{x}_0^{1/2}T$$

Finally, we can bound the quantity $\sum_{t=0}^{T-1} \hat{x}_t^I$

$$
\begin{aligned}
\sum_{t=0}^{T-1} \hat{x}_t^I &\le \int_0^T \frac{\hat{x}_t^I}{\left(1 - \alpha(I-1)\hat{x}_t^{I-1}t\right)^{\frac{I}{I-1}}} dt \\
&= \hat{x}_t \alpha^{-1} \left( \frac{1}{\left(1 - \alpha(I-1)\hat{x}_t^{I-1}T\right)^{\frac{1}{I-1}}} - 1 \right) \\
&\le x_0^+ \alpha^{-1} (\Delta/4)^{-\frac{1}{I-1}}.
\end{aligned}
$$

Let us consider the $\sigma_Z^2$ condition. Plugging in (11), it suffices to take

$$
\sigma_Z^2 \le \frac{x_0^I \varepsilon^2 \delta_{\mathbb{P}} \alpha}{\log(4/\Delta)} = \frac{x_0^I \varepsilon^2 \delta_{\mathbb{P}} \cdot 4I\hat{\sigma}_{2I}^2 \eta a_{\pi(q)}}{\log(4/\Delta)}
$$

Plugging in $\sigma_Z^2 \lesssim \eta^2 \|a\|_1^2$, and noting $x_0 \ge \frac{1}{2d}$, this is satisfied if we take

$$
\eta \lesssim \frac{a_{\min_*} \hat{\sigma}_{2I}^2 d^{-I} \Delta^2 \|a\|_1^{-2} \delta_{\mathbb{P}}}{I \log(4/\Delta)}.
$$

Next, for the $\Xi_1$ constraint, we require

$$
\begin{aligned}
\Xi_1 \le \frac{\varepsilon x_0}{8 \sum_{t=0}^{T-1} \hat{x}_t^{1/2}} &\Longleftarrow \eta \delta_{\text{error}} \lesssim \frac{\varepsilon x_0}{T\hat{x}_0^{1/2}} \\
&\Longleftarrow \eta \delta_{\text{error}} \lesssim a_{\min_*} \hat{\sigma}_{2I}^2 \eta d^{-(I-1)} \Delta x_0^{1/2} \\
&\Longleftarrow \delta_{\text{error}} \lesssim a_{\min_*} \hat{\sigma}_{2I}^2 d^{-I+1/2} \Delta
\end{aligned}
$$

For $\Xi_2$, plugging in (11) we require

$$
\begin{aligned}
\Xi_2 \le \frac{\varepsilon x_0^{I-1} \alpha}{8 \log(4/\Delta)} = \frac{\varepsilon x_0^{I-1} \cdot I\hat{\sigma}_{2I}^2 \eta a_{\pi(q)}}{2 \log(4/\Delta)} \\
\Longleftarrow \eta^2 d \|a\|_1^2 \log^{\tilde{Q}}(md/\delta_{\mathbb{P},\xi}) \ll \frac{\varepsilon x_0^{I-1} \cdot I\hat{\sigma}_{2I}^2 \eta a_{\pi(q)}}{2 \log(4/\Delta)} \\
\Longleftarrow \eta \ll \frac{a_{\min_*} \hat{\sigma}_{2I}^2 d^{-I} \|a\|_1^{-2} \Delta}{\log(4/\Delta) \log^{\tilde{Q}}(md/\delta_{\mathbb{P},\xi})}
\end{aligned}
$$

For $\Xi_3$, we require

$$
\begin{aligned}
\Xi_3 \le \frac{\varepsilon x_0}{8 \sum_{t=0}^{T-1} \hat{x}_t^I} &\Longleftarrow C_\sigma^2 \eta a_{\pi(q)} \varepsilon_0 \lesssim (\Delta/I)^{\frac{I}{I-1}} \alpha \\
&\Longleftarrow C_\sigma^2 \eta a_{\pi(q)} \varepsilon_0 \lesssim \Delta^{\frac{I}{I-1}} I^{-\frac{1}{I-1}} \hat{\sigma}_{2I}^2 \eta a_{\pi(q)} \\
&\Longleftarrow \varepsilon_0 \lesssim C_\sigma^{-2} \Delta^{\frac{I}{I-1}} \hat{\sigma}_{2I}^2,
\end{aligned}
$$

which is indeed true since $\varepsilon_0 \le d^{-1/2} \ll C_\sigma^{-2} \Delta^{\frac{I}{I-1}} \hat{\sigma}_{2I}^2$. Finally, for $\Xi_4$, we require

$$
\begin{aligned}
\Xi_4 \le \frac{\varepsilon x_0}{8T} &\Longleftarrow \eta^2 \|a\|_1^2 \log^{\tilde{Q}}(md/\delta_{\mathbb{P},\xi}) \ll \Delta d^{-1} \cdot \eta a_{\min_*}(I-1)\hat{\sigma}_{2I}^2 d^{-(I-1)} \\
&\Longleftarrow \eta \ll \frac{a_{\min_*}(I-1)\hat{\sigma}_{2I}^2 d^{-I} \|a\|_1^{-2} \Delta}{\log^{\tilde{Q}}(md/\delta_{\mathbb{P},\xi})}.
\end{aligned}
$$

Therefore the conditions of Lemma D.18 are satisfied, and so with probability $1 - T\delta_{\mathbb{P},\xi} - \delta_{\mathbb{P}}$ we have $X_t \le \hat{x}_t \le \varepsilon_0/2$ for all $t \le T$.

**Part 2** $(T_{(k,\pi(q))} \le t \le T_{P_*})$   We now show that $\bar{v}_{k,\pi(q)}^2$ doesn't increase too much in the time interval $[T_{(k,\pi(q))}, T_{P_*}]$. The case where $k, q > P_*$ is trivially true.

Consider the case when $q < k, q \in [P_*]$, so that $T = T_q$. By Induction Hypothesis D.1(b), when $t \geq T_q$, we have that $\left\|v_q(t)\right\|^2 = a_{\pi(q)} \pm \varepsilon_R$. When $\bar{v}_{k,\pi(q)}^2(t) \leq \varepsilon_0$, we have that

$$\bar{v}_{k,\pi(q)}^2(t+1) \leq \bar{v}_{k,\pi(q)}^2(t) + 4I\hat{\sigma}_{2I}^2 \eta \varepsilon_R \bar{v}_{k,\pi(q)}^{2I} + Z(t+1) + \xi(t+1),$$

where $\mathbb{E}[Z(t+1) \mid \mathcal{F}_t] \lesssim \eta^2 \|a\|_1^2 v_{k,\pi(q)}^2(t)$, and

$$|\xi(t+1)| \lesssim \eta^2 \|a\|_1^2 \varepsilon_0 d \log^{\tilde{Q}}(md/\delta_{\mathbb{P},\xi}) + C_\sigma^2 \eta a_{\pi(q)} \varepsilon_0^{I+1} + \eta \varepsilon_0^{1/2} \delta_{\text{error}}.$$

We would like to apply Lemma D.18 to the process $X_t = \bar{v}_{k,\pi(q)}^2(t + T_q)$ up to time $T_{P^*}$, with $\alpha = 4I\hat{\sigma}_{2I}^2 \eta \varepsilon_R, \varepsilon = 0.5$. We see that $X_0 \leq \frac{\varepsilon_0}{2} := x_0$, and so setting $\hat{x}_0 = 1.5x_0$, we have that

$$t \leq T_{P_*} \leq \frac{d^{I-1}}{2I(I-1)\hat{\sigma}_{2I}^2 \eta a_{\min_*}},$$

and thus as long as $\varepsilon_R \lesssim \frac{a_{\min_*}}{8 \cdot 1.5^{I-1} d^{\gamma(I-1)}}$, we have

$$\alpha(I-1)\hat{x}_0^{I-1} t \leq 2 \cdot 1.5^{I-1} \varepsilon_0^{I-1} d^{I-1} \varepsilon_R a_{\min_*}^{-1} = 2 \cdot 1.5^{I-1} d^{\gamma(I-1)} \varepsilon_R a_{\min_*}^{-1} \leq 1/4 \leq 1 - (3/4)^{I-1}$$

$$\implies \hat{x}_t \leq \frac{\hat{x}_0}{\left(1 - \alpha(I-1)\hat{x}_0^{I-1} t\right)^{\frac{1}{I-1}}} = \frac{1.5x_0}{\left(1 - \alpha(I-1)\hat{x}_0^{I-1} t\right)^{\frac{1}{I-1}}} \leq 2x_0 = \varepsilon_0$$

We next verify that the conditions of Lemma D.18 hold . We first require

$$\Xi_4 \leq \frac{x_0}{16T}$$

$$\Longleftarrow \eta^2 \|a\|_1^2 \varepsilon_0 d \log^{\tilde{Q}}(md/\delta_{\mathbb{P},\xi}) + C_\sigma^2 \eta a_{\pi(q)} \varepsilon_0^{I+1} + \eta \varepsilon_0^{1/2} \delta_{\text{error}} \lesssim I(I-1)\hat{\sigma}_{2I}^2 \eta a_{\min_*} d^{-(I-1)} \varepsilon_0$$

$$\Longleftarrow \eta \lesssim \frac{a_{\min_*} d^{-I} I(I-1)\hat{\sigma}_{2I}^2 \|a\|_1^{-2}}{\log^{\tilde{Q}}(md/\delta_{\mathbb{P},\xi})}, \quad \varepsilon_0^I \ll I(I-1)\hat{\sigma}_{2I}^2 a_{\min_*} d^{-(I-1)},$$

$$\text{and} \quad \delta_{\text{error}} \ll \frac{a_{\min_*} I(I-1)\hat{\sigma}_{2I}^2 \varepsilon_0^{1/2}}{d^{I-1}}.$$

Clearly the condition on $\eta$ is satisfied. Next, plugging in $\varepsilon_0 = d^{-(1-\gamma)}$, we require

$$d \gg \left(I(I-1)\hat{\sigma}_{2I}^2 a_{\min_*}\right)^{-\frac{1}{1-I\gamma}}.$$

Finally, the condition on $\delta_{\text{error}}$ is indeed satisfied, since we already have

$$\delta_{\text{error}} \lesssim a_{\min_*} I(I-1)\hat{\sigma}_{2I}^2 d^{-I+1/2} \Delta \ll a_{\min_*} I(I-1)\hat{\sigma}_{2I}^2 d^{-I+1} \varepsilon_0^{1/2}$$

Additionally, since we can bound $\sum_{t=1}^{T} \hat{x}_t \leq T\varepsilon_0$, we require

$$\sigma_Z^2 \lesssim \frac{x_0^2 \delta_{\mathbb{P}}}{T\varepsilon_0} \Longleftarrow \eta^2 \|a\|_1^2 \lesssim \varepsilon_0 \delta_{\mathbb{P}} I(I-1)\hat{\sigma}_{2I}^2 \eta a_{\min_*} d^{-(I-1)}$$

$$\Longleftarrow \eta \lesssim a_{\min_*} \varepsilon_0 d^{-(I-1)} I(I-1)\hat{\sigma}_{2I}^2 \|a\|_1^{-2} \delta_{\mathbb{P}},$$

which is again satisfied by our choice of $\eta$. Altogether, we have $X_t \leq \hat{x}_t \leq \varepsilon_0$ for all $t \leq T_{P_*}$.

Finally, consider the case when $k < q, k \in [P_*]$, so that $T = T_k$. By Induction Hypothesis D.1(a), when $t \geq T_k$, we have that $\bar{v}_{k,\pi(k)}(t)^2 \geq 1 - \bar{\varepsilon}$, and thus $\bar{v}_{k,\pi(q)}^2(t) \leq \bar{\varepsilon} \leq \varepsilon_0$, as desired. $\square$

### D.3.2 Upper Bounds on the Norm Growth

We start with an upper bound on the norm of the unused neurons, i.e., $v_k$ with $k > P_*$.

**Lemma D.20** (Bound on the unused neurons). *Inductively assume that Induction Hypothesis C.2(b) is true. Suppose that we choose*

$$\eta \lesssim \frac{a_{\min_*} d^{-I} I(I-1)\hat{\sigma}_{2I}^2 \|a\|^{-2} \delta_{\mathbb{P}}}{\log^{\tilde{Q}}(Tmd/\delta_{\mathbb{P}})}.$$

*Then, for any $k \in [m]$ with $k > P_*$, with probability at least $1 - \delta_{\mathbb{P}}$ we have $\|v_k\|^2 \leq O(\sigma_0^2) \ll \sigma_1^2$ throughout training.*

*Proof.* By the proof of Lemma C.14 along with Lemma D.1, we have

$$\|v_k(t+1)\|^2 \le \left(1 + 4\eta\varepsilon_0^I \|a\|_1\right) \|v_k\|^2 - 2\eta\langle v_k, H_k(t+1)\rangle + \xi_{k,R}(t+1)$$

The total running time of SGD is $T = \frac{1+\Delta/4}{4I(I-1)\hat\sigma_{2I}^2 \eta a_{\min_*} \bar{v}_{P_*,\pi(P_*)}^{2I-2}(0)} \lesssim \frac{d^{I-1}}{4I(I-1)\hat\sigma_{2I}^2 a_{\min_*}\eta}$. Therefore

$$4\eta\varepsilon_0^I \|a\|_1 \cdot T \lesssim \frac{\varepsilon_0^I \|a\|_1 d^{I-1}}{I(I-1)\hat\sigma_{2I}^2 a_{\min_*}} = \frac{d^{-(1-\gamma I)}\|a\|_1}{I(I-1)\hat\sigma_{2I}^2 a_{\min_*}} \ll 1,$$

since $d \gtrsim \left(\frac{\|a\|_1}{I(I-1)\hat\sigma_{2I}^2 a_{\min_*}}\right)^{\frac{1}{1-\gamma I}}$. Thus $\left(1 + 4\eta\varepsilon_0^I \|a\|_1\right)^T \lesssim 1$. In addition, by Lemma B.1, we have

$$\mathbf{Var}\left(2\eta\|v_k\|\langle\bar{v}_k, H_k(t+1)\rangle\right) \lesssim \eta^2 \|a\|_1^2 \|v_k\|^4.$$

Hence, using the language of Lemma F.6 of [RL24], we have

$$\alpha = \Theta\left(\eta\varepsilon_0^I \|a\|_1\right), \quad \sigma_Z^2 = O\left(\eta^2 \|a\|_1^2 \sigma_0^4\right)$$

$$\Xi = O\left(\eta^2 d \|a\|_1^2 \log^{\tilde{Q}}\left(\frac{Tmd}{\delta_{\mathbb{P}}}\right)\sigma_0^2\right), \quad T = O\left(\frac{d^{I-1}}{4I(I-1)\hat\sigma_{2I}^2 a_{\min_*}\eta}\right).$$

To satisfy the condition of that lemma, it suffices to choose

$$\sigma_Z^2 \lesssim \alpha\delta_{\mathbb{P}}\sigma_0^4 \quad \Leftarrow \quad \eta \lesssim \varepsilon_0^{-I} \|a\|^{-1} \delta_{\mathbb{P}}$$

$$\Xi \lesssim \frac{\sigma_0^2}{T} \quad \Leftarrow \quad \eta \lesssim \frac{a_{\min_*} d^{-I} I(I-1)\hat\sigma_{2I}^2 \|a\|^{-2}}{\log^{\tilde{Q}}(Tmd/\delta_{\mathbb{P}})}.$$

$\square$

Then, we consider $k = p \le P_*$. Unlike those unused neurons, since $v_p$ will eventually converge to $e_{\pi(p)}$, its norm cannot stay small. Our strategy here will be coupling its norm growth with the tangent movement. We will use the following extension to Lemma F.11 of [RL24]. The proof of this lemma can be found in Section D.5.

**Lemma D.21.** *Suppose that $(X_t)_t$ satisfies*

$$X_{t+1} = X_t + \alpha_t(X_t)X_t + \xi_{t+1} + Z_{t+1}, \quad X_0 = x_0 > 0,$$

*where $\alpha_t : \mathbb{R} \to \mathbb{R}_{\ge 0}$ is an $\mathcal{F}_t$-measurable non-decreasing function, $(\xi_t)_t$ is an adapted process, and $(Z_t)_t$ is a martingale difference sequence. Let $\varepsilon > 0$ be given and define the process*

$$\hat{X}_{t+1} = \hat{X}_t + \alpha_t(\hat{X}_t), \quad \hat{X}_0 = (1+\varepsilon)x_0.$$

*Fix $T > 0$, $\delta_{\mathbb{P}} \in (0,1)$. Suppose that there exists $\Xi, \sigma_Z > 0$ and $\delta_{\mathbb{P},\xi} \in (0,1)$ such that when $X_t \le \hat{X}_t$, we have $|\xi_{t+1}| \le \Xi$ with probability at least $1 - \delta_{\mathbb{P},\xi}$, and $\mathbb{E}[Z_{t+1} \mid \mathcal{F}_t] \le \sigma_Z^2$. Then, if*

$$\Xi \le \varepsilon_0 x_0/(2T) \quad \text{and} \quad \sigma_Z^2 \le \varepsilon^2 x_0^2 \delta_{\mathbb{P}}/(4T),$$

*we have $X_t \le \hat{X}_t$ for all $t \le T$.*

The following lemma verifies Induction Hypothesis C.2(a) for $\sigma_1 = O(\sigma_0 \bar\varepsilon^{-C/2})$ for some constant $C$.

**Lemma D.22** (Bound on $\|v_P\|^2$). *Suppose that $d \gg \left(\frac{\|a\|_1}{I(I-1)\hat\sigma_{2I}^2 a_{\min_*}}\right)^{\frac{1}{1-I\gamma}}$ and $\eta \lesssim \frac{a_{\min_*} I(I-1)\hat\sigma_{2I}^2 d^{-I} \|a\|_1^{-2}}{\log^{\tilde{Q}}(md/\delta_{\mathbb{P},\xi})}$. Then there exists a constant $C_{\exp}$ such that $\|v_P\|^2 \le O\left(\sigma_0^2 \bar\varepsilon^{-C_{\exp}}\right)$ as long as $\bar{v}_{p,\pi(p)}^2$ has not reached $1 - \bar\varepsilon$.*

*Proof.* By the proof of Lemma C.15, when Induction Hypothesis C.2(b) holds we have

$$\|v_P(t+1)\|^2 \le \|v_P\|^2 + 4\eta(a_{\pi(p)}\bar{v}_{p,\pi(p)}^{2I} + \|a\|_1 \varepsilon_0^I)\|v_P\|^2 - Z_{p,R}(t+1) + \xi_{p,R}(t+1),$$

where, by Lemma B.1 and Lemma D.1, the conditional variance of $Z_{p,R}$ is bounded by $O\left(\eta^2 \|a\|_1^2 \|v_p\|^4\right)$ and we have

$$|\xi_{p,R}(t+1)| \lesssim \eta^2 d \|a\|_1^2 \log^{\tilde{Q}}\left(\frac{md}{\delta_{\mathbb{P}}}\right) \|v_p\|^2 \quad \text{with probability at least } 1 - \delta_{\mathbb{P}}.$$

First, consider the situation where $\bar{v}^2_{p,\pi(p)} \leq 0.9$. We prove by stochastic induction that $\|v_p\|^2 \leq O(\sigma_0^2)$. Under this induction hypothesis, using the language of Lemma D.21 with $\varepsilon = 0.5$, we have

$$\sigma_Z^2 = O(\eta^2 \|a\|_1^2 \sigma_0^4), \quad \Xi = O\left(\eta^2 d \|a\|_1^2 \log^{\tilde{Q}}\left(\frac{md}{\delta_{\mathbb{P}}}\right)\sigma_0^2\right), \quad T = O\left(\frac{d^{I-1}}{I(I-1)\hat{\sigma}^2_{2I} a_{\min_*}\eta}\right).$$

Hence, to meet the condition of Lemma D.21, it suffices to choose

$$\sigma_Z^2 \lesssim \frac{\sigma_0^4 \delta_{\mathbb{P}}}{T} \quad \Leftarrow \quad \eta \lesssim a_{\min_*} d^{-(I-1)} I(I-1)\hat{\sigma}^2_{2I} \|a\|^{-2} \delta_{\mathbb{P}},$$

$$\Xi \lesssim \frac{\sigma_0^2}{T} \quad \Leftarrow \quad \eta \lesssim \frac{a_{\min_*} d^{-(I-1)} I(I-1)\hat{\sigma}^2_{2I} \|a\|^{-2}}{\log^{\tilde{Q}}\left(\frac{md}{\delta_{\mathbb{P}}}\right)}.$$

When these hold, then we have with probability at least $1 - O(\delta_{\mathbb{P}})$ that $\|v_p(t)\|^2 = (1 \pm 0.5) N^2(t)$ for any $t \leq T$, where $N^2$ is defined via

$$N^2(t+1) := N^2(t) + 4\eta\left(a_{\pi(p)} \bar{v}^{2I}_{p,\pi(p)}(t) + \|a\|_1 \varepsilon_0^I\right) N^2(t), \quad N^2(0) = 1.5 \|v_p(0)\|^2.$$

Now, we analyze the process $N^2$. First, note that

$$N^2(t) \leq N^2(0) \prod_{s=0}^{t-1}\left(1 + 4\eta\left(a_{\pi(p)} \bar{v}^{2I}_{p,\pi(p)}(s) + \|a\|_1 \varepsilon_0^I\right)\right)$$

$$\leq 1.5\sigma_0^2 \exp\left(4\eta T \|a\|_1 \varepsilon_0^I\right) \exp\left(4\eta a_{\pi(p)} \sum_{s=0}^{t} \bar{v}^{2I}_{p,\pi(p)}(s)\right).$$

First, we see that

$$4\eta T \|a\|_1 \varepsilon_0^I \leq \frac{d^{I-1} \|a\|_1 \varepsilon_0^I}{I(I-1)\hat{\sigma}^2_{2I} a_{\min_*}} = \frac{d^{I\gamma - 1} \|a\|_1}{I(I-1)\hat{\sigma}^2_{2I} a_{\min_*}} \ll 1,$$

since $d \gg \left(\frac{\|a\|_1}{I(I-1)\hat{\sigma}^2_{2I} a_{\min_*}}\right)^{\frac{1}{1-I\gamma}}$.

Next, By the proof of Lemma D.7, when $\bar{v}^2_{p,\pi(p)} \leq 0.9$, we have

$$\bar{v}^2_{p,\pi(p)}(t+1) \geq \bar{v}^2_{p,\pi(p)}(t) + 2\eta a_{\pi(p)} I \hat{\sigma}^2_{2I} \bar{v}^{2I}_{p,\pi(p)}(t) + Z_{t+1} + \xi_{t+1},$$

where with probability $1 - \delta_{\mathbb{P},\xi}$ we have $|\xi_{t+1}| \lesssim \eta^2 d \|a\|_1^2 \log^{\tilde{Q}}(md/\delta_{\mathbb{P},\xi})$, and the martingale term $Z_{t+1}$ satisfies $\mathbb{E}[Z_{t+1}^2 \mid \mathcal{F}_t] \lesssim \eta^2 \|a\|_1^2$. Therefore

$$\bar{v}^2_{p,\pi(p)}(t+1) \geq \bar{v}^2_{p,\pi(p)}(0) + 2\eta a_{\pi(p)} I \hat{\sigma}^2_{2I} \sum_{s=0}^{t} \bar{v}^{2I}_{p,\pi(p)}(s) + \sum_{s=0}^{t} \xi_{s+1} + \sum_{s=0}^{t} Z_{s+1}.$$

We first have

$$\left|\sum_{s=0}^{t} \xi_{s+1}\right| \lesssim T\eta^2 d \|a\|_1^2 \log^{\tilde{Q}}(md/\delta_{\mathbb{P},\xi}).$$

Since $\eta T \leq O\left(\frac{d^{I-1}}{I(I-1)\hat{\sigma}^2_{2I} a_{\min_*}}\right)$, we thus have $\left|\sum_{s=0}^{t} \xi_{s+1}\right| \leq 1$ whenever $\eta \lesssim \frac{a_{\min_*} I(I-1)\hat{\sigma}^2_{2I} d^{-I} \|a\|_1^{-2}}{\log^{\tilde{Q}}(md/\delta_{\mathbb{P},\xi})}$. Next, by Doob's submartingale inequality, we have

$$\mathbb{P}\left[\sup_{r \leq t}\left|\sum_{s=1}^{r} Z_s\right| \geq 1\right] \lesssim T\eta^2 \|a\|_1^2 \lesssim \eta \cdot \frac{d^{I-1} \|a\|_1^2}{I(I-1)\hat{\sigma}^2_{2I} a_{\min_*}}$$

and thus if $\eta \lesssim \frac{a_{\min_*} I(I-1)\hat\sigma_{2I}^2 \|a\|_1^{-2} \delta_P}{d^{I-1}}$ we have that $\sup_{r \leq t} \left|\sum_{s=1}^{r} Z_s\right| \leq 1$ with probability $1 - \delta_{\mathbb{P}}$. Altogether, on these events we have that

$$\eta a_{\pi(p)} I \hat\sigma_{2I}^2 \sum_{s=0}^{t} \bar v_{p,\pi(p)}^{2I}(s) \leq 1.5$$

As a result,

$$N^2(t) \leq 1.5\sigma_0^2 \exp\left(4\eta T \|a\|_1 \varepsilon_0^I\right) \exp\left(\frac{6}{I\hat\sigma_{2I}^2}\right) = O(\sigma_0^2),$$

In other words, we have $\|v_p\|^2 = O(\sigma_0^2)$ when $\bar v_{p,\pi(p)}^2 \leq 0.9$.

Now, consider the situation where $\bar v_{p,\pi(p)}^2 \in [0.9, 1 - \bar\varepsilon]$. By the proof of Lemma D.10, it takes at most $\frac{3^I \log(2/\bar\varepsilon)}{I\hat\sigma_{2I}^2 \eta a_{\pi(p)}}$ iterations for $\bar v_{p,\pi(p)}^2$ to grow from 0.9 to $1 - \bar\varepsilon$. In this stage, we have

$$\|v_p(t+1)\|^2 \leq \|v_p\|^2 + 4.1\eta a_{\pi(p)} \|v_p\|^2 - Z_{p,R}(t+1) + \xi_{p,R}(t+1).$$

Let the corresponding deterministic process be $M^2(t+1) = M^2(t) + 4.1\eta a_{\pi(p)} M^2(t)$ with $M^2(T_0) = O(\sigma_0^2)$ where $T_0$ is the time $\bar v_{p,\pi(p)}^2$ reaches 0.9. Using the language of Lemma F.6 of [RL24], we have

$$\alpha = 4.1\eta a_{\pi(p)}, \quad \sigma_Z^2 = O(\eta^2 \|a\|_1^2 \sigma_0^4), \quad \Xi = O\left(\eta^2 d \|a\|_1^2 \log^{\tilde Q}\left(\frac{md}{\delta_{\mathbb{P}}}\right)\sigma_0^2\right).$$

Therefore, to meet the condition of Lemma F.6 of [RL24], it suffices to require

$$\Xi \lesssim \frac{x_0}{T} \quad \Longleftarrow \quad \eta \lesssim \frac{a_{\pi(p)} I \hat\sigma_{2I}^2 \|a\|_1^{-2}}{d 3^I \log^{\tilde Q}\left(\frac{md}{\delta_{\mathbb{P}}}\right)\log(2/\bar\varepsilon)},$$

$$\sigma_Z^2 \lesssim \delta_{\mathbb{P}} \alpha x_0^2 \quad \Longleftarrow \quad \eta \lesssim a_{\pi(p)} \delta_{\mathbb{P}} \|a\|_1^{-2}.$$

Meanwhile, we have

$$M^2(T_1) \leq M^2(T_0) \exp\left((T_1 - T_0) \cdot 4.1\eta a_{\pi(p)}\right) \leq O\left(\sigma_0^2 \bar\varepsilon^{-C_{\exp}}\right),$$

for $C_{\exp} = \frac{4.1 \cdot 3^I}{I\hat\sigma_{2I}^2}$. $\qquad\qquad\qquad\qquad\qquad\qquad\qquad\qquad\qquad\qquad\qquad\square$

### D.4 Proof of Theorem 2.1

*Proof.* First, by Lemma B.2, with probability $1 - \delta_{\mathbb{P}^*}/2$, Assumption C.1 holds at initialization, with $\Delta := \min(\delta_r, \delta_c, \delta_t) = O(\frac{\delta_{\mathbb{P}^*}}{mP\max(m,P)})$.

Define $T_{\max} = \max_{p \in [P_*]}(1 + \Delta/4)T_p \lesssim \frac{d^{I-1}}{I(I-1)\hat\sigma_{2I}^2 \eta a_{\min_*}}$. We will show that, with probability $1 - \delta_{\mathbb{P}}^*/2$, that Induction Hypotheses C.2 and D.1 hold for all $t \leq T_{\max}$ with choice of parameters
We do so by union bounding over the consequence of the following lemmas:

- (Directional convergence) Lemma D.2 for all $p \in [P_*]$, with $\delta_{\mathbb{P}} = \frac{\delta_{\mathbb{P}}^*}{16P_* \log\log d}, \delta_{\mathbb{P},\xi} = \frac{\delta_{\mathbb{P}}^*}{16T_{\max}P_*}$. This implies the first half of part (b).

- (Convergence of norm) Lemma D.12 for all $p \in [P_*]$, with $\delta_{\mathbb{P}} = \frac{\delta_{\mathbb{P}}^*}{16P_*}, \delta_{\mathbb{P},\xi} = \frac{\delta_{\mathbb{P}}^*}{16T_{\max}P_*}$. This implies the second half of part (b).

- (Bound on the failed coordinates) Lemma D.19 for all $(k, \pi(q)) \notin \{(p, \pi(p))\}_{p \in [P_*]}$, with $\delta_{\mathbb{P}} = \frac{\delta_{\mathbb{P}}^*}{16mP}, \delta_{\mathbb{P},\xi} = \frac{\delta_{\mathbb{P}}^*}{16T_{\max}mP}$. This verifies that Induction Hypothesis C.2(b) holds throughout training.

- (Bound on unused neurons) Lemma D.20 for all $k \in [m] \setminus [P_*]$ with $\delta_{\mathbb{P}} = \frac{\delta_{\mathbb{P}}^*}{16m}$. This implies part (a).

- (Upper bound on norm growth) Lemma D.22 for all $p \in [P_*]$, with $\delta_{\mathbb{P}} = \frac{\delta_{\mathbb{P}}^*}{16 P_*}$. This implies part (c).

Next, we verify that our choice of $\varepsilon_0, \bar{\varepsilon}, \sigma_0, \sigma_1$ indeed satisfy the conditions of the lemmas. First, Lemma D.2 requires the conditions on C.3 to hold. Recall that we have chosen $\delta_T = \frac{\Delta^2}{CI^2}$ for sufficiently large constant $C$, and we will select $\gamma \le \frac{1}{4I}$. We thus require

$$d^{-1/2} \le \frac{\delta_T I \hat{\sigma}_{2I}^2}{C_\sigma^2} \Longleftarrow d \gtrsim \frac{C_\sigma^4 I^2}{\hat{\sigma}_{2I}^4 \Delta^4}, \quad \frac{d}{\log^4 d} \gtrsim I^2 \Delta^{-2}$$

$$m\sigma_1^2 \lesssim \frac{\Delta^2 \hat{\sigma}_{2I}^2 a_{\min_*}}{I^2 2^{3I} C_\sigma^2 d^{I-1/2}}$$

$$\bar{\varepsilon} \lesssim \left( \frac{\Delta^2 \hat{\sigma}_{2I}^2}{I^2 2^{3I+4} C_\sigma^2} \right)^2 \cdot \frac{1}{d^{1+2\gamma(I-1)}}$$

$$d \gtrsim \left( \frac{\hat{\sigma}_{2I}^2 a_{\min_*} \Delta^2}{I^2 2^{3I} C_\sigma^2 \|a\|_1} \right)^{-\frac{2}{1-2\gamma I}}$$

$$\varepsilon_D \ge \frac{2^{3I+7} 3^I C_\sigma^2}{\hat{\sigma}_{2I}^2} \left\{ \bar{\varepsilon}^{1/2} \varepsilon_0^{I-1} \vee \frac{m\sigma_1^2}{a_{\min_*}} \vee \frac{\|a\|_1}{a_{\min_*}} \varepsilon_0^I \right\} \Longleftarrow \begin{cases} \bar{\varepsilon} & \le \left( \frac{\hat{\sigma}_{2I}^2}{2^{3I+7} 3^I C_\sigma^2} \right)^2 \varepsilon_D^2 d^{2(1-\gamma)(I-1)} \\ m\sigma_1^2 & \le \frac{\hat{\sigma}_{2I}^2 a_{\min_*} \varepsilon_D}{2^{3I+7} 3^I C_\sigma^2} \\ \varepsilon_D & \ge \frac{2^{3I+7} 3^I C_\sigma^2 \|a\|_1}{\hat{\sigma}_{2I}^2 a_{\min_*}} d^{-I(1-\gamma)} \end{cases}$$

$$\eta \le \frac{a_{\pi(p)} \hat{\sigma}_{2I}^2 \|a\|_1^{-2} \delta_{\mathbb{P}}}{C \log(512 I/\Delta) \log^{\bar{Q}}(md/\delta_{\mathbb{P},\xi})} \min(d^{-I} \Delta^2, 3^{-I} \varepsilon_D^2).$$

Next, Lemma D.12 requires

$$\eta \lesssim \frac{\|a\|_1^{-2}}{\log(2a_k/\sigma_0^2)} \min\left( \frac{a_{\min_*} d^{-1} \varepsilon_R}{\log^{\bar{Q}}(md/\delta_{\mathbb{P},\xi})}, \varepsilon_R^2 \delta_{\mathbb{P}} \right)$$

$$\varepsilon_R \gtrsim \log(2a_k/\sigma_0^2) \left( C_\sigma^2 a_{\pi(p)} \bar{\varepsilon} + \|a\|_1 2^{2I} \varepsilon_0^I + m\sigma_1^2 \right)$$

Next, Lemma D.19 requires the conditions on Lemma C.9 and Lemma C.10 to hold, which are

$$\varepsilon_D \ge \frac{2^{3I+7} 3^I C_\sigma^2}{\hat{\sigma}_{2I}^2} \frac{\|a\|_1}{a_{\min_*}} \frac{1}{d^{(1-\gamma)I}}, \quad \varepsilon_R \ge 12 \|a\|_1 2^{2I} d^{-(1-\gamma)I}, \quad \Delta^2 \ge \frac{CI^2 2^{3I+4} C_\sigma^2}{\hat{\sigma}_{2I}^2} \frac{\|a\|_1}{a_{\min_*}} \frac{1}{d^{1/2-\gamma I}},$$

$$m\sigma_1^2 \le \frac{\hat{\sigma}_{2I}^2 a_{\min_*}}{2^{3I+7} C_\sigma^2} \left( 3^{-I} \varepsilon_D \wedge \frac{\Delta^2}{CI^2 d^{I-1/2}} \right) \wedge \frac{\varepsilon_R}{12},$$

$$\bar{\varepsilon} \le \left( \frac{\hat{\sigma}_{2I}^2}{2^{3I+7} 3^I C_\sigma^2} \right)^2 \varepsilon_D^2 d^{2(1-\gamma)(I-1)} \wedge \left( \frac{\Delta^2 \hat{\sigma}_{2I}^2}{CI^2 2^{3I+4} C_\sigma^2} \right)^2 \frac{1}{d^{1+2\gamma(I-1)}} \wedge \frac{\varepsilon_R}{12 C_\sigma^2 a_{\pi(p)}}.$$

and

$$\bar{\varepsilon} \le \left( \frac{\hat{\sigma}_{2I}^2}{2^{3I+4} C_\sigma^2} \frac{\Delta}{24} \right)^2 \frac{1}{d^{1+2\gamma(I-1)}}, \quad m\sigma_1^2 \le \frac{\hat{\sigma}_{2I}^2}{2^{3I+4} C_\sigma^2} \frac{a_{\min_*}}{2(\log d)^{2I-2} d^{I-1/2}} \frac{\Delta}{24},$$

$$\frac{d}{(\log^2 d)^{1/\gamma}} \ge \left( \frac{\Delta}{4} \right)^{-\frac{1}{\gamma(I-1)}}, \quad \frac{d}{(\log^2 d)^{\frac{I-1}{1/2-\gamma I}}} \ge \left( \frac{\hat{\sigma}_{2I}^2}{2^{3I+4} C_\sigma^2} \frac{a_{\min_*}}{\|a\|_1 2^{2I-2}} \frac{\Delta}{24} \right)^{-\frac{1}{1/2-\gamma I}}, \quad \frac{\Delta^2}{CI^2} \le \frac{\Delta}{240}.$$

Moreover D.19 additionally requires

$$\bar{\varepsilon}^{1/2} \varepsilon_0^{I-1} \vee m\sigma_1^2 \vee \|a\|_1 \varepsilon_0^I \ll \frac{a_{\min_*} \hat{\sigma}_{2I}^2 d^{-I+1/2} \Delta}{I2^{3I+6} C_\sigma^2} \Longleftarrow \begin{cases} \bar{\varepsilon} & \lesssim \left( \frac{a_{\min_*} \hat{\sigma}_{2I}^2 \Delta}{I2^{3I+6} C_\sigma^2} \right)^2 \frac{1}{d^{1+2\gamma(I-1)}} \\ m\sigma_1^2 & \lesssim \frac{a_{\min_*} \hat{\sigma}_{2I}^2 d^{-I+1/2} \Delta}{I2^{3I+6} C_\sigma^2} \\ d & \gtrsim \left( \frac{a_{\min_*} \hat{\sigma}_{2I}^2 \Delta}{I2^{3I+6} C_\sigma^2 \|a\|_1} \right)^{-\frac{2}{1-2I\gamma}} \end{cases}$$

$$\varepsilon_R \lesssim \frac{a_{\min_*}}{1.5^I d^{\gamma(I-1)}}$$

$$\frac{d}{\log^{2/\gamma} d} \geq 2^{1/\gamma} (4/\Delta)^{\frac{1}{\gamma(I-1)}}$$

Finally, Lemma D.22 requires

$$d \gg \left( \frac{\|\boldsymbol{a}\|_1}{\hat{\sigma}_{2I}^2 a_{\min_*}} \right)^{\frac{1}{1-I\gamma}}$$

$$\eta \lesssim \frac{a_{\min_*} I(I-1) \hat{\sigma}_{2I}^2 d^{-I} \|\boldsymbol{a}\|_1^{-2}}{\log^{\tilde{Q}}(md/\delta_{\mathbb{P},\xi})}$$

$$\sigma_1^2 \gtrsim \sigma_0^2 \bar{\varepsilon}^{-C_{\exp}}.$$

Assume that $\frac{d}{\log^{8I} d} \geq 2^{4I} (4/\Delta)^{\frac{4I}{I-1}}$. Then by choosing $\gamma$ to be the solution to $\frac{d}{\log^{2/\gamma}} = 2^{1/\gamma}(4/\Delta)^{\frac{1}{\gamma(I-1)}}$, we know that $\gamma \leq \frac{1}{4I}$. The constraints on $d$ then become:

$$d \gtrsim_\sigma \Delta^{-4} \vee \log^4 d \Delta^{-2} \vee \left( \|\boldsymbol{a}\|_1 \Delta^{-2} a_{\min_*}^{-1} \right)^4 \vee \log^{8(I-1)}(d) \left( \|\boldsymbol{a}\|_1 \Delta^{-1} a_{\min_*}^{-1} \right)^4$$

$$\Longleftarrow \frac{d}{\log^{8I} d} \gtrsim_\sigma \|\boldsymbol{a}\|_1^4 \Delta^{-8} a_{\min_*}^{-4}.$$

The conditions on the target accuracies $\varepsilon_R, \varepsilon_D$ become

$$\varepsilon_D \gtrsim_\sigma \frac{\|\boldsymbol{a}\|_1}{a_{\min_*}} \frac{1}{d^{I-1/4}}$$

$$\varepsilon_R \gtrsim_\sigma \frac{\|\boldsymbol{a}\|_1}{d^{I-1/4}}$$

$$\varepsilon_R \lesssim_\sigma \frac{a_{\min_*}}{d^{\gamma(I-1)}} =_\sigma \frac{a_{\min_*} \Delta}{\log^{2(I-1)} d}$$

Next, the constraints on $\bar{\varepsilon}$ become (substituting $d^\gamma = 2 \log^2 d (4/\Delta)^{\frac{1}{I-1}}$):

$$\bar{\varepsilon} \lesssim_\sigma \frac{\Delta^6}{d \log^{4(I-1)} d} \wedge \frac{\varepsilon_D^2 d^{2(I-1)}}{\log^{4(I-1)} d} \Delta^2 \wedge \frac{\varepsilon_R}{\log(1/\sigma_0^2)},$$

where we note we must also have $\bar{\varepsilon} \geq \varepsilon_D$. We can therefore choose $\bar{\varepsilon} = \varepsilon_D$, and observe that the conditions become

$$\frac{\Delta^6}{d \log^{4(I-1)} d} \gtrsim_\sigma \varepsilon_D \gtrsim_\sigma \frac{\|\boldsymbol{a}\|_1}{a_{\min_*} d^{I-1/4}}$$

$$\frac{a_{\min_*} \Delta}{\log^{2(I-1)d}} \gtrsim_\sigma \varepsilon_R \gtrsim_\sigma \varepsilon_D \log(1/\sigma_0^2) \vee \frac{\|\boldsymbol{a}\|_1}{d^{I-1/4}}$$

The condition on $m\sigma_1^2$ becomes

$$m\sigma_1^2 \lesssim_\sigma \frac{a_{\min_*} \Delta^2}{d^{I-1/2}} \wedge a_{\min_*} \varepsilon_D \wedge \frac{\varepsilon_R}{\log(1/\sigma_0^2)} \wedge \frac{a_{\min_*} \Delta}{d^{I-1/2} \log^{2I-2} d}.$$

We additionally require $\sigma_0^2 \lesssim_\sigma \sigma_1^2 \varepsilon_D^{C_{\exp}}$. Therefore it suffices to pick $\sigma_0 = d^{-C}, \sigma_1 = d^{-C'}$, where $C > C' > 0$ are sufficiently large constants depending only on $I, \sigma$.

Next, we choose the learning rate $\eta$. It suffices to set $\eta$ as

$$\eta \lesssim_\sigma \frac{a_{\min_*} \|\boldsymbol{a}\|_1^{-2} m^{-1} P^{-1} \delta_{\mathbb{P}}^*}{\log(512I/\Delta) \log^{\tilde{Q}} \left( \frac{md}{\delta_{\mathbb{P},\xi}} \right)} \min(\Delta^2 d^{-I}, \varepsilon_D^2).$$

Finally, we prove part (d), and bound the population loss $\mathcal{L}$ at time $t$. Recall that $\mathcal{L} = \sum_{i \geq I} \hat{\sigma}_{2i}^2 \mathcal{L}_i$, where

$$\mathcal{L}_i := \frac{1}{2} \|\boldsymbol{a}\|^2 - \sum_{p=1}^{P} \sum_{k=1}^{m} a_p \|\boldsymbol{v}_k\|^2 \bar{v}_{k,p}^{2i} + \frac{1}{2} \sum_{k,l=1}^{m} \|\boldsymbol{v}_k\|^2 \|\boldsymbol{v}_l\|^2 \langle \bar{\boldsymbol{v}}_k, \bar{\boldsymbol{v}}_l \rangle^{2i}.$$

Recall that $L := \{p \in [P] : \|\boldsymbol{v}_p\| \geq \sigma_1\}$. By parts (b) and (c), we must have $L = [k_*]$ for some integer $k_*$, and $\bar{v}_{p,\pi(p)}^2 \geq 1 - \bar{\varepsilon}$ for $\boldsymbol{v}_p \in L$. We can decompose the loss as follows:

$$\mathcal{L}_i = \frac{1}{2} \|\boldsymbol{a}\|^2 - \sum_{k \in L} \sum_{p \in [P]} a_p \|\boldsymbol{v}_k\|^2 \bar{v}_{k,p}^{2i} + \frac{1}{2} \sum_{k \in L} \|\boldsymbol{v}_k\|^4$$

$$+ \sum_{k,j \in L, k \neq j} \|\boldsymbol{v}_k\|^2 \|\boldsymbol{v}_j\|^2 \langle \bar{\boldsymbol{v}}_k, \bar{\boldsymbol{v}}_j \rangle^{2i} - \sum_{k \notin L} \sum_{p \in [P]} a_p \|\boldsymbol{v}_k\|^2 \bar{v}_{k,p}^{2i} + \frac{1}{2} \sum_{k \notin L} \sum_{j=1}^{m} \|\boldsymbol{v}_k\|^2 \|\boldsymbol{v}_j\|^2 \langle \bar{\boldsymbol{v}}_k, \bar{\boldsymbol{v}}_j \rangle^{2i}$$

The terms with $k \notin L$ are straightforward to bound, as

$$\sum_{k \notin L} \sum_{p \in [P]} a_p \|\boldsymbol{v}_k\|^2 \bar{v}_{k,p}^{2i} \leq m\sigma_1^2 \|\boldsymbol{a}\|_1$$

$$\frac{1}{2} \sum_{k \notin L} \sum_{j=1}^{m} \|\boldsymbol{v}_k\|^2 \|\boldsymbol{v}_j\|^2 \langle \bar{\boldsymbol{v}}_k, \bar{\boldsymbol{v}}_j \rangle^{2i} \leq \frac{1}{2} m\sigma_1^2 \sum_{j=1}^{m} \|\boldsymbol{v}_j\|^2 \leq m\sigma_1^2 \|\boldsymbol{a}\|_1.$$

Next, for $k \neq j \in L$, $\langle \bar{\boldsymbol{v}}_k, \bar{\boldsymbol{v}}_j \rangle^{2i} \leq \bar{\varepsilon}^i$, and thus

$$\sum_{k,j \in L, k \neq j} \|\boldsymbol{v}_k\|^2 \|\boldsymbol{v}_j\|^2 \langle \bar{\boldsymbol{v}}_k, \bar{\boldsymbol{v}}_j \rangle^{2i} \leq \bar{\varepsilon}^i \left( \sum_{k \in [m]} \|\boldsymbol{v}_k\|^2 \right)^2 \leq 4 \|\boldsymbol{a}\|_1^2 \bar{\varepsilon}^i.$$

Finally, we track the dominant loss term. We have

$$\frac{1}{2} \|\boldsymbol{a}\|^2 - \sum_{k \in L} \sum_{p \in [P]} a_p \|\boldsymbol{v}_k\|^2 \bar{v}_{k,p}^{2i} + \frac{1}{2} \sum_{k \in L} \|\boldsymbol{v}_k\|^4$$

$$= \frac{1}{2} \sum_{k \notin L} a_{\pi(k)}^2 + \frac{1}{2} \sum_{k \in L} \left( a_{\pi(k)}^2 - \|\boldsymbol{v}_k\|^2 \sum_{p \in P} a_p \bar{v}_{k,p}^{2i} + \|\boldsymbol{v}_k\|^4 \right)$$

We can bound

$$\|\boldsymbol{v}_k\|^2 \sum_{p \neq \pi(k)} a_p \bar{v}_{k,p}^{2i} \leq \|\boldsymbol{v}_k\|^2 \sum_{p \neq \pi(k)} a_p \bar{\varepsilon}^i \leq \bar{\varepsilon}^i \|\boldsymbol{v}_k\|^2 \|\boldsymbol{a}\|_1.$$

Moreover, $1 - \bar{v}_{k,p}^{2i} \leq 2i\bar{\varepsilon}$. Altogether,

$$\mathcal{L}_i = \frac{1}{2} \sum_{k \notin L} a_{\pi(k)}^2 + \frac{1}{2} \sum_{k \in L} \left( a_{\pi(k)} - \|\boldsymbol{v}_k\|^2 \right)^2 \pm O(\bar{\varepsilon}),$$

and since $\sum_l \hat{\sigma}_{2l}^2 = 1$, we have

$$\mathcal{L} = \frac{1}{2} \sum_{k \notin L} a_{\pi(k)}^2 + \frac{1}{2} \sum_{k \in L} \left( a_{\pi(k)} - \|\boldsymbol{v}_k\|^2 \right)^2 \pm O(\bar{\varepsilon})$$

as well. Next, if $t \leq (1 - \Delta/4)T_p$, then $p \notin L$, and thus

$$\mathcal{L} \geq \frac{1}{2} \sum_{k \notin L} a_{\pi(k)}^2 - O(\bar{\varepsilon}) \geq \frac{1}{2} - \frac{1}{2} \sum_{p \in P_*} a_{\pi(p)}^2 \cdot \mathbb{1}\left( t \geq (1 - \Delta/4)T_p \right) - O(\bar{\varepsilon}).$$

On the other hand, if $t \geq (1 + \Delta/4)T_p$, then $p \in L$ and $\left| a_{\pi(p)} - \|\boldsymbol{v}_p^2\| \right| \leq \varepsilon_R$, and thus

$$\mathcal{L} \leq \frac{1}{2} - \frac{1}{2} \sum_{p \in P_*} a_{\pi(p)}^2 \cdot \mathbb{1}\left( t \geq (1 + \Delta/4)T_p \right) + O(P_* \varepsilon_R^2 + \bar{\varepsilon}),$$

where the desired claim follows by additionally choosing $\varepsilon_R^2 \leq P_*^{-1} \varepsilon_D$. $\qquad\square$

## D.5 Deferred Proofs

*Proof of Lemma D.4.* Assume WLOG that the bounds on $X_t$ always hold. Inductively unroll the recursion as

$$X_t = X_0 P_{0,t} + \sum_{s=1}^{t} P_{s,t} \xi_s + \sum_{s=1}^{t} P_{s,t} Z_s,$$

where $P_{s,t} := \prod_{r=s}^{t-1} (1 + \alpha X_r^{I-1}) \geq 1$. As such,

$$P_{0,t}^{-1} X_t = X_0 + \sum_{s=1}^{t} P_{0,s}^{-1} \xi_s + \sum_{s=1}^{t} P_{0,s}^{-1} Z_s.$$

The error term gets bounded as

$$\left| \sum_{s=1}^{t} P_{0,s}^{-1} \xi_s \right| \leq \sum_{s=1}^{t} |\xi_s| \leq \Xi_1 \sum_{t=0}^{T-1} (x_t^+)^I + \Xi_2 \sum_{t=0}^{T-1} x_t^+ + T\Xi_3$$

with high probability for all $t$. We can bound each term by $x_0 \varepsilon / 6$. The martingale term can be controlled by Doob's inequality,

$$\mathbb{P} \left[ \sup_{r \leq t} \left| \sum_{s=1}^{t} P_{0,s}^{-1} Z_s \right| \geq M \right] \leq M^{-2} \sum_{s=1}^{t} \mathbb{E}[Z_s^2] \leq M^{-2} \sigma_Z^2 \sum_{t=0}^{T-1} x_t^+ \leq \delta_{\mathbb{P}},$$

when we take $M = x_0 \varepsilon / 2$. Altogether, we have that $P_{0,t}^{-1} X_t \geq X_0 - x_0 \varepsilon$, and thus

$$X_t \geq P_{0,t}(1 - \varepsilon) x_0 = \prod_{s=1}^{t-1} (1 + \alpha X_r^{I-1}) x_0^- \geq \prod_{s=1}^{t-1} \left( 1 + \alpha \left( x_r^- \right)^{I-1} \right) x_0^- = x_t^-.$$

Similarly, we have $P_{0,t}^{-1} X_t \geq X_0 + x_0 \varepsilon$, and thus

$$X_t \leq P_{0,t}(1 + \epsilon) x_0 = \prod_{s=1}^{t-1} (1 + \alpha X_r^{I-1}) x_0^+ \leq \prod_{s=1}^{t-1} \left( 1 + \alpha \left( x_r^+ \right)^{I-1} \right) x_0^+ = x_t^+,$$

as desired. $\qquad \square$

*Proof of Lemma D.6.* Assume that the bounds on $X_t$ always hold. If $\sup_{s \leq t} X_s > \delta$ then we are done; otherwise, unroll the recursion as

$$X_t = X_0 P_{0,t} + \sum_{s=1}^{t} P_{s,t} Z_s + \sum_{s=1}^{t} P_{s,t} \xi_s,$$

where $P_{s,t} := \prod_{r=s}^{t-1} (1 + \alpha X_r^{I-1}) \geq 1$. As such,

$$P_{0,t}^{-1} X_t = X_0 + \sum_{s=1}^{t} P_{0,s}^{-1} \xi_s + \sum_{s=1}^{t} P_{0,s}^{-1} Z_s.$$

The error term is bounded as

$$\left| \sum_{s=1}^{t} P_{0,s}^{-1} \xi_s \right| \leq \sum_{s=1}^{t} |\xi_s| \leq \Xi T \leq \frac{x_0}{4}$$

for high probability for all $t \leq T$. Next, we bound the martingale term by Doob's inequality:

$$\mathbb{P} \left[ \sup_{r \leq t} \left| \sum_{s=1}^{t} P_{0,s}^{-1} Z_s \right| \geq M \right] \leq M^{-2} \sum_{s=1}^{t} \mathbb{E}[Z_s^2] \leq M^{-2} \sigma_Z^2 T \leq \delta_{\mathbb{P}},$$

when we take $M = x_0 / 4$. Altogether,

$$X_t \geq P_{0,t} x_0 / 2 \geq \hat{x}_t,$$

as desired. $\qquad \square$

*Proof of Lemma D.9.* Expanding the recursion,

$$X_t \leq (1-\alpha)^t X_0 + \sum_{s=0}^{t-1}(1-\alpha)^s \xi_{t-s} + \sum_{s=0}^{t-1}(1-\alpha)^s Z_{t-s}.$$

We can bound the error term by

$$\left|\sum_{s=0}^{t-1}(1-\alpha)^s \xi_{t-s}\right| \leq \Xi \sum_{s=0}^{t-1}(1-\alpha)^s \leq \Xi\alpha^{-1} \leq \frac{\varepsilon}{4}$$

and by Doob's inequality bound the martingale by

$$\mathbb{P}\left[\sup_{r\leq t}\left|\sum_{s=0}^{t-1}(1-\alpha)^s Z_{t-s}\right| \geq M\right] \leq M^{-2}\sum_{s=0}^{t-1}(1-\alpha)^{-2s}\mathbb{E}[Z_{t-s}^2] \leq M^{-2}\sigma_Z^2\alpha^{-1} \leq \delta_{\mathbb{P}},$$

since we take $M = \varepsilon/4$. Therefore

$$X_t \leq (1-\alpha)^t X_0 + \varepsilon/2 \leq (1-\alpha)^t x_0 + \varepsilon/2.$$

$\square$

*Proof of Lemma D.13.* Expanding the recursion,

$$X_t = (1+\alpha)^t X_0 + \sum_{s=0}^{t-1}(1+\alpha)^s \xi_{t-s} + \sum_{s=0}^{t-1}(1+\alpha)^s Z_{t-s}$$

$$\implies (1+\alpha)^{-t} X_t = X_0 + \sum_{s=1}^{t}(1+\alpha)^{-s}\xi_s + \sum_{s=1}^{t}(1+\alpha)^{-s}Z_s.$$

We can bound the error term by

$$\left|\sum_{s=1}^{t}(1+\alpha)^{-s}\xi_s\right| \leq \Xi\sum_{s=1}^{t}(1+\alpha)^{-s}\cdot(1+\alpha)^s x_0 = \Xi T x_0 \leq \frac{x_0}{4}.$$

By Doob's inequality, we can bound the martingale term by

$$\mathbb{P}\left[\sup_{t\leq T}\left|\sum_{s=0}^{t-1}(1+\alpha)^{-s}Z_s\right| \geq M\right] \leq M^{-2}\sigma_Z^2\sum_{s=0}^{t-1}(1+\alpha)^{-2s}\cdot(1+\alpha)^{2s}x_0^2 = M^{-2}\sigma_Z^2 T x_0^2 \leq \delta_{\mathbb{P}},$$

since we chose $M = x_0/4$. Altogether,

$$(1+\alpha)^{-t}X_t = x_0 \pm 0.5x_0 \implies X_t = (1\pm0.5)x_t,$$

as desired. $\square$

*Proof of Lemma D.15.* Define $P_{s,t} := \prod_{r=s}^{t-1}(1+\alpha(X_r))$. Expanding the recursion,

$$X_t = P_{0,t}X_0 + \sum_{s=0}^{t-1}P_{t-s,t}\xi_{t-s} + \sum_{s=0}^{t-1}P_{t-s,t}Z_{t-s}.$$

We can bound the error term by

$$\left|\sum_{s=0}^{t-1}P_{t-s,t}\xi_{t-s}\right| \leq \Xi\sum_{s=0}^{t-1}(1-\alpha_-)^s \leq \Xi\alpha_-^{-1} \leq \varepsilon/4.$$

By Doob's inequality, we can bound the martingale term by

$$\mathbb{P}\left[\sup_{t\leq T}\left|\sum_{s=0}^{t-1}P_{t-s,t}Z_{t-s}\right| \geq M\right] \leq M^{-2}\sigma_Z^2\sum_{s=0}^{t-1}(1-\alpha_-)^{2s} \leq M^{-2}\sigma_Z^2\alpha_- \leq \delta_{\mathbb{P}},$$

since we chose $M = \varepsilon/4$. Therefore

$$X_t \leq P_{0,t}X_0 + \varepsilon/2 \leq (1-\alpha_-)^t x_0 + \varepsilon/2$$
$$X_t \geq P_{0,t}X_0 - \varepsilon/2 \geq (1-\alpha_+)^t x_0 - \varepsilon/2,$$

as desired. $\square$

*Proof of Lemma D.18.* Assume WLOG that the bounds on $X_t$ always hold. $(X_t)_t$ is stochastically dominated by the process where $X_{t+1} = X_t + \alpha X_t^I + \xi_{t+1} + Z_{t+1}$, so we can WLOG track this latter process. Expanding out the recursion, we get that

$$X_t = X_0 P_{0,t} + \sum_{s=1}^{t} P_{s,t}(\xi_s + Z_s),$$

where $P_{s,t} := \prod_{r=s}^{t-1}(1 + \alpha X_r^{I-1})$. Since $X_r \geq 0$, $P_{0,s} \geq 1$ and thus

$$P_{0,t}^{-1} X_t = X_0 + \sum_{s=1}^{t} P_{0,s}^{-1} \xi_s + \sum_{s=1}^{t} P_{0,s}^{-1} Z_s.$$

The error term gets bounded as

$$\left| \sum_{s=1}^{t} P_{0,s}^{-1} \xi_s \right| \leq \sum_{s=1}^{t} |\xi_s| \leq \Xi_1 \sum_{t=0}^{T-1} \hat{x}_t^{1/2} + \Xi_2 \sum_{t=0}^{T-1} \hat{x}_t + \Xi_3 \sum_{t=0}^{T-1} \hat{x}_t^I + T\Xi_4$$

with high probability for all $t$. We can bound each term by $x_0 \varepsilon / 8$. The martingale term can be controlled by Doob's inequality:

$$\mathbb{P}\left[ \sup_{r \leq t} \left| \sum_{s=1}^{t} P_{0,s}^{-1} Z_s \right| \geq M \right] \leq M^{-2} \sum_{s=1}^{T} \mathbb{E}[Z_s^2] \leq M^{-2} \sigma_Z^2 \sum_{t=0}^{T-1} \hat{x}_t \leq \delta_{\mathbb{P}},$$

when we take $M = x_0 \epsilon / 2$. Altogether, we get

$$X_t \leq P_{0,t} x_0 (1 + \epsilon) = P_{0,t} \hat{x}_0 \leq \hat{x}_t,$$

as desired. $\qquad\square$

*Proof of Lemma D.21.* We may assume w.l.o.g. that the bounds on $\xi_t$ and the conditional variance of $Z_{t+1}$ always hold. Define

$$P_{s,t}(X) := \begin{cases} \prod_{r=s}^{t-1}(1 + \alpha_r(X_r)), & t > s, \\ 1, & t = s. \end{cases}$$

Note that since $\alpha_r > 0$, we have $P_{s,t} \geq 1$. Then, we can unroll the recurrence relationship as

$$X_t = X_0 P_0(X) + \sum_{s=1}^{t} P_{s,t}(X) \left( \xi_{s-1} + Z_{s-1} \right).$$

Divide both sides with $P_{0,t}$, and we obtain

$$P_{0,t}^{-1}(X) X_0 = X_0 + \sum_{s=1}^{t} P_{0,s}^{-1}(X) \xi_{s-1} + \sum_{s=1}^{t} P_{0,s}^{-1}(X) Z_{s-1}.$$

For the second term, we have

$$\left| \sum_{s=1}^{t} P_{0,s}^{-1}(X) \xi_{s-1} \right| \leq \sum_{s=1}^{t} |\xi_{s-1}| \leq T\Xi,$$

for all $t \leq T$ with probability at least $1 - T\delta_{\mathbb{P},\xi}$. For the RHS to be bounded by $\varepsilon x_0 / 2$, it suffices to choose $\Xi \leq \varepsilon_0 x_0 / (2T)$. Meanwhile, by Doob's submartingale inequality, for any $M > 0$, we have

$$\mathbb{P}\left[ \sup_{r \leq t} \left| \sum_{s=1}^{t} P_{0,s}^{-1} Z_{s-1} \right| \geq M \right] \leq M^{-2} \sum_{s=1}^{t} \mathbb{E}\left[ P_{0,s}^{-2} Z_{s-1}^2 \right] \leq \frac{\sigma_Z^2 T}{M^2}.$$

Choose $M = \varepsilon x_0 / 2$. Then, the RHS becomes $\frac{4\sigma_Z^2 T}{\varepsilon^2 x_0}$. For it to be bounded by $\delta_{\mathbb{P}}$, we need $\sigma_Z^2 \leq \varepsilon^2 x_0^2 \delta_{\mathbb{P}} / (4T)$. The above two results imply that with the conditions on $\xi$ and $Z$ stated in the lemma, we have, with probability at least $1 - \delta_{\mathbb{P}} - T\delta_{\mathbb{P},\xi}$, that

$$X_t = P_{0,t}(X)(1 \pm \varepsilon) x_0 \leq P_{0,t}(X) \hat{X}_0 \leq P_{0,t}(\hat{X}) \hat{X}_0 \leq \hat{X}_t,$$

where the second inequality comes from the monotonicity of $x \mapsto \alpha_t(x)$. $\qquad\square$

**Lemma D.23.** *Let $(x_t)_t \in [0, 1]$ follow the update*

$$\hat{x}_{t+1} = \hat{x}_t + \alpha \hat{x}_t^I.$$

*Then*

$$\hat{x}_t \leq \frac{\hat{x}_0}{\left(1 - \alpha(I - 1)\hat{x}_0^{I-1}t\right)^{\frac{1}{I-1}}}.$$

*Proof.* Define the continuous time process $x(t)$ be the ODE $\dot{x}(t) = \alpha x(t)^I$ with initial condition $\hat{x}_0 = x(0)$. We prove by induction that $\hat{x}_t \leq x(t)$. Observe that both processes are monotonically increasing. Therefore

$$\hat{x}_{t+1} = \hat{x}_t + \alpha \hat{x}_t^I \leq x(t) + \alpha x(t)^I \leq x(t) + \int_t^{t+1} \alpha x(s)^I \, ds = x(t+1).$$

The desired result is obtained by solving the ODE for $x(t)$ with initial condition $x(0) = x_0$. $\qquad\square$

**Lemma D.24.** *Let $(x_t)_t \in [0, 1]$ follow the update*

$$x_{t+1} = x_t + \alpha x_t^I.$$

*Then*

$$x_t \geq \frac{x_0}{\left(1 - \alpha(I - 1)\exp(-\alpha I)x_0^{I-1}t\right)^{\frac{1}{I-1}}}.$$

*Proof.* We have that

$$
\begin{aligned}
\alpha &= \frac{x_t - x_{t-1}}{(x_{t-1})^I} \\
&= \frac{(x_t)^I}{(x_{t-1})^I} \cdot \frac{x_t - x_{t-1}}{(x_t)^2} \\
&\leq \frac{(x_t)^I}{(x_{t-1})^I} \int_{x_{t-1}}^{x_t} \frac{1}{x^I} dx \\
&= \frac{(x_t)^I}{(I-1)(x_{t-1})^I}\left(\frac{1}{x_{t-1}^{I-1}} - \frac{1}{x_t^{I-1}}\right) \\
&= (I-1)^{-1}(1 + \alpha x_{t-1}^{(I-1)})^I\left(\frac{1}{x_{t-1}^{I-1}} - \frac{1}{x_t^{I-1}}\right) \\
&\leq (I-1)^{-1}\exp(\alpha I)\left(\frac{1}{x_{t-1}^{I-1}} - \frac{1}{x_t^{I-1}}\right).
\end{aligned}
$$

Therefore

$$\frac{1}{x_t^{I-1}} \leq \frac{1}{x_{t-1}^{I-1}} - \alpha(I-1)\exp(-\alpha I),$$

so summing and solving for $x_t$ yields

$$x_t \geq \frac{x_0}{\left(1 - \alpha(I - 1)\exp(-\alpha I)x_0^{I-1}t\right)^{\frac{1}{I-1}}}.$$

$\qquad\square$

# E Scaling Law Derivations

We have shown that direction $e_{\pi(p)}$ will be learned at time $(1 \pm o(1))T_p$ where $T_p$ is defined by

$$T_p := \left(4I(I-1)\hat{\sigma}_{2I}^2 a_{\pi(p)} \eta \bar{v}_{p,\pi(p)}^2(0)\right)^{-1}.$$

Suppose that the signal follows the power law $a_p = p^{-\beta}/Z$ where $\beta > 1/2$ and $Z = \sum_{p=1}^{P} p^\beta$ is the normalizing constant. In Section 3.1, we informally derive the scaling law $\mathcal{L}(t) \propto t^{-(2\beta-1)/\beta}$. In this section, we prove that this is true up to a multiplicative constant (cf. Corollary 2.2).

To this end, it suffices to (1) argue that teacher neurons $p$ with large signal strength $a_p$ are likely to lie in the set of learned neurons $\{\pi(p) : p \in [P_*]\}$, and (2) bound the fluctuations of $\bar{v}_{p,\pi(p)}^2(0)$. A lower bound on the fluctuations is given in Lemma B.2(d). The following lemma shows that neurons with large signal strength do indeed get learned.

**Lemma E.1.** *Assume that $a_p \propto p^{-\beta}$ for $\beta > 1/2$. Let $\delta_\mathbb{P} = 1/\text{poly}(m)$ be the target failure probability. Then there exists a universal constant $C$ so that, with probability $1 - \delta_\mathbb{P}$, all teacher neurons $q$ satisfying $a_q \geq Ca_{P_*}$ lie in the set of learned neurons, i.e $q \in \{\pi(p) : p \in [P_*]\}$.*

*Proof.* Let $z_1, \ldots, z_m$ be independent $\mathcal{N}(0, I_d)$ variables. We remark that $\{\bar{v}_i\}_{i\in[m]}$ is equal in distribution to $\{z_i/\|z_i\|\}_{i\in[m]}$. First, with probability $1 - 2m\exp(-Cd)$, we have that $\|z_i\|^2 = (1 \pm 0.5)d$ for all $i \in [m]$. Moreover, $\mathbb{P}(\max_{k\in[m],p\in[P_*]} |z_{k,p}| \geq z) \leq 2mP_* e^{-z^2/2}$, and therefore $\max_{k\in[m],p\in[P_*]} z_{k,p}^2 \leq 2\log(2mP_*/\delta_\mathbb{P})$ with probability $1 - \delta_\mathbb{P}$. Let us condition on these two events.

Let $\gamma \geq 1$ be some threshold. We begin by computing $\mathbb{P}(\max_{k\in[m],p>P_*} a_p Z_{k,p}^2 \geq a_{P_*}\gamma)$. By standard Gaussian tail bounds and a union bound, we have that

$$\mathbb{P}\left(\max_{k\in[m],p>P_*} a_p Z_{k,p}^2 \geq a_{P_*}\gamma\right) \leq \sum_{p>P_*} 2m\exp\left(-\frac{a_{P_*}\gamma}{2a_p}\right)$$

Substituting $a_p = p^{-\beta}/Z$ for $\beta > \frac{1}{2}$, we get that

$$\sum_{p>P_*} \exp\left(-\frac{a_{P_*}\gamma}{2a_p}\right) = \sum_{p>P_*} \exp\left(-\frac{\gamma}{2}\left(\frac{p}{P_*}\right)^\beta\right) \leq \int_{P_*}^\infty \exp\left(-\frac{\gamma}{2}\left(\frac{p}{P_*}\right)^{1/2}\right) dp$$

$$= \frac{4\sqrt{P_*}}{\gamma}\exp(-\gamma/2)\sqrt{P_*} + \frac{8P_*}{\gamma^2}\exp(-\gamma/2) \leq 12P_*\exp(-\gamma/2).$$

Therefore

$$\mathbb{P}\left(\max_{k\in[m],p>P_*} a_p Z_{k,p}^2 \geq a_{P_*}\gamma\right) \leq 24P_* m\exp(-\gamma/2) \leq \delta_\mathbb{P}$$

for $\gamma = 2\log(24mP_*/\delta_\mathbb{P})$.

Next, we aim to upper bound the quantity $a_{\pi(P_*)}\bar{v}_{P_*,\pi(P_*)}^2$. The first case is when $\{\pi(p) : p \in [P_*]\} = [P_*]$. Since $\max_{k\in[m],p\in[P_*]} z_{k,p}^2 \leq 2\log(2mP_*/\delta_\mathbb{P})$, it is clear that $a_{\pi(P_*)}\bar{v}_{P_*,\pi(P_*)}^2 \leq 4a_{P_*}\log(2mP_*/\delta_\mathbb{P})/d$. Otherwise, there exists some $q \in [P_*]$ such that $\pi(q) > P_*$. We then have that $a_{\pi(P_*)}\bar{v}_{P_*,\pi(P_*)}^2 \leq a_{\pi(q)}\bar{v}_{q,\pi(q)}^2 \leq 2a_{P_*}\gamma/d = 4a_{P_*}\log(24mP_*/\delta_\mathbb{P})/d$.

Let $e_q$ be some teacher neuron which was not selected by the greedy maximum selection process, i.e $q \notin \{\pi(p) : p \in [P_*]\}$. Then we must have $a_q \bar{v}_{p,q}^2 \leq a_{\pi(P_*)}\bar{v}_{P_*,\pi(P_*)}^2$ for all $p > P_*$. Therefore

$$\mathbb{P}(q \notin \{\pi(p) : p \in [P_*]\}) \leq \mathbb{P}\left(\cup_{p>P_*} a_q \bar{v}_{p,q}^2 \leq a_{\pi(P_*)}\bar{v}_{P_*,\pi(P_*)}^2\right)$$

$$\leq \mathbb{P}\left(\cup_{p>P_*} z_{p,q}^2 \leq \frac{6a_{P_*}}{a_q}\log(24mP_*/\delta_\mathbb{P})\right).$$

For $\gamma > 1$, one can bound $\mathbb{P}(Z_i \geq \gamma) \geq \frac{1}{\sqrt{2\pi}}\frac{z}{1+z^2}e^{-z^2/2} \geq \frac{1}{\sqrt{2\pi}}e^{-3z^2/2}$. Therefore

$$\mathbb{P}(q \notin \{\pi(p) : p \in [P_*]\}) \leq \left(1 - \frac{1}{\sqrt{2\pi}}\exp\left(-\frac{9a_{P_*}}{a_q}\log(24mP_*/\delta_\mathbb{P})\right)\right)^{m-P_*}$$

$$\leq \left(1 - \frac{1}{\sqrt{2\pi}} \left(\frac{24mP_*}{\delta_{\mathbb{P}}}\right)^{-\frac{9a_{P_*}}{a_q}}\right)^{m/2}$$

$$\leq \exp\left(-\frac{m}{2\sqrt{2\pi}} \left(\frac{24mP_*}{\delta_{\mathbb{P}}}\right)^{-\frac{9a_{P_*}}{a_q}}\right).$$

If $a_q$ satisfies

$$a_q \geq a_{P_*} \cdot \frac{9\log(24mP_*/\delta_{\mathbb{P}})}{\log(\frac{m}{2\sqrt{2\pi}}) - \log\log(P/\delta_{\mathbb{P}})},$$

then plugging in we obtain $\mathbb{P}(q \notin \{\pi(p) : p \in [P_*]\}) \leq \delta_{\mathbb{P}}/P$. Finally, since $P_* \leq m$, for $\delta_{\mathbb{P}} = 1/\text{poly}(m)$ we can upper bound $\frac{9\log(24mP_*/\delta_{\mathbb{P}})}{\log(\frac{m}{2\sqrt{2\pi}}) - \log\log(P/\delta_{\mathbb{P}})} \leq C$ for some universal constant $C$. Union bounding over all $q$ yields the desired result. $\qquad\square$

Now, we are ready to prove our main theorem on the scaling law.

*Proof of Proposition 2.2.* By Theorem 2.1, we know that with probability at least $1 - o(1)$, we have

$$1 - \sum_{p \in [P_*]} a_{\pi(p)}^2 \mathbb{1}\left(t \geq (1 - \Delta/4)T_p\right) - O(\varepsilon_D) \leq \mathcal{L}(t) \leq 1 - \sum_{p \in [P_*]} a_{\pi(p)}^2 \mathbb{1}\left(t \geq (1 + \Delta/4)T_p\right) + O(\varepsilon_D).$$

It suffices to estimate the LHS and RHS. For the RHS, by Lemma E.1 we have that $\{q : a_q \geq Ca_{P_*}\} \subset \{\pi(p) : p \in [P_*]\}$, and by Lemma B.2 we have $\min_{p \in P_*} \bar{v}_{p,\pi(p)}^2 \geq (\log P_*)/d$, and thus

$$\sum_{p \in [P_*]} a_{\pi(p)}^2 \mathbb{1}\left(t \geq \frac{1 + o(1)}{4I(I-1)\hat{\sigma}_{2I}^2 \eta a_{\pi(p)} \bar{v}_{p,\pi(p)}^{2I-2}(0)}\right) \geq \sum_{p \in [P_*]} a_{\pi(p)}^2 \mathbb{1}\left(t \geq \frac{\tilde{C}d^{I-1}}{\eta a_{\pi(p)} \log^{2I-2} P_*}\right)$$

$$\geq \sum_{p=1}^{P_* C^{-1/\beta}} a_p^2 \mathbb{1}\left(t \geq \frac{\tilde{C}d^{I-1}}{\eta a_p \log^{2I-2} P_*}\right).$$

Therefore, letting $K = \eta Z^{-1}\tilde{C}^{-1} \log^{2I-2} P_*$, we have

$$\text{RHS}(t) \leq \frac{1 + o(1)}{2Z^2} \sum_{p=1}^{P} p^{-2\beta} \mathbb{1}\left\{t \geq \frac{d^{I-1}}{Kp^{-\beta}} \vee p \geq P_* C^{-1/\beta}\right\} + O(\varepsilon_D)$$

$$\leq \frac{1 + o(1)}{2Z^2} \sum_{p=1}^{P} p^{-2\beta} \mathbb{1}\left\{p \geq \left(Kt/d^{I-1}\right)^{1/\beta} \wedge P_* C^{-1/\beta}\right\} + O(\varepsilon_D)$$

$$\leq \frac{1 + o(1)}{2Z^2} \left[\left(\frac{Kt}{d^{I-1}}\right)^{-2} + P_*^{-2\beta} C^2\right] + \frac{1 + o(1)}{2Z^2} \int_{(Kt/d^{I-1})^{1/\beta} \wedge P_* C^{-1/\beta}}^{\infty} q^{-2\beta} \, \mathrm{d}q + O(\varepsilon_D)$$

$$\leq \frac{1 + o(1)}{2Z^2} \left[\left(\frac{Kt}{d^{I-1}}\right)^{-2} + P_*^{-2\beta} C^2\right] + \frac{1 + o(1)}{2Z^2} \frac{1}{2\beta - 1} \left[\left(\frac{Kt}{d^{I-1}}\right)^{-(2\beta-1)/\beta} \vee P_*^{-(2\beta-1)} C^{\frac{2\beta-1}{\beta}}\right] + O(\varepsilon_D).$$

When $\beta > 1/2$, we have $0 < 2\beta - 1 \leq 2\beta$. Hence, when $t \geq d^{I-1}/K, P_* \geq C^{1/\beta}$ the first term can be merged into the first term. Therefore,

$$\text{RHS}(t) \leq C_\beta \left[\left(\frac{Kt}{d^{I-1}}\right)^{-(2\beta-1)/\beta} \vee P_*^{-(2\beta-1)}\right] + O(\varepsilon_D).$$

We next consider the LHS. In Lemma E.1, we proved that $a_{\pi(P_*)} \bar{v}_{P_*,\pi(P_*)}^2 \leq 4a_{P_*} \log(24mP_*/\delta_{\mathbb{P}})/d$ with probability $1 - \delta_{\mathbb{P}}$. Repeating the argument for all $p \in [P_*]$ and union bounding, with probability $1 - \delta_{\mathbb{P}}$ we have that $a_{\pi(p)} \bar{v}_{p,\pi(p)}^2 \leq 4a_p \log(24mP_*^2/\delta_{\mathbb{P}})/d$ for $p \in [P_*]$. We can therefore upper bound the LHS as

$$\sum_{p \in [P_*]} a_{\pi(p)}^2 \mathbb{1}\left(t \geq \frac{1 - o(1)}{4I(I-1)\hat{\sigma}_{2I}^2 \eta a_{\pi(p)} \bar{v}_{p,\pi(p)}^{2I-1}(0)}\right) \leq \sum_{p \in [P_*]} a_{\pi(p)}^2 \mathbb{1}\left(t \geq \frac{\tilde{c}d^{I-1}}{\eta a_p \log^{2I-2} m}\right)$$

$$\leq \sum_{p \in [P_*]} a_p^2 \mathbb{1}\left(t \geq \frac{\tilde{c}d^{I-1}}{\eta a_p \log^{2I-2} m}\right).$$

Letting $k = \eta Z^{-1} \tilde{c}^{-1} \log^{2I-2} m$, we can similarly write

$$\text{LHS}(t) \geq \frac{1}{2Z^2} \sum_{p=1}^{P} p^{-2\beta} \mathbb{1}\left\{t \leq \frac{d^{I-1}}{kp^{-\beta}} \vee p \geq P_*\right\} - O(\varepsilon_D)$$

$$\geq \frac{1}{2Z^2} \sum_{p=1}^{P} p^{-2\beta} \mathbb{1}\left\{p \geq (kt/d^{I-1})^{1/\beta} \wedge P_*\right\} - O(\varepsilon_D)$$

$$\geq \frac{1}{2Z^2} \int_{(kt/d^{I-1})^{1/\beta} \wedge P_*}^{P} q^{-2\beta}\, \mathrm{d}q - O(\varepsilon_D)$$

$$\geq \frac{1}{2Z^2} \frac{1}{2\beta - 1}\left(\left(\frac{kt}{d^{I-1}}\right)^{-(2\beta-1)/\beta} \vee P_*^{-(2\beta-1)} - P^{1-2\beta}\right) - O(\varepsilon_D).$$

When $t \leq 2^{-\beta/(2\beta-1)} P^\beta d^{I-1}/k$, the last term can be merged into the second last term. This gives the lower bound

$$\text{LHS}(t) \geq c_\beta \left[\left(\frac{kt}{d}\right)^{-(2\beta-1)/\beta} \vee P_*^{-(2\beta-1)}\right] - O(\varepsilon_D).$$

Altogether, the desired claim in part (b) follows from choosing $P_* = \Theta(\frac{m}{\log m})$.

Finally, we observe that Lemma E.1 implies that all directions $e_p$ with $p \leq P_* C^{-1/\beta} = \tilde{\Theta}(\frac{m}{\log m})$ are learned, and Theorem 2.1 implies that this learning happens at time $\tilde{\Theta}(p^\beta d^{I-1}\eta^{-1})$. The conclusion in part (a) directly follows. □

