# OpenReview forum: "Emergence and scaling laws in SGD learning of shallow neural networks"
_NeurIPS.cc/2025/Conference — NeurIPS 2025 poster_

### Official Review · Reviewer_uVcd · 2025-06-30

**Clarity:** 4
**Significance:** 1
**Originality:** 2
**Rating:** 4
**Confidence:** 2

**Summary:**

The authors study the scaling laws exponents for the MSE for learning 2 later neural networks with even activations of information exponent larger than 2 and power law weights in the second layer. The analysis starts by looking at the population loss and showing that the different hidden units decouple. The MSE is then the sum of the error on each of the hidden units. Having a large separation of scale on the weights in the second layer will imply an effective power-law scaling in the MSE.

**Questions:**

1. Would there still be additivity in the tasks if you were to consider the gradient flow on the empirical loss?
2. Could you sketch what you expect to happen for lower information exponents?

**Ethical Concerns:**

["NO or VERY MINOR ethics concerns only"]

**Final Justification:**

The authors clarified my questions, especially regarding the choice of activation function. I happily raised my rating to 4.

**Limitations:**

Yes

**Paper Formatting Concerns:**

No concerns

**Quality:**

3

**Strengths And Weaknesses:**

The paper is clear and well written. The separation between the learning of the different hidden units is well supported by the proof, and the final result is very intuitive.
I believe the biggest weaknesses of this work lie in the poor applicability of the results and in the limitation of the model under study. Common activations used in practice do not have high information exponent, and it's not clear to me that the same additivity property that the authors exploit will translate to arbitrary functions. I am unsure about the significance of this work. On one hand, there are not many works on power-law scalings for linear-width 2-layer networks, yet the activations choice is so restrictive that the problem is essentially reduced to a stack of generalised linear model, studied in a limit where they interact very weakly.

---

> ### Author Rebuttal · Authors · 2025-07-31
>
> Thank you for your detailed review. We respond to your questions and concerns below.
>
> >“...the activations choice is so restrictive that the problem is essentially reduced to a stack of generalised linear model, studied in a limit where they interact very weakly.”
>
> - We would first like to emphasize that our analysis is not conducted in any limit of weak interaction between neurons. Indeed, there is no reason why the neurons should not interact a priori, and a technical contribution of our work is that we prove SGD implicitly decouples the learning of the different neurons. Moreover, if $v$ is the student neuron closest (in the weighted sense) to teacher neuron $e_p$ at initialization, it is not necessarily true that $v$ will converge to $e_p$. This is in contrast to prior works which explicitly assume that there is no interaction between the different tasks, and this is why we can remove the exponential dependence on the condition number.
>
> - Next, in the deep learning theory literature, the high information exponent regime has received significant prior interest [BAGJ21, DNGL23, OSSW24, SBH24], both since varying the information exponent controls the optimization hardness of the non-convex learning problem, and due to the connections between the IE>2 regime and higher-order tensor decompositions [LMZ20, GRWZ21]. Our work is the first to remove the exponential dependence on the condition number via a natural gradient-based algorithm, which we believe is a significant contribution in its own right.
>
> - Finally, while activation functions in practice do not necessarily have high information exponents, our work models how emergence and scaling laws can arise due to gradient-based feature learning. The “linear representation hypothesis” posits that in pretrained language models, individual skills and concepts correspond to single directions in parameter space [DDH+21, EHO+22, PSZA23]. The additive model target which we study is an abstraction for LLM pretraining -- each individual “skill” is captured by a single neuron, while the high information exponent indicates that the learning of each skill has a long search phase. We thus believe that our work is an interesting model for understanding both emergence of single skills and overall scaling laws in LLMs more generally.
>
>
> > “Would there still be additivity in the tasks if you were to consider the gradient flow on the empirical loss?”
>
> - We provide a complete analysis for gradient flow on the population loss in Appendix C. Via a uniform concentration bound on the gradient, our results will extend to gradient flow on the empirical loss as long as the sample size is $d^{\Theta(I)}$.
>
> - We also remark that studying online SGD allows us to equate the sample complexity and the number of optimization steps, as is done in many prior works [AAM23, BAGJ21, DNGL23]. This also more closely matches the LLM pretraining paradigm, where each sample is repeated only a few times.
>
> > “Could you sketch what you expect to happen for lower information exponents?”
> - We first note that we have replicated the experiment in Figure 2 for the ReLU activation, and observe that after the lower Hermite modes are learned, the same scaling exponent is observed. We will add these plots to the next revision.
> - Theoretically, the situation is more complicated when the information exponent is 2 (the lowest possible for even link functions). Our argument does not work directly in this case, but we expect the overall behavior to be similar.
>   - If the 2nd order (IE=2) terms are isotropic, then the model will first identify the relevant P-dimensional subspace and the distribution of the learner neurons will be close the standard Gaussian in the relevant subspace (see [RL24] for details). After learning the subspace, we can essentially pretend that the relevant subspace is the ambient space. For technical reasons, our proof does not directly work when $P_* = P = d$. However, we believe that when $P_* \ll P = d$, the same argument, after some minor modifications, yields the same dynamics and scaling laws.
>   - The case where the 2nd order terms are anisotropic is tricker. First, we can no longer use small initialization in this case, as the neurons will all collapse to the largest direction (think of the behavior of the matrix power method). Now, suppose that we initialize the neurons so that the 0th order terms match. Then, one should expect that, again, the subspace will get learned first, and the distribution of the learner neurons will become the Gaussian distribution (in that subspace) with covariance matrix determined by the 2nd order terms of the target function. (This behavior is shown in [LMZ20], though their bounds are very loose.) Similar to the previous case, we assume that $P_* \ll P = d$ and scale down the norm of the neurons, so that our argument can be applied here. Then, in this case, the initial correlation with direction $p$ will have order $(a_p v_p^2)^{I-1}$, instead of $v_p^{2I-2}$, and this direction will be learned around time $( a_p^I v_p^{2I-2} )$, instead of $( a_p v_p^{2I-2} )$. Effectively, this changes the power law of $(a_p)_p$ from $p^{-\beta}$ to $p^{- I \beta}$.
>
>
> [AAM23] Emmanuel Abbe, Enric Boix Adsera, and Theodor Misiakiewicz. SGD learning on neural networks: leap complexity and saddle-to-saddle dynamics. In The Thirty Sixth Annual Conference on Learning Theory, pages 2552–2623. PMLR, 2023.
>
> [BAGJ21] Gerard Ben Arous, Reza Gheissari, and Aukosh Jagannath. Online stochastic gradient descent on non-convex losses from high-dimensional inference. The Journal of Machine Learning Research, 22(1):4788–4838, 2021.
>
> [DDH+21] Damai Dai, Li Dong, Yaru Hao, Zhifang Sui, Baobao Chang, and Furu Wei. Knowledge neurons in pretrained transformers. arXiv preprint arXiv:2104.08696, 2021.
>
> [DNGL23] Alex Damian, Eshaan Nichani, Rong Ge, and Jason D. Lee. Smoothing the landscape boosts the signal for SGD: Optimal sample complexity for learning single index models. In Thirty-seventh Conference on Neural Information Processing Systems, 2023.
>
> [EHO+22] Nelson Elhage, Tristan Hume, Catherine Olsson, Nicholas Schiefer, Tom Henighan, Shauna Kravec, Zac Hatfield-Dodds, Robert Lasenby, Dawn Drain, Carol Chen, et al. Toy models of superposition. arXiv preprint arXiv:2209.10652, 2022.
>
> [GRWZ21] Rong Ge, Yunwei Ren, Xiang Wang, and Mo Zhou. Understanding deflation process in over-parametrized tensor decomposition. In Proceedings of the 35th International Conference on Neural Information Processing Systems, 2021.
>
> [LMZ20] Yuanzhi Li, Tengyu Ma, Hongyang R. Zhang. Learning Over-Parametrized Two-Layer ReLU Neural Networks beyond NTK. 2020
>
> [OSSW24] Kazusato Oko, Yujin Song, Taiji Suzuki, and Denny Wu. Learning sum of diverse features: computational hardness and efficient gradient-based training for ridge combinations. In Conference on Learning Theory. PMLR, 2024.
>
> [PSZA23] Abhishek Panigrahi, Nikunj Saunshi, Haoyu Zhao, and Sanjeev Arora. Task-specific skill localization in fine-tuned language models. arXiv preprint arXiv:2302.06600, 2023.
>
> [RL24] Yunwei Ren, Jason D. Lee. Learning Orthogonal Multi-Index Models: A Fine-Grained Information Exponent Analysis. 2024.
>
> [SBH24] Berfin Simsek, Amire Bendjeddou, and Daniel Hsu. Learning gaussian multi-index models with gradient flow: Time complexity and directional convergence. arXiv preprint arXiv:2411.08798, 2024.
>
> ---
>
> We would be happy to answer any follow-up questions in the discussion period.

---

> > ### Comment · Reviewer_uVcd · 2025-08-02
> >
> > Thanks for your detailed answer. I am happy to raise my score

---

### Official Review · Reviewer_yyk4 · 2025-07-01

**Clarity:** 3
**Significance:** 3
**Originality:** 3
**Rating:** 6
**Confidence:** 4

**Summary:**

This paper provides a precise analysis of online stochastic gradient descent (SGD) for training two-layer neural networks on isotropic Gaussian data, in the challenging “extensive-width” regime $P\gg1$. The target function is

$f^*(x) = \sum_{p=1}^P a_p \sigma ( \langle v_p^\*, x \rangle ), x \sim \mathcal{N} (0,I_d)$,

where the activation $\sigma$ is even with information exponent $k^\*>2$, the signal directions $v_p^\*$ are orthonormal, and the second-layer weights $a_p$ may follow a power law $a_p\asymp p^{-\beta}$. The authors identify **sharp transition times** at which each teacher neuron is recovered, and—by overlaying these abrupt emergences—derive **smooth neural scaling laws** for the cumulative mean-squared error (MSE) as a function of sample size, training steps, and width.

Key results include:

- **Complexity bounds** (Theorem 2.1) quantifying when SGD exits a long “search phase” and enters a rapid “descent phase” for each feature.
- **Scaling-law exponents** $\alpha = \frac{1-2\beta}{1+\beta}$ characterizing how the MSE decays as a power law in compute, data, or width under power-law weight tails.
- **Simulations** (Section 2.4) confirming the theoretical slopes for Hermite-activated networks and illustrating the compute-optimal frontier.

**Questions:**

## Questions for the Authors
1. How might the scaling-law exponents change if $x$ comes from a non-Gaussian or anisotropic distribution? What barriers exist to extending the analysis beyond $\mathcal{N}(0,I)$?
2. Can the techniques handle ReLU or GeLU activations, which are neither even nor finite-Hermite? Which parts of the proof would break?
3. The theory requires $\eta$ to scale inversely with $d^2 a_p$ to control fluctuations. In practice, did you find reasonable rates that match theory, or is there a gap?
4. Could a simple decay schedule (e.g.\ $\eta_t\propto t^{-\gamma}$) recover the anytime scaling law without knowing the total number of steps in advance? What are the main technical hurdles?
5. Do you foresee obstacles to applying your emergent-plus-scaling-law framework to multi-layer or residual architectures? What conceptual changes would be required?
6. Figure 2 shows matching slopes between idealized and empirical loss curves for $\sigma = h_4$. Have the authors tried other Hermite orders or common activations (e.g. ReLU)? How consistent is the predicted exponent $(1 - 2\beta)/(1 + \beta)$ in those cases?

**Ethical Concerns:**

["NO or VERY MINOR ethics concerns only"]

**Final Justification:**

This is an excellent paper, and the authors solved all my concerns.

**Limitations:**

yes

**Paper Formatting Concerns:**

The reviewer has no formatting concerns.

**Quality:**

4

**Strengths And Weaknesses:**

## Strengths
- **Rigorous theory connecting emergent dynamics to scaling laws.** Bridges sharp feature-recovery transitions with smooth empirical power laws.
- **Extensive-width regime treatment.** Handles $P\gg1$ and allows diverging condition numbers, covering realistic over-parameterization scenarios.
- **Sharp, non-asymptotic bounds.** Provides concrete rates for plateaus and descents in terms of problem parameters $(d,\beta,k^*,\eta)$.
- **Empirical validation.** Figure 2 demonstrates close alignment between idealized theory and SGD simulations, including error bars and absolute loss curves.

## Weaknesses
- **Gaussian and orthonormal assumptions.** All results hinge on $x\sim\mathcal{N}(0,I_d)$ and exactly orthonormal $v_p^*$, limiting applicability to more structured or correlated data.
- **Even activation with high Hermite order.** Requires $\sigma$ be even with information exponent $k^*\ge 3$, excluding common odd or non-polynomial activations (e.g.\ ReLU, GeLU).
- **Learning-rate constraints.** Ensuring martingale fluctuations remain small demands $\eta\ll d^{-2}a_p$; such small rates may be impractical for moderate $d$.
- **Idealized two-layer setting.** Extensions to deeper architectures, non-additive targets, or realistic neural networks remain unaddressed.
- **Limited exploration of learning-rate schedules.** Corollary 2.3 requires knowing the total number of SGD steps in advance; adaptive or decaying schedules are not analyzed.

---

> ### Author Rebuttal · Authors · 2025-07-31
>
> Thank you for your detailed review, and for the positive evaluation of our paper. We respond to your questions below.
>
> **How might the scaling-law exponents change if x comes from a non-Gaussian or anisotropic distribution?**
>
> - One challenge is that when the input distribution is not Gaussian/spherically symmetric, very little is known even for the simplest single-index case. The only work we are aware of that deals with reasonably general input distributions is [BPVZ23]. They use a universality argument to show that similar convergence/sample complexity guarantees hold when the input distribution is close to the standard Gaussian. While it might be possible to extend our results to this setting, it is likely quite difficult to derive similar results for general non-Gaussian input distributions. In fact, when the target direction and the input distribution are sufficiently misaligned, even recovering a single-index model is not guaranteed.
>
> - Our results can potentially be generalized to non-isotropic Gaussians. If one assumes the spectrum of the covariance matrix also follows a power law (with a potentially different exponent), one might be able to get a two-parameter scaling law. However, we will need to use elliptical normalization, as in [MHWSE23], to ensure the preactivation is still isotropic.
>
> **Can the techniques handle ReLU or GeLU activations, which are neither even nor finite-Hermite?**
>
> - *Functions with IE=1:* First, we wish to mention that if we assume the target function is symmetric in the sense that both $v_k^\*$ and $-v_k^\*$ are present, then the target will automatically be even and contain no first-order terms. If the target is not symmetrized and has a linear term, then all neurons will first be attracted to the direction that is the weighted average of all teacher neurons. This cannot be cleanly fixed by using, say, a large initialization, as the effect of the first-order terms is shifting instead of rescaling. One trick to solve this issue is to learn the first-order terms first and then explicitly remove it from the target function. The time/samples needed for this step is much smaller than the later stages, so this will not change the scaling law. So far, we do not know any other fix that is more natural.
>
> - *Functions with IE=2:* Theoretically, the situation is more complicated when the information exponent is 2 (the lowest possible for even link functions). Our argument does not work directly in this case, but we expect the overall behavior to be similar.
>   - If the 2nd order (IE=2) terms are isotropic, then the model will first identify the relevant P-dimensional subspace and the distribution of the learner neurons will be close the standard Gaussian in the relevant subspace (see [RL24] for details). After learning the subspace, we can essentially pretend that the relevant subspace is the ambient space. For technical reasons, our proof does not directly work when $P_* = P = d$. However, we believe that when $P_* \ll P = d$, the same argument, after some minor modifications, yields the same dynamics and scaling laws.
>   - The case where the 2nd order terms are anisotropic is tricker. First, we can no longer use small initialization in this case, as the neurons will all collapse to the largest direction (think of the behavior of the matrix power method). Now, suppose that we initialize the neurons so that the 0th order terms match. Then, one should expect that, again, the subspace will get learned first, and the distribution of the learner neurons will become the Gaussian distribution (in that subspace) with covariance matrix determined by the 2nd order terms of the target function. (This behavior is shown in [LMZ20], though their bounds are very loose.) Similar to the previous case, we assume that $P_* \ll P = d$ and scale down the norm of the neurons, so that our argument can be applied here. Then, in this case, the initial correlation with direction $p$ will have order $(a_p v_p^2)^{I-1}$, instead of $v_p^{2I-2}$, and this direction will be learned around time $( a_p^I v_p^{2I-2} )$, instead of $( a_p v_p^{2I-2} )$. Effectively, this changes the power law of $(a_p)_p$ from $p^{-\beta}$ to $p^{- I \beta}$.
>
> - *Non-even functions:* For non-even functions, the issue is that there will be a ½ chance that the “good” neuron lies in the wrong hemisphere, and there will be no way to detect this early on when the IE is even. Unfortunately, our analysis is quite sensitive to these type of bad neurons.
>
> - *Non-polynomial function:* The only place we need the link function to be a polynomial is to get a tail bound on the per-sample gradient. We have realized that this condition can be relaxed to requiring the derivative of the link function to be polynomially bounded. We will switch to this condition in the revision.
>
> **Learning rate constraint.**
>
> - The learning rate scaling we consider of $\eta \lesssim d^{-I}$ is a common learning rate for feature learning via online SGD, and leads to the “optimal” rate for learning single-index models via online SGD [BAGJ21].
>
> **Could a simple decay schedule recover the anytime scaling law?**
>
> - It is possible, but not entirely clear, that a decaying learning rate schedule, where $\eta$ is proportional to $a_p$ during the time period where $v_p$ is learned, could recover the anytime scaling law. One advantage of choosing a small constant learning rate is that this allows partial progress to be made from the beginning, even for the smaller directions. The technical challenge of using a decaying learning rate is ensuring that the total contribution of the martingale term is negligible relative to the initialization, which requires a subtle argument since a priori a large learning rate early in training could blow up the martingale term for the later neurons. One could expect to recover the anytime scaling law if these noises can somehow be controlled. Otherwise, one should expect to get a distorted scaling law, as the learning of a small direction starts only after the learning rate becomes sufficiently small. We leave understanding this to future work.
>
> **Do you foresee obstacles to applying your emergent-plus-scaling-law framework to multi-layer or residual architectures?**
>
> - There are currently extremely few guarantees for feature learning via SGD in architectures beyond two-layer networks, such as multi-layer networks with residual connections. Since results for SGD in far simpler settings do not exist for such architectures, it seems quite challenging to rigorously derive a scaling law in the extensive width regime.
>
> - However, we remark that the “linear representation hypothesis” posits that in pretrained neural networks, individual skills and concepts correspond to single directions in parameter space [DDH+21, EHO+22, PSZA23]. The additive model target which we study is an abstraction for LLM pretraining, which requires the learning of many distinct skills. We thus believe that, at least conceptually, our emergent-plus-scaling-law framework applies to other neural network architectures more generally.
>
> **Figure 2 shows matching slopes between idealized and empirical loss curves for He4. Have the authors tried other Hermite orders or common activations (e.g. ReLU)?**
>
> - We have replicated Figure 2 for the ReLU activation, and observe that after the lower Hermite modes are learned, the same scaling exponent is observed. We will add these plots to the next revision.
>
> [BAGJ21] Gerard Ben Arous, Reza Gheissari, and Aukosh Jagannath. Online stochastic gradient descent on non-convex losses from high-dimensional inference. The Journal of Machine Learning Research, 22(1):4788–4838, 2021.
>
> [BPVZ23] Joan Bruna, Loucas Pillaud-Vivien, Aaron Zweig. On Single Index Models beyond Gaussian Data. 2023
>
> [DDH+21] Damai Dai, Li Dong, Yaru Hao, Zhifang Sui, Baobao Chang, and Furu Wei. Knowledge neurons in pretrained transformers. arXiv preprint arXiv:2104.08696, 2021.
>
> [EHO+22] Nelson Elhage, Tristan Hume, Catherine Olsson, Nicholas Schiefer, Tom Henighan, Shauna Kravec, Zac Hatfield-Dodds, Robert Lasenby, Dawn Drain, Carol Chen, et al. Toy models of superposition. arXiv preprint arXiv:2209.10652, 2022.
>
> [MHWSE23] Alireza Mousavi-Hosseini, Denny Wu, Taiji Suzuki, and Murat A. Erdogdu. Gradient-based feature learning under structured data. In Thirty-seventh Conference on Neural Information Processing Systems (NeurIPS 2023), 2023.
>
> [PSZA23] Abhishek Panigrahi, Nikunj Saunshi, Haoyu Zhao, and Sanjeev Arora. Task-specific skill localization in fine-tuned language models. arXiv preprint arXiv:2302.06600, 2023.
>
> ---
>
> We would be happy to answer any follow-up questions in the discussion period.

---

> > ### Comment · Reviewer_yyk4 · 2025-08-06
> >
> > I'm more than happy to raise my score. This is an excellent paper!

---

### Official Review · Reviewer_vSW9 · 2025-07-01

**Clarity:** 2
**Significance:** 3
**Originality:** 3
**Rating:** 4
**Confidence:** 2

**Summary:**

This paper studies SGD in a very simple context and provides strong results characterizing scaling laws of SGD as a function of all the main things of interest (# training samples, amount of training, # of parameters).  A core analysis result is that scaling laws emerge from superposition of neurons with different timescales of learning dynamics.

**Questions:**

-- The main thing I'd want to see is some kind of empirical investigation in a much more real-world situation, e.g. like Resnet-18 learning imagenet categorization, or a GPT-2 learning process.   Is there some way to compare the theory to the empirical results in those kinds of cases?  I'm not stuck on that particular evaluation, but I'm open to anything that would take these results out of such a simple highly restricted domain of assumptions.

-- Is there some way to indicate how the scaling laws would change in other cases, like deep convnets or transformers?

**Ethical Concerns:**

["NO or VERY MINOR ethics concerns only"]

**Final Justification:**

I think basically this is a good paper but I am not that competent to judge it.    My main complaint is that it seems pretty far from being applicable to anything I can imagine people working on practically.

**Quality:**

3

**Strengths And Weaknesses:**

Strengths:
-- the motivation for knowing the forms of these scaling curves is very clear
-- the analysis presented is very complete and clear
-- the P>>1 regime is often not studied so this fills in an important gap in the literature
-- emergence result is very interesting.

Weaknesses:
-- the setting is so simple it seems to me quite unlikely the methods will generalize to real-world situations.  so it's hard to know how applicable results will be to real situations.
-- there is no real attempt to measure empirically if similar outcomes arise in more complex real-world situations, to which the theory can at least be compared

---

> ### Author Rebuttal · Authors · 2025-07-31
>
> Thank you for your detailed review. We respond to your questions and concerns below.
>
> > “the setting is so simple it seems to me quite unlikely the methods will generalize to real-world situations. so it's hard to know how applicable results will be to real situations.”
>
> The goal of our work is to reconcile the phenomena of emergent learning curves and smooth power law decay in the loss, both of which have been robustly observed in practice. We show that both phenomena arise in the SGD learning of a shallow neural network, thus demonstrating how scaling laws can arise in the feature learning regime. We note that:
> - Our studied model is already more “realistic” of a model for scaling laws than prior theoretical works, where most of the optimization analyses are restricted to linear regression with no learned features (see Appendix A for discussion). Importantly, it is well known that high-dimensional SGD in the feature learning regime typically exhibits emergent / staircase-like risk curves; this motivates us to consider the extensive-width and large condition number regime (in light of the additive model hypothesis) which requires significant technical novelty.
> - Our work provides theoretical justification to the additive model hypothesis of neural scaling laws in the context of SGD learning of neural networks, where unlike prior works [MLGT24, NFLL24], we do not assume explicitly decoupled tasks.
>
> > “The main thing I'd want to see is some kind of empirical investigation in a much more real-world situation, e.g. like Resnet-18 learning imagenet categorization, or a GPT-2 learning process. Is there some way to compare the theory to the empirical results in those kinds of cases?”
>
> - Firstly, we would like to emphasize that our goal is to show how emergence and scaling laws -- two phenomena which are commonly observed in practice -- arise via gradient-based feature learning; see the above point for additional discussion.
> - In addition, it is difficult to reasonably decompose real-world tasks into individual tasks, so that our analysis can be directly applied. However, we note that the prior work [MLGT24] provides empirical evidence that for language models, the overall loss can be decomposed into a superposition of emergent learning curves. Our work complements these empirical results by providing theoretical justification in the setting of SGD learning of a two-layer network.
>
> > “Is there some way to indicate how the scaling laws would change in other cases, like deep convnets or transformers?”
>
> - It is indeed an interesting question to understand theoretically how scaling laws vary for different architectures. We note, however, that there are extremely few guarantees for gradient descent on transformers, all of which are restricted to one or two self-attention layers. Even for two-layer neural networks, our work is the first polynomial width guarantee in the diverging condition number and extensive width regime. We believe that establishing such a scaling law for gradient-based feature learning in two-layer networks is a necessary first step before deriving architecture-specific scaling laws for transformers, and is a significant contribution in its own right.
>
> - We would also like to remark that the “linear representation hypothesis” posits that in pretrained language models, individual skills and concepts correspond to single directions in parameter space [DDH+21, EHO+22, PSZA23]. The additive model target which we study is an abstraction for LLM pretraining, which requires the learning of many distinct skills, and we thus believe that this is an interesting model for both the emergence of single skills and overall scaling laws in LLMs more generally.
>
> [DDH+21] Damai Dai, Li Dong, Yaru Hao, Zhifang Sui, Baobao Chang, and Furu Wei. Knowledge neurons in pretrained transformers. arXiv preprint arXiv:2104.08696, 2021.
>
> [EHO+22] Nelson Elhage, Tristan Hume, Catherine Olsson, Nicholas Schiefer, Tom Henighan, Shauna Kravec, Zac Hatfield-Dodds, Robert Lasenby, Dawn Drain, Carol Chen, et al. Toy models of superposition. arXiv preprint arXiv:2209.10652, 2022.
>
> [MLGT24] Eric Michaud, Ziming Liu, Uzay Girit, and Max Tegmark. The quantization model of neural scaling. Advances in Neural Information Processing Systems, 36, 2024.
>
> [NFLL24] Yoonsoo Nam, Nayara Fonseca, Seok Hyeong Lee, and Ard Louis. An exactly solvable model for emergence and scaling laws. arXiv preprint arXiv:2404.17563, 2024.
>
> [PSZA23] Abhishek Panigrahi, Nikunj Saunshi, Haoyu Zhao, and Sanjeev Arora. Task-specific skill localization in fine-tuned language models. arXiv preprint arXiv:2302.06600, 2023.
>
> ---
>
> We would be happy to answer any follow-up questions in the discussion period.

---

> > ### Comment · Reviewer_vSW9 · 2025-08-03
> > **Resposne to rebuttal**
> >
> > What I am sure I can say effectively as that as I really not a theorist, it is very hard for me to evaluate this work properly in context.
> >
> > Frankly, this work seems very carefully done and solid technically.   That is my subjective impression as a non-expert, but also seems confirmed by the other reviewers who are obviously much more expert that I am.
> >
> > I think where I can say something as a AI/ML practitioner, I have a very very hard time understanding how this work will ever apply to anything I think about.   Maybe that is not a very important criterion for this kind of work.   I tend to view the comments by the authors in rebuttal as basically saying I'm applying an unfair criterion to this whole domain.  That could be true.   But I don't view the rebuttal as really attempting to answer my concerns, probably because the authors view my concerns as effectively unanswerable, at least not in the timeframe of a rebuttal.    I am not sure what else to say.
> >
> > I feel comfortable keeping my scored as it is, because of the above thoughts.   But perhaps the AC will feel freer disgregarding my thoughts, as I (admit) they are not very expert.

---

### Official Review · Reviewer_cXnh · 2025-07-03

**Clarity:** 4
**Significance:** 3
**Originality:** 4
**Rating:** 5
**Confidence:** 4

**Summary:**

The paper studies the online SGD dynamic for two-layer neural networks, in the limit where the width $P\gg 1$, data is generated for a Gaussian distribution, and labeled through another, fixed, teacher network, with orthogonal weights.
If the target function is hard enough (information exponent $>2$), the authors are able to prove the convergence and sharp transition of student neurons to teacher ones.
Moreover, by assuming a scaling on the second layer of the teacher, the paper predicts a scaling law of the population loss minimum, compared with the "total compute" measured as the product of time steps with the width of the student.

**Questions:**

- Can the mean field limit be connected to this result in some weight? I know the classical _mean field regime_ \[1,2\] is usually done for finite $d$, but there are some works that consider the limit $P\to+\infty$ before going $d\to+\infty$, such as \[3\]. I guess here you are considering the limit in the opposite order, but maybe there is still a connection.
- You say that "online SGD exhibits a long plateau" (line 131) for $IE>2$. Isn't the plateau present also for $IE=2$? Does the separations of timescales appear also in this case? I expect not, so there is no scaling law for $IE=2$, right?
###### Rereference
> \[1\] Song Mei, Andrea Montanari, and Phan-Minh Nguyen. A mean field view of the landscape of two-layer neural networks. Proceedings of the National Academy of Sciences

>\[2\] Lenaic Chizat and Francis Bach. On the global convergence of gradient descent for over-
parameterized models using optimal transport.

> \[3\] Luca Arnaboldi, Ludovic Stephan, Florent Krzakala, Bruno Loureiro. From high-dimensional & mean-field dynamics to dimensionless ODEs: A unifying approach to SGD in two-layers networks

**Ethical Concerns:**

["NO or VERY MINOR ethics concerns only"]

**Final Justification:**

My evaluation of the paper is definitely positive. The authors present a clear and formal theory for modeling scaling laws using a two-layer neural network. The authors replied promptly to my questions. I think the paper deserves to be accepted, but I'm not sure if it deserves to be in the Spotlight, given that the theory does not cover cases with $IE\le 2$.

**Limitations:**

yes

**Paper Formatting Concerns:**

No concerns

**Quality:**

3

**Strengths And Weaknesses:**

- The main theorem provides a neat result about the separation of the times where the neurons are learnt, that is unprecedented in the literature, to my best knowledge. Being able to characterize this for $P\to\infty$ is very surprising and remarkable.
- The formal math prediction of the scaling law is matched by numerical experiments, in Figure 2. It would be nice to append the same plot for different values of $\beta$ though.
- The setting is a bit artificial for two reasons: the assumption $IE>2$ is strong enough to exclude any reasonable or commonly used activation function, and the scaling laws emerge only when the teacher is crafted ad hoc to have power-law decaying weights in the second layer.

---

> ### Author Rebuttal · Authors · 2025-07-31
>
> Thank you for your detailed review, and for the positive evaluation of our paper. We respond to your questions below.
>
> **Can the mean field limit be connected to this result?**
>
> - Thank you for the interesting suggestion. In short, this depends on whether we require *polynomial-width* learnability.
>   - If we have infinitely/exponentially many neurons and use the homogeneous parameterization (which gives a Fisher-Rao type dynamics), then MF gradient flow can learn single-index models without the extensive search phase. Intuitively, this is because with that many neurons, all directions can be covered at initialization, whence it suffices to amplify the norm of the neurons that happen to be aligned with a ground truth direction. See Section 5.3 in [GWB25] for discussions.
>   - If one wishes to do a finite/polynomial width mean-field analysis, then one needs to properly truncate the mean-field initialization to remove those neurons that are close to the ground truth directions (the total mass of them is exponentially small). Then, one could try to control the difference between the infinite-width mean-field network and its finite-width counterpart. This is the approach adopted in [LMZ20]. We believe in principle, this strategy can lead to results similar to ours in some potentially more general settings. However, controlling the difference between an infinite-width network and a polynomial-width network for $\Omega(d)$ time is extremely technically challenging and does not seem to lead to the tight bounds.
>
> **Isn't the plateau present also for IE=2? Does the separations of timescales appear also in this case? Are there scaling laws for IE=2?**
>
> - There is indeed a plateau even when IE=2. In this case, however, there are no sharp transitions, which are usually defined as super-exponential blowups. To see this, consider the equation $\dot{v}\_{p, t} = a\_p x\_{p, t}$, which roughly captures the signal growth of the IE=2 case. Note that the solution to it is $v\_{p, t} = v\_{p, 0} e^{a\_p t}$. We can make two observations from it. First, if $v\_{p, 0} = (1 + c) v\_{p', 0}$, then their ratio will remain $1 + c$ throughout training. Second, if $a\_p = (1 + c) a\_q$, then $v\_{p, t} / v\_{p, 0} = ( v\_{q, t} / v\_{q, 0} )^{1+c}$. In other words, even though the gap from the initialization and different growth rates can be amplified, the amplification will not be strong enough to create a big separation when $c$ is small. This is in contrast to the high-IE setting, where small gaps can lead to very strong separations. See the discussions in Section 3.1 for details.
>
> - When IE=2, it might still be possible to derive some scaling laws, although the situation is more complicated and our argument does not work directly in this case. For example, if the 2nd order (IE=2) terms are isotropic, then we expect the model will first identify the relevant P-dimensional subspace and the distribution of the learner neurons will be close the standard Gaussian in the relevant subspace (see [RL24] for details). After learning the subspace, we can essentially pretend that the relevant subspace is the ambient space. For some technical reasons, our proof does not directly work when $P_* = P = d$. However, we believe that when $P_* \ll P = d$, the same argument, after some minor modifications, yields the same dynamics and scaling laws.
>
> [GWB25] Margalit Glasgow, Denny Wu, and Joan Bruna. Propagation of chaos in one-hidden-layer neural networks beyond logarithmic time. arXiv preprint arXiv:2504.13110, 2025.
>
> [LMZ20] Yuanzhi Li, Tengyu Ma, and Hongyang R Zhang. Learning over-parametrized two-layer neural networks beyond ntk. In Conference on learning theory, pages 2613–2682. PMLR, 2020.
>
> [RL24] Yunwei Ren, Jason D. Lee. Learning Orthogonal Multi-Index Models: A Fine-Grained Information Exponent Analysis. 2024.
>
> ---
>
> We would be happy to answer any follow-up questions in the discussion period.

---

> > ### Comment · Reviewer_cXnh · 2025-08-01
> >
> > I thank the authors for the answer and the references; I confirm my acceptance recommendation for this paper.

---

### Decision · Program_Chairs · 2025-09-17

**Decision:**

Accept (poster)

**Comment:**

This paper studies a two-layer NN with an even activation with information exponent $k_*>2$, and its
emergent learning dynamics with sharp transition and scaling law (from the juxtaposition of emergent curves) with respect to the number of training samples/SGD steps/parameters.

The reviewers appreciate the **strengths** of the paper:
(cXnh) neat result, very surprising and remarkable, formal math prediction is matched by numerical experiments,
(vSW9) clear motivation, complete and clear analysis, fills in an important gap in the literature,
(yyk4) rigorous theory connecting emergent dynamics to scaling laws, realistic over-parameterization scenarios, sharp, non-asymptotic bounds, empirical validation,
(uVcd) clear and well written, well supported by the proof, intuitive result.

The reviewers also find the **weaknesses** of the paper:
(cXnh) a bit artificial setting,
(vSW9) no attempt to empirically validate the results in real-world situation,
(yyk4) limited assumptions, excluding common odd or non-polynomial activations, idealized two-layer settings.
(uVcd) poor applicability of the results and the limitation of the model (e.g,. activations).

**After the discussion**, the reviewers agreed to accept the paper as it gives a theoretical contribution despite its limited setting.